

# Systematics of the Rubidgeinae (Therapsida: Gorgonopsia)

Christian F. Kammerer

Museum für Naturkunde, Leibniz-Institut für Evolutions- und Biodiversitätsforschung, Berlin, Germany

## ABSTRACT

The subfamily Rubidgeinae, containing the largest known African gorgonopsians, is thoroughly revised. Rubidgeinae is diagnosed by the absence of a blade-like parasphenoid rostrum and reduction or absence of the preparietal. Seven rubidgeine species from the Karoo Basin of South Africa are recognized as valid: *Aelurognathus tigriceps*, *Clelandina rubidgei*, *Dinogorgon rubidgei*, *Leontosaurus vanderhorsti*, *Rubidgea atrox*, *Smilesaurus ferox*, and *Sycosaurus laticeps*. Rubidgeines are also present in other African basins: *A. tigriceps* and *S. laticeps* occur in the Upper Madumabisa Mudstone Formation of Zambia, and *D. rubidgei*, *R. atrox*, and the endemic species *Ruhuhucerberus haughtoni* comb. nov. and *Sycosaurus nowaki* comb. nov. occur in the Usili Formation of Tanzania. *Aelurognathus nyasaensis* from the Chiweta Beds of Malawi also represents a rubidgeine, but of uncertain generic referral pending further preparation. No rubidgeine material is known outside of Africa: the purported Russian rubidgeine *Leogorgon klimovensis* is not clearly referable to this group and may not be diagnosable. Phylogenetic analysis of rubidgeines reveals strong support for a clade (Rubidgeini) of advanced rubidgeines including *Clelandina*, *Dinogorgon*, *Leontosaurus*, and *Rubidgea*. Support for *Smilesaurus* as a rubidgeine is weak; it may, as previous authors have suggested, represent an independent evolution of large body size from an *Arctops*-like ancestor. Temporally, rubidgeines are restricted to the Late Permian, first appearing in the *Tropidostoma* Assemblage Zone and reaching highest diversity in the *Cistecephalus* and *Daptocephalus* assemblage zones of the Beaufort Group.

## INTRODUCTION

Gorgonopsians are among the most iconic of Permian animals, and feature prominently in popular literature on the period (e.g., *Ward, 2004*). This popular attention, however, belies a remarkable lack of scientific interest. In the last 50 years, only a handful of papers have been published on the African record of Gorgonopsia, our primary source of data on the group (*Kemp, 1969*; *Sigogneau, 1968*; *Sigogneau, 1970*; *Cruickshank, 1973*; *Parrington, 1974*; *Sigogneau-Russell, 1989*; *Laurin, 1998*; *Maisch, 2002*; *Gebauer, 2014*; *Kammerer, 2014*; *Kammerer, 2015*; *Kammerer et al., 2015*). No new South African gorgonopsian taxa have been named since 1959, despite the subsequent discovery of hundreds of new specimens in the Karoo Basin. This sad state of affairs can be attributed

Corresponding author
Christian F. Kammerer,
christian.kammerer@mfn-berlin.de

almost entirely to the chaotic state of gorgonopsian systematics. Although the Karoo therapsid fauna in general was badly oversplit by Robert Broom (*Wyllie, 2003*), the homomorphism of gorgonopsian crania has made revision of this group particularly difficult. The confusion surrounding gorgonopsian alpha taxonomy has also hindered higher level systematic study: no published phylogeny of Gorgonopsia exists, and its position within Therapsida remains volatile (*Rubidge & Sidor, 2001*).

Although the relationships of taxa within Gorgonopsia are largely unknown, one distinctive subclade has long been recognized within the group. *Watson & Romer (1956)* recognized a close relationship between the gigantic, heavily-pachyostosed African gorgonopsian genera *Rubidgea* and *Dinogorgon* and *Sigogneau (1970)* also included *Clelandina* and *Sycosaurus* in this group, as the subfamily Rubidgeinae *Broom, 1938*. Rubidgeines include the largest African gorgonopsians, with basal skull lengths exceeding 40 cm in several genera. Members of this subfamily constitute the top predators of African terrestrial ecosystems in the Late Permian, and their fossils are common in rocks of the *Cistecephalus* and *Daptocephalus* (sensu *Viglietti et al., 2016*; formerly *Dicynodon*) assemblage zones (AZs) of the Karoo (*Smith, Rubidge & van der Walt, 2012*).

A few attempts have been made at reining in the unsatisfactory state of gorgonopsian systematics. Of particular import are two monographic revisions of Gorgonopsia, both of which substantially altered rubidgeine alpha taxonomy, based on the doctoral dissertations of *Sigogneau (1970*; expanded to non-South African taxa in *Sigogneau-Russell 1989)* and *Gebauer (2007)*. *Sigogneau (1970)* recognized six genera and 17 species of rubidgeines: *Broomicephalus* (containing one species: *B. laticeps*), *Clelandina* (containing two species: *C. rubidgei* and *C. scheepersi*), *Dinogorgon* (containing three species: *D. rubidgei*, *D. quinquemolaris*, and *D. pricei*), *Prorubidgea* (containing five species: *P. maccabei*, *P. alticeps*, *P. brinki*, *P. brodiei*, and *P. robusta*), *Rubidgea* (containing three species: *R. atrox*, *R. platyrhina*, and *R. majora*), and *Sycosaurus* (containing three species: *S. laticeps*, *S. vanderhorsti*, and ?*S. kingoriensis*). *Sigogneau-Russell (1989)* followed *Tatarinov (1977)* in including the Russian taxon *Niuksenitia sukhonensis* in Rubidgeinae, but questioned this referral and suggested that this species may have closer affinities with burnetiamorphs, a hypothesis borne out by more recent research (*Ivakhnenko et al., 1997*; *Sidor & Welman, 2003*). *Sigogneau (1970)* considered *Aelurognathus*, '*Cephalicustriodus*' (UMZC T891), and *Leontocephalus* to lie outside of Rubidgeinae, and considered *Clelandina major*, *Gorgonognathus maximus*, *Gorgonorhinus luckhoffi*, and *Rubidgea kitchingi* to be nomina dubia.

*Gebauer (2007)* significantly revised the previous generic taxonomy, synonymizing *Prorubidgea* with *Aelurognathus*, *Cephalicustriodus* and *Leontocephalus* with *Sycosaurus*, and *Broomicephalus* with *Clelandina*. She also synonymized *Ruhuhucerberus*, established by *Maisch (2002)* for the Cambridge '*Cephalicustriodus*' specimen (UMZC T891), with *Sycosaurus*, albeit as a valid species (*S. terror*). Additionally, she considered the type species of *Dinogorgon* (*D. rubidgei*) to be indeterminate, and transferred the remaining species to *Rubidgea*. Altogether, *Gebauer (2007)* recognized four genera and 16 species of rubidgeines: *Aelurognathus* (containing six species: *A. tigriceps*, *A. alticeps*, *A. brodiei* (misspelled '*broodiei*'), *A. kingwilli*, *A. ferox*, and *A. maccabei*), *Clelandina* (containing

three species: *C. rubidgei*, *C. laticeps*, and *C. scheepersi*), *Rubidgea* (containing three species: *R. atrox*, *R. quinquemolaris*, and *R. pricei*), and *Sycosaurus* (containing four species: *S. laticeps*, *S. kingoriensis*, *S. terror*, and ?*S. intactus*). Finally, in an unpublished MSc thesis, *Norton (2012)* reviewed the species of *Aelurognathus*, synonymizing all species recognized by Gebauer with the type species, *A. tigriceps*.

Although these revisions have improved our understanding of gorgonopsian taxonomy from the days of Broom, it is clear that the group is still highly oversplit relative to more intensely-studied Permo-Triassic therapsid groups (see, e.g., *Hopson & Kitching, 1972*; *Keyser, 1975*; *King & Rubidge, 1993*; *Kammerer, 2011*; *Kammerer, Angielczyk & Fröbisch, 2011*). In particular, the existing taxonomic framework for gorgonopsians (*Sigogneau-Russell, 1989*; *Gebauer, 2007*) makes it very difficult to identify new specimens to species, as the majority of species distinctions within genera are still based on minor differences in proportions that frequently vary with size and taphonomic distortion. Here, I present a new, comprehensive revision of rubidgeine taxonomy. This paper is part of a series of contributions aiming to resolve the alpha taxonomy of Gorgonopsia, establish biologically meaningful and easily identifiable morphospecies, and place these taxa in a phylogenetic context.

## MATERIALS

Each specimen referenced in this paper, including every known rubidgeine type, was examined personally by the author. Additionally, specimens of the following non-rubidgeine gorgonopsians were examined for comparative purposes and to provide codings for the phylogenetic analysis: *Arctognathus curvimola* (B 452; BP/1/5668; CGS AF 126–83; CGS S 33; NHMUK 47339; NMQR 857; RC 110; RC 308; RC 454; RC 492; SAM-PK-3329; SAM-PK-9345), *Arctops willistoni* (BP/1/698; NHMUK R4099), *Eriphostoma microdon* (AM 3751; AMNH FARB 5524; BP/1/7275; NMQR 3006; SAM-PK-2754; SAM-PK-5598; SAM-PK-11846; SAM-PK-11849; SAM-PK-12220; SAM-PK-K208; SAM-PK-K230; SAM-PK-K11164), *Gorgonops torvus* (AMNH FARB 5515; BP/1/1992; BP/1/4089; NHMUK R1647; SAM-PK-K11143), *Inostrancevia alexandri* (PIN 2005/1587; PIN 2005/1774; PIN 2005/1856), and *Lycaenops ornatus* (AMNH FARB 2240; BP/1/2470; CGS FL 17; NMQR 3075).

### Institutional abbreviations

**AM**, Albany Museum, Grahamstown, South Africa; **AMNH FARB**, American Museum of Natural History, Fossil Amphibian, Reptile, and Bird Collection, New York, USA; **B**, Bremner Collection, Graaff-Reinet Museum, Graaff-Reinet, South Africa; **BP**, Evolutionary Studies Institute (formerly the Bernard Price Institute for Palaeontological Research), University of the Witwatersrand, Johannesburg, South Africa; **CGS** (also **CGP**), Council for Geoscience, Pretoria, South Africa; **GPIT**, Paläontologische Sammlung, Eberhard Karls Universität Tübingen, Tübingen, Germany; **NHMUK**, the Natural History Museum, London, UK; **NMQR**, National Museum, Bloemfontein, South Africa; **PIN**, Paleontological Institute of the Russian Academy of Sciences, Moscow, Russia; **RC**, Rubidge Collection, Wellwood, Graaff-Reinet, South Africa; **SAM**, Iziko: South African Museum, Cape Town, South Africa; **TM**, Ditsong, the National Museum of Natural

History (formerly the Transvaal Museum), Pretoria, South Africa; **UCMP**, University of California Museum of Paleontology, Berkeley, USA; **UMZC**, University Museum of Zoology, Cambridge, UK.

## SYSTEMATIC PALEONTOLOGY

**Therapsida *Broom, 1905***

**Gorgonopsia *Seeley, 1894***

**Gorgonopidae *Lydekker, 1890***

**Rubidgeinae *Broom, 1938***

Rubidgeidae *Broom, 1938*:529

Sycosauridae *Watson & Romer, 1956*:60

Rubidgeinae *Sigogneau, 1970*:255

Broomicephalinae *Tatarinov, 1974*:100

Sycosaurinae *Tatarinov, 1974*:60

*Type genus*: Rubidgea *Broom, 1938*.

    *Included genera*: Aelurognathus *Haughton, 1924*; Clelandina *Broom, 1948*; Dinogorgon *Broom, 1936*; Leontosaurus *Broom & George, 1950*; Rubidgea *Broom, 1938*; Ruhuhucerberus *Maisch, 2002*; Smilesaurus *Broom, 1948*; Sycosaurus *Haughton, 1924*.

    *Diagnosis*: Large gorgonopsians characterized by the following unique autapomorphies: absence of blade-like parasphenoid rostrum and relatively tall suborbital portion of the zygomatic arch. Also characterized by the following features shared with *Arctognathus curvimola*, which are here reconstructed as homoplasies: preparietal reduced or absent, reduction of the palatal boss of the pterygoid, absence of teeth on the transverse process of the pterygoid, and massive dentary symphysis. Rubidgeines other than *Smilesaurus ferox* are further characterized by the following unique autapomorphies: frontals excluded from orbital margin, postorbital bar anteroposteriorly expanded, and circumorbital and supratemporal margins rugose.

## DESCRIPTION

*Kemp (1969)* provided a thorough description of the rubidgeine skull, based on acid-prepared specimens of *Sycosaurus nowaki* from Tanzania (Kemp described this material as *Leontocephalus intactus* and *Arctognathus* sp.; for referral to *S. nowaki*, see species account below). Nevertheless, an overview of rubidgeine cranial anatomy is warranted here, to enumerate typical features of the group as a whole and provide frame of reference for the morphologies of individual taxa. Some autapomorphic features of individual rubidgeines are mentioned in this overview where appropriate, but the majority are detailed in their respective species accounts. This section is intended to be applicable to all taxa, but for ease of reference, figure callouts refer to the skull reconstruction of the first taxon detailed below, *Aelurognathus tigriceps* (Figs. 1 and 2).

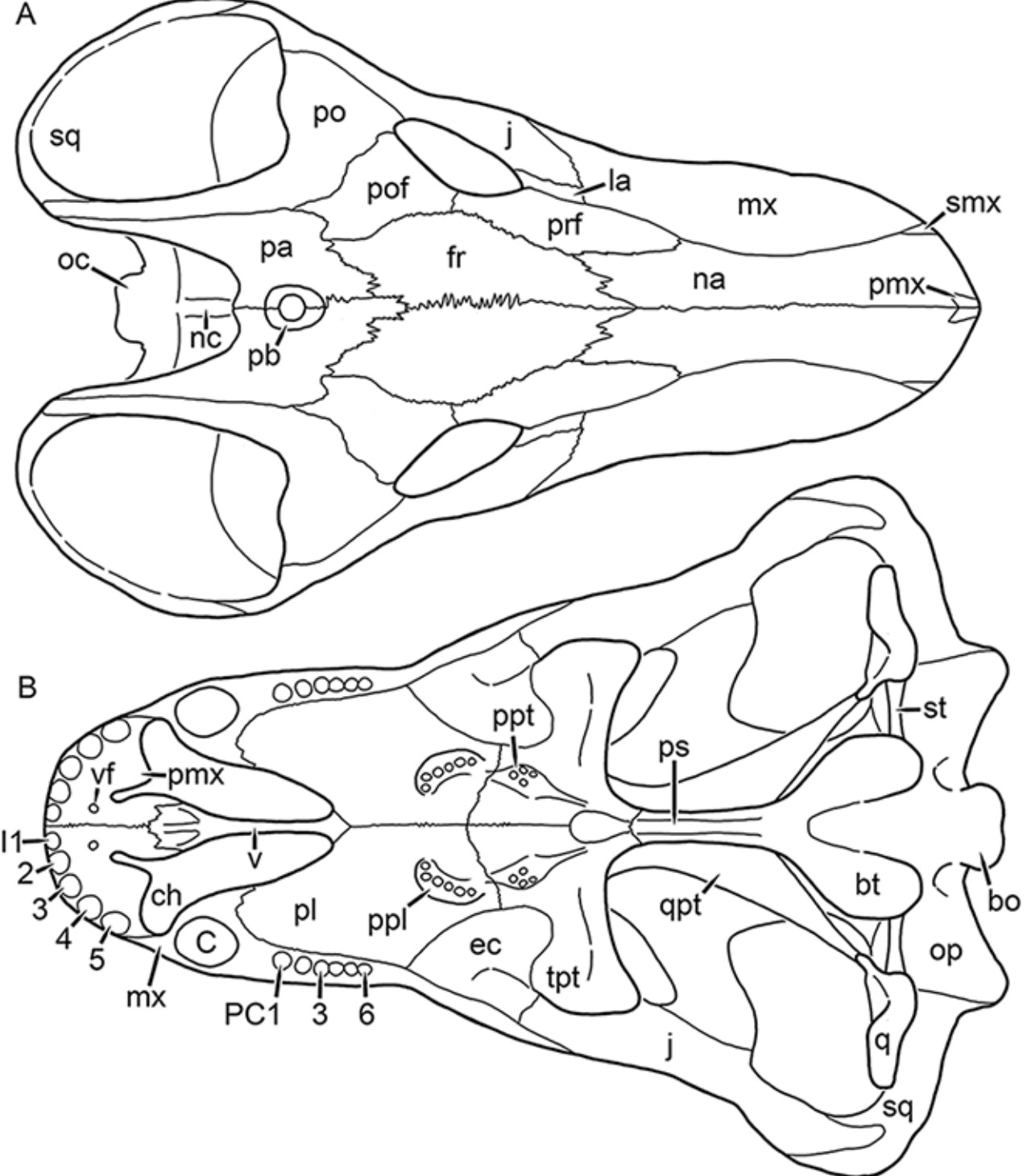

**Figure 1 Reconstruction of the skull of *Aelurognathus tigriceps* (*Broom & Haughton, 1913*) in (A) dorsal and (B) ventral views.** Reconstructions based primarily on BP/1/1566, BP/1/3464, RC 35, and SAM-PK-2342. Abbreviations: bo, basioccipital; bt, basal tuber; C, upper canine; ch, choana; ec, ectopterygoid; fr, frontal; I, upper incisor; j, jugal; la, lacrimal; mx, maxilla; na, nasal; nc, nuchal crest; oc, occipital condyle; op, opisthotic; pa, parietal; pb, pineal boss; PC, upper postcanine; pl, palatine; pmx, premaxilla; po, postorbital; pof, postfrontal; ppl, palatal boss of palatine; ppt, palatal boss of pterygoid; prf, prefrontal; ps, parasphenoid; q, quadrate; qpt, quadrate ramus of pterygoid; smx, septomaxilla; sq, squamosal; st, stapes; tpt, transverse process of pterygoid; v, vomer; vf, ventral premaxillary foramen.

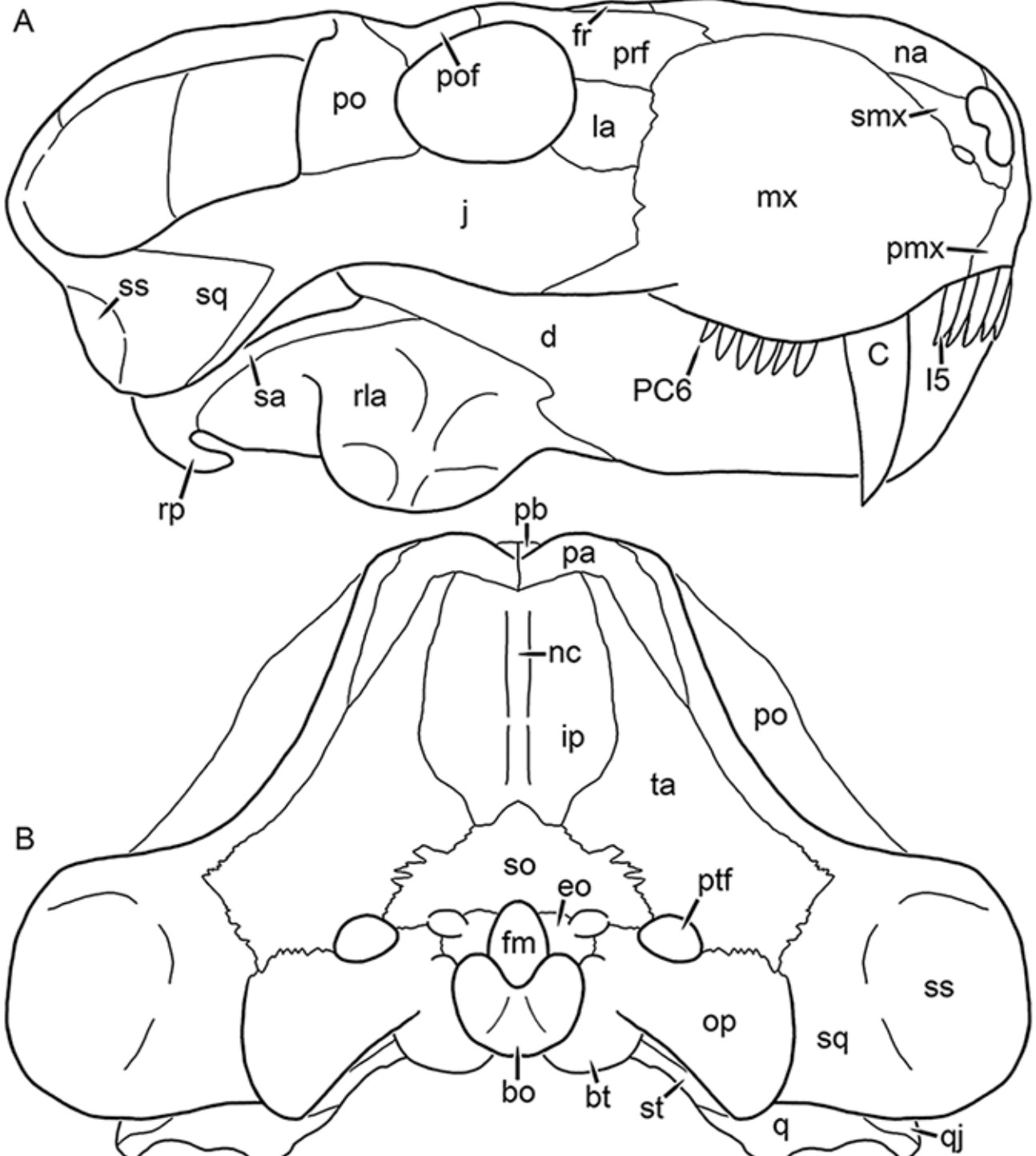

**Figure 2 Reconstruction of the skull of *Aelurognathus tigriceps* (*Broom & Haughton, 1913*) in (A) lateral and (B) occipital views.** Reconstructions based primarily on BP/1/813, BP/1/1566, BP/1/3464, and SAM-PK-2342. Abbreviations: bo, basioccipital; bt, basal tuber; C, upper canine; d, dentary; eo, exoccipital; fm, foramen magnum; fr, frontal; I, upper incisor; ip, interparietal; j, jugal; la, lacrimal; mx, maxilla; na, nasal; nc, nuchal crest; op, opisthotic; pa, parietal; pb, pineal boss; PC, upper postcanine; pmx, premaxilla; po, postorbital; pof, postfrontal; prf, prefrontal; ptf, post-temporal fenestra; q, quadrate; qj, quadratojugal; rla, reflected lamina of angular; rp, retroarticular process; sa, surangular; smx, septomaxilla; so, supraoccipital; sq, squamosal; ss, squamosal sulcus; st, stapes; ta, tabular.

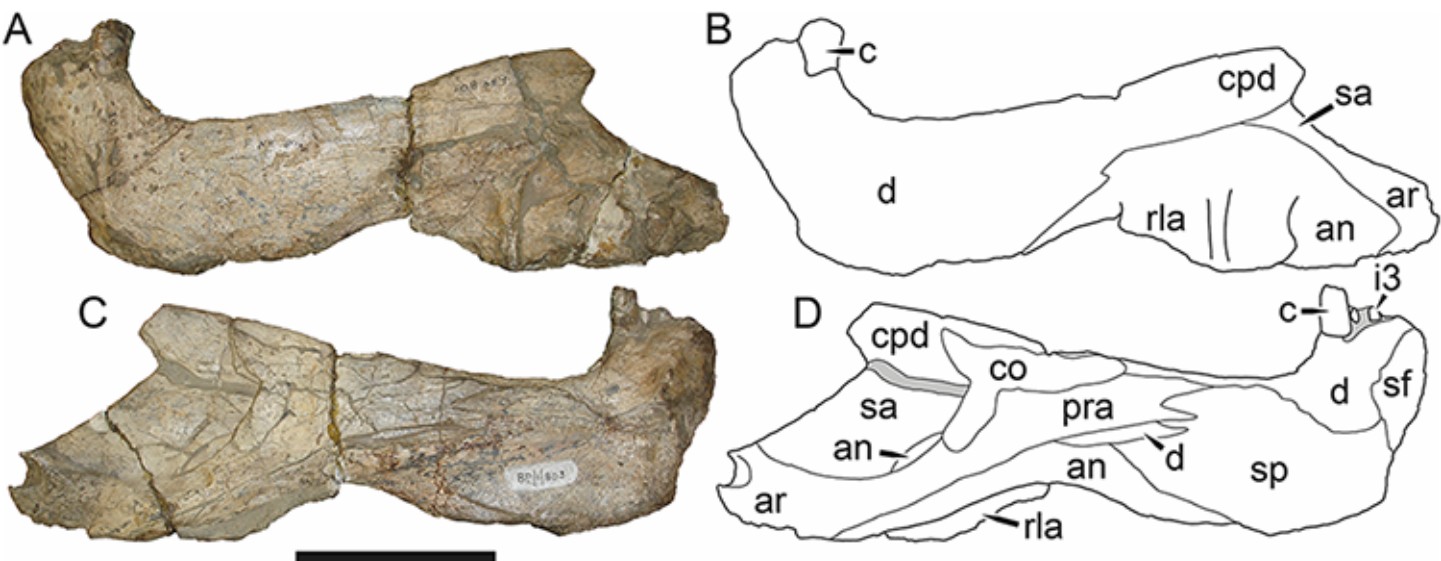

**Figure 3** Left mandibular ramus of a referred specimen (BP/1/803) of *Leontosaurus vanderhorsti Broom & George, 1950* in (A) lateral and (C) medial view (with (B) and (D) interpretive drawings). Holotype of *Rubidgea platyrhina Brink & Kitching, 1953*. Abbreviations: an, angular; ar, articular; c, lower canine; co, coronoid; cpd, coronoid process of dentary; d, dentary; i, lower incisor; pra, prearticular; rla, reflected lamina of angular; sa, surangular; sf, symphysial facet (mid-dentary suture); sp, splenial. Gray indicates matrix. Scale bar equals 10 cm.

*Aelurognathus* is the most abundant and morphologically thoroughly-known rubidgeine, and also probably represents a good approximation of what the ancestral rubidgeine would have looked like. For the lower jaw, the lateral reconstruction of *Aelurognathus* is supplemented by figures of the two best-prepared rubidgeine mandibles, BP/1/803 (Fig. 3, referred specimen of *Leontosaurus vanderhorsti*) and UMZC T877 (Fig. 4, referred specimen of *Sycosaurus nowaki*).

The cranial reconstructions presented herein represent idealized adult skulls based on information from multiple specimens. Because these reconstructions are based on specimen composites instead of individual exemplars, no scale bars are provided for them—refer to figures illustrating actual specimens for sizes. Different views of the reconstructions (dorsal, ventral, lateral, and occipital) are not to scale; each view is presented at maximum size for ease of observation. In figures illustrating actual specimens, however, all views of a specimen are to the same scale (unless explicitly shown otherwise by the presence of multiple scale bars). For dorsal reconstructions, anterior is right, whereas for ventral, anterior is left (so as to optimize figure space). All lateral reconstructions are presented in right lateral view; occipital reconstructions represent the posterior view of a skull in standard horizontal orientation.

The premaxilla of rubidgeines has only limited exposure on the dorsolateral surface of the skull (Figs. 1A and 2A). Laterally, is is covered by an anterior lamina of the maxilla, such that the premaxillary-maxillary suture is always anterior to the fifth upper incisor (Fig. 2A). The internarial bar is a tall, narrow structure that is often broken off in rubidgeine specimens. Paired anterior premaxillary foramina are present at the base of the internarial bar. The ascending process of the premaxilla is relatively short (compared to the primitive condition in therapsids), terminating above the nares (Fig. 1A).

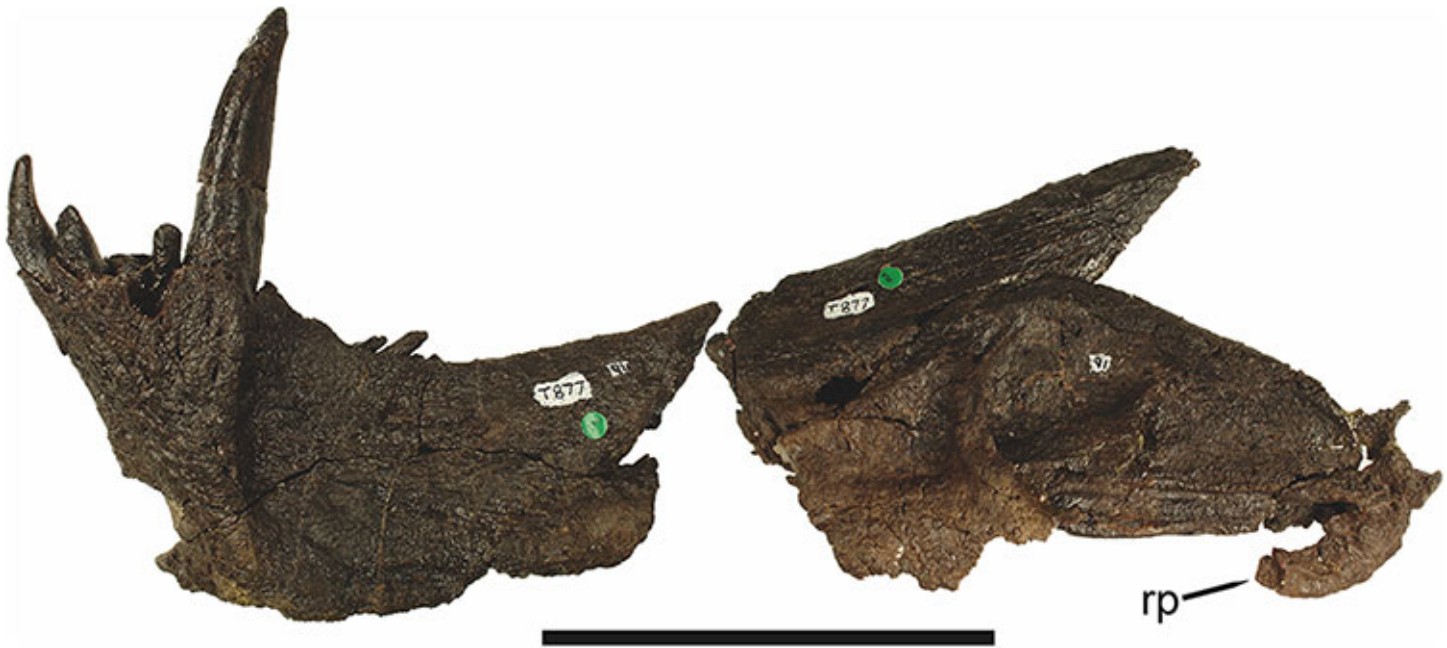

**Figure 4 Left mandibular ramus of a referred specimen (UMZC T877) of *Sycosaurus nowaki* (*Broili & Schröder, 1936*) in lateral view.** Abbreviation: rp, retroarticular process. Scale bar equals 10 cm.

Palatally, the premaxilla forms a broad plate behind the incisor alveoli (Fig. 1B). All known rubidgeines have five upper incisors, the typical number for gorgonopsians (but reduced to four in *Inostrancevia*) (*Sigogneau-Russell, 1989*). Incisor dimensions vary among rubidgeine taxa, but these teeth are always weakly spatulate with mesiodistal serrations. The ventral mid-premaxillary suture is weakly interdigitated. Paired ventral premaxillary foramina are present near the mid-premaxillary suture, as is typical of gorgonopsians (although absent in *Arctognathus* (*Kammerer, 2015*)). These foramina are frequently present in biarmosuchians, therocephalians, and cynodonts (*van den Heever, 1994*; *Sidor, 2003*; *Sidor & Smith, 2007*), but are absent in anomodonts and dinocephalians (C Kammerer, personal observations). The ventral premaxillary foramen in theriodonts communicates with a dorsal premaxillary foramen (exiting within the nasal capsule) through a thin canal (*Brink, 1960*; *Kemp, 1969*; *Fourie, 1974*; *van den Heever, 1994*), which probably housed the terminal branch of the maxillary artery (as in extant squamates (*Oelrich, 1956*)). In the cynodont *Thrinaxodon*, this canal branches so that it exits through both the dorsal and anterior premaxillary foramina (*Fourie, 1974*), and it is likely the same is true of gorgonopsians.

The posterior border of the ventral premaxillary plate is deeply invaginated by an anterior extension of the choana, which separates the vomerine process of the premaxilla from the premaxillary plate (Fig. 1B). The vomerine process of the premaxilla is a broad structure overlapping the vomer ventrally and making a major contribution to the expanded interchoanal body. The vomerine process of the premaxilla is relatively short medially, but extends further as an elongate process at its lateral margin, sheathing the lateral surface of the vomer.
The maxilla is a large bone with broad lateral exposure on the snout (nearly excluding the nasals from lateral view in some rubidgeines, e.g., *Leontosaurus*) (Fig. 2A). In well-preserved rubidgeine skulls the lateral surface of the maxilla is weakly rugose, with numerous small foramina associated with the canine root. The dorsal margin of the maxilla is always gently rounded (Fig. 1A). The posterior margin of the maxilla varies in shape between rubidgeines: it may be nearly straight, gently rounded, or strongly invaginated by anterior processes of the prefrontal and lacrimal. Posteroventrally, the maxilla forms a lengthy posterior process which extends beneath the jugal before terminating near the midpoint of the orbit (Figs. 1 and 2A). The base of this process is offset laterally from the underlying portion of the maxilla. In most gorgonopsians, the degree of this offset is relatively weak, but in some rubidgeines the underlying portion of the maxilla is strongly depressed, forming a distinct maxillary emargination above the postcanine region. This is developed to the greatest extent in *Clelandina* and *Rubidgea*, in which the postcanine teeth are reduced or absent.

Palatally, the maxilla originates immediately behind the fifth upper incisor, in the form of a thin process extending lateral to the expanded portion of the choana (where it accommodates the lower canine) (Fig. 1B). Posteriorly, the maxilla expands around the canine alveolus, then is constricted by the palatine and tapers off before terminating lateral to the transverse process of the pterygoid. The upper canine is the largest tooth in the skull, a massive, blade-like tooth with well-developed mesiodistal serrations. Anterior and posterior upper canine alveoli are present in most rubidgeine skulls, with tooth replacement alternating between them. A single erupted canine is usually present on each side of the skull. The rarity of rubidgeine skulls preserved with the replacement canine partially erupted and pushing out the old canine suggests relatively rapid tooth replacement in this group, unlike in basal therocephalians where the old and replacement canines are frequently present simultaneously (*van den Heever, 1980*). Rubidgeine postcanines are conical and often mesiodistally serrated. Tooth count varies extensively in the group as a whole, with every postcanine number from zero to seven being represented. However, within rubidgeine species postcanine counts are fairly conservative, typically only varying by one tooth position.

The septomaxilla is a narrow, irregular bone largely confined to the naris (Fig. 2A). It forms a ventral footplate within the external naris, between the maxilla and premaxilla. It then narrows into a thin bar posterodorsally before expanding into a broad transverse lamina that separates the naris into dorsal and ventral compartments. Posteriorly, a narrow septomaxillary process leaves the naris and extends between the maxilla and nasal, tapering off and terminating above the level of the canine.

The nasal is a plate-like bone making up the dorsal surface of the snout (Fig. 1A). It is broadest anteriorly, where it makes up part of the dorsal margin of the naris (Figs. 1A and 2A). It is constricted at mid-length by the dorsal margin of the maxilla, and slightly expands posteriorly before being wedged between the prefrontals (Fig. 1A). The fronto-nasal suture is highly irregular and strongly interdigitated, and is located anterior to the orbits. The dorsal surface of the nasals is covered with fine longitudinal sculpturing in most well-preserved rubidgeine skulls (e.g., UMZC T891).

The prefrontal of rubidgeines is an elongate, trapezoidal bone, always extending anterior to the lacrimal and jugal (unlike in *Arctognathus*, in which the anterior margins of the prefrontal and jugal are at the same point on the skull (*Kammerer, 2015*)) (Fig. 2A). It forms the anterodorsal margin of the orbit. The anterior tip of the prefrontal is usually tapered into a point in rubidgeines, with the exception of *Ruhuhucerberus*. In all rubidgeines other than *Smilesaurus*, the prefrontal contacts the postfrontal posteriorly, excluding the frontal from the orbital margin (Fig. 1A). The dorsal surface of the prefrontal is fairly rugose in most rubidgeines (with again, the exception being *Smilesaurus*), and this is developed to an extreme degree in *Clelandina*, *Dinogorgon*, and *Rubidgea*, where a massive, pachyostosed supraorbital boss extends across the prefrontal and postfrontal.

The lacrimal is a small, usually rectangular bone at the anterior edge of the orbit (Fig. 2A). A lacrimal foramen is present on its posterior surface, where it makes up part of the orbital wall. A second lacrimal foramen (probably connected to the former through an internal channel) exits onto the facial surface of the lacrimal in *Clelandina*, *Dinogorgon*, and *Rubidgea*.

The jugal is an elongate bone forming part of the lateral surface of the snout and most of the zygomatic arch (Figs. 1 and 2A). The facial portion is defined as the section of the jugal anterior to the orbits, where it underlies the lacrimal. This portion is usually roughly quadrangular and has a flat-to-concave surface (Fig. 2A). The zygomatic portion of the jugal underlies the orbit, postorbital bar, and most of the temporal fenestra. Unlike in therocephalians (*van den Heever, 1994*), it does not contribute to the postorbital bar. The proportions of the zygomatic portion of the jugal vary extensively among rubidgeines. In all rubidgeines the dorsoventral height of the jugal is lowest beneath the postorbital bar, but extreme narrowing of the jugal at this point is characteristic of *Sycosaurus*. Transversely, the jugal is typically narrow in gorgonopsians (Fig. 1A), but is broadly expanded in a variety of rubidgeines (*Clelandina*, *Dinogorgon*, *Leontosaurus*, *Rubidgea*, and to a lesser extent *Sycosaurus*). The jugal is deflected beneath the temporal fenestra in all rubidgeines (as is also the case in some non-rubidgeine gorgonopsians, e.g., *Lycaenops* (*Sigogneau, 1970*)) (Fig. 2A). This occurs to an extreme degree in *Clelandina*, *Dinogorgon*, *Leontosaurus*, and *Rubidgea*. Laterally, the jugal bears a facet for the anterior process of the zygomatic ramus of the squamosal. Ventrally, the jugal divides the lateral and medial zygomatic portions of the squamosal (Fig. 1B), and extends far enough posteriorly to be visible in occipital view in some taxa (e.g., *Dinogorgon*, *Rubidgea*).

The frontals of rubidgeines are relatively narrow compared to other gorgonopsians, because of their exclusion from the orbital margin (Fig. 1A). The mid-frontal suture is strongly interdigitated. Near the mid-length of the mid-frontal suture, this interdigitation is exceptionally intense, and often associated with short interorbital ridge. Posteriorly, the frontal forms a pointed process extending between the postfrontal and the median process of the parietal.

The postfrontal is a large, triangular-to-quadrangular element in rubidgeines (Fig. 1A). It usually broadly contacts the parietal posteriorly (although this contact is minimal in *Ruhuhucerberus*). The shape of the posteromedial portion of the postfrontal is variable
within the group: in *Leontosaurus* and *Sycosaurus* it forms a distinct, 'tab'-like process. In *Clelandina*, *Dinogorgon*, and *Rubidgea*, the circumorbital portion of the postfrontal is extremely pachyostosed and bears a rugose supraorbital boss. Posteromedially, however, the postfrontal is as flat and unornamented in these taxa as in other rubidgeines.

The postorbital bone consists of two parts: a ventral ramus making up the postorbital bar and a dorsal ramus making up the medial margin of the temporal fenestra (Figs. 1A and 2A). The postorbital bar is anteroposteriorly expanded in all rubidgeines other than *Smilesaurus* and small, probably juvenile individuals of *Aelurognathus*. In *Clelandina*, *Dinogorgon*, *Leontosaurus*, and *Rubidgea* the postorbital bar is massively expanded (equal or greater in width to the orbit) and pachyostosed.

The preparietal is completely absent in most rubidgeines. A small preparietal is definitely present in smaller skulls of *Aelurognathus* and *Smilesaurus*, but absent in larger specimens. However, in those larger specimens an anterior process of the parietal, in the same position as the preparietal, extends between the frontals. It is likely that this bone was present at birth and fused with the parietal during development (as is also probably the case in *Arctognathus* (*Kammerer, 2015*)).

The parietal is a relatively short component of the skull roof, but bears an elongate posterior process that typically mirrors the dorsal ramus of the postorbital (Fig. 1A). This process extends onto the occiput, between the tabular and squamosal, in *Aelurognathus* (Fig. 2B), *Smilesaurus*, and *Sycosaurus*. A well-developed pineal boss is usually present at the mid-parietal suture, near the end of the skull roof (Fig. 1A). At their posterior midpoint, the parietals weakly bulge out above the occipital plate, forming the dorsal tip of the nuchal ridge.

The squamosal forms the posterior margin of the temporal fenestra and the lateral margin of the occiput (Figs. 1A and 2B). The zygomatic ramus of the squamosal bears a tapering anterior process that overlaps the jugal laterally (Fig. 2A). Ventrally, the squamosal forms a thickened, curved bar extending between the jugal and the opisthotic (Fig. 1B). The posterior face of the squamosal is typically the largest element of the occiput (Fig. 2B). The occipital dimensions of this bone are extremely variable. A squamosal sulcus (homologous to the external auditory meatus of mammals (*Sidor & Hopson, 1998*)) is present on the lower lateral edge of the occipital portion of the squamosal, and extends forward onto the zygoma in most species (albeit not *Clelandina*, *Dinogorgon*, *Leontosaurus*, or *Rubidgea*). In most gorgonopsians, the occipital portion of the squamosal is very narrow dorsal to the squamosal sulcus (*Sigogneau-Russell, 1989*; *Kammerer, 2015*), and this condition is retained in *Aelurognathus* (Fig. 2B). In all other rubidgeines, however, the squamosal remains broadly expanded dorsal to the sulcus.

The tabular is a tall, broad paired element situated between the interparietal and supraoccipital medially and parietal laterally (Fig. 2B). It forms the dorsolateral margin of the post-temporal fenestra and partially overlaps the opisthotic dorsally (at the lateral edge of the paroccipital process). The tabular sutures are typically densely interdigitated, especially with the supraoccipital and lower part of the squamosal.

The interparietal (also known as the postparietal) is a median element near the top of the occiput (Fig. 2B). It is typically roughly quadrangular in rubidgeines. Its midline bears

a well-developed nuchal crest, which extends downwards from the parietals. It lies above another median element, the supraoccipital, which forms the dorsal margin of the foramen magnum and also contributes to the dorsolateral margin of the post-temporal fenestra. The supraoccipital is a broad bone, always wider than tall.

The vomer is almost entirely confined to the internal choana in gorgonopsians and is always unpaired (Fig. 1B). The post-choanal portion forms an extremely short, triangular plate anterior to the mid-palatine suture. Anterior to this, the vomer forms a narrow rod, which eventually expands into a broad interchoanal body. In most rubidgeines, this expansion occurs in the anterior half of the choana, near the point where the vomer contacts the vomerine process of the premaxilla. In *Sycosaurus* (and to a lesser degree *Smilesaurus*), however, the vomer begins expanding in a relatively posterior position. The anterior margin of the vomer has a trident-like morphology (three tips, with one long central process and a pair of shorter lateral process) where it contacts the premaxilla. All rubidgeines also have the typical gorgonopsian set of three vomerine ridges (one central, two lateral), although the relative positions and robusticity of these ridges vary between species.

The palatine is the largest bone in the rubidgeine palate (Fig. 1B). Anteriorly, it is broad but tapering, terminating in a rounded edge abutting the maxilla immediately posterior to the upper canine. Laterally, it broadly overlaps the maxilla, nearly reaching the postcanine alveoli. Posteriorly, the palatine forms a broad plate bearing a discrete palatine boss. In rubidgeines, this boss is reniform (i.e., 'kidney' or 'bean'-shaped) and bears a variable number of teeth (1–7), typically in a single row.

The ectopterygoid is a semi-ovoid bone situated between the posterior process of the maxilla (laterally) and the palatine-pterygoid complex (medially). It is a simple, edentulous element making up the anterior base of the transverse process (Fig. 1B).

The pterygoid is a complex element composed of three distinct rami: palatal, transverse, and quadrate (Fig. 1B). The palatal portion of the pterygoid is broad and flattened, like the palatine that it borders anteriorly, and bears a palatal boss. In *Aelurognathus* and *Smilesaurus*, this structure is a discrete boss bearing a cluster of small teeth, as is the case in most gorgonopsians (*Sigogneau-Russell, 1989*). In all other rubidgeines, however, the boss is reduced to a thin ridge (toothless in all taxa except *Ruhuhucerberus*) extending posteromedially from the palatine boss. The transverse process is always edentulous in rubidgeines. The long axis of this process is usually transversely straight, but it is 'backswept' in *Leontosaurus* (as is also the case in some non-rubidgeine gorgonopsians, e.g., *Aelurosaurus* and *Gorgonops* (*Sigogneau, 1970*)). An interpterygoid vacuity is sometimes present between the transverse processes. Posteriorly, the pterygoid makes a small contribution to the anterior tip of the basicranial girder, at its contact with the parasphenoid. The quadrate ramus of the pterygoid extends posterolaterally from the edge of the basicranial girder. It forms a broad, thin sheet of bone hugging the edges of the parabasisphenoid, before detaching as an elongate process anterior to the basal tubera. This process extends posterolaterally (with the degree of lateral angulation differing among species) before contacting the quadrate at tip.

The parasphenoid and basisphenoid are fused into a single element, the parabasisphenoid. From comparisons with other therapsids, it is probable that the basicranial girder is composed primarily of parasphenoid (Fig. 1B), with the basisphenoid making up the anterior portion of the basal tuber. The basicranial girder of gorgonopsians is typically dominated by a tall, blade-like parasphenoid rostrum (*Sigogneau-Russell, 1989*; *Kammerer et al., 2015*). Uniquely among gorgonopsians, rubidgeines lack this structure, and instead have reverted to the primitive therapsid condition: a low basicranial girder with an elongate ventral depression between the edges of the parasphenoid.

The basal tubera are paired, typically ovoid structures at the base of the braincase (Fig. 1B), which accommodate the medial end of the stapes. The stapes is rarely preserved in rubidgeines; when present it accords in morphology with other gorgonopsians, being a robust rod with a distinct dorsal process and large stapedial foramen (*Kemp, 1969*; *Sigogneau-Russell, 1989*). In addition to forming the posterior half of the basal tuber, the basioccipital makes up the floor of the braincase and the ventral, median portion of the occipital condyle. The lateral portions of the occipital condyle are made up of the paired exoccipitals, which also form part of the occipital plate lateral to the foramen magnum. A bulbous exoccipital process is present on the edge of this plate in all rubidgeine specimens with a well-preserved occiput.

The opisthotic is a stout element extending laterally in the form of a paroccipital process (Figs. 1B and 2B). Unfortunately the anterodorsal portion of the opisthotic is very rarely exposed in rubidgeines, and comparative data on their inner ear is lacking. The epipterygoid, prootic, and orbitosphenoid bones are also rarely exposed in rubidgeine skulls, and it was not possible to compare their morphologies between the taxa under consideration here. They are fully-prepared and suturally distinct only in the acid-prepared specimens of *Sycosaurus nowaki* described by *Kemp (1969)*.

The quadrate-quadratojugal complex of rubidgeines (Figs. 1B and 2B) is typical for gorgonopsians: they are not sutured to the squamosal, but rather lodged in an anteroventral squamosal depression (*Kemp, 1969*; *Kammerer, 2015*). The quadrate is the larger of the two elements, and a large quadrate foramen is situated between them. This complex is difficult to study in rubidgeines, as it is usually either absent (if only the cranium is preserved) or obscured by the lower jaw (if it is preserved in articulation).

The dentary of rubidgeines is massive, with a very robust symphysis accommodating the enlarged lower canine (Figs. 2A, 3 and 4). The dentaries are tightly sutured at the symphysis, producing a mandible more similar to that of eucynodonts (in which the dentaries fuse) than therocephalians (*van den Heever, 1994*). The anterior face of the symphysis is steeply sloping and very tall: the incisor and canine bases are elevated well above the postcanine tooth row. A distinct longitudinal ridge is present on the lateral edge of the symphysis, immediately followed by a depression accommodating the upper canine (Fig. 3A). Four lower incisors are present, identical in morphology to the uppers. The number of lower postcanines is variable, but always fewer than the uppers. No lower postcanines are present in *Clelandina*, *Leontosaurus*, or *Rubidgea*. Although lower posterior to the symphysis, the dentary overall remains proportionally taller in

rubidgeines than in most other gorgonopsians (with *Arctognathus* being an exception (*Sigogneau-Russell, 1989*; *Kammerer, 2015*)). Medially, the dentary is mostly obscured by the other mandibular bones, but has a narrow exposure between the prearticular and splenial (Fig. 3D). Posteriorly, the dentary detaches from the rest of the mandibular ramus to form a free-standing coronoid process.

The splenial is a tall, laminar bone restricted to the base of the mandibular symphysis and the medial face of the anterior mandibular ramus (Fig. 3D). At the base of the symphysis, it forms a distinct posteriorly-directed process. At its posterodorsal edge, the splenial has a zig-zag suture with the prearticular, a thin, ribbon-like bone angled posteroventrally that eventually fuses with the articular. Dorsal to the prearticular is a single coronoid. The coronoid is typically triangular in gorgonopsians (*Sigogneau-Russell, 1989*), but in rubidgeines where this region is exposed, it is a triradiate structure, with an elongate longitudinal portion and a descending ventral process (Fig. 3D).

Laterally, the postdentary region is composed primarily of the angular (Figs. 2A and 3B). Medially, the angular has a narrow anterior process extending far anteriorly, nearly reaching the symphysis (Fig. 3D). Laterally, it is dominated by the reflected lamina (Fig. 2A). Like other gorgonopsians, the reflected lamina of rubidgeines is not free dorsally and bears a robust dorsoventral ridge (*Sigogneau-Russell, 1989*). Posterior to the reflected lamina, the main body of the angular is exposed, separating the lamina from the articular. A thin, curved portion of the surangular overlies the angular in lateral view (Fig. 3B). Medially, this element is exposed more broadly, forming a rhomboidal plate between the dentary, coronoid, angular, prearticular, and articular (Fig. 3D). The articular is restricted to the posterior tip of the jaw, and bears a deep glenoid fossa for articulation with the upper jaw (Fig. 3D). The glenoid fossa is topped with a dorsal process. The ventral edge of the articular bears a large, hook-like retroarticular process (Fig. 4).

## SPECIES ACCOUNTS

### Aelurognathus *Haughton, 1924*

*Gorgonorhinus Broom, 1937*:141
*Leontocephalus Broom, 1940b*:174
*Prorubidgea Broom, 1940b*:169
*Tigricephalus Broom, 1948*:599

*Type species*: *Scymnognathus tigriceps Broom & Haughton, 1913*.
   *Diagnosis*: As for the type and only recognized species.

### Aelurognathus tigriceps (*Broom & Haughton, 1913*) (Reconstruction Figs. 1–2, Specimen Figs. 5–15)

*Scymnognathus tigriceps Broom & Haughton, 1913*:26
*Scymnognathus serratidens Haughton, 1915*:88
*Aelurognathus serratidens Haughton, 1924*:505

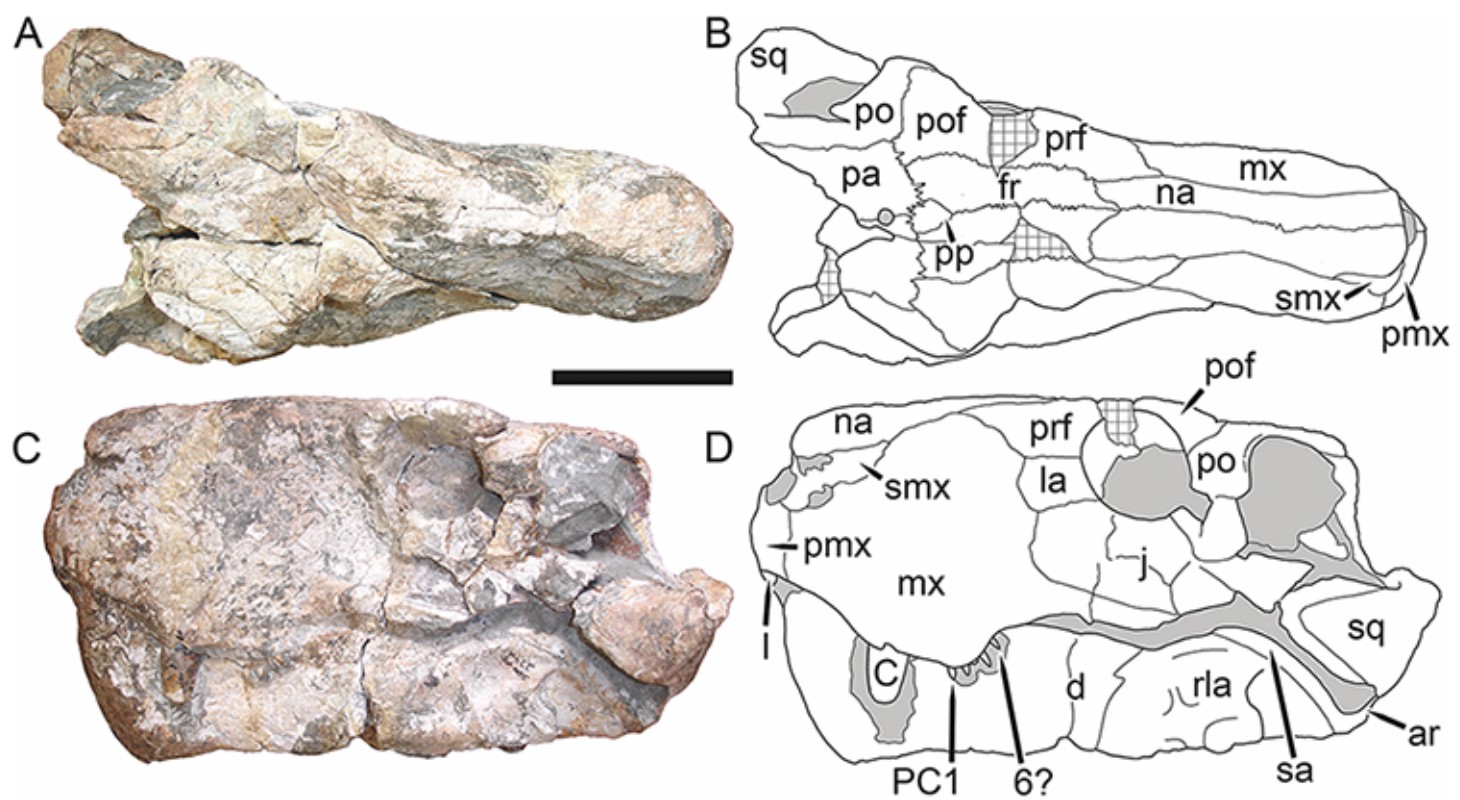

**Figure 5 Holotype (SAM-PK-2342) of *Aelurognathus tigriceps* (*Broom & Haughton, 1913*) in (A) dorsal and (C) left lateral view (with (B) and (D) interpretive drawings).** Abbreviations: ar, articular; C, upper canine; d, dentary; fr, frontal; I, upper incisor; j, jugal; la, lacrimal; mx, maxilla; na, nasal; pa, parietal; PC, upper postcanine; pmx, premaxilla; po, postorbital; pof, postfrontal; pp, preparietal; prf, prefrontal; rla, reflected lamina of angular; sa, surangular; smx, septomaxilla; sq, squamosal. Gray indicates matrix, hatching indicates plaster. Scale bar equals 10 cm.

*Aelurognathus tigriceps* Haughton, 1924:505

*Gorgonorhinus luckhoffi* Broom, 1937:141

*Leontocephalus cadlei* Broom, 1940b:174

*Prorubidgea maccabei* Broom, 1940b:169

*Sycosaurus brodiei* Broom, 1941:198

*Clelandina major* Broom, 1948:591

*Gorgonorhinus minor* Broom, 1948:597

*Tigricephalus kingwilli* Broom, 1948:599

*Lycaenops alticeps* Brink & Kitching, 1953:22

*Prorubidgea brinki* Manten, 1959:67

*Arctops? minor* Sigogneau, 1970:146

*Lycaenops kingwilli* Sigogneau, 1970:198

*Prorubidgea alticeps* Sigogneau, 1970:269

*Prorubidgea brodiei* Sigogneau, 1970:278

*Aelurognathus alticeps* Gebauer, 2007:187

*Aelurognathus broodiei* (sic) Gebauer, 2007:187

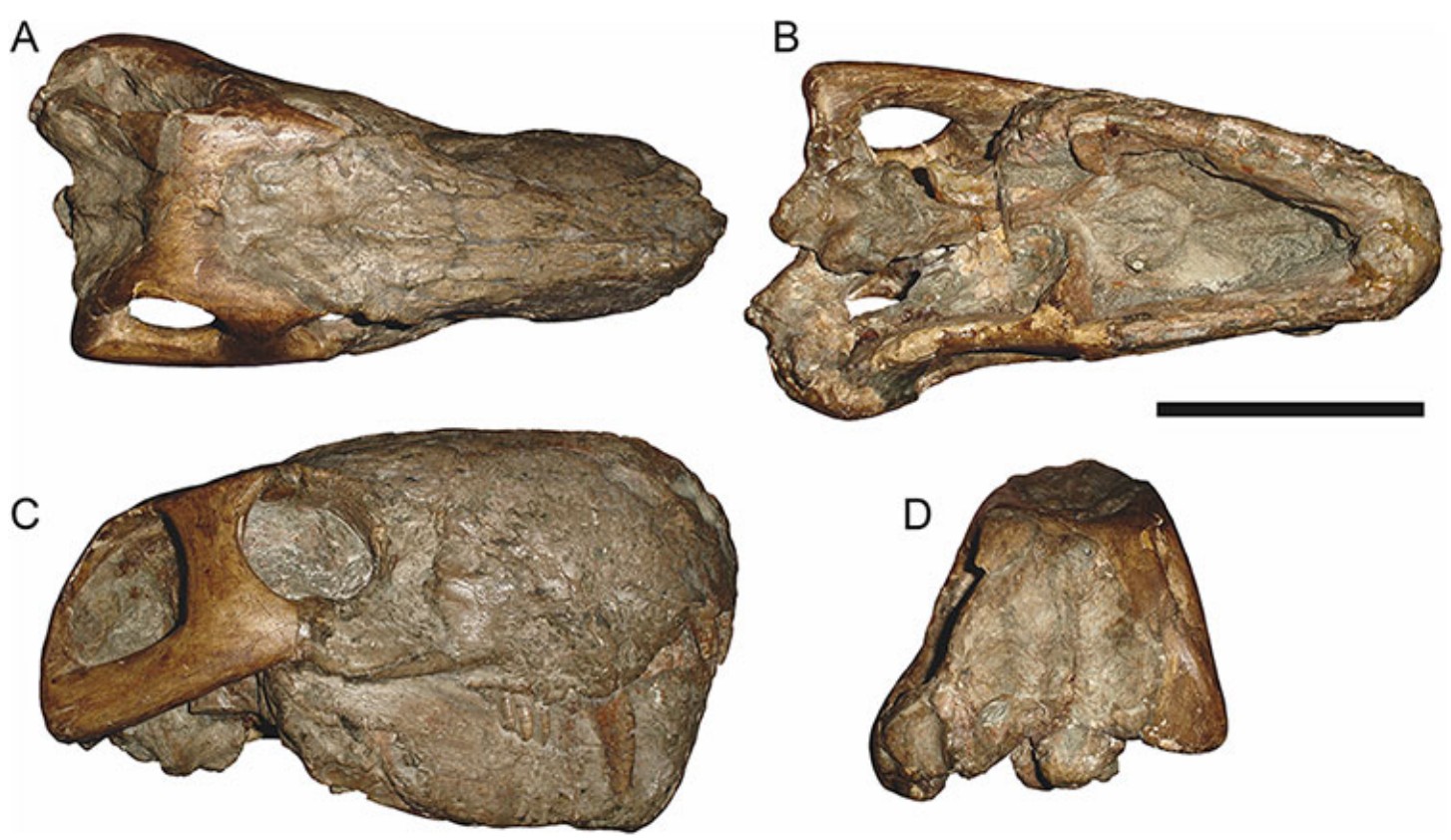

**Figure 6 Referred specimen (BP/1/813) of *Aelurognathus tigriceps* (*Broom & Haughton, 1913*) in (A) dorsal, (B) ventral, (C) right lateral, and (D) occipital view.** Holotype of *Lycaenops alticeps Brink & Kitching, 1953*. Scale bar equals 10 cm.

*Aelurognathus kingwilli Gebauer, 2007*:186

*Aelurognathus maccabei Gebauer, 2007*:186

*Holotype*: SAM-PK-2342 (Fig. 5), a complete but poorly-prepared skull and lower jaws from Dunedin, Beaufort West, South Africa.

   *Referred specimens*: BP/1/813 (Fig. 6; a partial skull, missing the temporal arches, and lower jaws from Hoeksplaas, Murraysburg, South Africa; holotype of *Lycaenops alticeps*); BP/1/1566 (Fig. 7; a complete skull and lower jaws from Ringsfontein, Murraysburg, South Africa; holotype of *Prorubidgea brinki*); BP/1/3464 (Fig. 8; a complete skull and lower jaws from *Drysdall & Kitching's (1963)* Locality 5 of the Luangwa Valley, Zambia); CGS R 163 (a crushed skull and lower jaws from Hoedemaker, Beaufort West, South Africa); CGS RMS 562 (a fragmentary skull and lower jaws from Groot Tafelbergsfontein, Beaufort West, South Africa); CGS WB 281 (a skull, missing the snout tip, and lower jaws from Weltevreden, Pearston, South Africa); RC 34 (Fig. 9; a complete skull and lower jaws and anterior three cervical vertebrae from St. Olives, Graaff-Reinet, South Africa; holotype of *Prorubidgea maccabei*); RC 35 (Fig. 10; a weathered snout from Weltevreden, Nieu Bethesda, South Africa; holotype of *Leontocephalus cadlei*); RC 60 (Fig. 11; a complete skull and lower jaws from Middlevlei, Murraysburg, South Africa; holotype of *Tigricephalus kingwilli*); RC 94 (Fig. 12; a poorly-preserved skull from Spandau Kop,

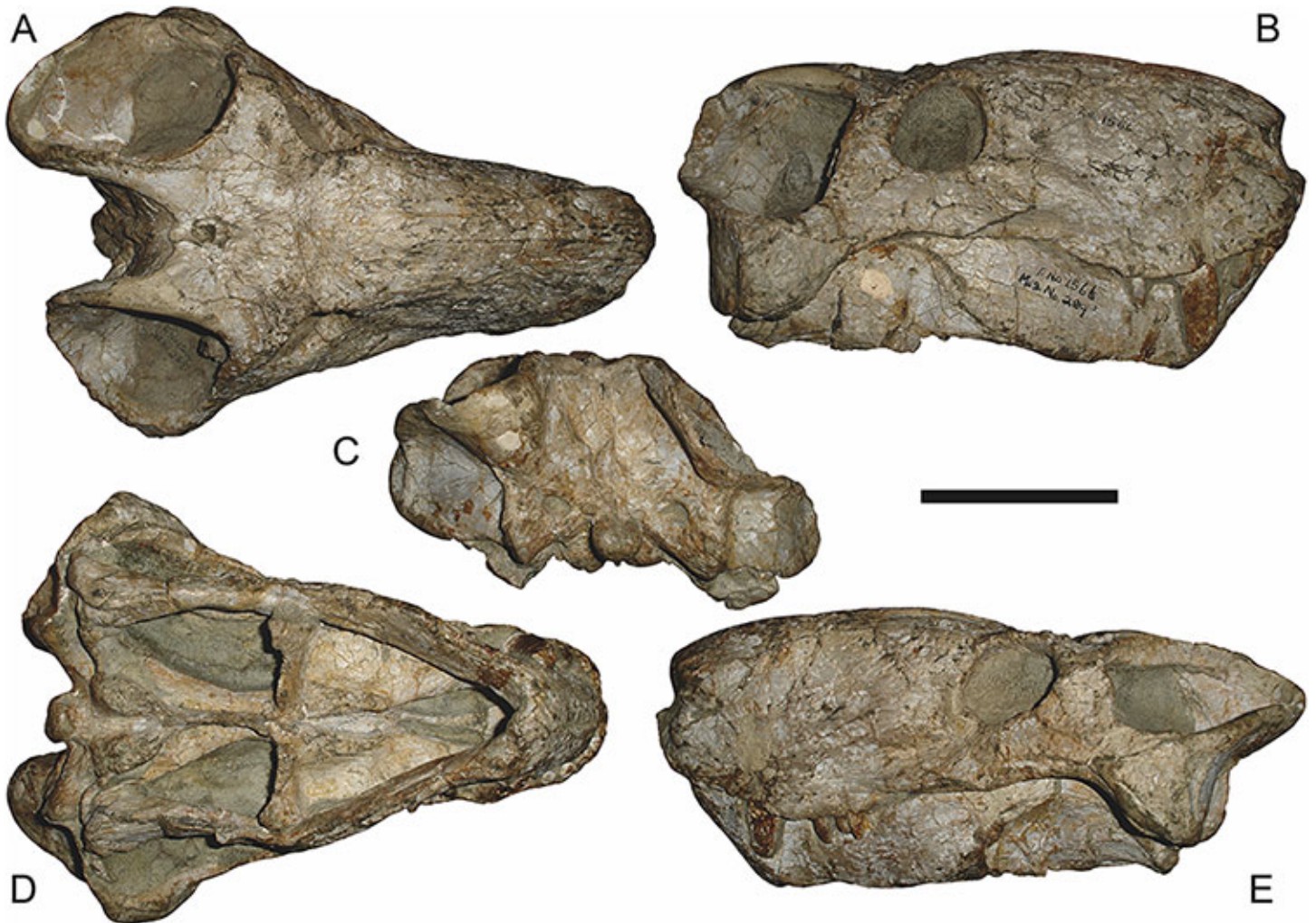

**Figure 7  Referred specimen (BP/1/1566) of *Aelurognathus tigriceps* (*Broom & Haughton, 1913*) in (A) dorsal, (B) right lateral, (C) occipital, (D) ventral, and (E) left lateral view.** Holotype of *Prorubidgea brinki Manten, 1959*. Scale bar equals 10 cm.

Graaff-Reinet, South Africa; holotype of *Clelandina major*); RC 110 (Fig. 13B; a partial skull and lower jaws from Zuurplaas, Graaff-Reinet, South Africa; holotype of *Gorgonorhinus minor*); RC 115 (Figs. 14A–14C; a skull and lower jaws from Ferndale, Graaff-Reinet, South Africa); RC 198 (a crushed partial skull and lower jaws from Graaff-Reinet Commonage, Graaff-Reinet, South Africa); RC 792 (a partial skull from Bulberg, Richmond, South Africa); SAM-PK-2672 (Fig. 14D; a snout and lower jaws from Dunedin, Beaufort West, South Africa; holotype of *Scymnognathus serratidens*); SAM-PK-10071 (a distorted but mostly complete skull from Dunedin, Beaufort West, South Africa); SAM-PK-11121 (a somewhat crushed skull and lower jaws from Rocklands, Beaufort West, South Africa); SAM-PK-K1220 (Fig. 13A; a crushed snout from Zuurplaas, Graaff-Reinet, South Africa; holotype of *Gorgonorhinus luckhoffi*); SAM-PK-K1302 (a partial snout and lower jaws from Bleak Hoose, Renosterkop, Beaufort West, South Africa); SAM-PK-K8558 (a complete skull and lower jaws from De Hoop 117,
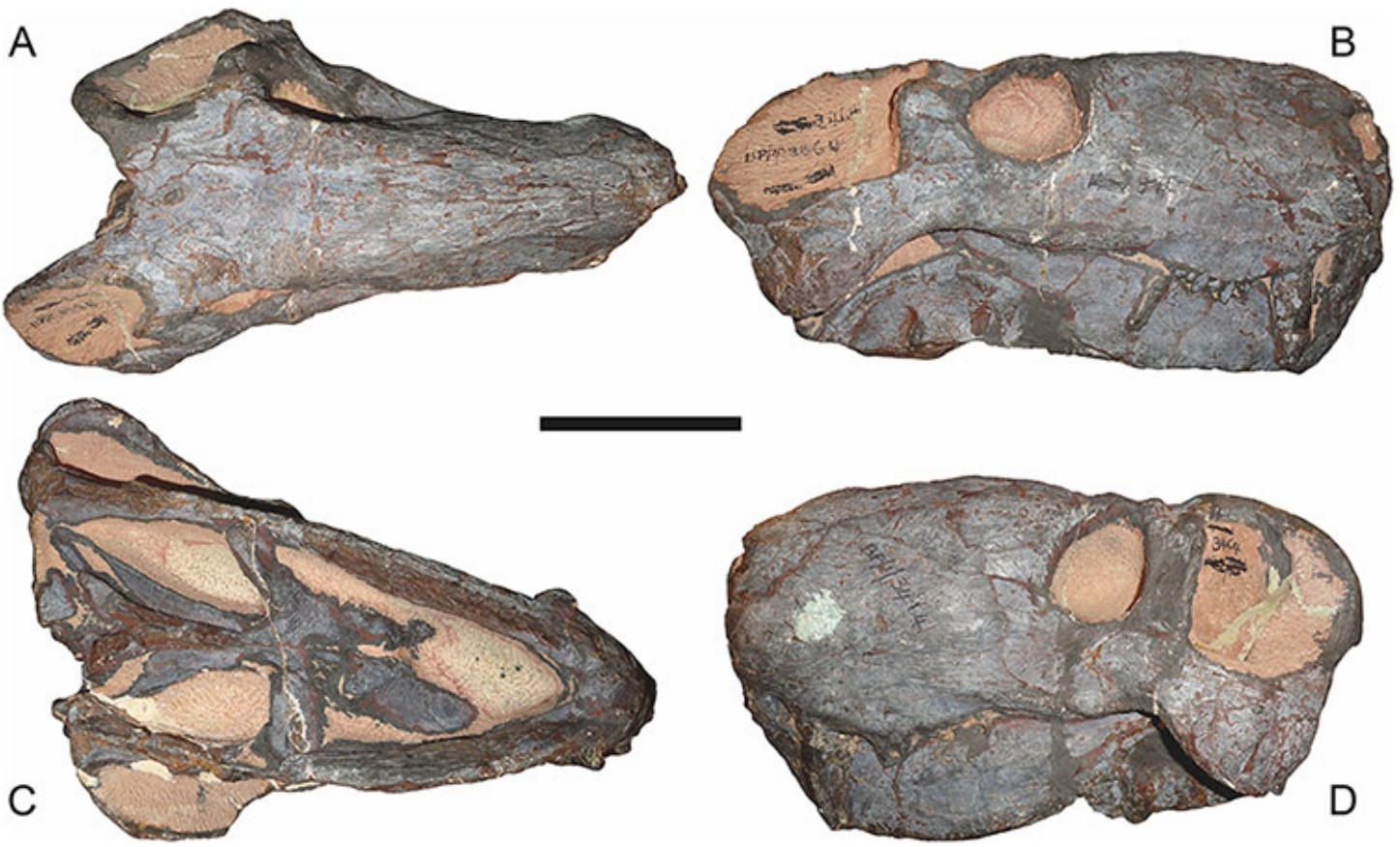

**Figure 8** Referred specimen (BP/1/3464) of *Aelurognathus tigriceps* (*Broom & Haughton, 1913*) in (A) dorsal, (B) right lateral, (C) ventral, and (D) left lateral view. Scale bar equals 10 cm.

Beaufort West, South Africa); TM 1493 (Fig. 15; a poorly-preserved skull and lower jaws from Houdconstant, Graaff-Reinet, South Africa; holotype of *Sycosaurus brodiei*).

*Diagnosis*: *Aelurognathus tigriceps* can be recognized as a rubidgeine by the combination of a low parasphenoid rostrum with median depression and reduction or absence of the preparietal. *Aelurognathus* can be distinguished from all rubidgeines other than *Smilesaurus* by the primitive retention of a tall, narrow occiput and discrete, dentigerous palatal boss of the pterygoid. *Aelurognathus* can be distinguished from *Smilesaurus* by the following features shared with all other rubidgeines: absence of a frontal contribution to the orbit, expanded postorbital bar, and thickened dorsal margin of the orbit and temporal fenestra. It can also be distinguished from *Smilesaurus* by the long, narrow parasphenoid rostrum (a primitive retention), proportionally smaller canine, bulbous snout, anteriorly bulbous interchoanal body, and presence of 4–6 upper postcanines.

*Comments*: *Broom & Haughton (1913)* originally described this taxon as *Scymnognathus tigriceps*, with the genus *Scymnognathus* serving as a wastebasket for medium-sized gorgonopsians at the time. *Haughton (1924)* re-examined the type specimen of *S. tigriceps* (SAM-PK-2342) and, concluding that it was not congeneric with *Scymnognathus whaitsi* (the type species of *Scymnognathus*, which is currently considered a junior synonym of

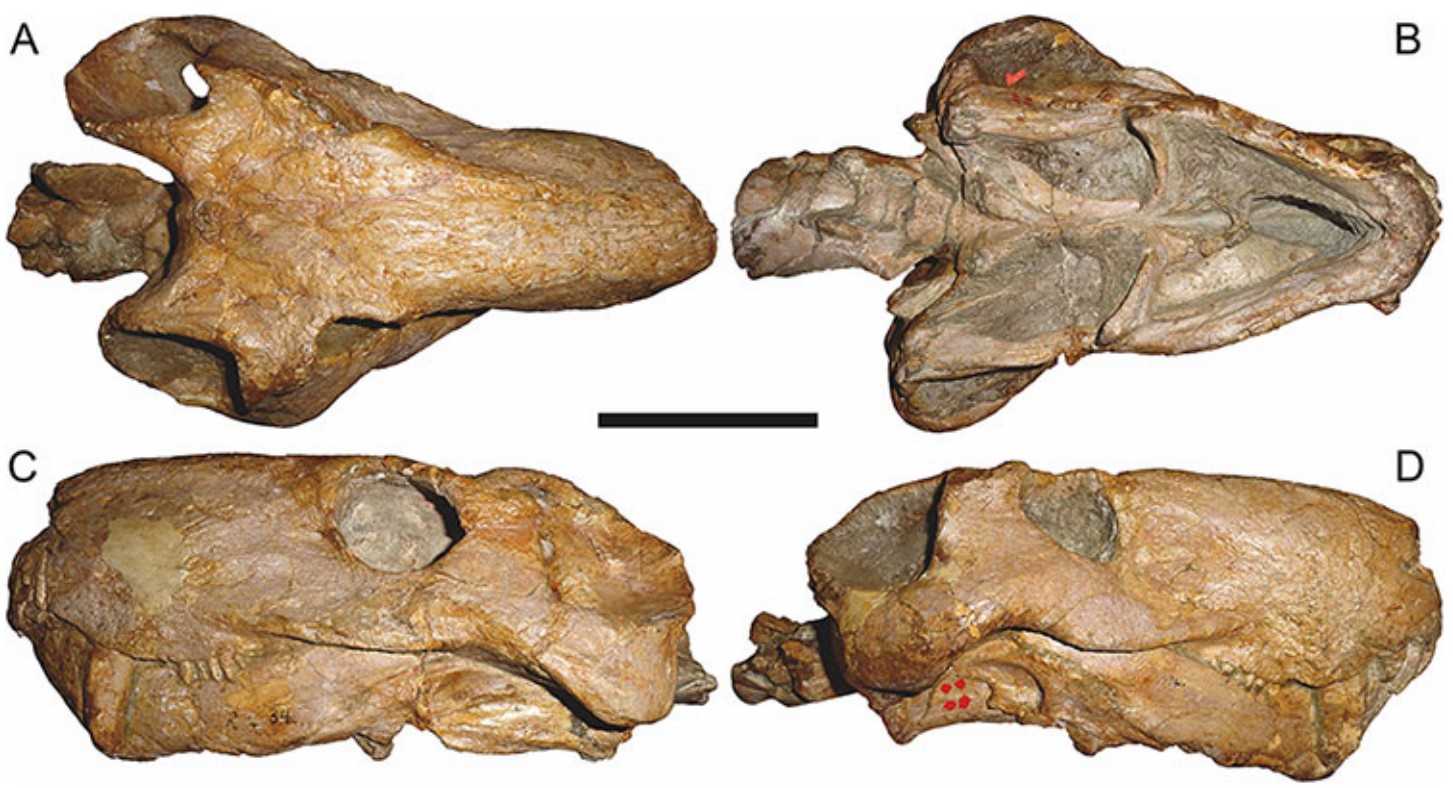

**Figure 9 Referred specimen (RC 34) of *Aelurognathus tigriceps* (*Broom & Haughton, 1913*) in (A) dorsal, (B) ventral, (C) left lateral, and (D) right lateral view.** Holotype of *Prorubidgea maccabei Broom, 1940b*. Scale bar equals 10 cm.

*Gorgonops* (*Sigogneau, 1970*)), established the new genus *Aelurognathus* for it. In the same paper, Haughton referred another of his previously-described *Scymnognathus* species, *S. serratidens*, to *Aelurognathus*. In subsequent years, *Aelurognathus* also became somewhat of a wastebasket, and had a variety of disparate new species referred to it (*Aelurognathus nyasaensis Haughton, 1926*; *Aelurognathus microdon Boonstra, 1934*; *Aelurognathus sollasi Broili & Schröder, 1935*; *Aelurognathus haughtoni Huene, 1950*; *Aelurognathus minor Brink & Kitching, 1953*).

In her monographic revision of South African gorgonopsians, *Sigogneau (1970)* maintained most of the nominal *Aelurognathus* species as valid, but removed *A. haughtoni* (which she referred to *Leontocephalus*), *A. microdon*, and *A. minor* (both of which she tentatively referred to *Lycaenops*) from the genus. She also questioned the validity of *A. nyasaensis*, referring to the holotype SAM-PK-7847 as *Aelurognathus* cf. *tigriceps*. Additionally, *Sigogneau (1970)* referred the east African gorgonopsian species *Dixeya quadrata Haughton, 1926* and *Scymnognathus parringtoni Huene, 1950* to *Aelurognathus*. *Sigogneau-Russell (1989)* largely followed the taxonomic scheme of *Sigogneau (1970)*, but resurrected *A. nyasaensis* for a total of six valid species of *Aelurognathus*: *A. quadrata*, *A. nyassaensis* (sic), ?*A. parringtoni*, *A. serratidens*, *A. sollasi*, and *A. tigriceps*.

*Gebauer (2007)* revised the genus *Aelurognathus* as part of her redescription of *Scymnognathus parringtoni* and broader study of gorgonopsian taxonomy.

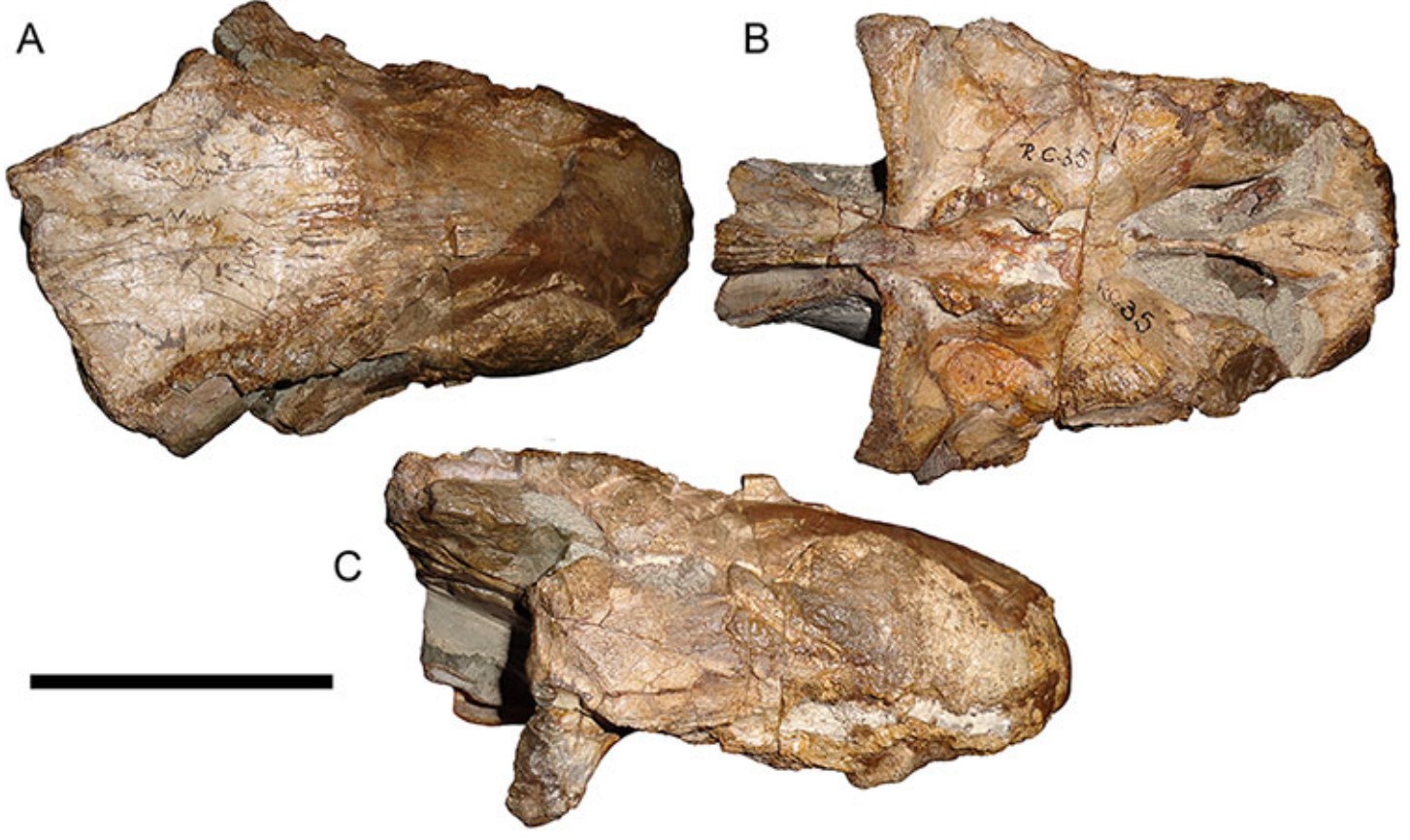

**Figure 10 Referred specimen (RC 35) of *Aelurognathus tigriceps* (*Broom & Haughton, 1913*) in (A) dorsal, (B) ventral, and (C) right lateral view.** Holotype of *Leontocephalus cadlei Broom, 1940b*. Scale bar equals 10 cm.

She synonymized *A. serratidens* with *A. tigriceps*, transferred *A. quadrata* and *A. sollasi* to *Lycaenops*, and transferred ?*A. parringtoni* to the otherwise Russian genus *Sauroctonus*. Additionally, *Gebauer (2007)* referred *Tigricephalus kingwilli Broom, 1948* (*Lycaenops kingwilli* in *Sigogneau (1970)*) and *Smilesaurus ferox Broom, 1948* (?*Arctops ferox* in *Sigogneau (1970)*) to *Aelurognathus*. Most importantly, she synonymized the genus *Prorubidgea Broom, 1940b* with *Aelurognathus*. *Prorubidgea* was originally established by *Broom (1940b)* for *P. maccabei*, a species known only from a large, well-preserved skull (RC 34) from Graaff-Reinet. Subsequent workers added additional species to *Prorubidgea* (*Prorubidgea robusta Brink & Kitching, 1953*; *Prorubidgea brinki Manten, 1959*) and *Sigogneau (1970)* transferred the species *Lycaenops alticeps Brink & Kitching, 1953* and *Sycosaurus brodiei Broom, 1941* to this genus. *Sigogneau-Russell (1989)* had recognized a close similarity between *Aelurognathus* and *Prorubidgea*, and noted that the former could be ancestral to the latter, but included only *Prorubidgea* in the Rubidgeinae. *Gebauer (2007)* took these observations to their logical conclusion, recognizing only a single genus for these species, for which the name *Aelurognathus* has priority. However, she retained most of the former *Prorubidgea* species as valid, synonymizing only *P. brinki* with her *Aelurognathus alticeps* and *P. robusta* with her *A. broodiei* (sic). So in total, *Gebauer (2007)*

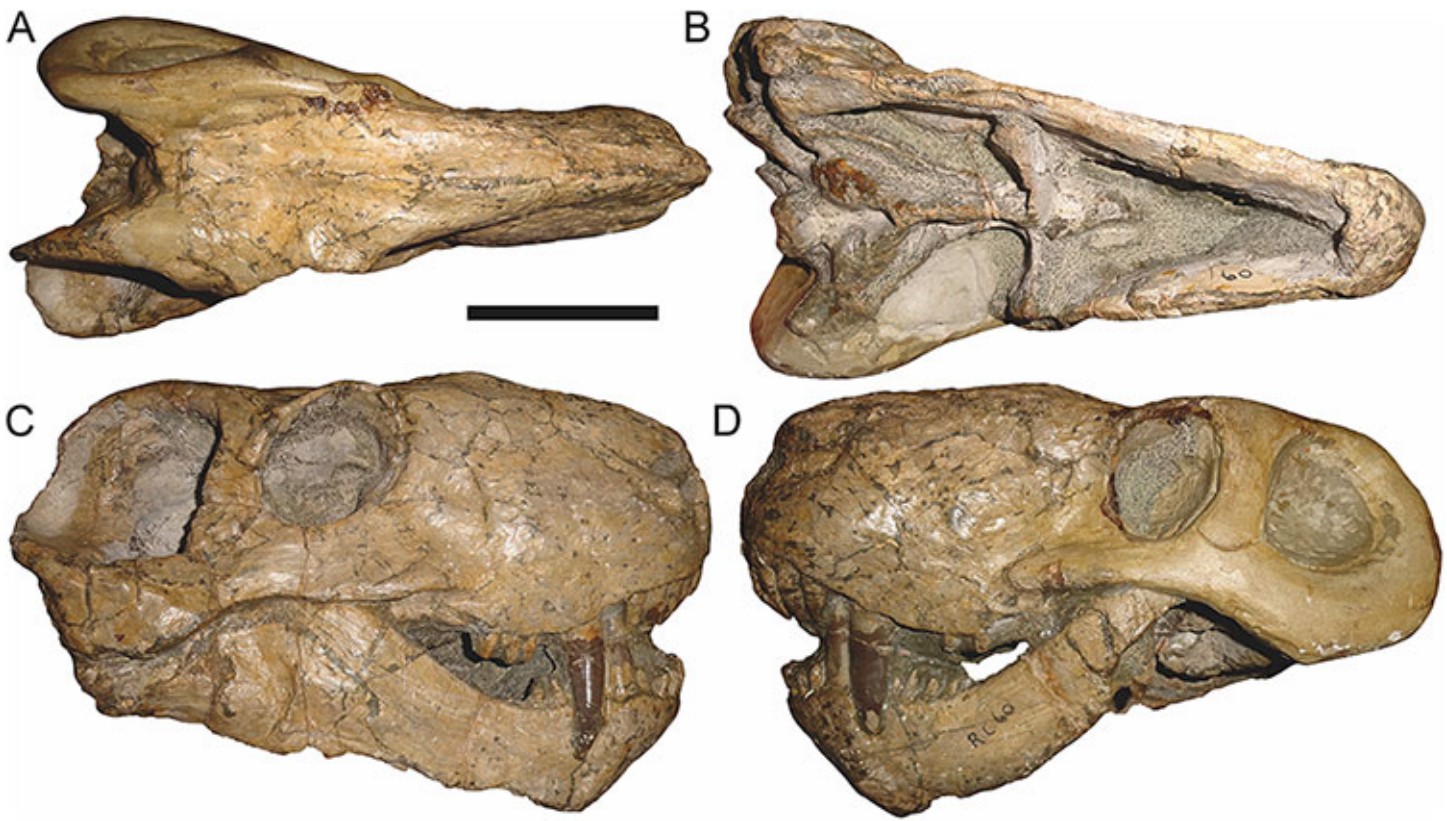

**Figure 11 Referred specimen (RC 60) of *Aelurognathus tigriceps* (*Broom & Haughton, 1913*) in (A) dorsal, (B) ventral, (C) right lateral, and (D) left lateral view.** Holotype of *Tigricephalus kingwilli Broom, 1948*. Scale bar equals 10 cm.

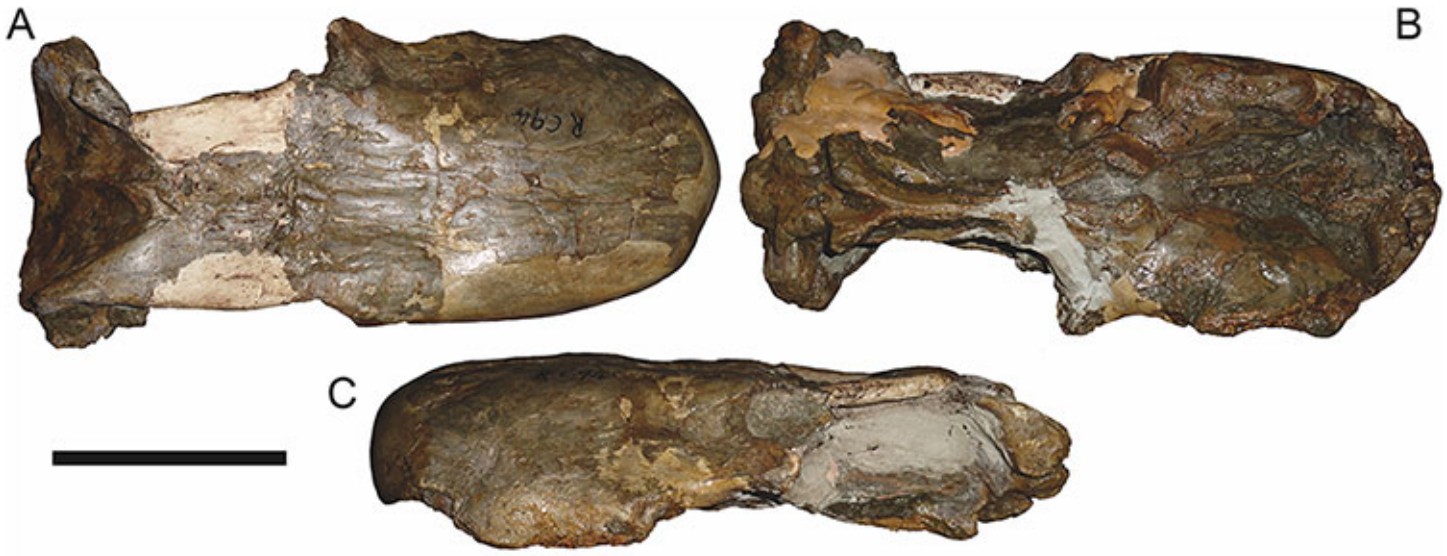

**Figure 12 Referred specimen (RC 94) of *Aelurognathus tigriceps* (*Broom & Haughton, 1913*) in (A) dorsal, (B) ventral, and (C) left lateral view.** Holotype of *Clelandina major Broom, 1948*. Scale bar equals 10 cm.

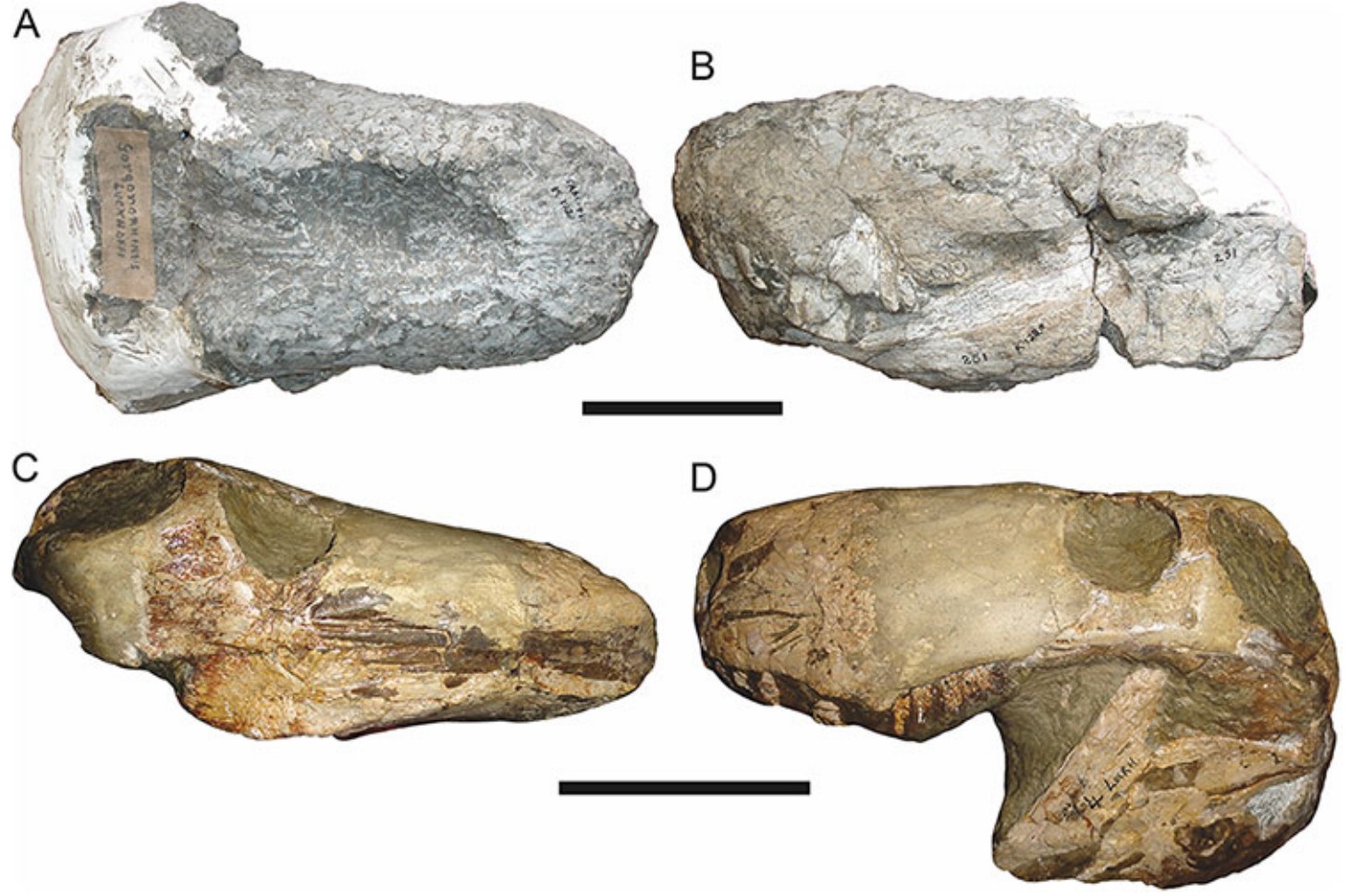

**Figure 13 Referred specimens of *Aelurognathus tigriceps* (*Broom & Haughton, 1913*).** SAM-PK-K1280 (holotype of *Gorgonorhinus luckhoffi Broom, 1937*) in (A) dorsal and (B) left lateral view; RC 110 (holotype of *Gorgonorhinus minor Broom, 1948*) in (C) dorsal and (D) left lateral view. Scale bars equal 10 cm.

also recognized six valid species of *Aelurognathus*: *A. alticeps*, *A. broodiei* (sic), *A. ferox*, *A. kingwilli*, *A. maccabei*, and *A. tigriceps*.

*Norton (2012)* examined 16 gorgonopsian specimens referred to *Aelurognathus* sensu *Gebauer (2007)* and used linear morphometrics to test specific variation in skull morphology. Unable to recover discrete species clusters within these data, he considered there to be only a single valid species of *Aelurognathus*, *A. tigriceps*, including the other five species recognized by Gebauer as junior synonyms.

My interpretation of these specimens accords with some of the previous work on *Aelurognathus*, but differs in a number of details. I concur with *Sigogneau (1970)* in excluding *A. haughtoni*, *A. microdon*, and *A. minor* from *Aelurognathus*. The status of *A. haughtoni* is dealt with in detail in the section on *Ruhuhucerberus* below. All specimens referred to *A. minor* (see *Sigogneau-Russell (1989)* for listings) have a tall, blade-like parasphenoid rostrum and numerous teeth on the transverse process of the pterygoid, indicating that they are not *Aelurognathus*. These specimens bear 3–4 close-packed upper

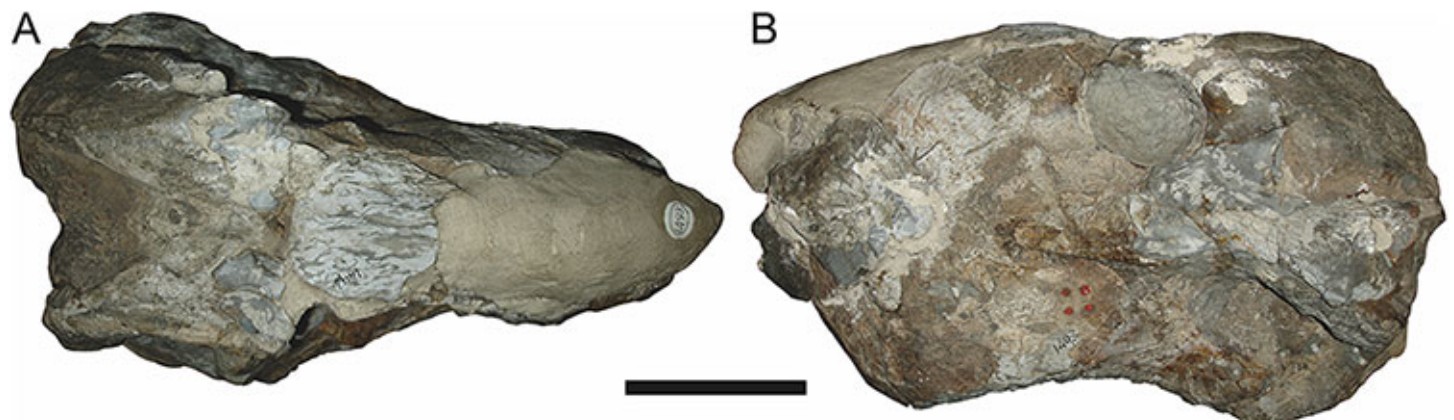

**Figure 14 Referred specimens of *Aelurognathus tigriceps* (*Broom & Haughton, 1913*).** RC 115 in (A) dorsal, (B) right lateral, and (C) left lateral view; SAM-PK-2672 (holotype of *Scymnognathus serratidens Haughton, 1915*) in (D) left lateral view. Scale bars equal 10 cm.

**Figure 15 Referred specimen (TM 1493) of *Aelurognathus tigriceps* (*Broom & Haughton, 1913*) in (A) dorsal and (B) left lateral view.** Holotype of *Sycosaurus brodiei Broom, 1941*. Scale bar equals 10 cm.

postcanines and have a deflected subtemporal bar, indicating that *Sigogneau's (1970)* referral of this species to *Lycaenops* is probably correct, although *Kammerer (2015)* noted that the validity of this species requires reconsideration. The generic position of *A. microdon* is more uncertain, but the large preparietal and low, straight snout of the holotype (SAM-PK-9344) do indicate that it is not *Aelurognathus*. The short row of small, close-packed postcanines in this specimen is very similar to that of *A. minor*, and they may be conspecific.

I concur with *Gebauer (2007)* in excluding *A. sollasi* and ?*A. parringtoni* from *Aelurognathus*. The palatal dentition of *A. sollasi* is more extensive than that of *Aelurognathus* (particularly on the transverse process) and ?*A. parringtoni* has a blade-like parasphenoid rostrum. *Aelurognathus* (originally *Dixeya*) *quadrata* is more problematic— the type specimen (SAM-PK-7856) is very poor, but it also appears to have a blade-like parasphenoid rostrum. I also agree with the synonymy of *A. serratidens* and *A. tigriceps*, which was originally proposed by *Broom (1932)*. The fact that even Robert Broom considered these specimens conspecific should be sufficient indication that these species are synonymous, but to expand slightly on the topic, their type specimens are from the same locality (Dunedin) and the only character differentiating them is the larger preparietal of *A. serratidens* (*Haughton, 1915*; *Sigogneau-Russell, 1989*). Preparietal size and shape varies extensively in therapsids (including gorgonopsians; *Kammerer et al., 2015*), and is not a robust indicator of taxonomic distinction. Other, minor proportional differences between *A. serratidens* and *A. tigriceps* are likely to be taphonomic in origin: in overall morphology SAM-PK-2672 (holotype of *A. serratidens*) is nearly identical to a similarly-preserved specimen referable to *A. tigriceps* (see comparisons in Fig. 14).

Dunedin is a *Tropidostoma* AZ locality (*Smith, 1993*), making SAM-PK-2342 and SAM-PK-2672 among the earliest known rubidgeines. Most other members of the subfamily, and the majority of other specimens herein referred to *Aelurognathus tigriceps*, are from later in the *Cistecephalus* and *Daptocephalus* AZs. As such, one may question the conspecificity of these early records with later specimens of '*A. tigriceps*', especially given the seemingly primitive retention of a large preparietal in these specimens (Fig. 5B). SAM-PK-10071, another specimen from Dunedin, has a much smaller preparietal, but the skull is otherwise very similar to SAM-PK-2342 and SAM-PK-2672. Given this variability, and the retention of a small preparietal in some stratigraphically higher *Aelurognathus* specimens (e.g., BP/1/813), I do not consider the presence of a preparietal in the *Tropidostoma* AZ material to indicate specific distinction. However, better-prepared *Aelurognathus* specimens from the *Tropidostoma* AZ are needed to properly evaluate this issue—the three known specimens are all badly damaged, limiting their utility for detailed comparison.

Although I concur with *Norton (2012)* in recognizing only a single species of *Aelurognathus*, I consider his synonymy of all six *Aelurognathus* species sensu *Gebauer (2007)* to be overzealous. *Aelurognathus ferox* is clearly a distinct taxon, as is dealt with in detail in the section on *Smilesaurus* below. Furthermore, BP/1/2190 (holotype of *Prorubidgea robusta*, which *Gebauer (2007)* considered synonymous with

*Aelurognathus broodiei* (sic)) can be referred to *Dinogorgon rubidgei* rather than *Aelurognathus*, as discussed in the section on *Dinogorgon*. The remaining four species (*A. alticeps*, *A. brodiei*, *A. kingwilli*, and *A. maccabei*) are best considered synonyms of *A. tigriceps* based on available data, however.

BP/1/813, the holotype of *Aelurognathus* (originally *Lycaenops*) *alticeps* (Fig. 6), is one of the smaller (23.0 cm basal skull length) known specimens of *Aelurognathus*. The prepariet al in this specimen is extremely reduced in size, and although the base of the skull is poorly prepared, the absence of a blade-like parasphenoid rostrum appears to be real. In addition to these general rubidgeine features, the presence of five postcanines, a tall, bulbous snout, and a tall, narrow occiput indicate that this specimen is referable to *Aelurognathus tigriceps*. *Sigogneau-Russell (1989)* and *Gebauer (2007)* retained this species as valid because of its relatively narrow intertemporal region, but intertemporal width is frequently an ontogenetically variable feature in therapsids (see *Kammerer, Angielczyk & Fröbisch, 2011*), and given the small size of this skull this is not sufficient grounds to recognize a separate species. *Gebauer (2007)* considered *Prorubidgea brinki* to be a synonym of *A. alticeps*, and argued that it has a proportionally smaller postfrontal than other species of *Aelurognathus*. The holotype of *P. brinki*, BP/1/1566, is a well-preserved skull that has suffered only minor distortion (Fig. 7). Intriguingly, it shows a small, rhomboidal anterior process of the parietals that is equivalent in size and position to the prepariet al in BP/1/813, suggesting fusion of that element with growth. Although generally well-preserved, the skull roof of BP/1/1566 has numerous cracks. My examination of this specimen suggests that the postfrontal-frontal 'suture' that *Gebauer (2007)* took to indicate an unusually small postfrontal is actually a crack, with the actual postfrontal-frontal suture being located more medially.

TM 1493, the holotype of *Aelurognathus* (originally *Sycosaurus*) *brodiei*, is a large (34.0 cm basal length), very poorly-preserved and prepared skull (Fig. 15). This specimen has a tall, short snout, massive lower jaw, and five postcanines. Based on these features alone it could represent either *Aelurognathus* or *Dinogorgon*, but the combination of a weakly-emarginated maxilla, only moderately expanded postorbital bar, and weakly deflected subtemporal bar indicates that it is referable to *A. tigriceps*. *Sigogneau-Russell (1989)* retained this species based on its narrow interorbital region, but given that the orbital margin is damaged on both sides of the skull this character is not reliable. *Gebauer (2007)* considered this species to have a straighter dorsal profile of the skull than is typical for *Aelurognathus*, but the snout of TM 1493 is mostly restored in plaster; the sole intact portion in front of the orbits is convex, indicating that the snout was bulbous. Gebauer's diagnosis for this species was based primarily on BP/1/2190 (holotype of *Prorubidgea robusta*), which, as noted above, I consider to be a specimen of *Dinogorgon rubidgei*.

RC 60, the holotype of *Aelurognathus* (originally *Tigricephalus*) *kingwilli* (Fig. 11), is slightly smaller (29.9 cm basal length) than TM 1493 but is extremely similar in its preserved anatomy (compare Figs. 11 and 15). *Gebauer (2007)* diagnosed this species based on the combination of a small lacrimal, wide occiput, relatively narrow subtemporal bar, and absence of a prepariet al. In all of these features, however, RC 60 is comparable to

other, similar-sized specimens of *A. tigriceps*, and is considered synonymous with that species here.

The most problematic of specimens herein referred to *Aelurognathus tigriceps* is RC 34, the holotype of *Prorubidgea maccabei*. This specimen is comparable in size (27.5 cm basal skull length) to RC 60, but has a significantly more expanded postorbital bar and a longer, lower snout. At present I consider these differences to most likely be due to a combination of taphonomic distortion and intraspecific variation. RC 60 has suffered some lateral crushing and RC 34 some dorsoventral, which may account for the differences in snout morphology between them. Of relevance to this issue is a specimen from Zambia (BP/1/3464; Fig. 8), here referred to *A. tigriceps*, that has suffered shear such that the two sides of its skull have been distorted in different ways. In BP/1/3464, the right side of the skull is similar in appearance to RC 34 (compare Figs. 8B and 9C), whereas the left is similar to RC 60 (compare Figs. 8D and 11C). Given this variability and the singleton status of RC 34, *P. maccabei* is considered synonymous with *A. tigriceps* here. If future discoveries show that the proportions of RC 34 are more broadly present in the record, this synonymy will need to be reconsidered: additional field work at the *P. maccabei* type locality (St. Olives, Graaff-Reinet) would be beneficial towards resolving this problem. Additional preparation of RC 34 (particularly to better expose the anterior vomer) would also be useful, as this specimen's snout and postorbital proportions are closer to those of *Sycosaurus* than other *Aelurognathus*, and it is possible this specimen will prove referable to the former genus.

In addition to the species discussed above, there are several nominal gorgonopsian taxa that have never been considered in the context of possible synonyms of *Aelurognathus*, but which my examination suggests are referable to *A. tigriceps*. The genus *Gorgonorhinus Broom, 1937* contains two nominal species: *G. luckhoffi Broom, 1937* (the type) and *G. minor Broom, 1948*. The type specimens of both species are exceedingly poor, but their preserved skull morphology accords with *Aelurognathus tigriceps*. The holotype of *Gorgonorhinus luckhoffi* (SAM-PK-K1220) is a very large (~22 cm snout length) specimen that is almost completely unprepared (Figs. 13A and 13B). Plaster obscures the orbital region, and the skull is broken off before the postorbital bar. *Sigogneau (1970)* and *Sigogneau-Russell (1989)* considered this specimen to be indeterminate. The referral of this specimen to *Aelurognathus tigriceps* is tentative, and based on the presence of five postcanines (preserved as roots on the left side, and whole crowns of PC1, 2, 4, and 5 on the right side) and the lower position of the incisor tooth row compared to *Dinogorgon*. This specimen appears to have suffered dorsoventral crushing, however, so this proportional difference may be artifactual. Preparation is needed to confirm the taxonomic attribution of *G. luckhoffi*. If it is an individual of *A. tigriceps*, it would be one of the largest specimens known from South Africa.

The second species of *Gorgonorhinus*, *G. minor*, was tentatively referred to *Arctops* by *Sigogneau (1970)*, in the new combination *Arctops? minor*. *Gebauer (2007)* considered the holotype (RC 110) indeterminate. RC 110 is badly worn and highly incomplete, missing much of the right side of the skull and with the left preorbital region reconstructed in plaster (Figs. 13C and 13D). The only visible sutures are in the

interorbital region and on the lateral surface of the snout tip. The preparietal is absent. The frontals appear to contribute to the orbital rim, but this is probably attributable to damage, as large portions of the circumorbital bones are broken or worn. Although damaged, the left postorbital bar is clearly broad at base and narrows dorsally. RC 110 has a high postcanine tooth count; *Sigogneau-Russell (1989)* tentatively listed this specimen as having seven upper postcanines. Only two tooth crowns (and a sliver-like posterior tooth fragment) are visible on the right side of the skull, but most of the maxillary alveolar surface is covered with matrix. A combination of broken crowns and tooth impressions do indicate the presence of seven teeth in the left maxilla, but the third tooth position consists solely of a narrow impression wedged between two well-developed crowns. I interpret this tooth position as the remains of a postcanine undergoing replacement (either PC2 or PC3), and suggest that this specimen had only six postcanines. The combination of six upper postcanines, absence of a preparietal, and postorbital morphology indicates that RC 110 is referable to *Aelurognathus tigriceps*. Although a high tooth count (6–7 upper postcanines) and lack of a preparietal also characterize the coeval non-rubidgeine gorgonopsian *Arctognathus curvimola*, RC 110 can be distinguished from *Arctognathus* by the relatively anterior termination of the tooth row (in *Arctognathus*, the tooth row terminates beneath the lacrimal, near the orbital margin) and the absence of a concave maxillary margin around the canine root (*Kammerer, 2015*).

*Leontocephalus cadlei Broom, 1940b* is the type species of *Leontocephalus*. *Sigogneau (1970)* considered this genus to be valid (but not a rubidgeine), and recognized four species: *L. cadlei Broom, 1940b*, *L. haughtoni* (*Huene, 1950*; originally *Aelurognathus*), ?*L. intactus Kemp, 1969*, and ?*L. rubidgei Broom, 1940a*; originally *Broomisaurus*. *Gebauer (2007)* considered RC 35, the holotype of *L. cadlei*, to be referable to *Sycosaurus* but indeterminate to species. This skull is very incomplete, worn, and dorsoventrally crushed, but the intertemporal skull roof and palate are quite well preserved (Fig. 10). The preparietal is absent. Although the edges of the orbits are poorly preserved, the frontal is clearly excluded from the orbital margin. The transverse process of the pterygoid is edentulous, but the palatal boss of the pterygoid is discrete and dentigerous, bearing 4–5 palatal teeth. At least four postcanines are present, and there were probably five total (as indicated by missing space between teeth). The interchoanal body of the vomer is bulbous anteriorly. Taken as a whole, this combination of characters is known only in *Aelurognathus tigriceps*, and despite its incompleteness, *L. cadlei* should be synonymized with that taxon. The better-known Tanzanian species *Leontocephalus intactus* is not referable to *Aelurognathus*, however—for coverage of this taxon refer to the section on *Sycosaurus nowaki* below.

The species *Clelandina major Broom, 1948* has largely been ignored by previous gorgonopsian workers—*Sigogneau (1970)* considered it *incertae sedis*, as she had not been able to examine the holotype (RC 94), and *Gebauer (2007)* did not mention it. RC 94 is a very badly crushed skull (strongly dorsoventrally compressed), but the skull roof and palate are well-preserved and reasonably prepared. The combination of five postcanines, an anteriorly bulbous interchoanal body of the vomer, discrete, dentigerous palatal boss of

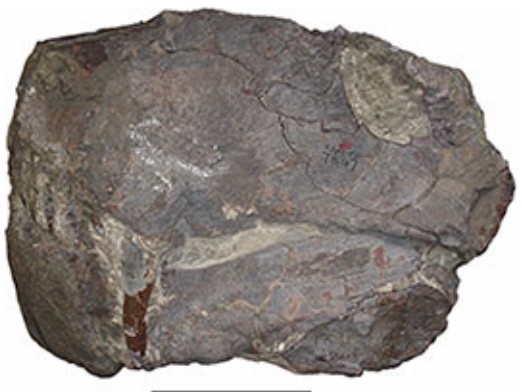

**Figure 16 Holotype (SAM-PK-7847) of *Aelurognathus nyasaensis Haughton, 1926* in left lateral view.** This specimen is of uncertain generic attribution, and requires additional preparation and study. Scale bar equals 10 cm.

the pterygoid, absence of a preparietal, and lack of a blade-like parasphenoid rostrum is sufficient to identify this as a specimen of *Aelurognathus tigriceps*.

*Sigogneau (1970)* and *Gebauer (2007)* both considered the Malawian species *Aelurognathus nyasaensis Haughton, 1926* to be synonymous with *Aelurognathus tigriceps* (although *Sigogneau-Russell (1989)* reversed this decision). *Aelurognathus nyasaensis* is known only from a strongly sheared partial skull (broken behind the postorbital bar) and lower jaws (SAM-PK-7847) from Chiweta (Fig. 16). This skull appears to lack a preparietal and has a very deep suborbital portion of the zygoma. The mandibular symphysis is massive and the snout relatively tall. The postcanine count cannot be taken with certainty because of incomplete preparation. Only two alveoli are visible in the right maxilla, and the tooth row is completely obscured on the left maxilla. Unfortunately, the exposed morphology of SAM-PK-7847 does not permit a specific attribution; it could represent a distorted specimen of either *Aelurognathus* or *Dinogorgon*, and the cranial proportions also somewhat evoke *Smilesaurus*. At present, *A. nyasaensis* must be considered indeterminate. Additional preparation of the holotype is required to resolve the status of this taxon.

### *Clelandina Broom, 1948*

*Tigrisaurus Broom & George, 1950*:188
*Dracocephalus Brink & Kitching, 1953*:5

*Type species*: *Clelandina rubidgei Broom, 1948*.
 *Diagnosis*: As for the type and only recognized species.

### *Clelandina rubidgei Broom, 1948* (Reconstruction Figs. 17–18, Specimen Figs. 19–23)

*Tigrisaurus pricei Broom & George, 1950*:188
*Dracocephalus scheepersi Brink & Kitching, 1953*:5

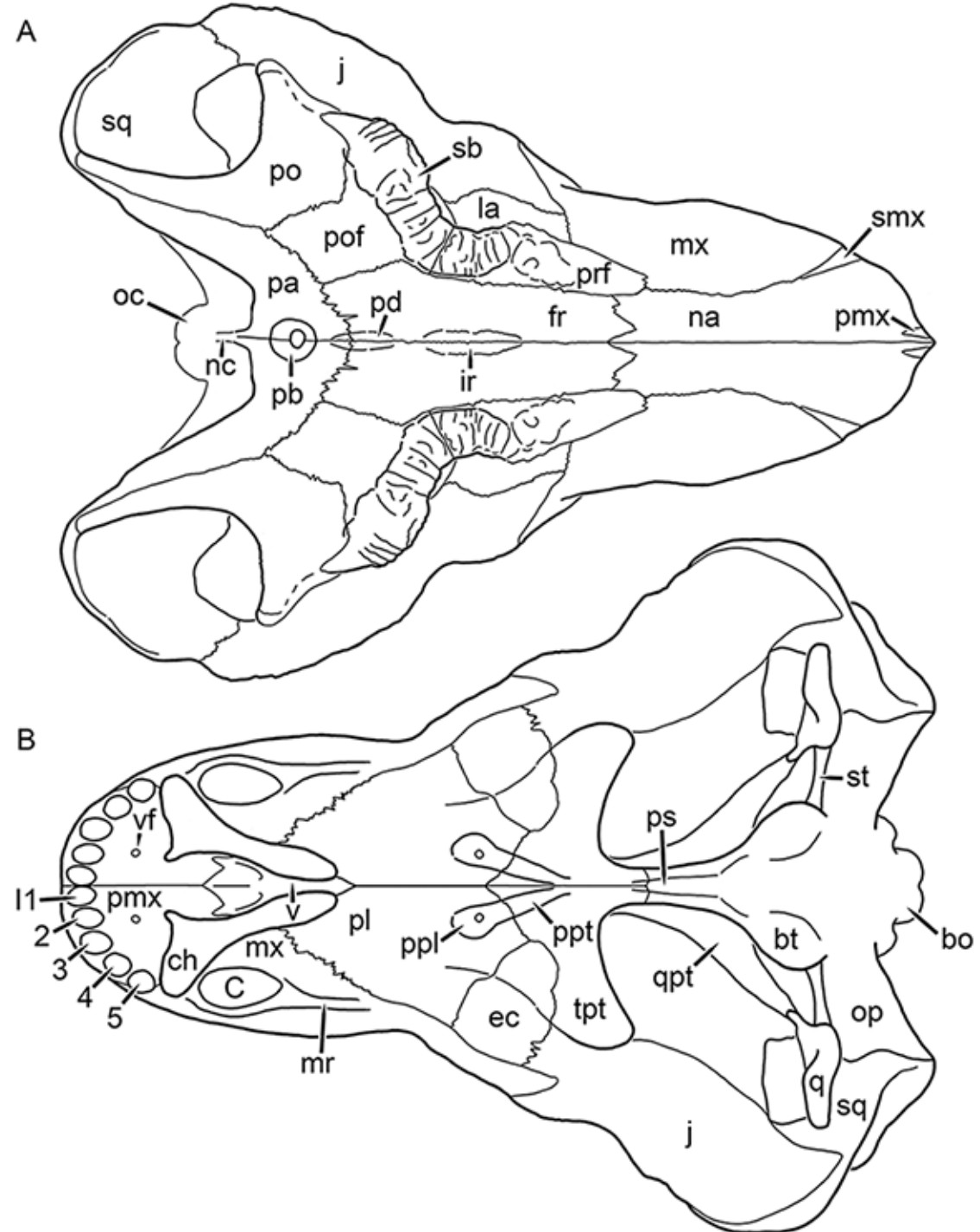

**Figure 17 Reconstruction of the skull of *Clelandina rubidgei Broom, 1948* in (A) dorsal and (B) ventral views.** Reconstructions based primarily on BP/1/742 and UCMP 35437. Abbreviations: bo, basioccipital; bt, basal tuber; C, upper canine; ch, choana; ec, ectopterygoid; fr, frontal; I, upper incisor; ir, interorbital ridge; j, jugal; la, lacrimal; mr, maxillary ridge; mx, maxilla; na, nasal; nc, nuchal crest; oc, occipital condyle; op, opisthotic; pa, parietal; pb, pineal boss; pd, pre-parietal depression; pl, palatine; pmx, premaxilla; po, postorbital; pof, postfrontal; ppl, palatal boss of palatine; ppt, palatal boss of pterygoid; prf, prefrontal; ps, parasphenoid; q, quadrate; qpt, quadrate ramus of pterygoid; sb, supraorbital boss; smx, septomaxilla; sq, squamosal; st, stapes; tpt, transverse process of pterygoid; v, vomer; vf, ventral premaxillary foramen.

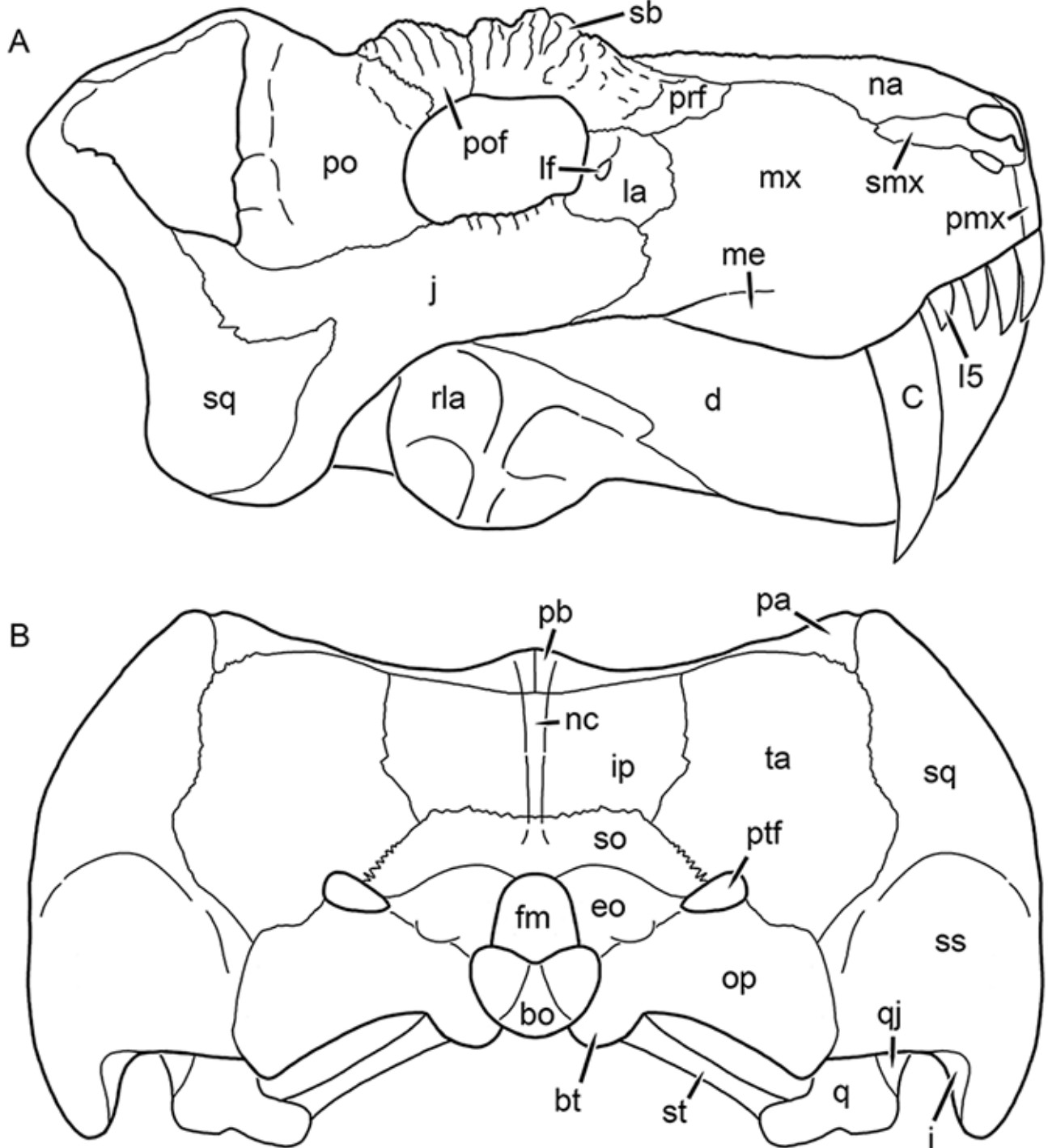

**Figure 18 Reconstruction of the skull of *Clelandina rubidgei Broom, 1948* in (A) lateral and (B) occipital views.** Reconstructions based primarily on BP/1/742, RC 102, and UCMP 35437. Abbreviations: bo, basioccipital; bt, basal tuber; C, upper canine; d, dentary; eo, exoccipital; fm, foramen magnum; I, upper incisor; ip, interparietal; j, jugal; la, lacrimal; lf, lacrimal foramen; me, maxillary emargination; mx, maxilla; na, nasal; nc, nuchal crest; op, opisthotic; pa, parietal; pb, pineal boss; pmx, premaxilla; po, postorbital; pof, postfrontal; prf, prefrontal; ptf, post-temporal fenestra; q, quadrate; qj, quadratojugal; rla, reflected lamina of angular; sb, supraorbital boss; smx, septomaxilla; so, supraoccipital; sq, squamosal; ss, squamosal sulcus; st, stapes; ta, tabular.

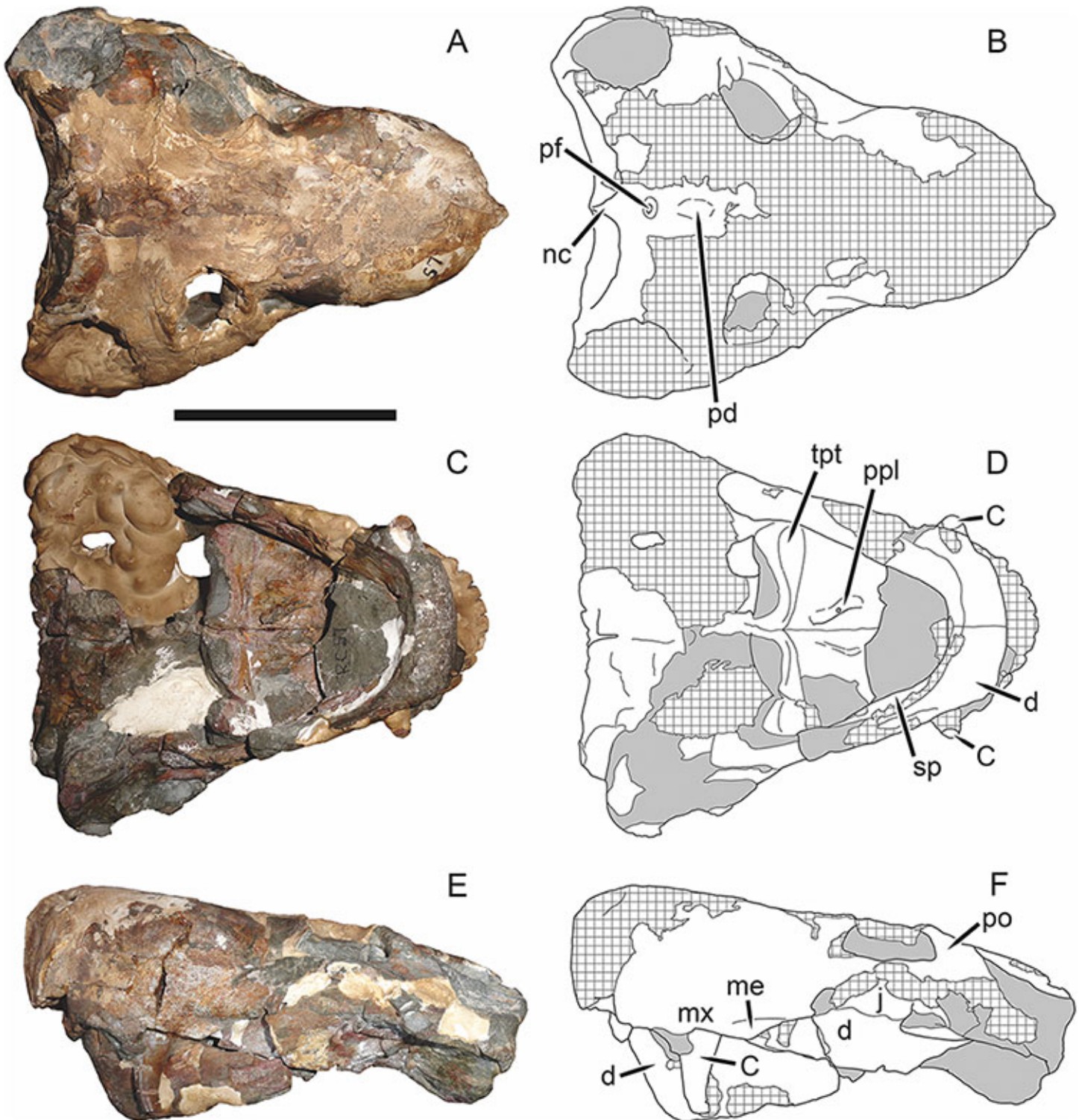

**Figure 19 Holotype (RC 57) of *Clelandina rubidgei* *Broom, 1948* in (A) dorsal, (C) ventral, and (E) left lateral view (with (B) (D) and (F) interpretive drawings).** Abbreviations: C, upper canine; d, dentary; j, jugal; me, maxillary emargination; nc, nuchal crest; pd, pre-parietal depression; pf, pineal foramen; po, postorbital; ppl, palatal boss of palatine; sp, splenial; tpt, transverse process of pterygoid. Gray indicates matrix, hatching indicates plaster. Scale bar equals 10 cm.

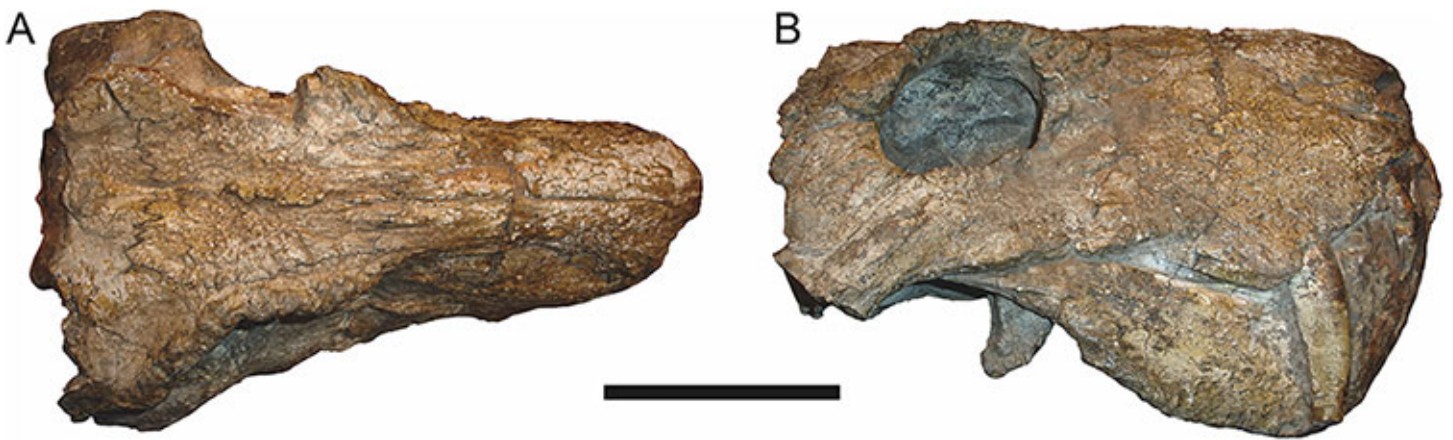

**Figure 20** Referred specimen (BP/1/742) of *Clelandina rubidgei Broom, 1948* in (A) dorsal and (B) right lateral view. Holotype of *Tigrisaurus pricei Broom & George, 1950*. Scale bar equals 10 cm.

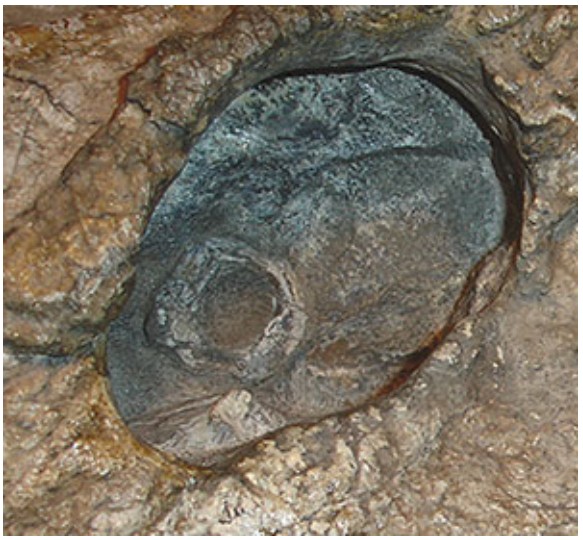

**Figure 21** Close-up on the right orbit of BP/1/742 (*Clelandina rubidgei*), showing the remarkably small sclerotic ring in relation to the orbit size.

*Dinogorgon* (*Dracocephalus*) *scheepersi Watson & Romer, 1956*:58

*Clelandina scheepersi Sigogneau, 1970*:284

*Dinogorgon pricei Sigogneau, 1970*:296

*Rubidgea pricei Gebauer, 2007*:223

*Holotype*: RC 57, a poorly-preserved skull and lower jaws (Fig. 19) from Adendorp, Graaff-Reinet, South Africa.

    *Referred specimens*: BP/1/742 (Figs. 20 and 21; a partial skull and lower jaws from Milton, Murraysburg, South Africa; holotype of *Tigrisaurus pricei*); RC 102 (Fig. 22; a crushed complete skull and lower jaws from Zuurplaas, Graaff-Reinet, South Africa;

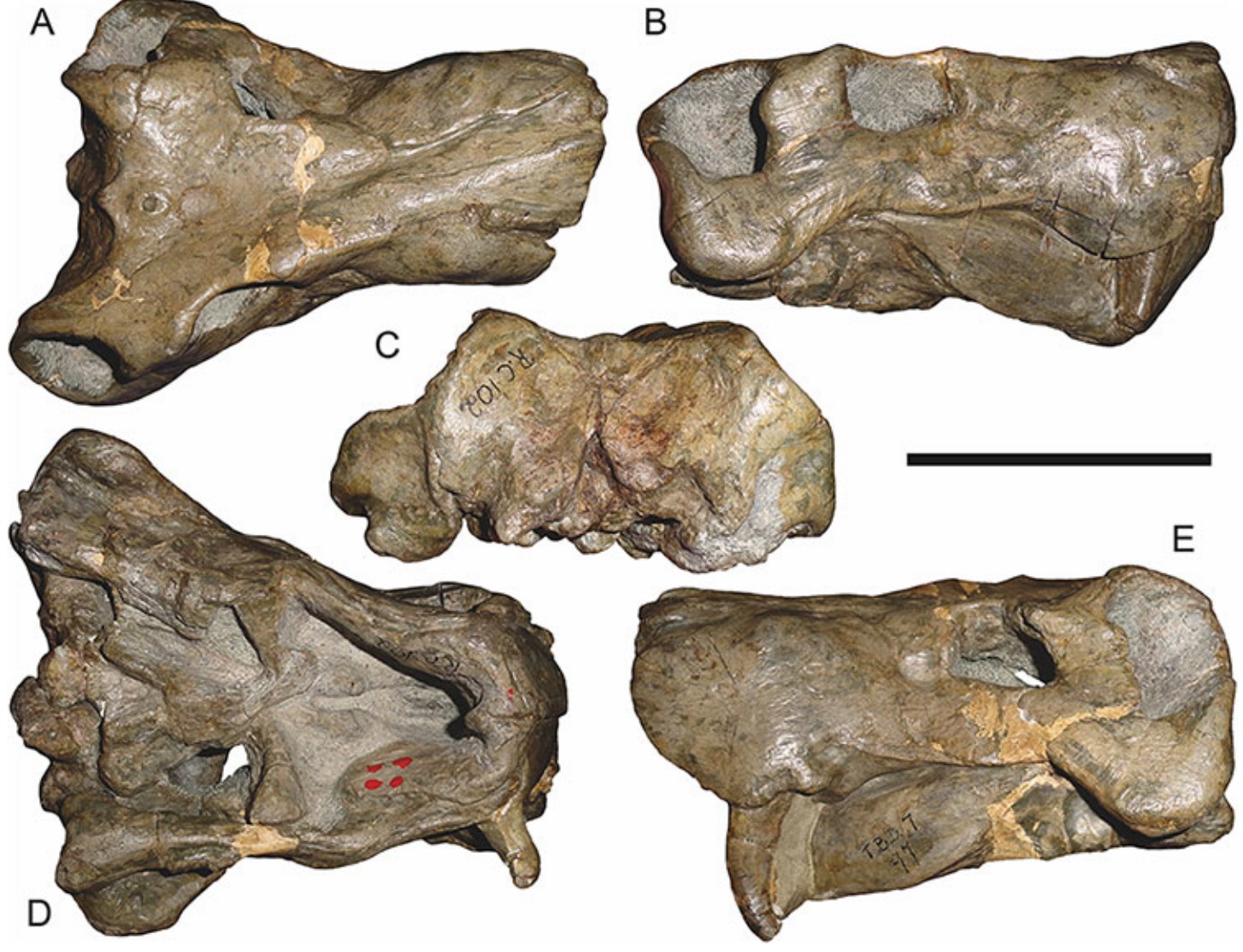

**Figure 22 Referred specimen (RC 102) of *Clelandina rubidgei* Broom, 1948 in (A) dorsal, (B) right lateral, (C) occipital, (D) ventral, and (E) left lateral view.** Holotype of *Dracocephalus scheepersi* Brink & Kitching, 1953. Scale bar equals 10 cm.

holotype of *Dracocephalus scheepersi*); UCMP 35437 (Fig. 23; a crushed complete skull from Waterval, Graaff-Reinet, South Africa).

*Diagnosis*: Large gorgonopsian (up to 36 cm basal skull length) distinguished from all other rubidgeines by the following autapomorphies: postcanine teeth completely absent, edentulous maxillary ridge present in the postcanine region, extremely reduced palatal dentition (1–2 teeth on palatine boss), and depression on skull roof between parietals and frontals.

*Comments*: *Clelandina* is one of the rarest rubidgeines, with only four skulls that can confidently be referred to this taxon. Like *Dinogorgon* and *Rubidgea*, discoveries of this taxon in South Africa have been limited to the region in and around Graaff-Reinet. *Clelandina* is unique among gorgonopsians in its complete lack of postcanine teeth.

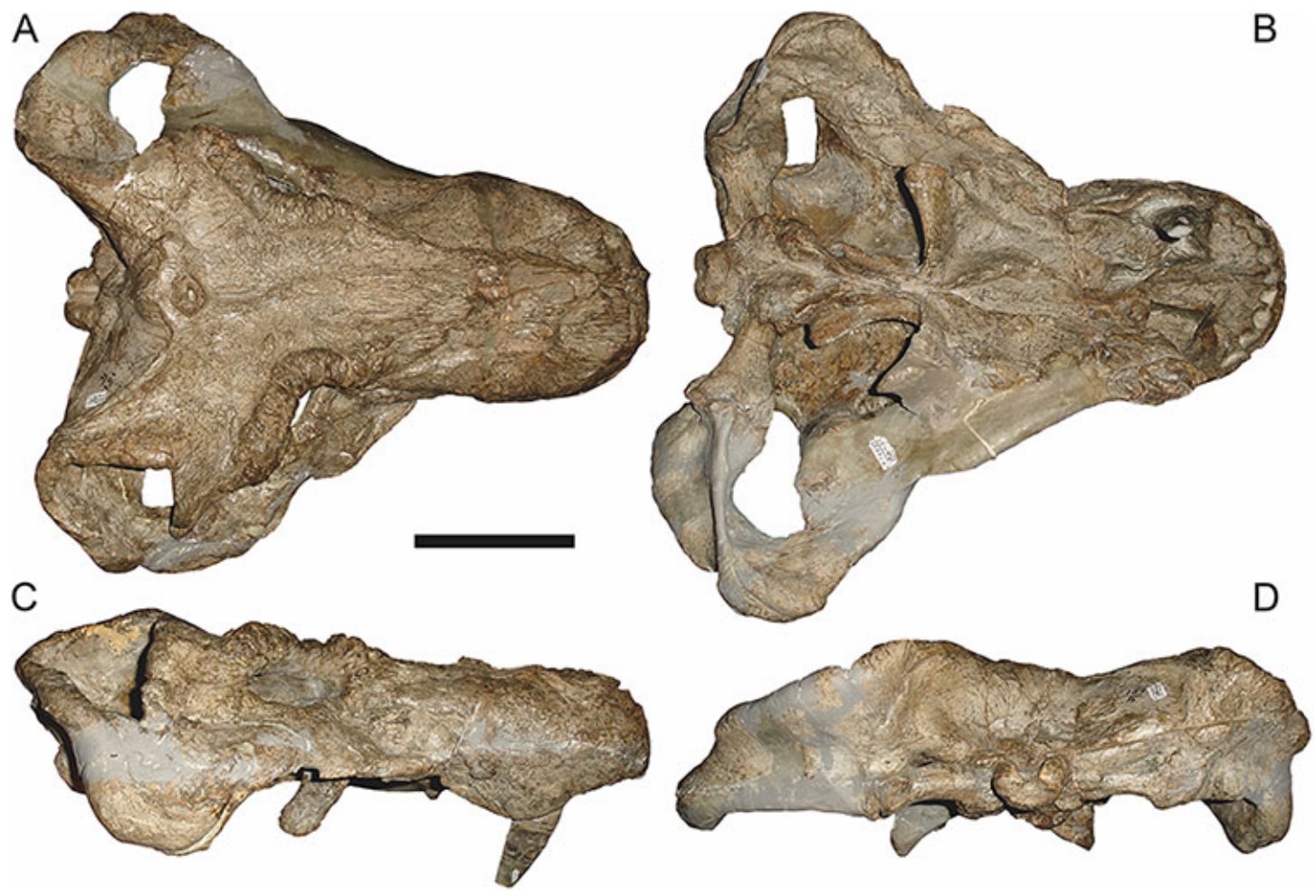

**Figure 23 Referred specimen (UCMP 35437) of *Clelandina rubidgei* *Broom, 1948* in (A) dorsal, (B) ventral, (C) right lateral, and (D) occipital view.** Scale bar equals 10 cm.

Postcanines are absent in the mandibles of *Leontosaurus*, *Rubidgea*, and *Inostrancevia*, but those genera retain at least a few maxillary postcanines. In *Clelandina*, a bony maxillary ridge occupies the edentulous postcanine margin of the maxilla (Fig. 17B), as in the coeval whaitsiid therocephalian *Theriognathus microps* (*Brink, 1980*). The convergent appearance of this feature in these distantly-related theriodonts is remarkable, especially considering the otherwise disparate morphology of their skulls (narrow, tapering snout and hugely enlarged temporal fenestrae in *Theriognathus* versus robust snout housing massive canine and short, pachyostosed temporal region in *Clelandina*). The functional significance of the replacement of postcanines with a bony ridge remains obscure; see the Discussion for further commentary on rubidgeine feeding habits.

The holotype of *Clelandina rubidgei* (RC 57) is a small (~19.0 cm basal skull length), very poor skull with attached lower jaws (Fig. 19). Most of the cranium is reconstructed with plaster, but part of the left side of the skull (including the postorbital bar and zygomatic arch) is intact, and the palate and occiput are preserved but poorly prepared. Despite its incompleteness, this specimen clearly shows that no postcanines are present in

the maxilla; instead, there is the characteristic labial emargination and maxillary ridge that allow *Clelandina* to be diagnosed (Figs. 19E and 19F). Additionally, this specimen shows proportions typical of an advanced rubidgeine, including a proportionally short parasphenoid rostrum.

BP/1/742, the holotype of *Tigrisaurus pricei Broom & George, 1950*, is missing part of the temporal region but is otherwise the best-preserved skull of *Clelandina* (Fig. 20). *Sigogneau (1970)* and *Gebauer (2007)* referred *T. pricei* to *Dinogorgon* and *Rubidgea* (respectively), but the complete absence of postcanines indicates that this species should instead be synonymized with *C. rubidgei*. Unlike in RC 57 and RC 102, the bone surface of BP/1/742 is relatively well preserved, showing extensive sculpturing across the dorsal skull roof (Fig. 20A) and zygomatic arch (Fig. 20B). This sculpturing is particularly well-developed on the supraorbital boss, taking the form of jagged furrows and rugosities. Unlike the other three known *Clelandina* specimens, the skull of BP/1/742 is largely undistorted, and can be taken as representative of the general skull shape for the taxon (Figs. 17 and 18). BP/1/742 preserves a remarkably small sclerotic ring (Fig. 21) in proportion to its orbit size (orbital dimensions: 6.1 × 8.7 cm; sclerotic dimensions: 1.4 cm (internal ring), 2.9 cm (external ring)). As sclerotic rings are not known for other rubidgeine taxa, it is unclear whether these proportions are autapomorphic for *Clelandina*.

*Sigogneau (1970)* considered *Dracocephalus scheepersi Brink & Kitching, 1953* to be referable to *Clelandina*, albeit as a valid species. *Gebauer (2007)* also recognized *Clelandina scheepersi* as valid, distinguishing it from *C. rubidgei* based on the absence of the preparietal. However, no distinct preparietal is present in RC 57—only a depression where the preparietal would usually be located. This depression is present in the same position in all four specimens of *Clelandina*, and is characteristic of the taxon. RC 102, the holotype of *Dracocephalus scheepersi*, is a complete skull and mandible that has suffered some dorsoventral compaction and left-right shear (Fig. 22). This specimen was poorly prepared, and almost no original bone surface remains. As such, the smooth texture of the skull roof (Fig. 22A) should not be taken as natural. Given the small size (19.0 cm basal skull length) of this specimen and its lack of supraorbital bosses, it is possible that the rugosities present in BP/1/742 and UCMP 35437 had not yet developed in RC 102. However, until a better-preserved small *Clelandina* skull is found, ontogenetic variation in bone surface texture in this taxon should be considered uncertain.

UCMP 35437 is the largest specimen of *Clelandina* (36.5 cm basal skull length) and is the most heavily pachyostosed, with extremely baroque sculpturing on the supraorbital bosses (Fig. 23). The supraorbital bosses of UCMP 35437 are the largest, both absolutely and proportionally, of any *Clelandina* specimen, but remain restricted to the postfrontal and prefrontal bones. However, rugose bone texture is also present on the postorbital, frontal, and anteromedial portion of the prefrontal (anterior to the supraorbital boss). This specimen has suffered extensive dorsoventral compaction, so the skull is probably somewhat wider in dorsal view and narrower in lateral view than would have been the case in life. The postorbital bar in this specimen is significantly broader than in the other, smaller *Clelandina* specimens; it is proportionally equivalent in size to that of large *Rubidgea* and *Dinogorgon* specimens. The palate of this specimen is

well-exposed and very similar to that of *Rubidgea*: the palatine and pterygoid bosses are reduced to a single, narrow ridge on each side, with only two tiny (and probably functionally useless) palatine teeth remaining. According to Charles Camp's field notes (stored at the University of California, Berkeley), UCMP 35437 was collected a half mile north of the Waterval ranch house, 200' above the road where the road enters the waterfall gate.

Other than in lacking postcanines, *Clelandina* is very similar to *Rubidgea*, albeit somewhat smaller in maximum size. Although this size disjunct may give reason for suspicion, *Clelandina* is unlikely to represent the juvenile morphology of *Rubidgea*. The smallest known specimen of *Clelandina* (RC 102) already has adult snout proportions (although it has not yet developed supraorbital bosses), whereas specimens herein identified as juveniles of *Rubidgea atrox* (which are larger than RC 102) have proportionally shorter, taller snouts than adults (e.g., BP/1/3857, RC 101). The largest known specimen of *Clelandina* (UCMP 35437) is very heavily pachyostosed, with intense bone surface rugosity, as is also the case in only the largest known specimens of *Rubidgea* (e.g., BP/1/699, RC 13) and *Dinogorgon* (GPIT K16). This suggests that UCMP 35437 represents a mature adult, despite its smaller size than presumed adults of other rubidgeine taxa. Sexual dimorphism is also unlikely to explain the differences between *Clelandina* and *Rubidgea*, which share the same features typically invoked as sexually selected in therapsid fossils (i.e., cranial bosses and rugosities). Based on this information, it is most parsimonious to conclude that *Clelandina* and *Rubidgea* are closely-related but distinct co-occurring taxa.

### *Dinogorgon* Broom, 1936

*Type species*: *Dinogorgon rubidgei* Broom, 1936.

  *Diagnosis*: As for the type and only recognized species.

### *Dinogorgon rubidgei* Broom, 1936
### (Reconstruction Figs. 24–25, Specimen Figs. 26–31)

*Dinogorgon quinquemolaris* Huene, 1950:81

*Dinogorgon oudebergensis* Brink & Kitching, 1953:6

*Prorubidgea robusta* Brink & Kitching, 1953:14

*Rubidgea quinquemolaris* Gebauer, 2007:222

*Holotype*: RC 1, a partial skull (complete from the orbits forward) and lower jaws (Fig. 26) from Wellwood, Graaff-Reinet, South Africa.

  *Referred specimens*: Bremner Collection unnumbered specimen (Fig. 27; snout and lower jaw collected at 3100′ in De Vrede, Graaff-Reinet, South Africa); BP/1/2167 (a partial snout and lower jaw from Ferndale, Graaff-Reinet, South Africa); BP/1/2190 (Figs. 28 and 29; a complete skull, lower jaws, and partial forelimb from Poortjie, Graaff-Reinet, South Africa; holotype of *Prorubidgea robusta*); BP/1/5322 (a weathered partial skull, missing the snout, from Dalham, Graaff-Reinet, South Africa); GPIT K16 (Fig. 30; a nearly complete skull and lower jaws from Kingori, Ruhuhu Basin, Tanzania;

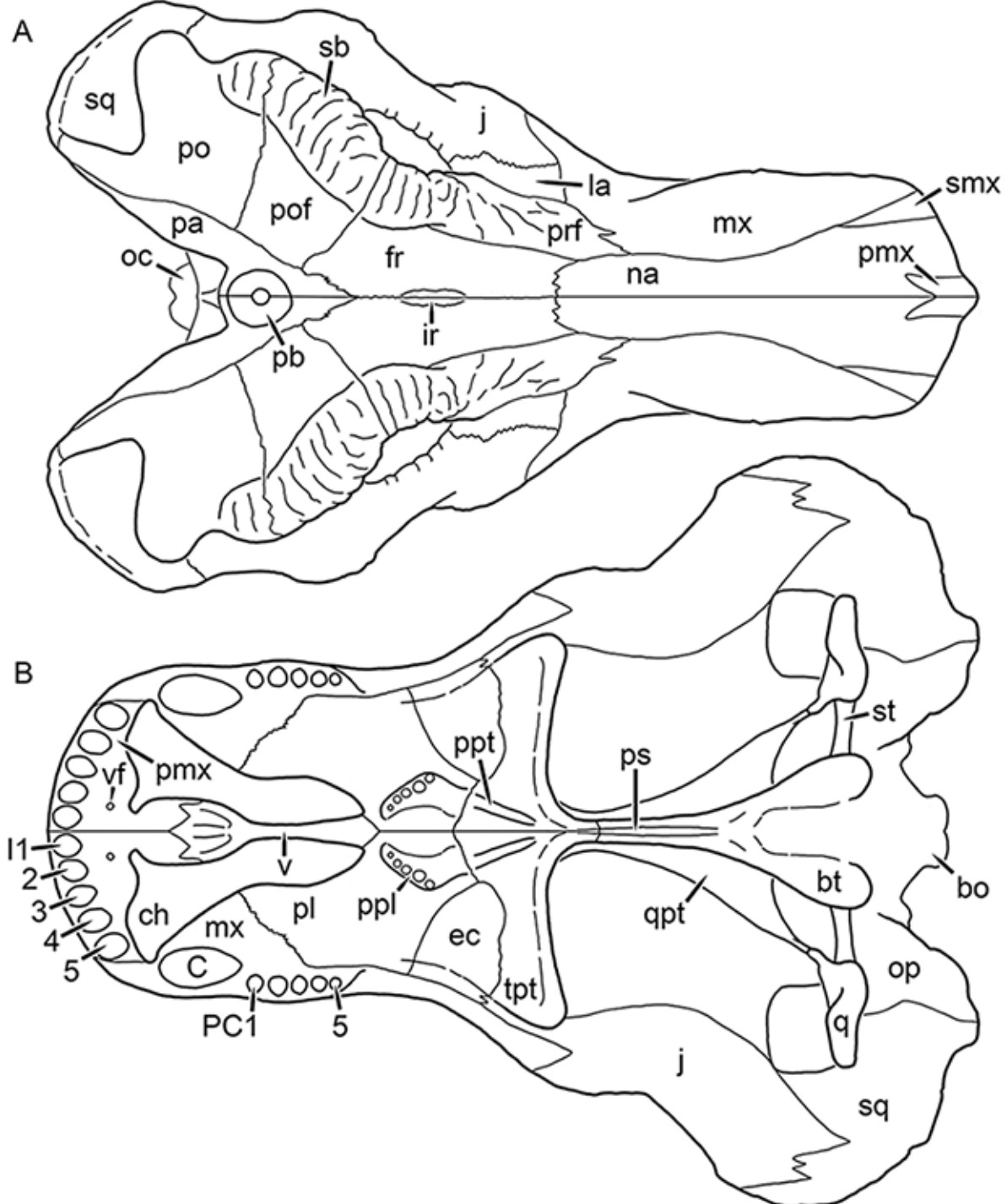

**Figure 24 Reconstruction of the skull of *Dinogorgon rubidgei Broom, 1936* in (A) dorsal and (B) ventral views.** Reconstructions based primarily on GPIT K16 and RC 103. Abbreviations: bo, basioccipital; bt, basal tuber; C, upper canine; ch, choana; ec, ectopterygoid; fr, frontal; I, upper incisor; ir, interorbital ridge; j, jugal; la, lacrimal; mx, maxilla; na, nasal; oc, occipital condyle; op, opisthotic; pa, parietal; pb, pineal boss; PC, upper postcanine; pl, palatine; pmx, premaxilla; po, postorbital; pof, postfrontal; ppl, palatal boss of palatine; ppt, palatal boss of pterygoid; prf, prefrontal; ps, parasphenoid; q, quadrate; qpt, quadrate ramus of pterygoid; sb, supraorbital boss; smx, septomaxilla; sq, squamosal; st, stapes; tpt, transverse process of pterygoid; v, vomer; vf, ventral premaxillary foramen.

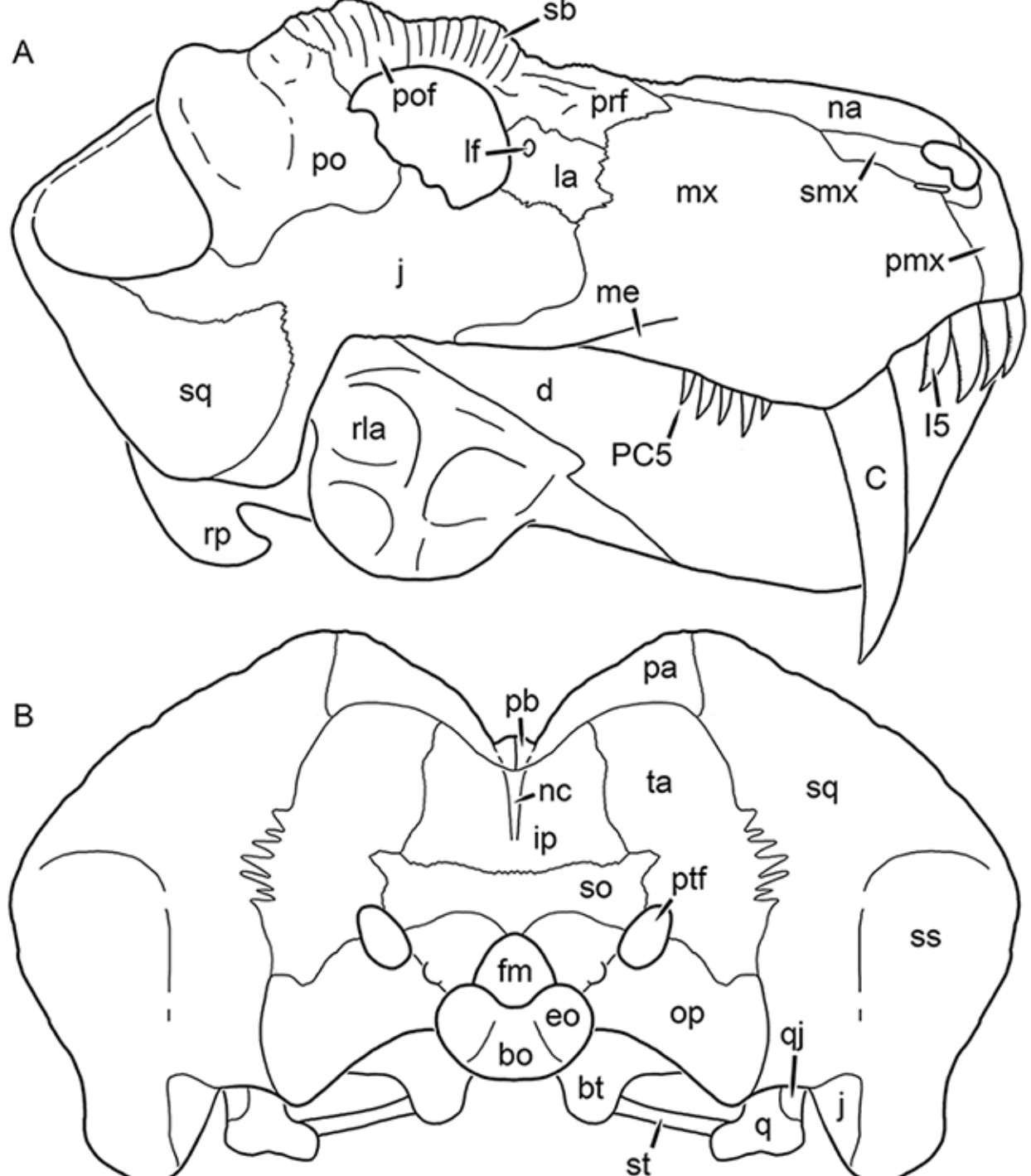

**Figure 25 Reconstruction of the skull of *Dinogorgon rubidgei Broom, 1936* in (A) lateral and (B) occipital views.** Reconstructions based primarily on GPIT K16 and RC 103. Abbreviations: bo, basioccipital; bt, basal tuber; C, upper canine; d, dentary; eo, exoccipital; fm, foramen magnum; I, upper incisor; ip, interparietal; j, jugal; la, lacrimal; lf, lacrimal foramen; me, maxillary emargination; mx, maxilla; na, nasal; nc, nuchal crest; op, opisthotic; pa, parietal; pb, pineal boss; PC, upper postcanine; pmx, premaxilla; po, postorbital; pof, postfrontal; prf, prefrontal; ptf, post-temporal fenestra; q, quadrate; qj, quadratojugal; rla, reflected lamina of angular; rp, retroarticular process; sb, suprorbital boss; smx, septomaxilla; so, supraoccipital; sq, squamosal; ss, squamosal sulcus; st, stapes; ta, tabular.

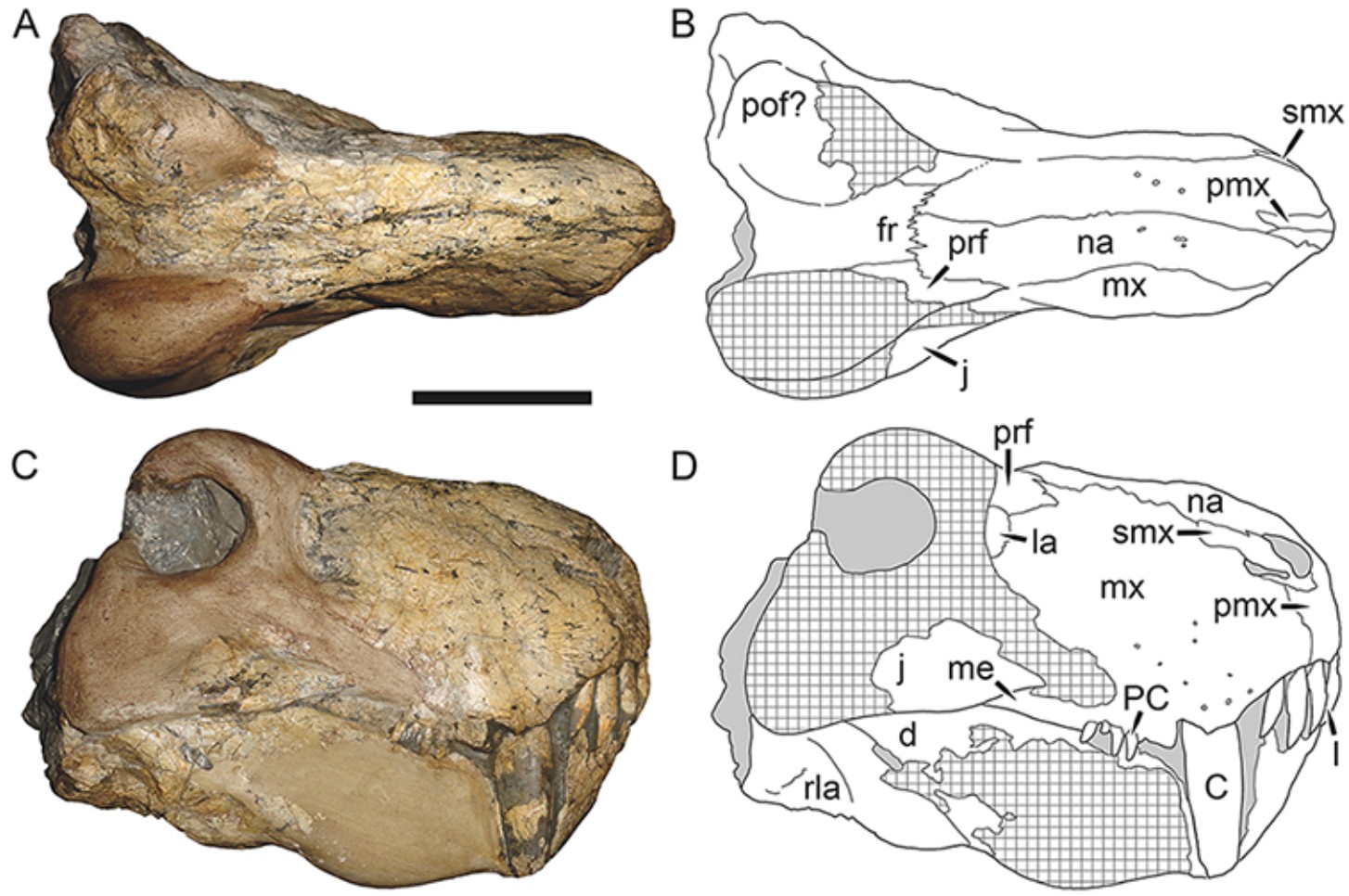

**Figure 26 Holotype (RC 1) of *Dinogorgon rubidgei Broom, 1936* in (A) dorsal and (C) right lateral view (with (B) and (D) interpretive drawings).** Abbreviations: C, upper canine; d, dentary; fr, frontal; I, upper incisor; j, jugal; la, lacrimal; me, maxillary emargination; mx, maxilla; na, nasal; PC, upper postcanine; pmx, premaxilla; pof, postfrontal; prf, prefrontal; rla, reflected lamina of angular; smx, septomaxilla. Gray indicates matrix, hatching indicates plaster. Scale bar equals 10 cm.

holotype of *Dinogorgon quinquemolaris*); GPIT/RE/7137 (specimen K12 of *Huene (1950)*, an isolated snout and lower jaw from Kingori, Ruhuhu Basin, Tanzania); RC 103 (Fig. 31; a nearly complete skull and lower jaws from the Oudeberg Plateau, Graaff-Reinet, South Africa; holotype of *Dinogorgon oudebergensis*); UMZC T880 (isolated snout and lower jaw from Stockley's Site B4/7, Katumbi Viwili, Ruhuhu Basin, Tanzania); UMZC T890 (isolated snout and lower jaw from Stockley's Site B19, Kingori, Ruhuhu Basin, Tanzania).

*Diagnosis*: Large gorgonopsian (up to ~40 cm basal skull length) diagnosed by the combination of massive, rugose supraorbital bosses (shared with *Clelandina* and *Rubidgea*), strongly convex canine margin of the maxilla (shared with *Clelandina* and *Rubidgea*), 4–5 upper and lower postcanine teeth, and a tall, transversely narrow snout (similar to that of *Aelurognathus*, narrower than *Clelandina* and *Rubidgea*).

*Comments*: Previous workers have recognized a close relationship between the genera *Dinogorgon Broom, 1936* and *Rubidgea Broom, 1938*. *Sigogneau (1970)* suggested that they could be synonymous, but refrained from formalizing this, so as to maintain the use of the

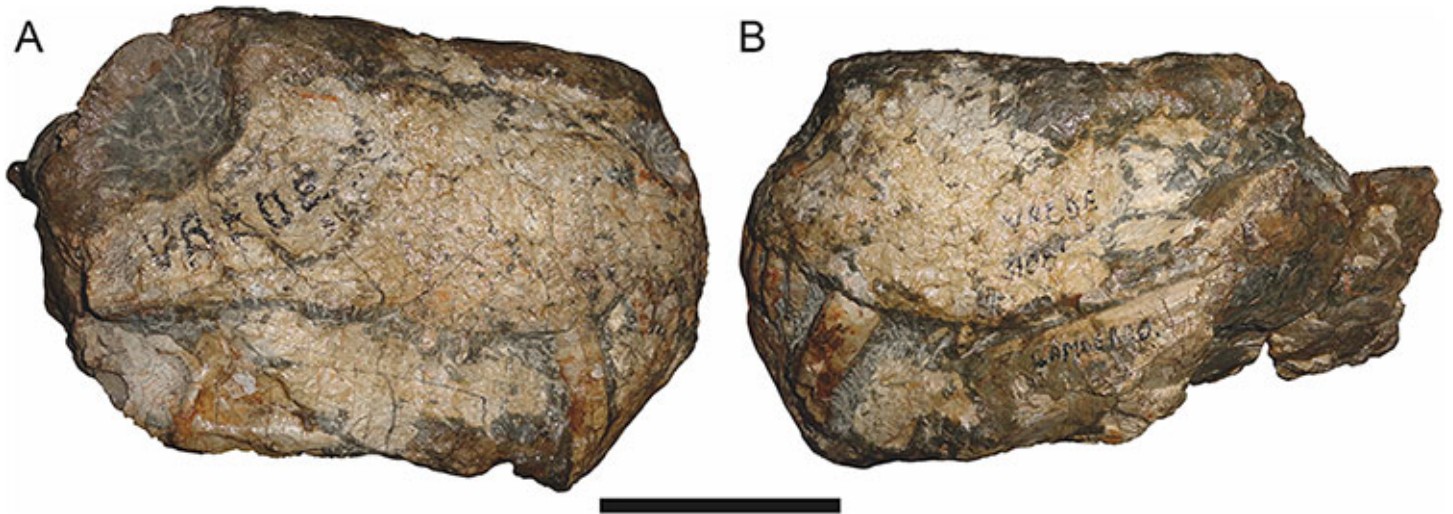

**Figure 27 Referred specimen (Bremner Collection, unnumbered) of *Dinogorgon rubidgei Broom, 1936* in (A) right lateral and (B) left lateral views.** Scale bar equals 10 cm.

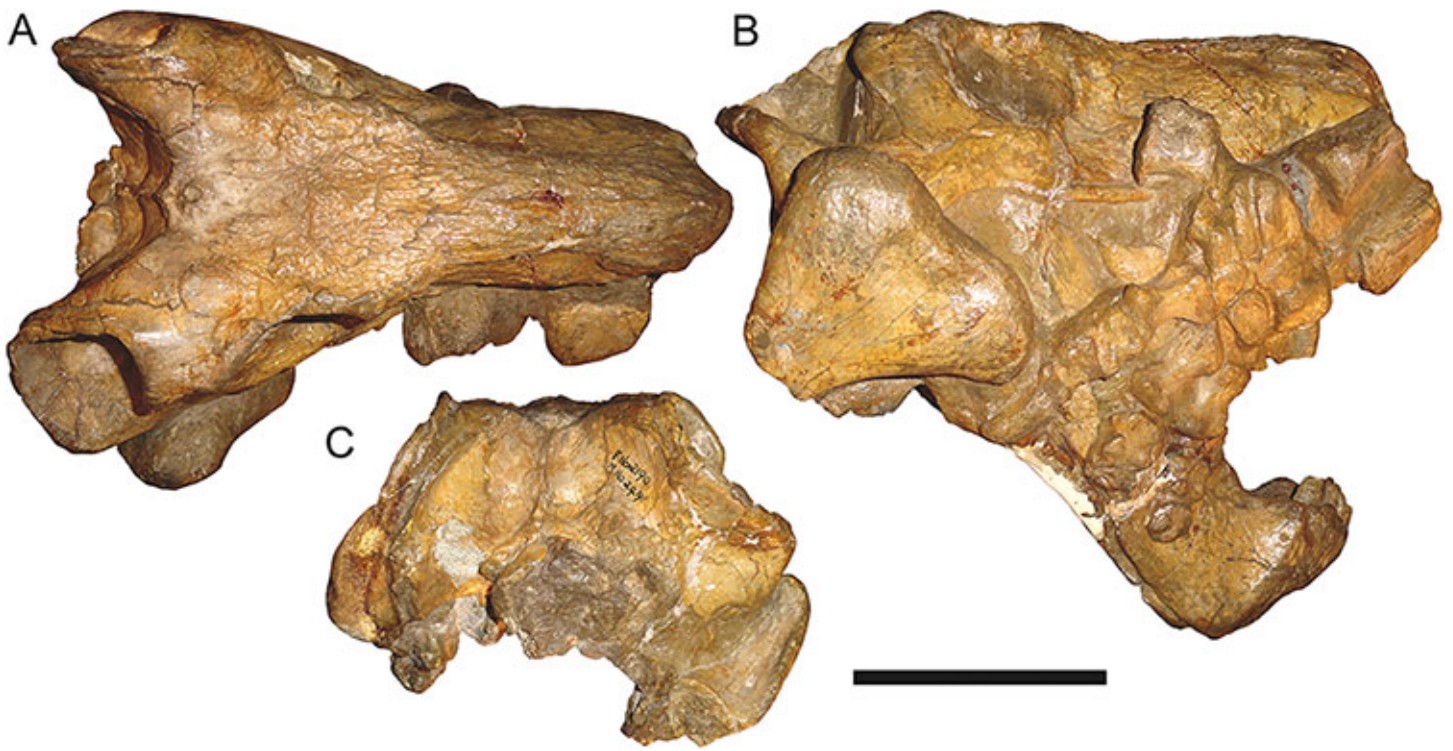

**Figure 28 Referred specimen (BP/1/2190) of *Dinogorgon rubidgei Broom, 1936* in (A) dorsal, (B) right lateral, and (C) occipital views.** Holotype of *Prorubidgea robusta Brink & Kitching, 1953*. Scale bar equals 10 cm.

better-known but junior genus *Rubidgea*. *Sigogneau-Russell (1989)* retained *Dinogorgon* as a valid genus, and considered it to be a morphological intermediate between *Prorubidgea* and *Rubidgea*. *Gebauer (2007)* considered the type species of *Dinogorgon*, *D. rubidgei*, to be indeterminate, and transferred the remaining species (*D. quinquemolaris*)

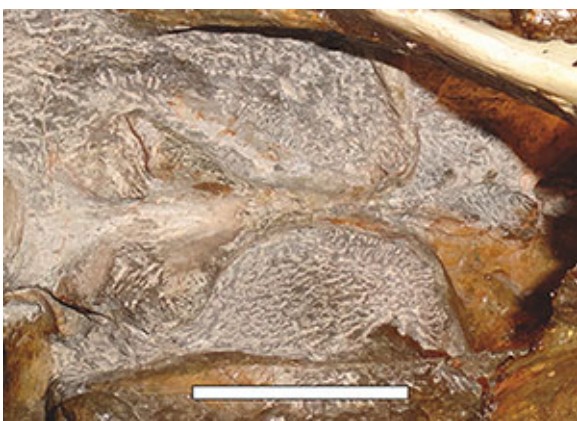

**Figure 29 Close-up of the basicranium of BP/1/2190 (*Dinogorgon rubidgei*), showing the extremely short parasphenoid rostrum.** Scale bar equals 5 cm.

A    B

C    D

**Figure 30 Referred specimen (GPIT K16) of *Dinogorgon rubidgei Broom, 1936* in (A) dorsal, (B) ventral, (C) right lateral, and (D) left lateral views.** Holotype of *Dinogorgon quinquemolaris Huene, 1950*. Scale bar equals 10 cm.

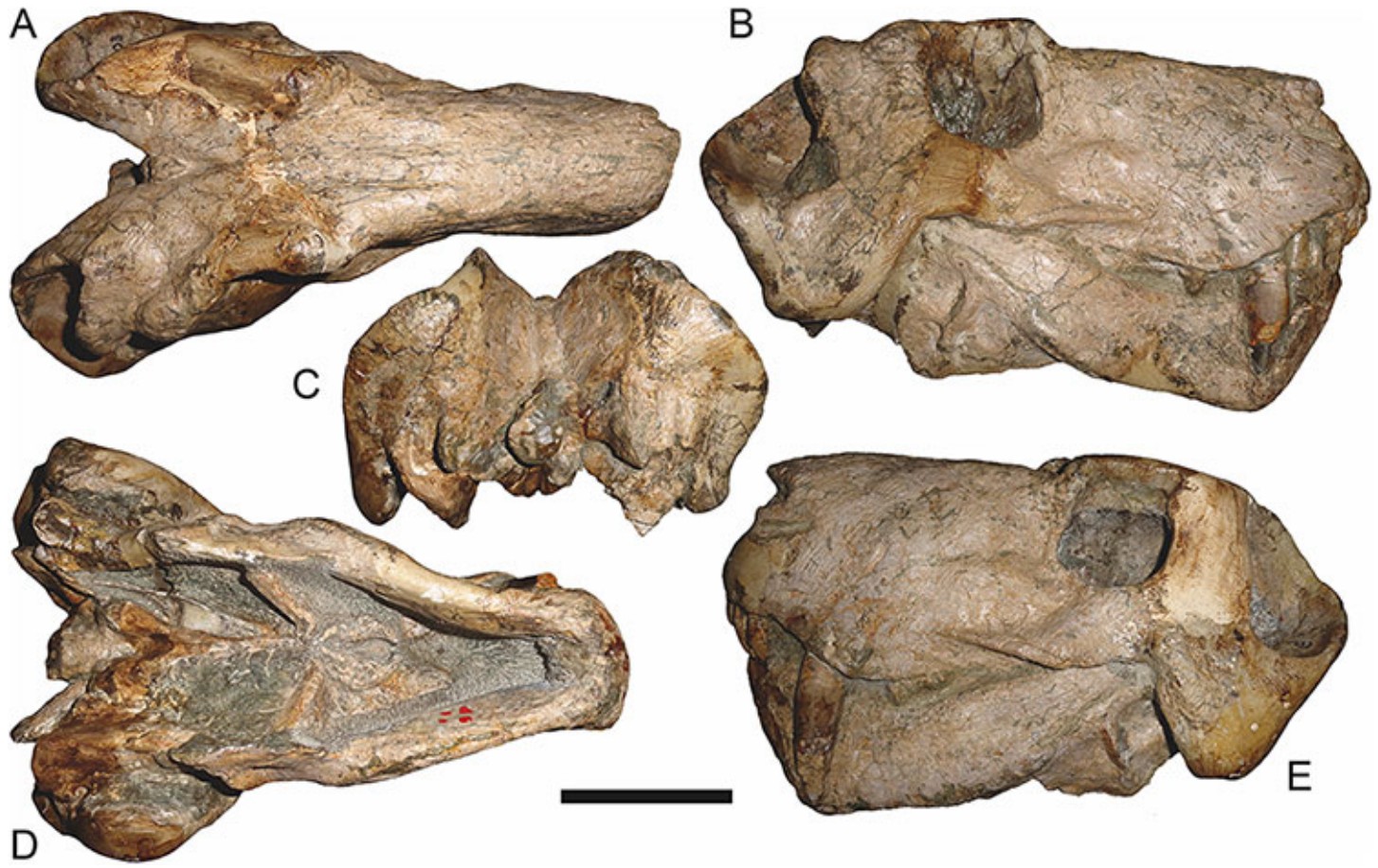

**Figure 31 Referred specimen (RC 103) of *Dinogorgon rubidgei Broom, 1936* in (A) dorsal, (B) right lateral, (C) occipital, (D) ventral, and (E) left lateral views.** Holotype of *Dinogorgon oudebergensis Brink & Kitching, 1953*. Scale bar equals 10 cm.

to *Rubidgea* (problematically, however, she still recognized RC 1, the holotype of *D. rubidgei*, as being referable to '*Rubidgea* sp.').

RC 1 consists of a large snout (22.4 cm long, height above canine 14.9 cm on right side) with partial orbital region and lower jaw (Fig. 26). At least four postcanines are present on the left side of the skull (but the maxilla is broken off posterior to PC4), and four are present on the right (PC2 and PC4 with crowns, PC1 and PC3 indicated by roots). The jaw in this specimen is short and massive (~10 cm tall at symphysis) and the left orbital region preserves a large supraorbital boss. Contra *Gebauer (2007)*, RC 1 is not indeterminate. Although poorly preserved, this specimen can clearly be distinguished from other rubidgeines: the relatively tall, transversely narrow snout and high postcanine count differentiates it from *Rubidgea* and *Clelandina* and the massively pachyostosed supraorbital region differentiates it from the remaining rubidgeine genera. This combination of features is also known in *Dinogorgon quinquemolaris Huene, 1950* and *Dinogorgon oudebergensis Brink & Kitching, 1953*, which are here considered synonyms of *D. rubidgei*.

*Huene (1950)* described *Dinogorgon quinquemolaris* on the basis of a nearly complete (missing the left temporal arcade) skull and lower jaws (Fig. 30) from Kingori, in the Usili

Formation of Tanzania. Both *Sigogneau-Russell (1989)* and *Gebauer (2007)* considered *D. quinquemolaris* to be a valid species, although the former suggested that it may eventually prove to be conspecific with *D. rubidgei*. The holotype of *D. quinquemolaris* (GPIT K16) is the best-preserved specimen of the genus, clearly showing the highly rugose bone surface of the skull roof also seen in *Rubidgea* and *Clelandina* (the bone surface is overprepared in both RC 1 and RC 103). In addition to GPIT K16, three additional specimens (all of which are isolated snouts and anterior lower jaws) from the Usili Formation can be referred to *Dinogorgon*: GPIT/RE/7137, UMZC T880, and UMCZ T890. These specimens all represent large gorgonopsians with massive dentary symphyses, well-developed maxillary emarginations, and five postcanines.

The holotype of *Dinogorgon oudebergensis Brink & Kitching, 1953* (RC 103) is a large (29.8 cm basal length) nearly complete skull (Fig. 31). Five postcanines are present on the left side of the skull, and partial crowns of PC1 and PC2, space for two further teeth, and the root of a presumed PC5 are present on the right. *Sigogneau (1970)* synonymized this species with *D. quinquemolaris*, a position followed by *Gebauer (2007)*. I agree that *D. quinquemolaris* and *D. oudebergensis* are best treated as conspecific (albeit going further and synonymizing them with the type species *D. rubidgei*). The only differences between these skulls are the degree of cranial rugosity (which, as noted above, can be attributed to overpreparation of RC 103) and the presence of postorbital bosses in *D. oudebergensis*. However, the absence of this boss in GPIT K16 (compare Figs. 30C with 31B) is probably artifactual: examination of this specimen has revealed that this region is broken off on the specimen and has been restored with plaster. Although it is not certain whether a postorbital boss was present, its existence is strongly indicated by the massive expansion of the posterior edge of the preserved portion of the postorbital bar in this specimen.

One additional species, not previously associated with *Dinogorgon*, can be referred to *D. rubidgei*. *Brink & Kitching (1953)* described *Prorubidgea robusta* on the basis of a complete, well-preserved skull and lower jaws and partial forelimb (BP/1/2190; Fig. 28). *Sigogneau (1970)* retained *P. robusta* as a valid species of *Prorubidgea*, but *Gebauer (2007)* synonymized it with *Aelurognathus* (originally *Sycosaurus*) *brodiei*. As discussed in the section on *Aelurognathus* above, the holotype of *S. brodiei* (TM 1493) requires further preparation, but appears referable to *Aelurognathus tigriceps*. BP/1/2190, by contrast, exhibits a series of characters at odds with identification as *Aelurognathus* but in accordance with advanced rubidgeines such as *Dinogorgon*. The postorbital bar is broadest dorsally, with swollen, rugose anterior and posterior margins with a well-developed fossa on the dorsolateral surface between them. The intertemporal region is proportionally broader than in any known *Aelurognathus* specimen and the basal tubera are elongate, with an extremely short parasphenoid (Fig. 29). A ridge is present on the zygoma and the lacrimal foramen exits onto the facial surface of the lacrimal. Finally, BP/1/2190 has considerably greater bone rugosity on the skull roof than even well-prepared specimens of *Aelurognathus*. Although this skull lacks massive supraorbital bosses, this may be attributable to its small size (24.2 cm dorsal skull length) compared with other *Dinogorgon* specimens: based on the growth series known for *Clelandina* and *Rubidgea*, the

development of these bosses occurs relatively late in ontogeny, being present only in mature adults. (An unnumbered specimen from the Bremner Collection (Fig. 27) of similar size closely matches BP/1/2190 in overall morphology, so it is probable that this is typical for subadult *Dinogorgon*). Among advanced rubidgeines, BP/1/2190 can be referred to *Dinogorgon rubidgei* by its high tooth count (five upper and four lower postcanines), tall and transversely narrow snout, and palatine boss with at least five teeth in a single row (distinguishing it from *Clelandina* and *Rubidgea*, albeit not *Leontosaurus*).

### Leontosaurus Broom & George, 1950

*Type species*: *Leontosaurus vanderhorsti* Broom & George, 1950.

*Diagnosis*: As for the type and only recognized species.

### Leontosaurus vanderhorsti Broom & George, 1950
### (Reconstruction Figs. 32–33, Specimen Figs. 3, 34–37)

*Rubidgea platyrhina* Brink & Kitching, 1953:12

*Sycosaurus vanderhorsti* Sigogneau, 1970:258

*Holotype*: BP/1/743, a dorsoventrally crushed skull and lower jaws (Figs. 34 and 35) from Swaelkrans, Murraysburg, South Africa.

*Referred specimens*: BP/1/803 (Fig. 36; complete skull and lower jaws from Swaelkrans, Murraysburg, South Africa; holotype of *Rubidgea platyrhina*); BP/1/3853 (complete skull and lower jaws, ventral surface unprepared, with associated vertebrae from Katbosch, Graaff-Reinet, South Africa); CGS AF 19-83 (a complete skull from Bloemhof, Richmond, South Africa); UCMP 42750 (Fig. 37; crushed skull missing the right temporal arch from Swaelkrans, Murraysburg, South Africa).

*Diagnosis*: Large gorgonopsian (up to ~40 cm basal skull length) distinguished from all other rubidgeines by the following autapomorphies: 'backswept' morphology of the transverse process of the pterygoid, with an anterior depression restricted to the pterygoid portion of the process, set off in slope from the ectopterygoid, and extremely tall maxilla, strongly constricting the nasals in dorsal view. Also diagnosed by the unique combination of an extremely expanded, deflected subtemporal bar with a lateral ridge (shared with *Clelandina*, *Dinogorgon*, and *Rubidgea*), numerous teeth in a single row on the palatine boss (present in most rubidgeines, but not *Clelandina* and *Rubidgea*), and a tab-like posterior portion of the postfrontal (shared with some *Sycosaurus nowaki* specimens).

*Comments*: *Leontosaurus* is perhaps the most problematic of the rubidgeine genera recognized herein, as it exhibits a perplexing mosaic of features seen in various other taxa (particularly *Rubidgea* and *Sycosaurus*). Sigogneau (1970) synonymized *Leontosaurus* with *Sycosaurus*, but retained the species *Sycosaurus vanderhorsti* as valid. Gebauer (2007) considered *S. vanderhorsti* to be synonymous with *Sycosaurus laticeps*. BP/1/743, the holotype of *L. vanderhorsti* (Fig. 34), is broadly similar to *Sycosaurus*. However, there are several peculiar features of this specimen which do not accord with *S. laticeps*: the zygoma is relatively tall beneath the postorbital bar (Fig. 34A), the maxillae are unusually tall, such that they sharply constrict the nasals (Fig. 34B), and the transverse processes of the

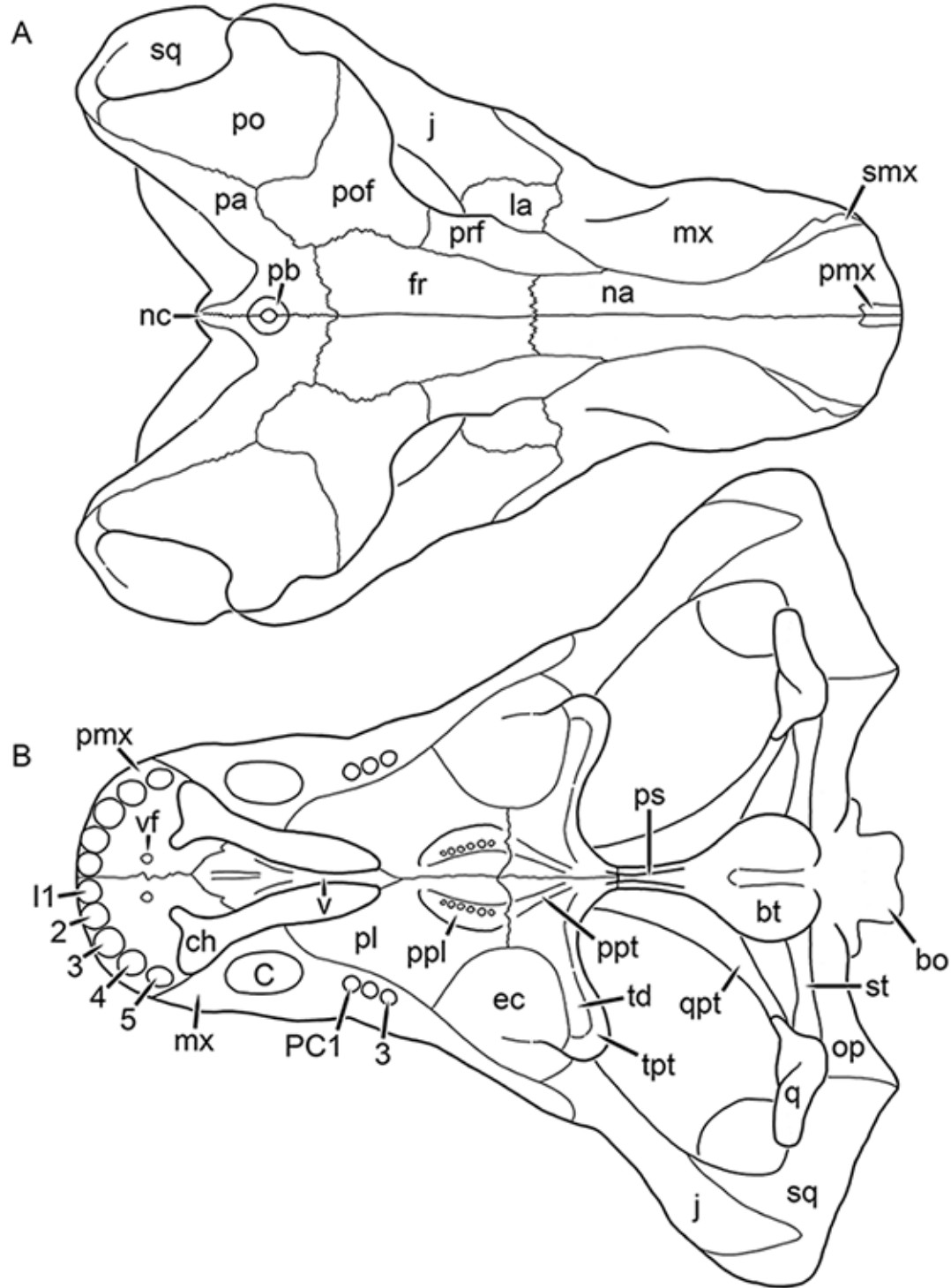

**Figure 32 Reconstruction of the skull of *Leontosaurus vanderhorsti* *Broom & George, 1950* in (A) dorsal and (B) ventral views.** Reconstructions based primarily on BP/1/743 and BP/1/803. Abbreviations: bo, basioccipital; bt, basal tuber; C, upper canine; ch, choana; ec, ectopterygoid; fr, frontal; I, upper incisor; j, jugal; la, lacrimal; mx, maxilla; na, nasal; nc, nuchal crest; op, opisthotic; pa, parietal; pb, pineal boss; PC, upper postcanine; pl, palatine; pmx, premaxilla; po, postorbital; pof, postfrontal; ppl, palatal boss of palatine; ppt, palatal boss of pterygoid; prf, prefrontal; ps, parasphenoid; q, quadrate; qpt, quadrate ramus of pterygoid; smx, septomaxilla; sq, squamosal; st, stapes; td, transverse depression on pterygoid; tpt, transverse process of pterygoid; v, vomer; vf, ventral premaxillary foramen.

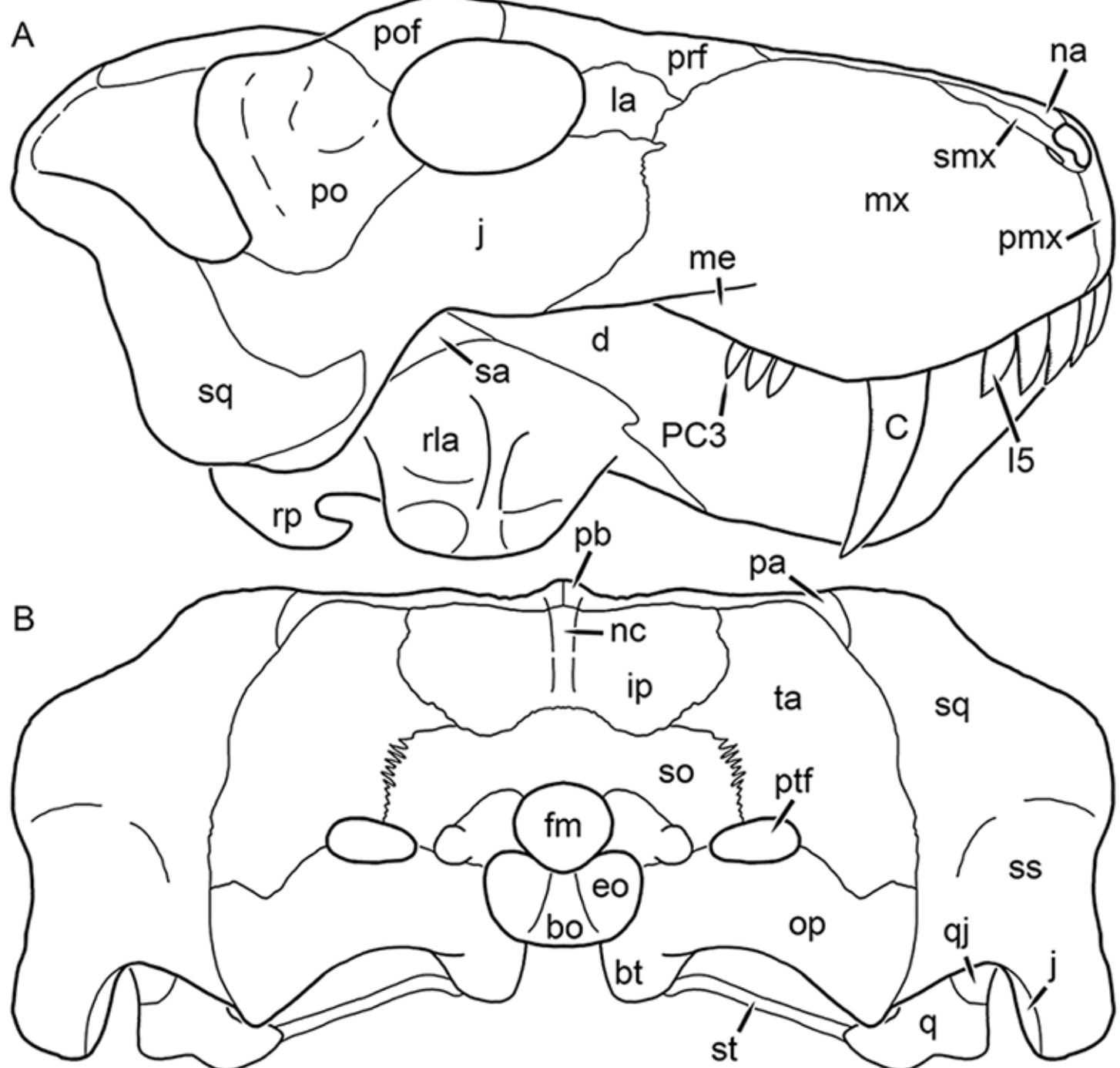

**Figure 33 Reconstruction of the skull of *Leontosaurus vanderhorsti* *Broom & George, 1950* in (A) lateral and (B) occipital views.** Reconstructions based primarily on BP/1/743 and BP/1/803. Abbreviations: bo, basioccipital; bt, basal tuber; C, upper canine; d, dentary; eo, exoccipital; fm, foramen magnum; I, upper incisor; ip, interparietal; j, jugal; la, lacrimal; me, maxillary emargination; mx, maxilla; na, nasal; nc, nuchal crest; op, opisthotic; pa, parietal; pb, pineal boss; PC, upper postcanine; pmx, premaxilla; po, postorbital; pof, postfrontal; prf, prefrontal; ptf, post-temporal fenestra; q, quadrate; qj, quadratojugal; rla, reflected lamina of angular; rp, retroarticular process; sa, surangular; smx, septomaxilla; so, supraoccipital; sq, squamosal; ss, squamosal sulcus; st, stapes; ta, tabular.

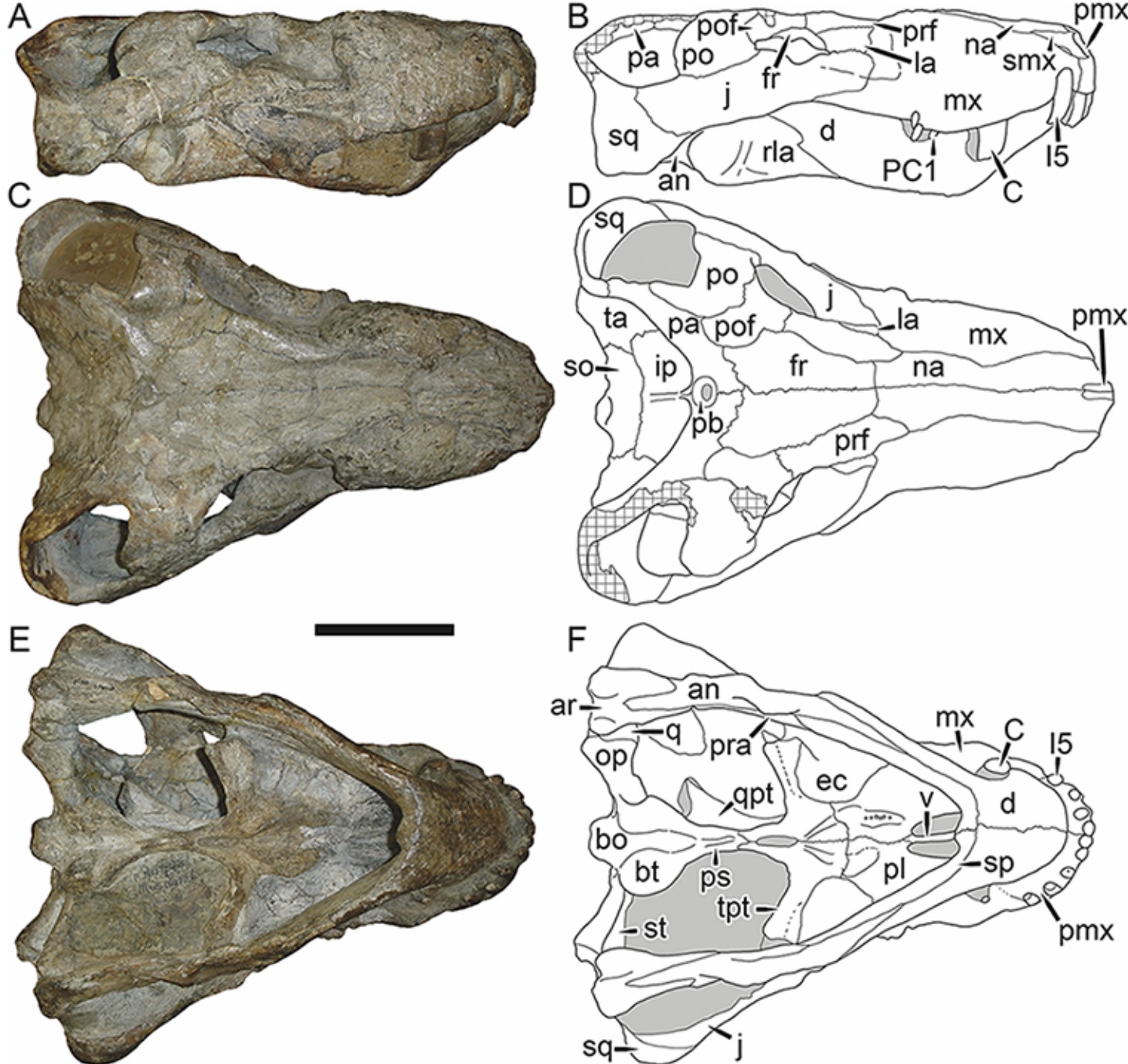

**Figure 34 Holotype (BP/1/743) of *Leontosaurus vanderhorsti* *Broom & George, 1950* in (A) right lateral, (C) dorsal, and (E) ventral view (with (B) (D) and (F) interpretive drawings).** Abbreviations: an, angular; ar, articular; bo, basioccipital; bt, basal tuber; C, upper canine; d, dentary; ec, ectopterygoid; fr, frontal; I, upper incisor; ip, interparietal; j, jugal; la, lacrimal; mx, maxilla; na, nasal; pa, parietal; pb, pineal boss; PC, upper postcanine; pl, palatine; pmx, premaxilla; po, postorbital; pof, postfrontal; pra, prearticular; prf, prefrontal; ps, parasphenoid; q, quadrate; qpt, quadrate ramus of pterygoid; rla, reflected lamina of angular; smx, septomaxilla; so, supraoccipital; sp, splenial; sq, squamosal; st, stapes; ta, tabular; tpt, transverse process of pterygoid; v, vomer. Gray indicates matrix, hatching indicates plaster. Scale bar equals 10 cm.

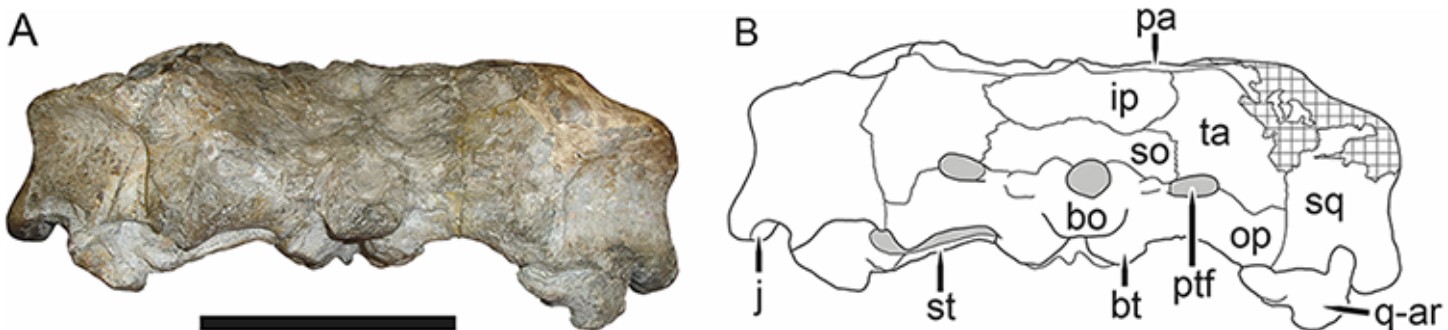

**Figure 35 Holotype (BP/1/743) of *Leontosaurus vanderhorsti* *Broom & George, 1950* in (A) occipital view (with (B) interpretive drawing).** Abbreviations: bo, basioccipital; bt, basal tuber; ip, interparietal; j, jugal; op, opisthotic; pa, parietal; ptf, post-temporal fenestra; q-ar, quadrate-articular complex; so, supraoccipital; sq, squamosal; st, stapes; ta, tabular. Gray indicates matrix, hatching indicates plaster. Scale bar equals 10 cm.

**Figure 36 Referred specimen (BP/1/803) of *Leontosaurus vanderhorsti* *Broom & George, 1950* in (A) dorsal, (B) ventral, (C) left lateral, and (D) right lateral view.** Holotype of *Rubidgea platyrhina* *Brink & Kitching, 1953*. Scale bar equals 10 cm.

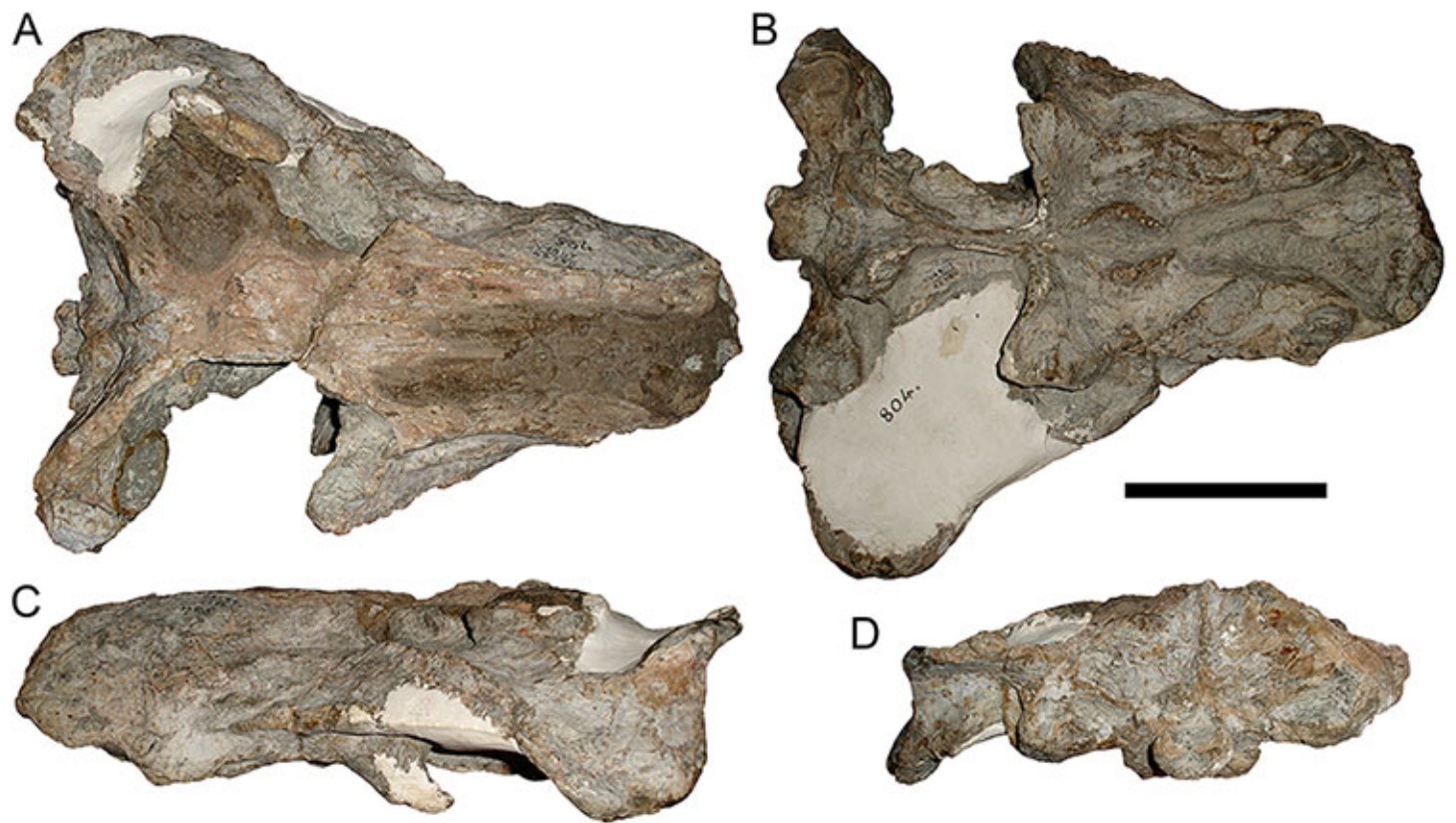

**Figure 37 Referred specimen (UCMP 42750) of *Leontosaurus vanderhorsti* *Broom & George, 1950* in (A)** dorsal, **(B)** ventral, **(C)** left lateral, **and (D)** occipital view. Scale bar equals 10 cm.

pterygoids are notably 'backswept' with a longitudinal depression on the anteroventral face of the process, before its suture with the ectopterygoid (Fig. 34E). Taken by themselves, each of these features could be explained away as individual variation or taphonomic distortion. However, this same suite of features is also found in two larger skulls from the same locality (Swaelkrans) as the holotype: BP/1/803 and UCMP 42750.

BP/1/803 is the holotype of *Rubidgea platyrhina* *Brink & Kitching, 1953*, and consists of a nearly complete skull and lower jaws (Fig. 36). *Sigogneau (1970)* retained *R. platyrhina* as a valid species, whereas *Gebauer (2007)* considered it synonymous with *Rubidgea atrox*. This specimen exhibits the extremely expanded zygoma, postorbital bar, and intertemporal region found in *Clelandina, Dinogorgon,* and *Rubidgea,* and of those genera most closely resembles *Rubidgea* in snout morphology. Strangely, however, it lacks well-developed bosses above the orbits (as in all of the aforementioned genera) or on the dentary (as is autapomorphic for *Rubidgea*). BP/1/803 is a large specimen (40.0 cm basal skull length), equal in size to the most heavily pachyostosed specimens of *Dinogorgon* and *Rubidgea*, so this distinction cannot be attributed to ontogeny. Also unlike *Rubidgea*, this specimen appears to have an extensive row of teeth on the palatine boss and three upper postcanines. The same palatine morphology is present in UCMP 42750, a slightly smaller (34.9 cm basal skull length) specimen that also features 'backswept' transverse processes with anterior depressions (Fig. 37). Finally, the posteromedial portion of the postfrontal is

distinctly 'tab'-like (with a rounded posterior portion separated from its supraorbital portion by a slight constriction) in both BP/1/743 and BP/1/803 (Figs. 34A and 36A). This morphology is also seen in some specimens of *Sycosaurus nowaki* (e.g., UMZC T878), but not in *Clelandina* or *Rubidgea*, where the postfrontal is rectangular.

It is possible that the absence of cranial bosses in these '*Leontosaurus* morph' specimens could be explained by sexual dimorphism, for example, if they were females of *Rubidgea atrox*. However, this cannot readily explain the differences in palatal morphology between them: in all other specimens of *Rubidgea*, the transverse processes are stout and straight, without anterior depressions. It is also possible that these specimens could represent adults of *Sycosaurus laticeps*, but both BP/1/803 and UCMP 42750 have the anteriorly bulbous interchoanal vomers typical of rubidgeines, instead of the gradually expanding vomers typical of *Sycosaurus*. Additional research on these specimens is required, but for now, I consider the most conservative approach to be recognizing *Leontosaurus vanderhorsti* as a valid taxon, related to but distinct from the *Clelandina-Dinogorgon-Rubidgea* group.

### *Rubidgea* Broom, 1938

*Broomicephalus* Brink & Kitching, 1953:3
*Titanogorgon* Maisch, 2002:248

*Type species*: *Rubidgea atrox* Broom, 1938.
  *Diagnosis*: As for the type and only recognized species.

### *Rubidgea atrox* Broom, 1938
### (Reconstruction Figs. 38–39, Specimen Figs. 40–48)

*Rubidgea kitchingi* Broom, 1938:529
*Rubidgea laticeps* Broom, 1940b:173
*Gorgonognathus maximus* Huene, 1950:86
*Broomicephalus laticeps* Brink & Kitching, 1953:3
*Rubidgea majora* Brink & Kitching, 1953:10
*Dinogorgon* (*Broomicephalus*) *laticeps* Watson & Romer, 1956:58
*Titanogorgon maximus* Maisch, 2002:248
*Clelandina laticeps* Gebauer, 2007:21

*Holotype*: RC 13, a complete skull and lower jaws (Figs. 40 and 41) from Dorsfontein, Graaff-Reinet, South Africa.
  *Referred specimens*: B 353 (complete skull from Doornplaas, Graaff-Reinet, South Africa); B 354 (Fig. 42A; laterally crushed skull from Vlakplaas, Graaff-Reinet, South Africa); BP/1/195 (Fig. 42C; partially restored skull and lower jaws from Hoeksplaas, Murraysburg, South Africa; referred to *Broomicephalus laticeps* by Brink & Kitching (1953), referred to *Rubidgea* cf. *platyrhina* by Sigogneau (1970)); BP/1/699

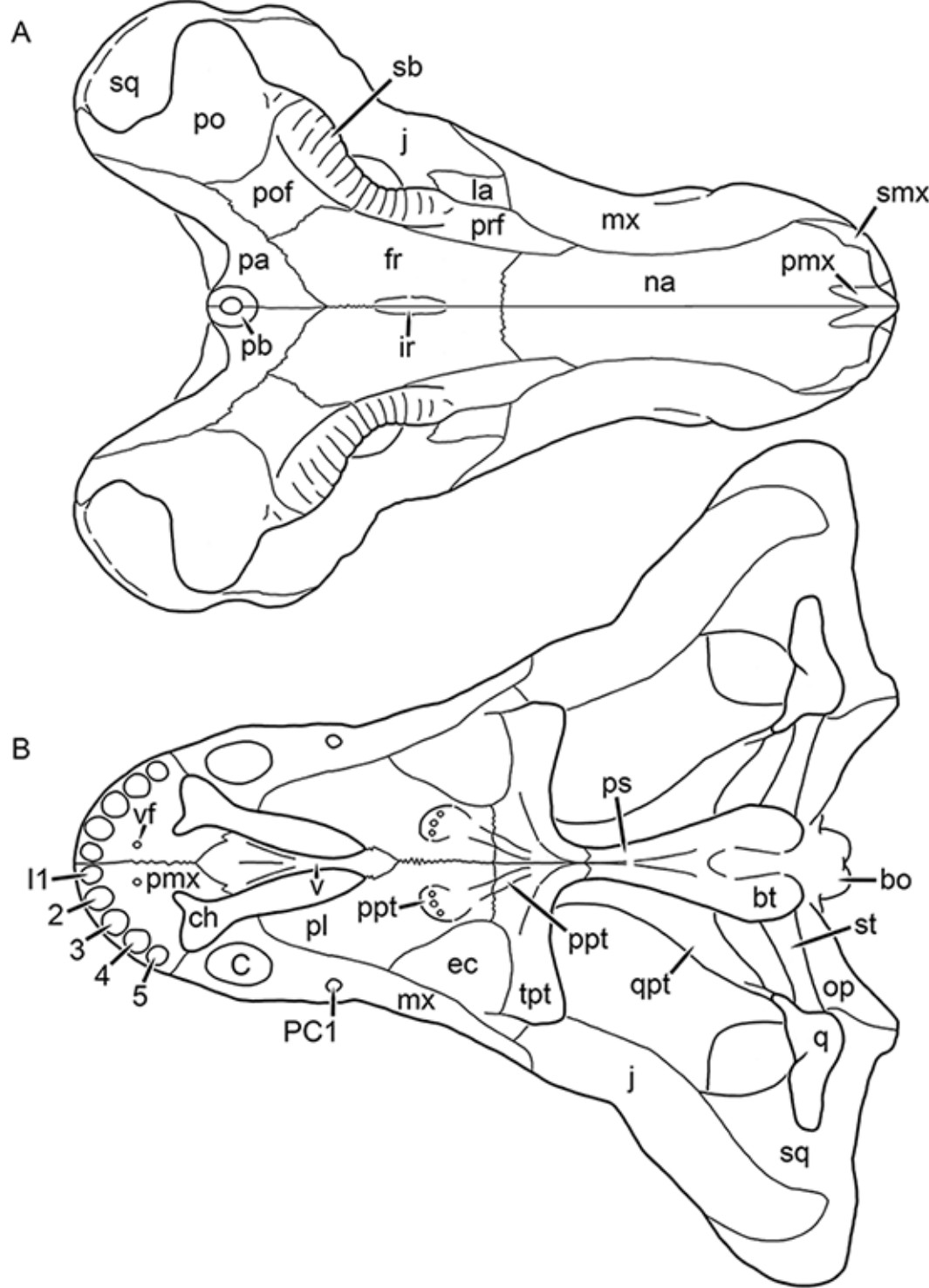

**Figure 38 Reconstruction of the skull of *Rubidgea atrox* *Broom, 1938* in (A) dorsal and (B) ventral views.** Reconstructions based primarily on RC 13 and RC 33. Abbreviations: bo, basioccipital; bt, basal tuber; C, upper canine; ch, choana; ec, ectopterygoid; fr, frontal; I, upper incisor; ir, interorbital ridge; j, jugal; la, lacrimal; mx, maxilla; na, nasal; op, opisthotic; pa, parietal; pb, pineal boss; PC, upper postcanine; pl, palatine; pmx, premaxilla; po, postorbital; pof, postfrontal; ppl, palatal boss of palatine; ppt, palatal boss of pterygoid; prf, prefrontal; ps, parasphenoid; q, quadrate; qpt, quadrate ramus of pterygoid; sb, supraorbital boss; smx, septomaxilla; sq, squamosal; st, stapes; tpt, transverse process of pterygoid; v, vomer; vf, ventral premaxillary foramen.

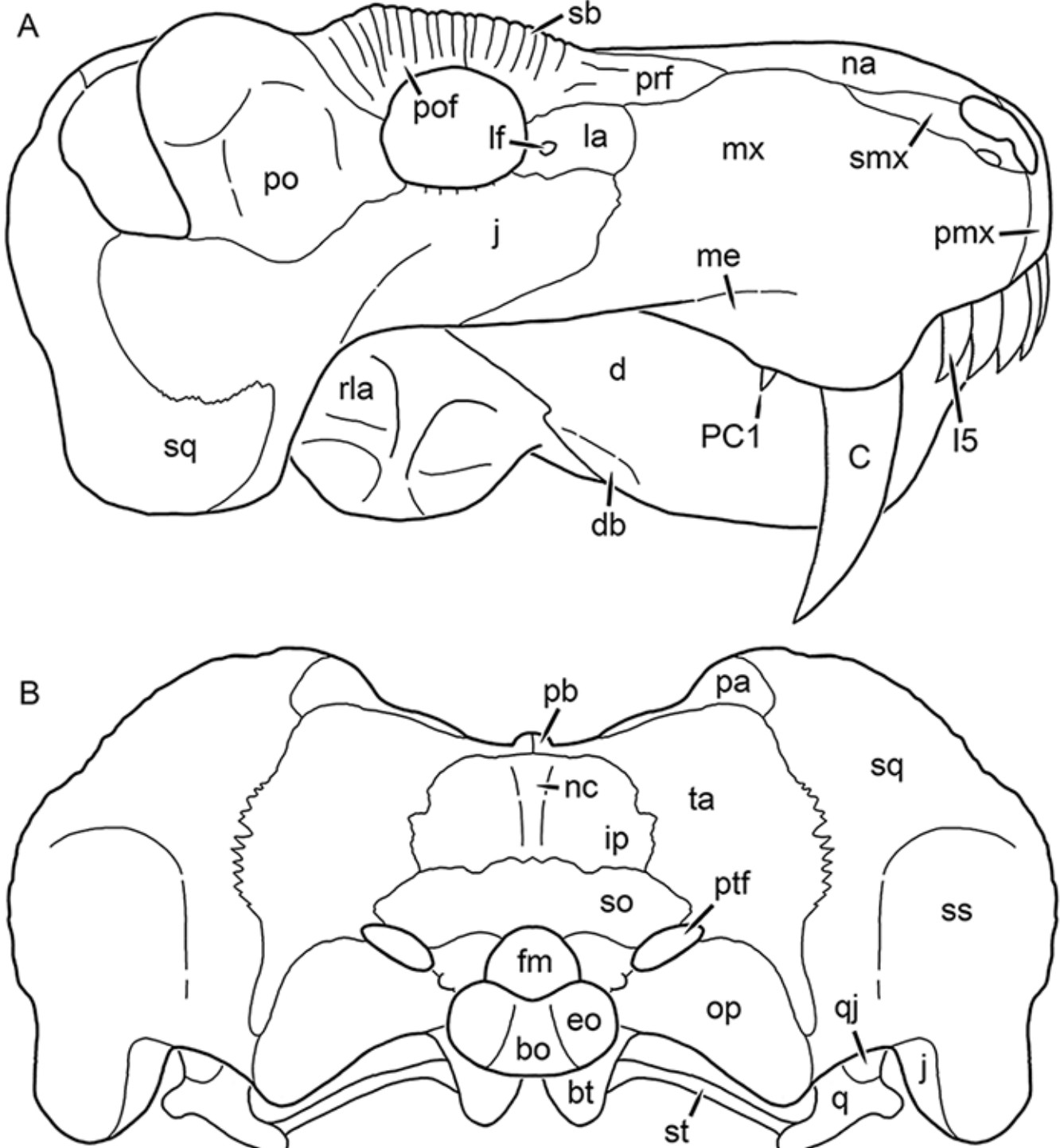

**Figure 39  Reconstruction of the skull of *Rubidgea atrox Broom, 1938* in (A) lateral and (B) occipital views.** Reconstructions based primarily on BP/1/699 and RC 13. Abbreviations: bo, basioccipital; bt, basal tuber; C, upper canine; d, dentary; db, dentary boss; eo, exoccipital; fm, foramen magnum; I, upper incisor; ip, interparietal; j, jugal; la, lacrimal; me, maxillary emargination; mx, maxilla; na, nasal; nc, nuchal crest; op, opisthotic; pa, parietal; pb, pineal boss; PC, upper postcanine; pmx, premaxilla; po, postorbital; pof, postfrontal; prf, prefrontal; ptf, post-temporal fenestra; q, quadrate; qj, quadratojugal; rla, reflected lamina of angular; sb, supraorbital boss; smx, septomaxilla; so, supraoccipital; sq, squamosal; ss, squamosal sulcus; st, stapes; ta, tabular.

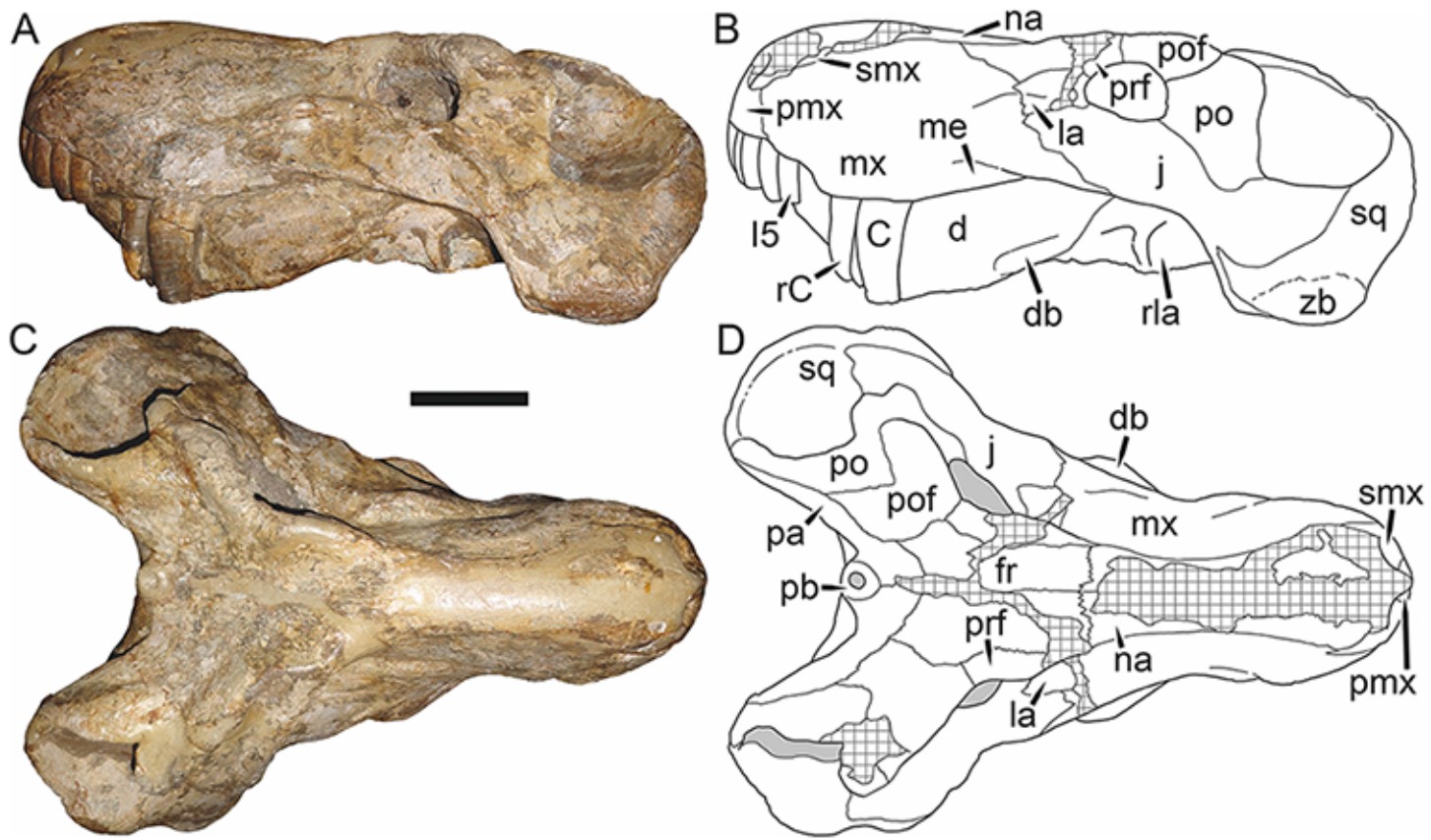

**Figure 40 Holotype (RC 13) of *Rubidgea atrox Broom, 1938* in (A) left lateral and (C) dorsal view (with (B) and (D) interpretive drawings).** Abbreviations: C, upper canine; d, dentary; db, dentary boss; fr, frontal; I, upper incisor; j, jugal; la, lacrimal; me, maxillary emargination; mx, maxilla; na, nasal; pa, parietal; pb, pineal boss; pmx, premaxilla; po, postorbital; pof, postfrontal; prf, prefrontal; rC, erupting replacement canine; rla, reflected lamina of angular; smx, septomaxilla; sq, squamosal; zb, zygomatic boss. Gray indicates matrix, hatching indicates plaster. Scale bar equals 10 cm.

(Fig. 43; nearly complete skull and lower jaws from Coetzeeskraal, Murraysburg, South Africa; holotype of *Rubidgea majora*); BP/1/3857 (Fig. 44; complete skull and lower jaws of a juvenile individual from Doornplaas, Graaff-Reinet, South Africa); CGS WB 235 (a partial skull, missing the temporal arches, from Zondagsrivier, Pearston, South Africa); GPIT K46 (Fig. 45A; the flattened right half of a skull from Kingori, Ruhuhu Basin, Tanzania; holotype of *Gorgonognathus maximus*); RC 33 (Fig. 46; dorsoventrally crushed, highly restored skull and lower jaws from Patrysfontein, Graaff-Reinet, South Africa; holotype of *Rubidgea laticeps*); RC 101 (Fig. 47; a nearly complete skull and lower jaws from Soetvlei, Richmond, South Africa; holotype of *Broomicephalus laticeps*); SAM-PK-K1235 (Fig. 42D; snout and lower jaws); TM 2002 (Fig. 48; poorly-preserved, mostly unprepared partial skull and lower jaws, much of it obscured by plaster, from Doornberg, Nieu Bethesda, South Africa; holotype of *Rubidgea kitchingi*); TM 4417 (Figs. 42D and 45B; complete but largely unprepared skull and some postcranial elements; locality data not available).

*Diagnosis*: Gigantic gorgonopsian (up to ~45 cm basal skull length) distinguished from all other rubidgeines by the following autapomorphies: elongate boss present on

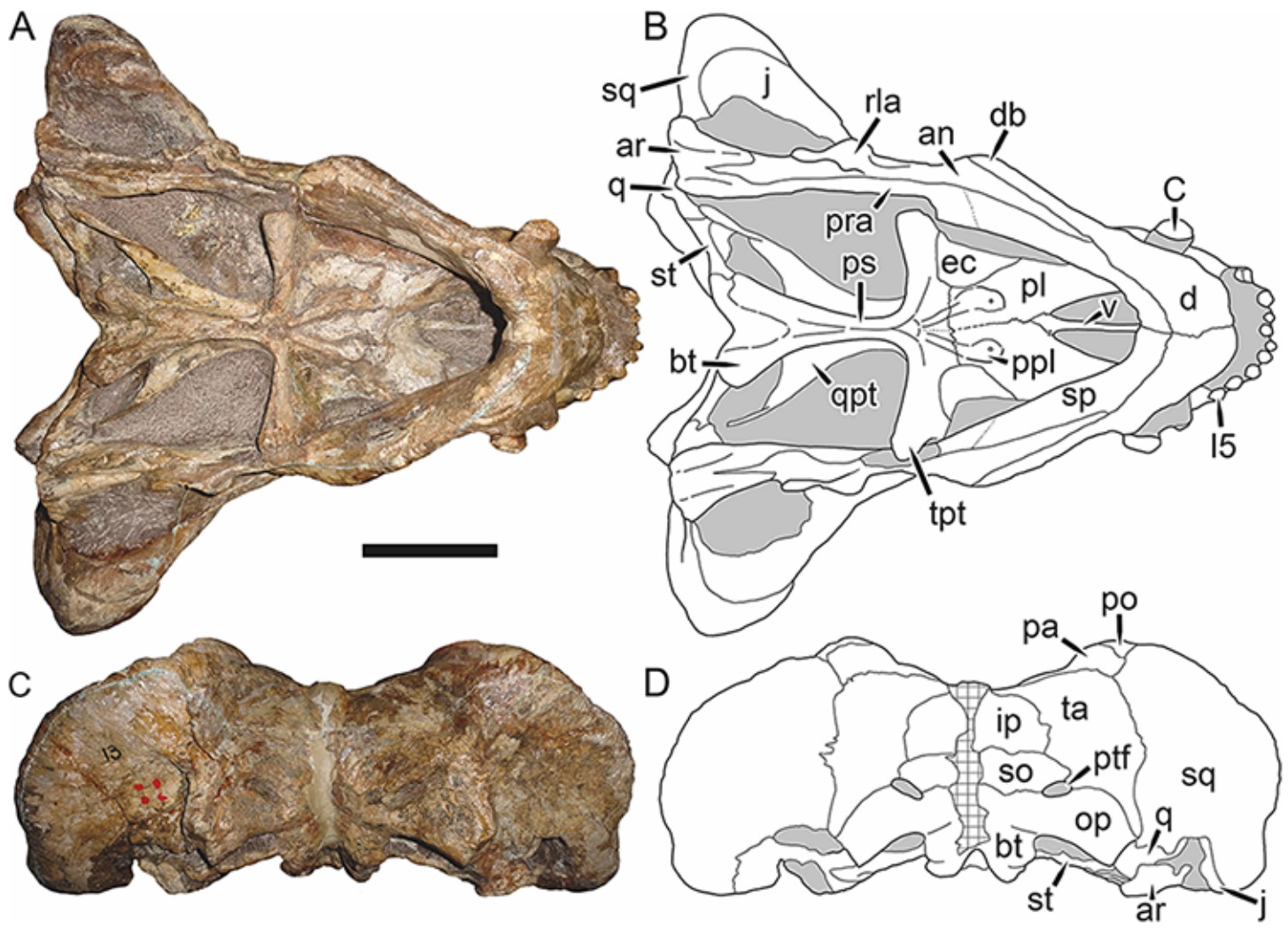

**Figure 41 Holotype (RC 13) of *Rubidgea atrox Broom, 1938* in (A) ventral and (C) occipital view (with (B) and (D) interpretive drawings).** Abbreviations: an, angular; ar, articular; bt, basal tuber; C, upper canine; d, dentary; db, dentary boss; ec, ectopterygoid; I, upper incisor; ip, interparietal; j, jugal; op, opisthotic; pa, parietal; pl, palatine; po, postorbital; ppl, palatal boss of palatine; pra, prearticular; ps, parasphenoid; ptf, post-temporal fenestra; q, quadrate; qpt, quadrate ramus of pterygoid; rla, reflected lamina of angular; so, supraoccipital; sp, splenial; sq, squamosal; st, stapes; ta, tabular; tpt, transverse process of pterygoid; v, vomer. Gray indicates matrix, hatching indicates plaster. Scale bar equals 10 cm.

ventrolateral edge of dentary, posterior flange of postorbital bar in form of massive, rounded boss, and jugal broadly exposed dorsal to squamosal in subtemporal bar. Also diagnosed by the unique combination of 1–2 upper postcanines (two upper postcanines present in some specimens of *Smilesaurus*), no lower postcanines (shared with *Clelandina* and *Leontosaurus*), dorsal skull roof pachyostosed, with rugose sculpturing and well-developed supraorbital bosses (shared with *Clelandina* and *Dinogorgon*), reduced dentition on the palatine boss (shared with *Clelandina*), flange-like maxillary alveolar region (shared with *Clelandina*), and bulbous snout (shared with *Ruhuhucerberus* and *Aelurognathus*). Ridge extending from posteroventral margin of orbit down ventrolateral edge of temporal arch very strongly developed. Intertemporal region extremely broad, comparable to *Clelandina* and *Dinogorgon*.

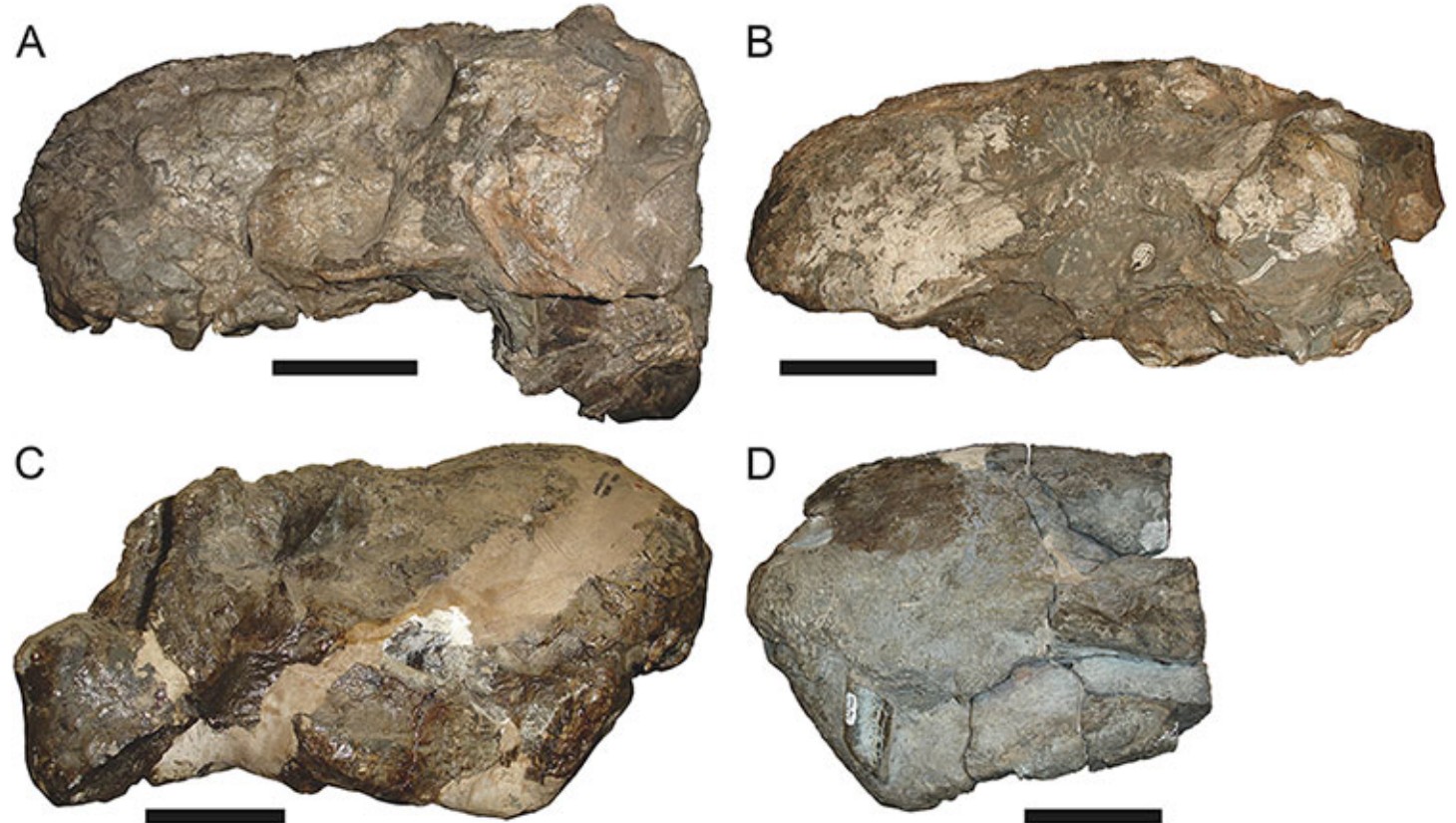

**Figure 42 Referred skulls of *Rubidgea atrox* Broom, 1938 in various preservational styles, all showing the bulbous snout morphology characteristic of this taxon.** (A) B 354, laterally crushed skull in left lateral view; (B) TM 4417, dorsoventrally crushed skull in left lateral view; (C) BP/1/195, dorsoventrally crushed (and somewhat anteriorly sheared) skull and lower jaws in right lateral view; (D) SAM-PK-K1235, slightly laterally crushed snout and lower jaws in left lateral view. Scale bars equal 10 cm.

*Comments*: *Rubidgea* is the largest known African gorgonopsian. The skull of this taxon approaches that of the Russian *Inostrancevia* in length, but is significantly more massive. The greatest development of cranial bosses among the gorgonopsians occurs in this genus, with well-developed supraorbital, postorbital, subtemporal, and dentary bosses all being present. Unlike the similarly pachyostotic dinocephalians and burnetiids, however, these bosses are confined to relatively narrow margins of the skull; they do not form a diffuse cranial 'dome.' The supraorbital and postorbital bosses give the orbitotemporal margin a distinctly 'wavy' appearance in dorsal view (Fig. 40C). *Broom (1938)* initially described two species of *Rubidgea*: the type, *R. atrox*, based on a nearly complete skull and jaws (RC 13), and *R. kitchingi*, based on a significantly more fragmentary specimen (TM 2002).

The holotype of *Rubidgea atrox* (RC 13) is a very large (40.2 cm basal length) skull (Figs. 40 and 41) with a single postcanine on both sides. This specimen has one of the better-preserved palatine bosses among *Rubidgea* skulls, and shows that the palatine had a small (1.5 cm width), reniform boss with three teeth in a transverse row, connecting to a thin, ridge-like, edentulous palatal boss of the pterygoid. This contrasts with the condition in *Dinogorgon*, in which the palatine boss is more elongate, with a single tooth row curving anteromedially. Well-developed, elongate bosses are present at the posteroventral edges of

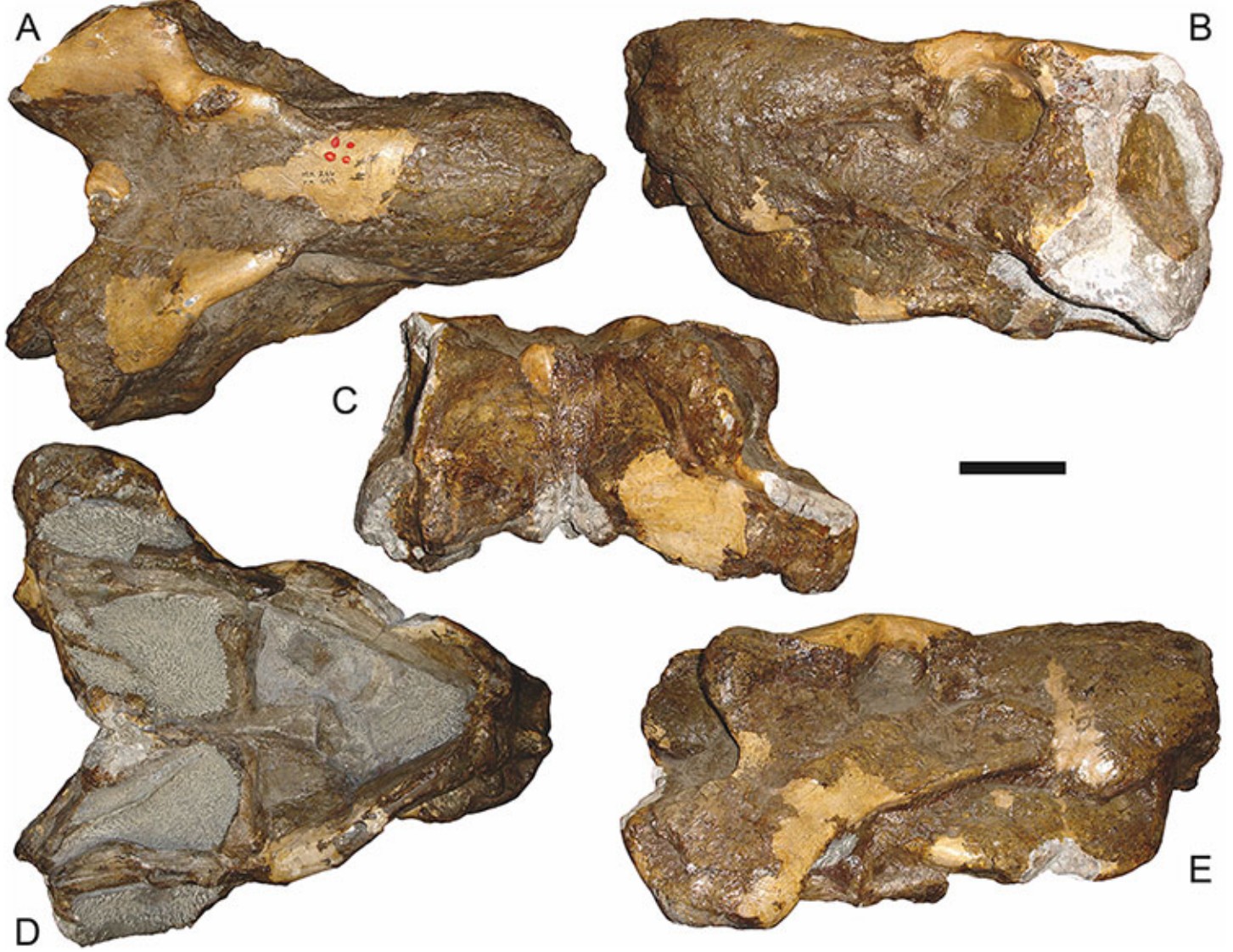

**Figure 43 Referred skull (BP/1/699) of *Rubidgea atrox* *Broom, 1938* in (A) dorsal, (B) left lateral, (C) occipital, (D) ventral, and (E) right lateral view.** Holotype of *Rubidgea majora* *Brink & Kitching, 1953*. Scale bar equals 10 cm.

the dentaries, giving them an especially 'bowed' appearance in ventral view (Fig. 41A). Thickening of the posterior base of the dentary is also present in *Dinogorgon*, but a discrete dentary boss is unique to *Rubidgea* among gorgonopsians.

In contrast to the specimen discussed above, the holotype of *Rubidgea kitchingi* (TM 2002) is extremely poor, missing the occiput and largely unprepared (Fig. 48). Plaster obscures most of the snout and right temporal region. *Sigogneau (1970)* considered this specimen indeterminate, and later (*Sigogneau-Russell, 1989*:114) remarked that it was "not a *Rubidgea*, if even a rubidgeine." *Gebauer (2007)* did not address this taxon. Although difficult to interpret, this specimen does appear to represent *Rubidgea atrox*. The intertemporal region is extremely broad, to a degree only seen in the advanced rubidgeines (*Rubidgea*, *Clelandina*, *Dinogorgon*, and *Leontosaurus*). Only two postcanine

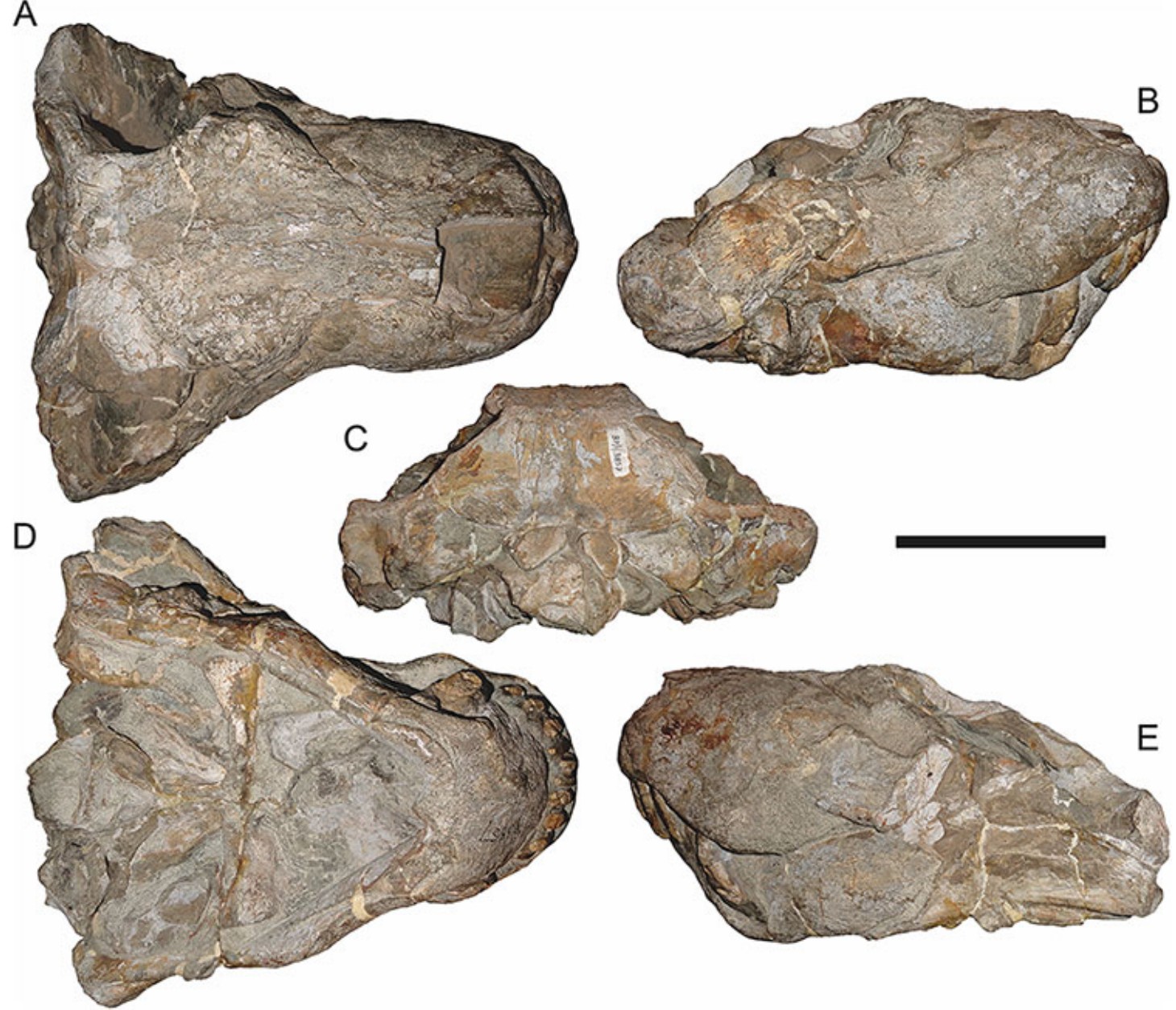

**Figure 44 Referred skull (BP/1/3857) of *Rubidgea atrox Broom, 1938* in (A) dorsal, (B) right lateral, (C) occipital, (D) ventral, and (E) left lateral view.** Scale bar equals 10 cm.

roots are visible in the left maxilla and the snout displays the typical bulbous morphology of *R. atrox*. A 'bulbous' snout is characteristic of *Rubidgea* among the gigantic rubidgeines: in *Dinogorgon* and *Leontosaurus*, the dorsal surface of the snout is relatively straight, and in *Clelandina* the dorsal convexity is not as well developed. This morphology is not an artifact of deformation: bulbous snouts are present in an array of *Rubidgea* specimens that have suffered various types of crushing (Fig. 42).

There is some confusion in the literature surrounding the similarly-named species *Rubidgea laticeps Broom, 1940b* (holotype: RC 33, Fig. 46) and *Broomicephalus laticeps*

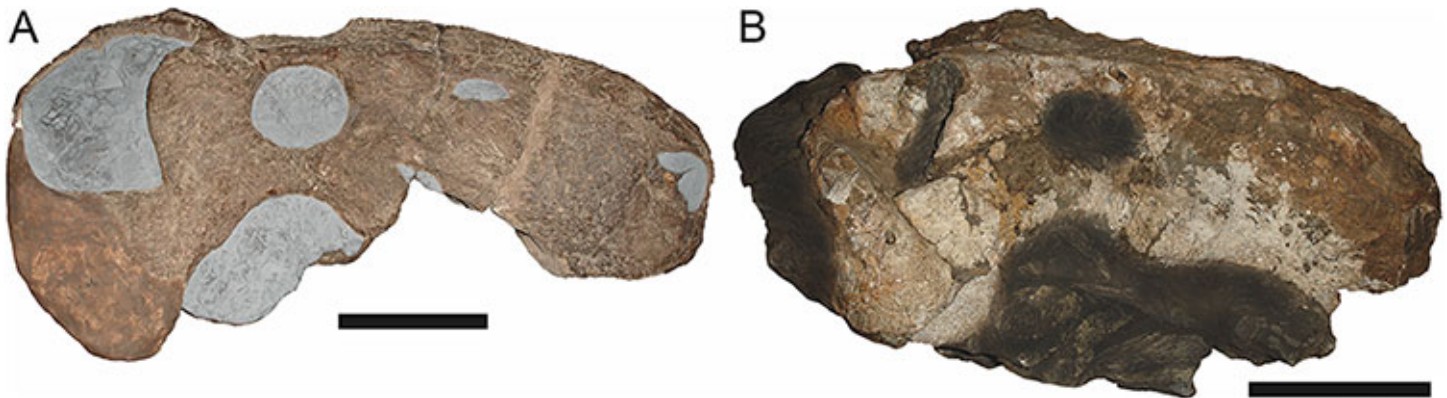

**Figure 45  Referred skulls of *Rubidgea atrox Broom, 1938*.** (A) GPIT K46 (holotype of *Gorgonognathus maximus Huene, 1950*) in right lateral view. (B) TM 4417 in right lateral view, with surrounding matrix darkened to show how the seemingly narrow suborbital zygoma of GPIT K46 can be produced through taphonomic artifact. Scale bars equal 10 cm.

*Brink & Kitching, 1953* (holotype: RC 101, Fig. 47). Both *Sigogneau (1970)* and *Gebauer (2007)* considered these specimens conspecific. *Sigogneau (1970)* recognized *Broomicephalus* as a valid, monospecific genus. *Sigogneau-Russell (1989)* listed the type species of *Broomicephalus* as *B. laticeps* (*Broom, 1940b*), attributing the generic referral to *Brink & Kitching (1953)*. However, *Brink & Kitching (1953)* specifically stated that the similarities between *Rubidgea laticeps* and their new taxon *Broomicephalus laticeps* were only superficial, the result of cranial deformation in the former taxon, and did not consider these species congeneric. *Gebauer (2007)* considered *Broomicephalus* to be synonymous with *Clelandina*, but as a valid species *C. laticeps* (*Brink & Kitching, 1953*; note, however, that because she considered RC 33 to also be referable to this species, the correct authorship should actually be *Broom, 1940b*). The presence of postcanines in these specimens indicates that they are not referable to *Clelandina*. RC 33 is a poorly-preserved, extensively reconstructed skull (Fig. 46), but its anatomy largely accords that of *Rubidgea atrox*. The supraorbital boss on the left side of this skull is relatively weak, but this is probably attributable to overpreparation.

Contra *Brink & Kitching (1953)*, RC 101 has quite clearly suffered anteroposterior crushing, as indicated by orbits that are significantly taller than wide (Fig. 47B). This deformation can be blamed in part for the extremely short snout and broad temporal region in this specimen. However, RC 101 is also a small skull (18.9 cm dorsal skull length), and its proportions may also be due to juvenile or subadult status. This idea is supported by another small specimen, BP/1/3857 (21.5 dorsal and 26.5 cm basal skull length) (Fig. 44). BP/1/3857 shows several diagnostic features of the taxon *Rubidgea atrox*, including the reduced upper postcanine count (two on each side), flange-like maxillary alveolar border, and a small dentary boss. However, it lacks the other cranial bosses (notably the supraorbital), which probably develop late in ontogeny (as discussed in the sections on *Clelandina* and *Dinogorgon* above). Interestingly, the dorsal surface of the skull is already rugosely sculptured (Fig. 44A), similar to that of '*Prorubidgea robusta*' (=*Dinogorgon*). BP/1/3857 also differs from larger *Rubidgea* specimens in having an extremely tall, short snout, similar to RC 101. Proportional increase of snout length with

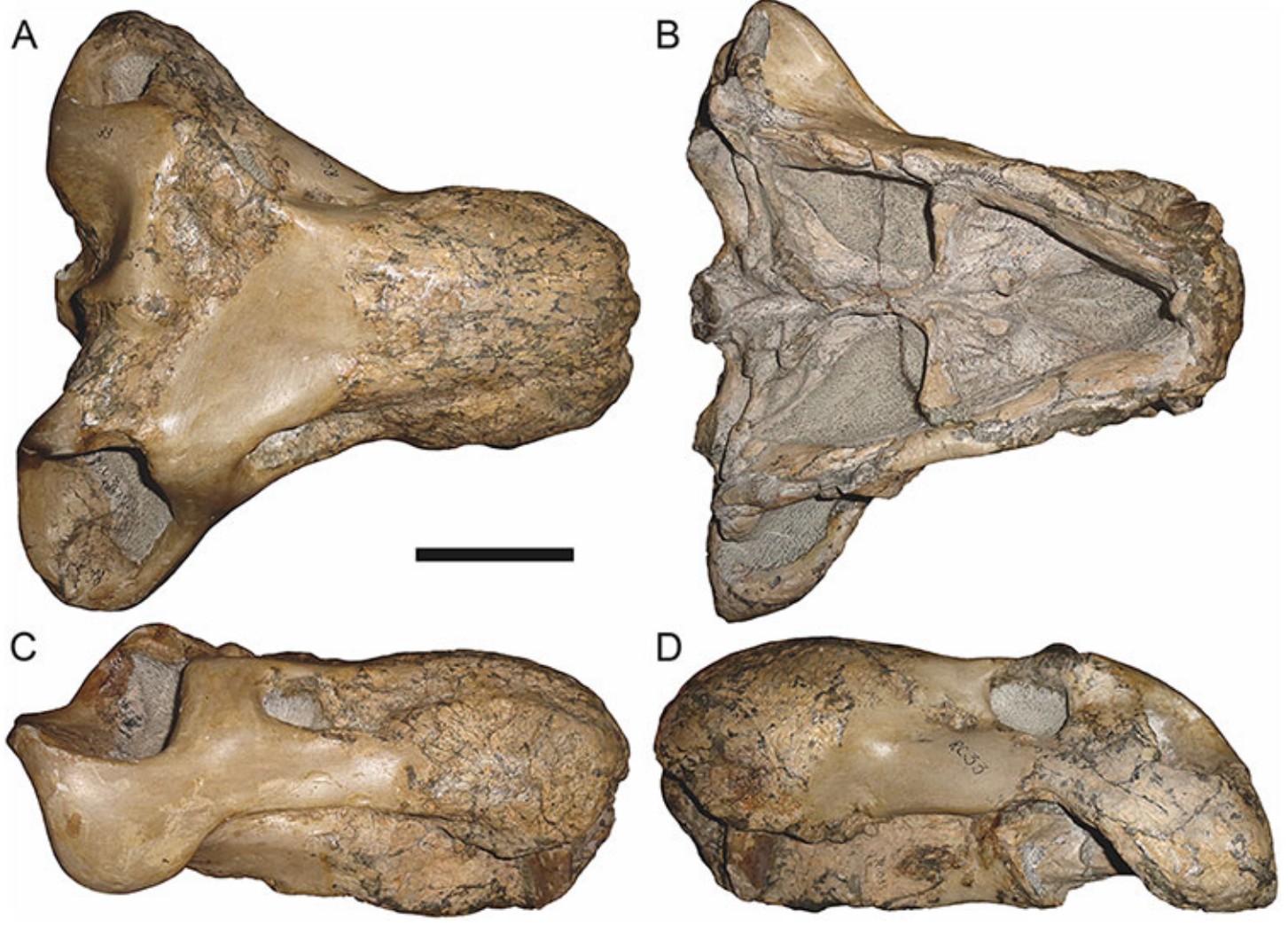

**Figure 46  Referred skull (RC 33) of *Rubidgea atrox Broom, 1938* in (A) dorsal, (B) ventral, (C) right lateral, and (D) left lateral view.** Holotype of *Rubidgea laticeps Broom, 1940b*. Scale bar equals 10 cm.

growth is a typical allometric feature of synapsids (and many other tetrapods), and short snouts in these specimens are here taken as correlates of youth rather than taxonomic distinction.

The most heavily pachyostosed specimens of *Rubidgea* are BP/1/699 (the holotype of *Rubidgea majora*; Fig. 43) and BP/1/195 (a specimen listed as *Rubidgea* cf. *R. platyrhina* by *Sigogneau (1970)*; Fig. 42C). Both skulls are partially restored with plaster, but BP/1/195 is significantly more fragmentary and crushed. These are the most massive gorgonopsian skulls known, with greatly expanded postorbital bars and zygomatic flanges and somewhat shorter snouts than RC 13. I consider it most probable that these represent fully mature specimens that have experienced some anteroposterior deformation (making the snout appear shorter) rather than a distinct species.

*Huene (1950)* described a gigantic partial skull (GPIT K46) from the Usili (formerly Kawinga) Formation of Tanzania as a new species of the genus *Gorgonognathus*:

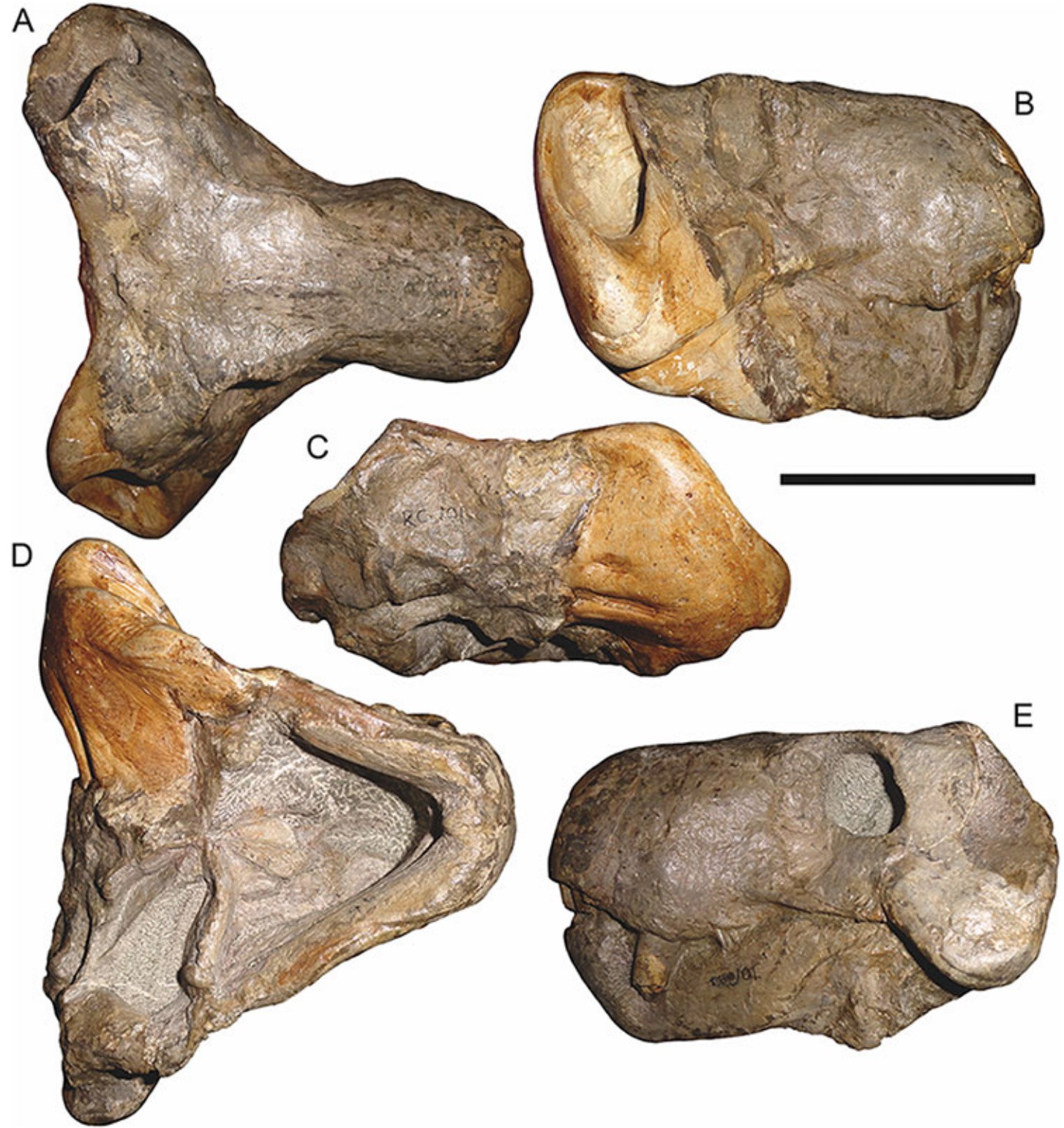

**Figure 47** Referred skull (RC 101) of *Rubidgea atrox Broom, 1938* in (A) dorsal, (B) right lateral, (C) occipital, (D) ventral, and (E) left lateral view. Holotype of *Broomicephalus laticeps Brink & Kitching, 1953*. Scale bar equals 10 cm.

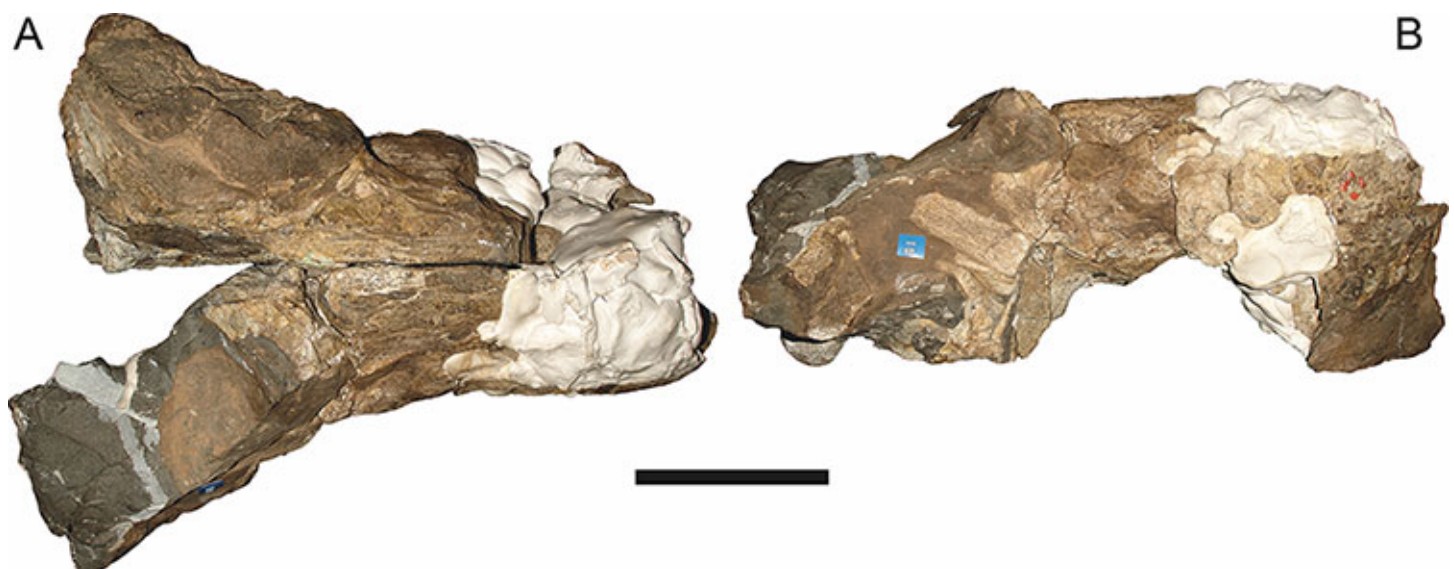

**Figure 48 Referred skull (TM 2002) of _Rubidgea atrox Broom, 1938_ in (A) dorsal and (B) right lateral view.** Holotype of _Rubidgea kitchingi Broom, 1938_. Scale bar equals 10 cm.

_G. maximus._ GPIT K46 is the right side of a laterally crushed skull with several large missing portions, notably in the maxilla (Fig. 45A). Huene distinguished this taxon from the type species of _Gorgonognathus_ (the South African _G. longifrons_) based on its greater size and expanded postorbital bar. _Sigogneau-Russell (1989)_ considered _G. maximus_ to be a _nomen dubium_, identifiable only as an indeterminate gorgonopid. _Maisch (2002)_ restudied GPIT K46, recognizing that it was clearly a rubidgeine (based on the posteroventrally expanded zygomatic arch) and that the relatively long, low snout of this specimen was most similar to that of _Rubidgea_ itself. However, he argued that it represented a distinct genus (which he named _Titanogorgon_) based on the presence of a "very slender and ventrally concave suborbital arch" (_Maisch, 2002_:246). He stressed that this concavity was a real feature of the skull and not an artifact of preservation or preparation. My comparisons between this specimen and skulls of _Rubidgea atrox_ strongly suggest that this concavity does not correspond to the actual ventral margin of the zygomatic arch, however. The ventral margin of the zygoma in GPIT K46 perfectly matches the location and curvature of the zygomatic ridge in _Rubidgea atrox_ (particularly in the holotype, RC 13; see Fig. 40A). The portion of the suborbital arch beneath this ridge is relatively thin and weak, and it is readily destroyed or obscured in poorly-preserved specimens. For example, in TM 4417, a definite South African specimen of _R. atrox_, a combination of distortion and damage to the skull has produced a zygomatic profile identical to that of GPIT K46 (Fig. 45B), with the illusion of a narrow, ventrally concave suborbital arch due to breakage and the actual ventral margin being obscured by matrix. Noteworthy is the fact that GPIT K46 is obscured with plaster beneath the suborbital concavity, hiding potential breakage beneath the zygomatic ridge. Given that the entire rest of the ventral margin of the skull is broken in this specimen, I consider breakage of the portion beneath the orbit

to be likely as well. Under this reinterpretation, GPIT K46 exhibits no unique features warranting recognition as a separate rubidgeine taxon. As *Maisch (2002)* noted, the dimensions of the snout accord with those of *Rubidgea* rather than *Dinogorgon*, the only other rubidgeine attaining this size. In the absence of any clear autapomorphies distinguishing GPIT K46 from South African specimens of *Rubidgea*, I consider *Titanogorgon maximus* to be a junior synonym of *R. atrox*.

### Ruhuhucerberus Maisch, 2002

*Type species*: *Ruhuhucerberus terror* *Maisch, 2002* (=junior subjective synonym of *Aelurognathus haughtoni* *Huene, 1950*).

   *Diagnosis*: As for the type and only recognized species.

### Ruhuhucerberus haughtoni (Huene, 1950) comb. nov. (Reconstruction Figs. 49–50, Specimen Figs. 51–53)

*Aelurognathus haughtoni* *Huene, 1950*:88

*Leontocephalus haughtoni* *Sigogneau, 1970*:249

*Cephalicustriodus kingoriensis* *Parrington, 1974*:51 (*partim*)

*Ruhuhucerberus terror* *Maisch, 2002*:247

*Sycosaurus terror* *Gebauer, 2007*:205

*Holotype*: GPIT/RE/7117 (K46B of *Huene (1950)*), a nearly complete skull (Fig. 51) from Kingori, Ruhuhu Basin, Tanzania.

   *Referred specimens*: UMZC T881 (Fig. 52; a fragmentary skull from Stockley's Site B4/7, Katumbi Viwili, Ruhuhu Basin, Tanzania); UMZC T891 (Fig. 53; a well-preserved, complete skull from Stockley's Site B4/7, Katumbi Viwili, Ruhuhu Basin, Tanzania; holotype of *Ruhuhucerberus terror*).

   *Diagnosis*: Large gorgonopsian (up to 33 cm basal skull length) distinguished from all other rubidgeines by the following autapomorphies: narrow prefrontal-postfrontal contact that barely excludes the frontals from the orbital margin, broad anterior margins of the prefrontals, such that they do not taper anteriorly in dorsal view, small, triangular postfrontal making minimal contact with the parietal, anteroposteriorly expanded postorbital bar lacking a posterior flange, and presence of a relatively tall, broad snout (shorter and transversely broader, proportionally, than in other tall-snouted rubidgeines such as *Aelurognathus tigriceps* and *Dinogorgon rubidgei*). Also diagnosed by the unique combination of four upper postcanines, presence of a small preparietal (shared with some specimens of *Aelurognathus tigriceps* and *Smilesaurus ferox*), well-developed interorbital ridge (shared with *Clelandina*, *Dinogorgon*, and *Rubidgea*), and palatal boss of the pterygoid that is narrow and ridge-like but still dentigerous (as in some subadult specimens of *Leontosaurus vanderhorsti* and *Sycosaurus laticeps*).

   *Comments*: UMZC T891, one of the finest gorgonopsian skulls known, has had a lengthy history of taxonomic flux. *Parrington (1974)* initially described this skull, referring

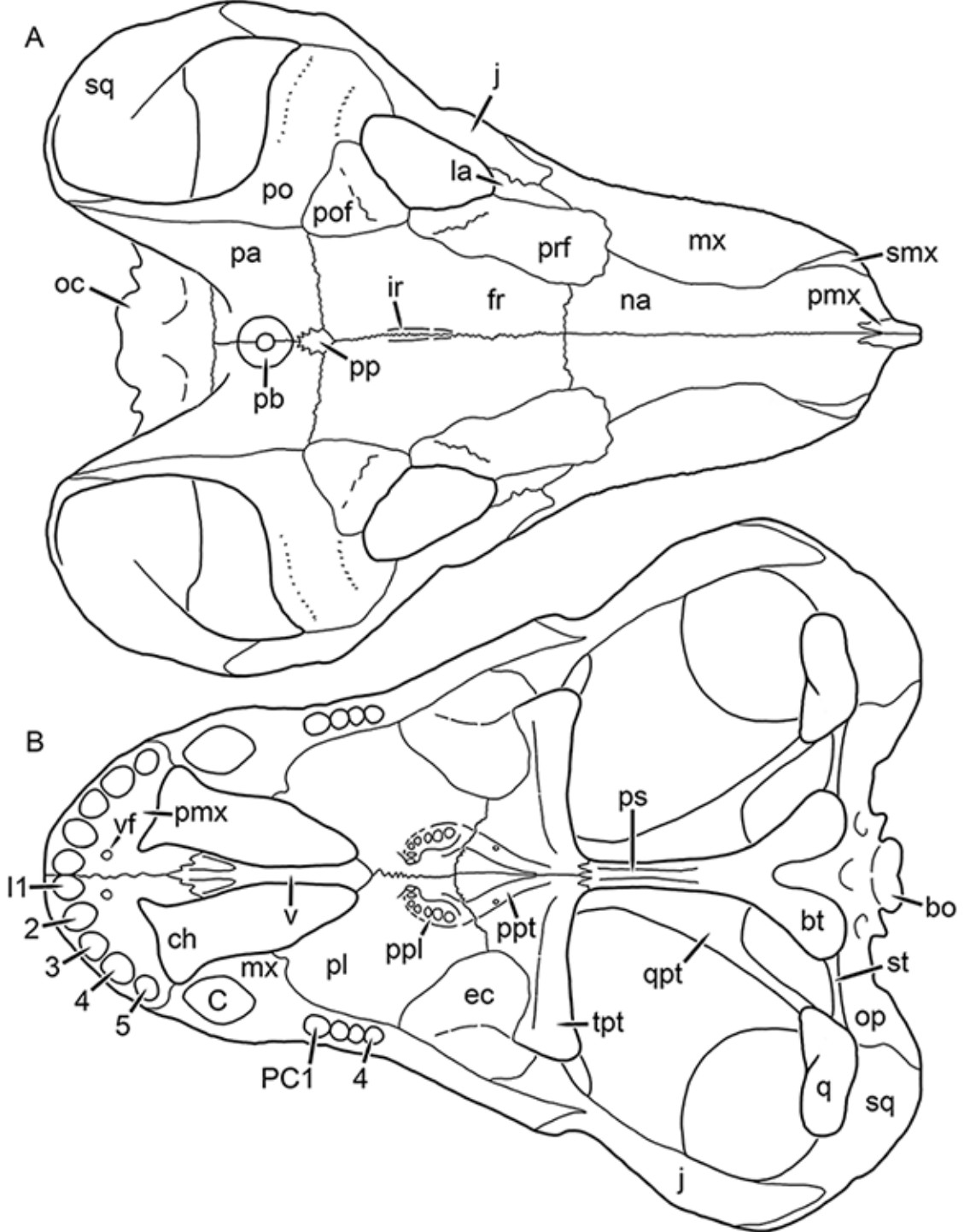

**Figure 49 Reconstruction of the skull of *Ruhuhucerberus haughtoni* (*Huene, 1950*) in (A) dorsal and (B) ventral views.** Reconstructions based primarily on GPIT/RE/7117 and UMZC T891. Abbreviations: bo, basioccipital; bt, basal tuber; C, upper canine; ch, choana; ec, ectopterygoid; fr, frontal; I, upper incisor; ir, interorbital ridge; j, jugal; la, lacrimal; mx, maxilla; na, nasal; op, opisthotic; pa, parietal; pb, pineal boss; PC, upper postcanine; pl, palatine; pmx, premaxilla; po, postorbital; pof, postfrontal; pp, preparietal; ppl, palatal boss of palatine; ppt, palatal boss of pterygoid; prf, prefrontal; ps, parasphenoid; q, quadrate; qpt, quadrate ramus of pterygoid; smx, septomaxilla; sq, squamosal; st, stapes; tpt, transverse process of pterygoid; v, vomer; vf, ventral premaxillary foramen.

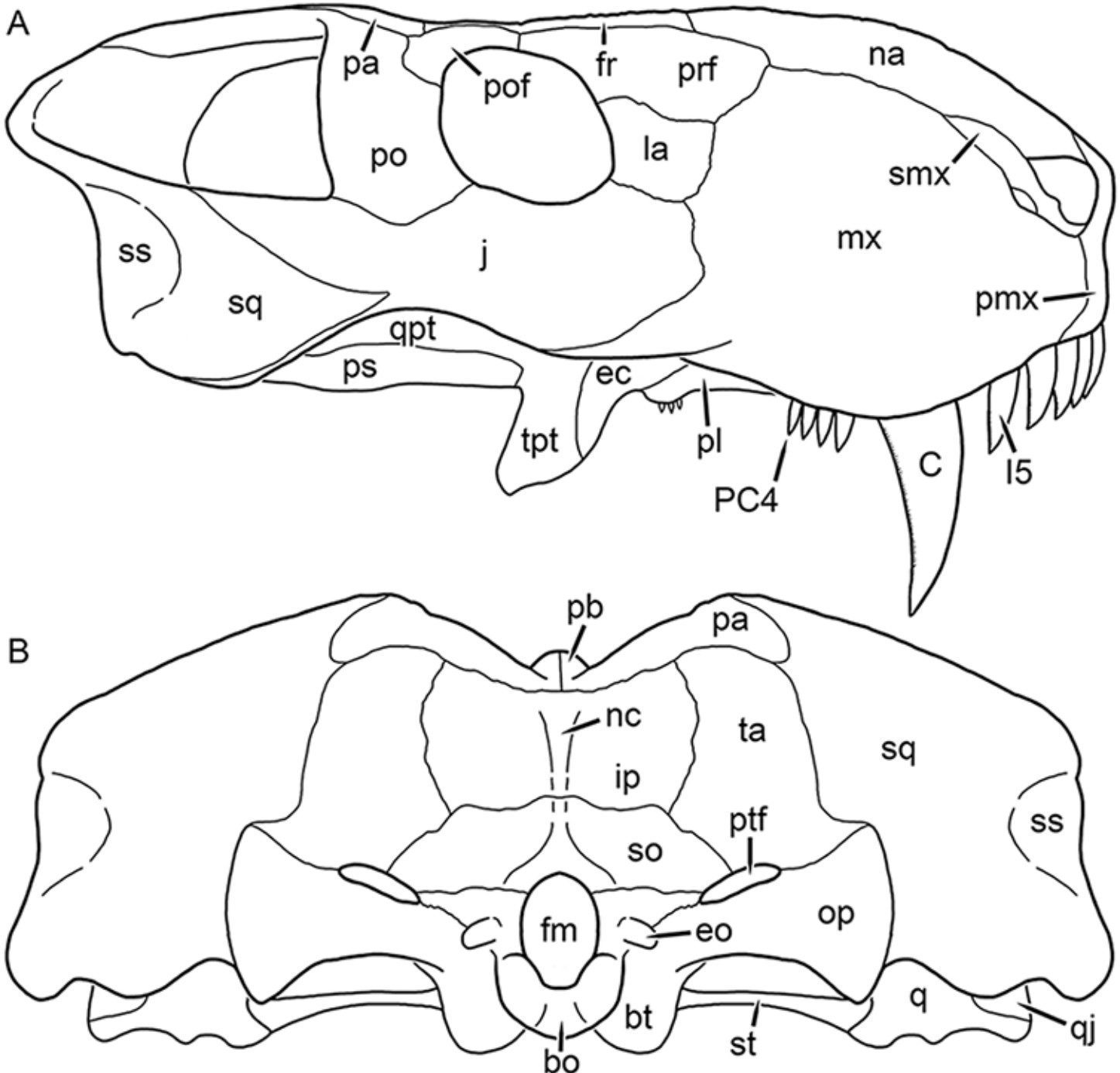

**Figure 50 Reconstruction of the skull of *Ruhuhucerberus haughtoni* (*Huene, 1950*) in (A) lateral and (B) occipital views.** Reconstructions based primarily on GPIT/RE/7117 and UMZC T891. Abbreviations: bo, basioccipital; bt, basal tuber; C, upper canine; ec, ectopterygoid; eo, exoccipital; fm, foramen magnum; fr, frontal; I, upper incisor; ip, interparietal; j, jugal; la, lacrimal; mx, maxilla; na, nasal; nc, nuchal crest; op, opisthotic; pa, parietal; pb, pineal boss; PC, upper postcanine; pl, palatine; pmx, premaxilla; po, postorbital; pof, postfrontal; prf, prefrontal; ps, parasphenoid; ptf, post-temporal fenestra; q, quadrate; qj, quadratojugal; qpt, quadrate ramus of pterygoid; smx, septomaxilla; so, supraoccipital; sq, squamosal; ss, squamosal sulcus; ta, tabular; tpt, transverse process of pterygoid.

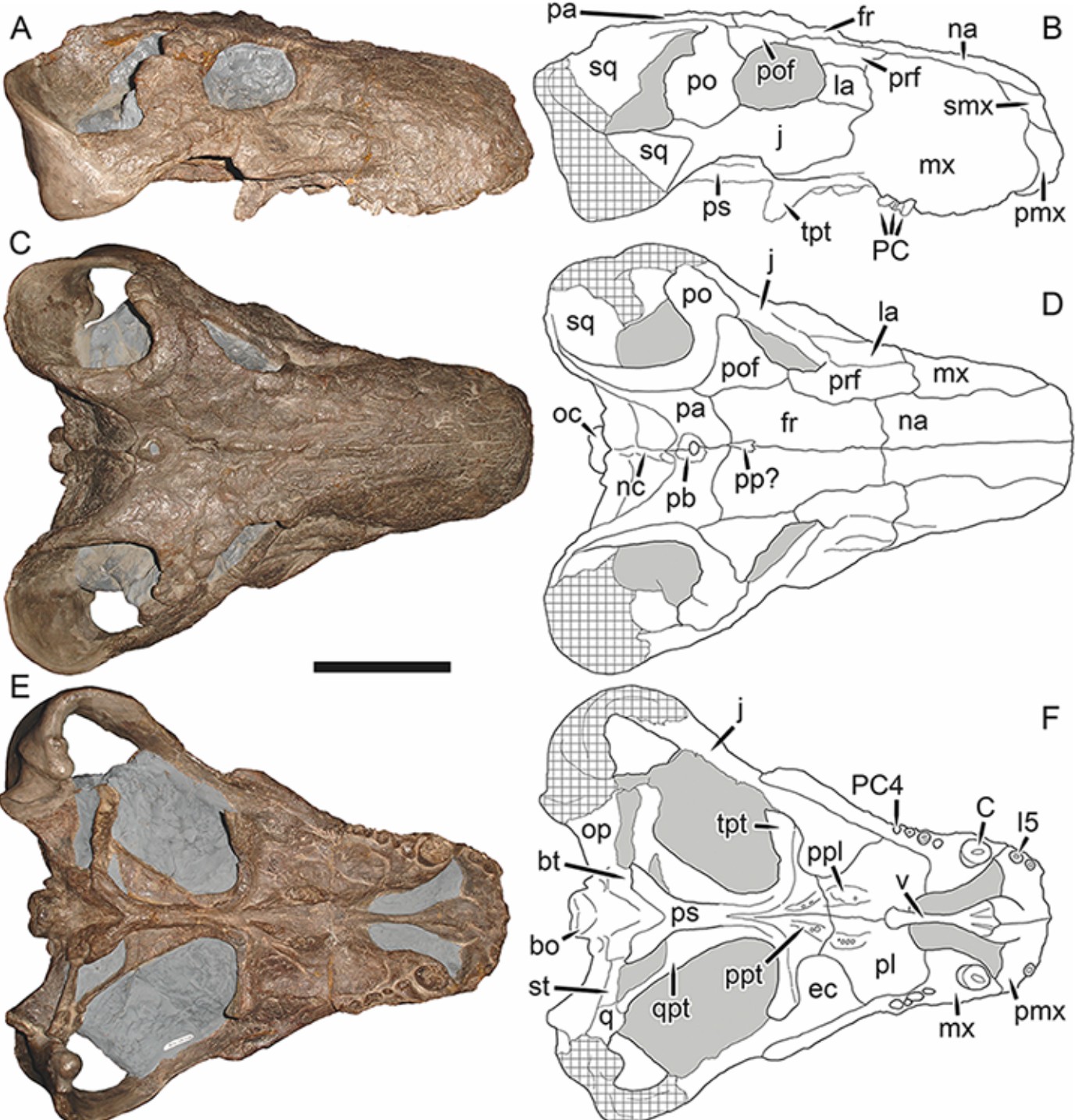

**Figure 51 Holotype (GPIT/RE/7117) of *Ruhuhucerberus haughtoni* (*Huene, 1950*) in (A) right lateral, (C) dorsal, and (E) ventral view (with (B) (D) and (F) interpretive drawings).** Abbreviations: bo, basioccipital; bt, basal tuber; C, upper canine; ec, ectopterygoid; fr, frontal; I, upper incisor; j, jugal; la, lacrimal; mx, maxilla; na, nasal; nc, nuchal crest; oc, occipital condyle; op, opisthotic; pa, parietal; pb, pineal boss; PC, upper postcanine; pmx, premaxilla; po, postorbital; pof, postfrontal; pp, preparietal; ppl, palatal boss of palatine; ppt, palatal boss of pterygoid; prf, prefrontal; ps, parasphenoid; q, quadrate; qpt, quadrate ramus of pterygoid; smx, septomaxilla; sq, squamosal; st, stapes; tpt, transverse process of pterygoid; v, vomer. Gray indicates matrix, hatching indicates plaster. Scale bar equals 10 cm.

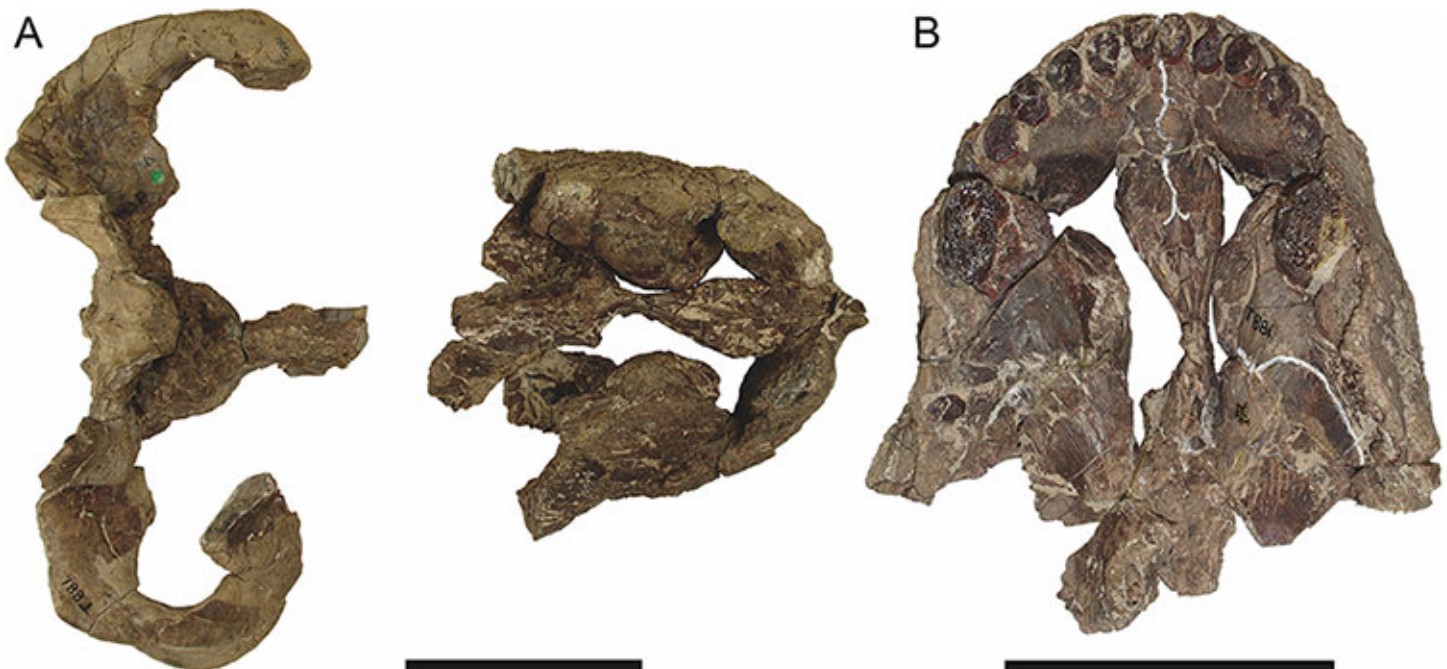

**Figure 52 Referred skull (UMZC T881) of *Ruhuhucerberus haughtoni* (*Huene, 1950*) in (A) dorsal and (B) ventral view.** Anterior is right in (A) up in (B). Scale bars equal 10 cm.

it to *Huene's (1950)* species *Lycaenops kingoriensis*, but on the basis of its distinct snout shape erected the new genus *Cephalicustriodus*. *Sigogneau-Russell (1989)* maintained *Cephalicustriodus kingoriensis* as a valid taxon, but only for UMZC T891—she referred GPIT/RE/7116 (holotype of *Lycaenops kingoriensis*) to *Sycosaurus*, as ?*S. kingoriensis*. *Maisch (2002)* noted that this nomenclatural scheme was untenable: since the type species of *Cephalicustriodus* was *Lycaenops kingoriensis*, that genus name is necessarily tied to GPIT/RE/7116, not UMZC T891. Thus, he bestowed the new genus and species name *Ruhuhucerberus terror* on the latter specimen. *Gebauer (2007)* recognized this species as valid, but considered it referable to *Sycosaurus*, as *S. terror*.

UMZC T891 is, as noted by *Parrington (1974)*, unique among gorgonopsians in the proportions of its snout, which is short, broad, and bulbous. Furthermore, this specimen is almost completely undistorted, so these proportions are not attributable to taphonomy. A second, significantly less complete skull (UMZC T881; Fig. 52) from the same locality has an identical snout shape. In addition to snout shape, UMZC T891 can be distinguished from other gorgonopsians by its broad anterior margin of the prefrontal, very narrow prefrontal-postfrontal contact, and small, triangular postfrontal that barely contacts the parietal. Like most other rubidgeines, the postorbital bar is anteroposteriorly expanded, but lacks the distinct posterior flange typical of other taxa; instead, it is evenly expanded throughout its height.

Although I concur with *Sigogneau-Russell (1989)*, *Maisch (2002)* and *Gebauer (2007)* that UMZC T891 is not referable to *Lycaenops kingoriensis*, it closely matches another Tanzanian gorgonopsian described by *Huene (1950)*: *Aelurognathus haughtoni*.

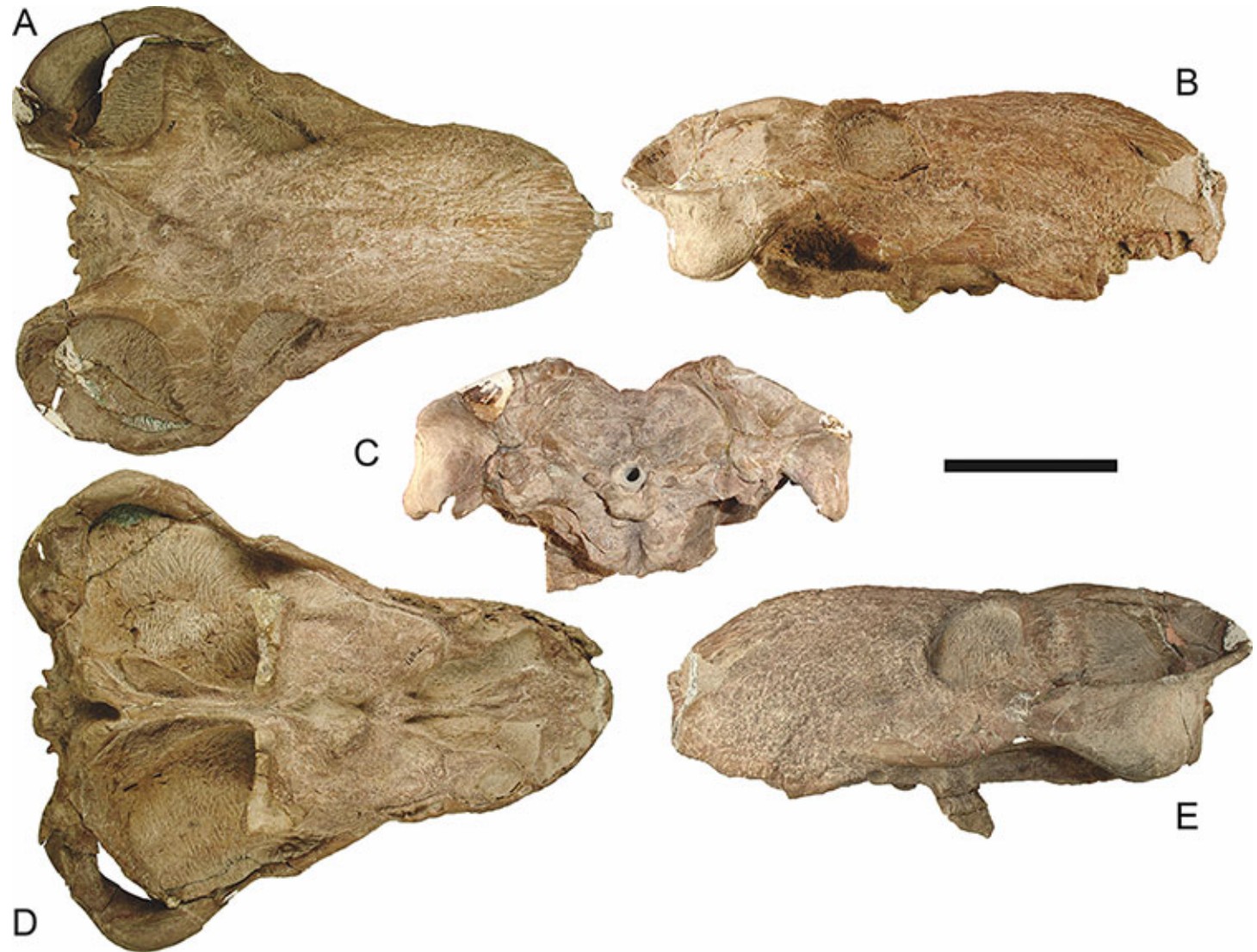

**Figure 53 Referred skull (UMZC T891) of *Ruhuhucerberus haughtoni* (*Huene, 1950*) in (A) dorsal, (B) right lateral, (C) occipital, (D) ventral, and (E) left lateral view.** Holotype of *Ruhuhucerberus terror Maisch, 2002*. Scale bar equals 10 cm.

*Sigogneau-Russell (1989)* considered *A. haughtoni* to be referable to *Leontocephalus*, whereas *Gebauer (2007)* synonymized it with *Sycosaurus* (originally *Lycaenops*) *kingoriensis*. However, the holotype of *A. haughtoni* (GPIT/RE/7117) shares all of the characteristic features of UMZC T891 listed above, as well as a well-developed interorbital ridge and anteriorly bulbous interchoanal vomer (which are not unique among rubidgeines, but do distinguish this taxon from *Sycosaurus*). The snout of GPIT/RE/7117 is not as bulbous as UMZC T891, but this can be attributed to dorsoventral crushing. As such, I consider *Ruhuhucerberus terror* to be a junior synonym of *Aelurognathus haughtoni*. As this species is not the sister taxon of *Aelurognathus tigriceps* (see Phylogenetic Analysis), however, the genus *Ruhuhucerberus* is retained.

### Smilesaurus *Broom, 1948*

*Pardocephalus* *Broom, 1948*:603

*Type species*: *Smilesaurus ferox* *Broom, 1948*.
   *Diagnosis*: As for the type and only recognized species.

### Smilesaurus ferox *Broom, 1948*
### (Reconstruction Figs. 54–55, Specimen Figs. 56–61)

*Pardocephalus wallacei* *Broom, 1948*:603
*Smilesaurus maccabei* *Broom, 1948*:601
*Arctops*? *ferox* *Sigogneau, 1970*:148
*Aelurognathus ferox* *Gebauer, 2007*:186

*Holotype*: RC 62, a distorted skull and lower jaws (Fig. 56) with postcranial elements from Graaff-Reinet Commonage, Graaff-Reinet, South Africa.
   *Referred specimens*: B 352 (a mostly unprepared, complete skull from Graaff-Reinet, South Africa); BP/1/2465 (Fig. 57; a complete, laterally crushed skull and right mandible from Oudeplaas, Richmond, South Africa; referred to *Arctops*? *ferox* by *Sigogneau (1970)*); BP/1/4409 (Fig. 58; a complete, distorted skull and lower jaws with attached postcranial fragments from Eselskop, Pearston, South Africa); BP/1/4410 (Fig. 59; a complete, small skull and lower jaws from Eselskop, Pearston, South Africa); CGS RS 176 (a nearly complete skull, missing the left temporal arch, and lower jaws from De Hoop annex No. 1, Kuilspoort, Beaufort West, South Africa); CGS S 231 (a snout and anterior lower jaws from Reiersvlei, Fraserburg, South Africa); CGS WB 22 (a nearly complete skull with lower jaws and anterior cervical vertebrae from Groote Riet Valley, Somerset East, South Africa); CGS WB 213 (a complete skull and lower jaws from Platrivier, Pearston, South Africa); NMQR 480 (a laterally crushed, incomplete skull and jaws from Quaggasfontein, Colesberg, South Africa); RC 81 (Fig. 60; a skull, missing the right temporal region, and lower jaws from Riverdale, Graaff-Reinet, South Africa; holotype of *Smilesaurus maccabei*); RC 82 (Fig. 61; an incomplete skull, missing the dorsal surface and temporal region, and lower jaws from Dalham, Graaff-Reinet, South Africa; holotype of *Pardocephalus wallacei*); TM 4986 (a complete, laterally crushed skull and lower jaws; locality data not available).
   *Diagnosis*: Large gorgonopsian (up to 31 cm basal skull length) distinguished from all other rubidgeines by the following autapomorphies: pineal boss situated in a diamond-shaped depression, large, ventrally-projecting reflected lamina of angular, and extremely long upper canine. Also diagnosed by the unique combination of 2–3 upper postcanines, a short parasphenoid (shared with all rubidgeines other than *Aelurognathus* and *Ruhuhucerberus*), a tall, narrow occiput (shared with *Aelurognathus*), lengthy posterior processes of frontals terminating medial to the temporal fenestra (shared with the non-rubidgeine gorgonopsian *Arctognathus* (*Kammerer, 2015*)), relatively large frontal contribution to the orbital margin (unlike other rubidgeines), absence of any cranial pachyostosis or bone rugosity, and relatively small orbits.

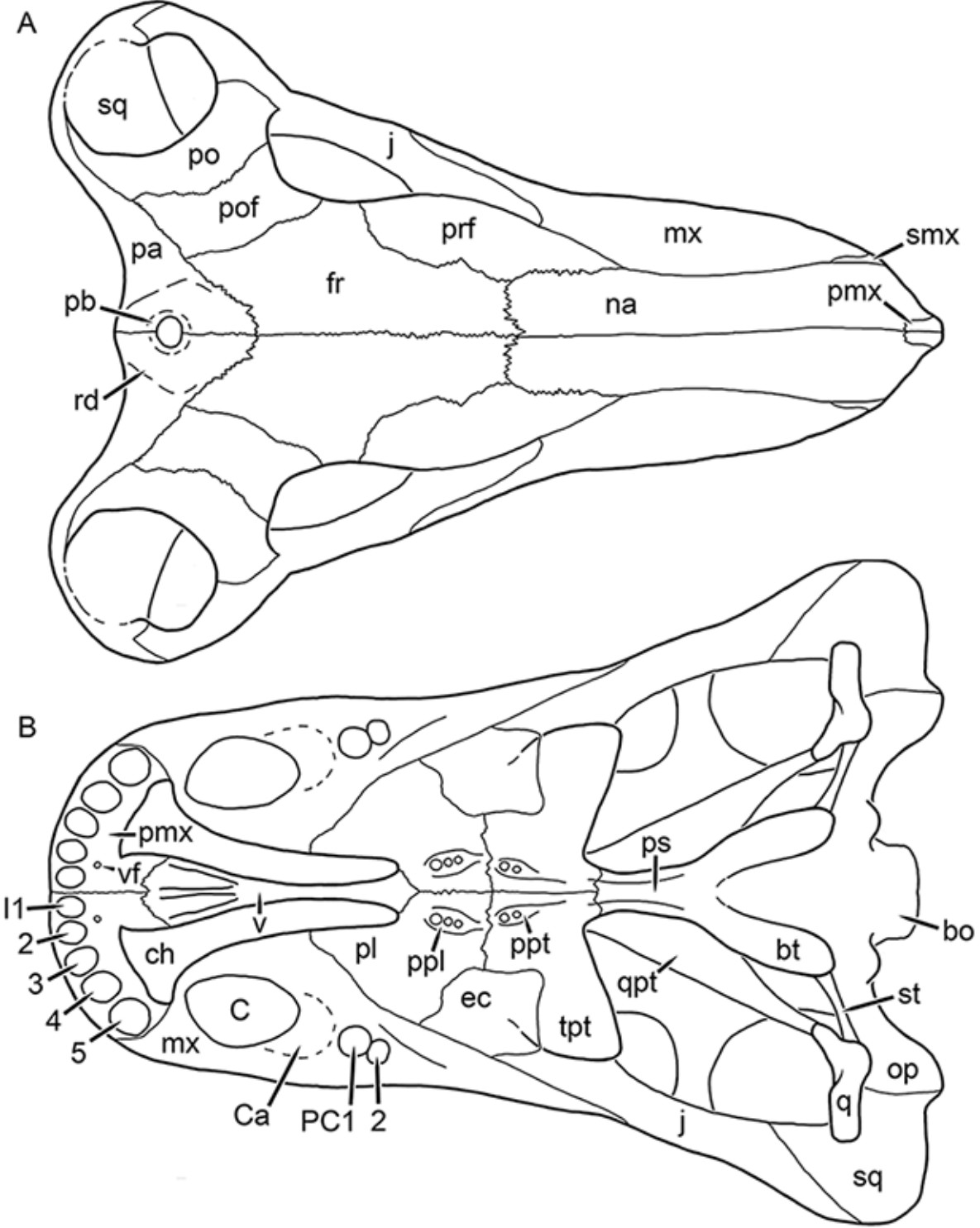

**Figure 54 Reconstruction of the skull of *Smilesaurus ferox Broom, 1948* in (A) dorsal and (B) ventral views.** Reconstructions based primarily on BP/1/2465, BP/1/4409, and RC 62. Abbreviations: bo, basioccipital; bt, basal tuber; C, upper canine; Ca, canine alveolus; ch, choana; ec, ectopterygoid; fr, frontal; I, upper incisor; j, jugal; mx, maxilla; na, nasal; op, opisthotic; pa, parietal; pb, pineal boss; PC, upper postcanine; pl, palatine; pmx, premaxilla; po, postorbital; pof, postfrontal; ppl, palatal boss of palatine; ppt, palatal boss of pterygoid; prf, prefrontal; ps, parasphenoid; q, quadrate; qpt, quadrate ramus of pterygoid; rd, rhomboidal depression; smx, septomaxilla; sq, squamosal; st, stapes; tpt, transverse process of pterygoid; v, vomer; vf, ventral premaxillary foramen.

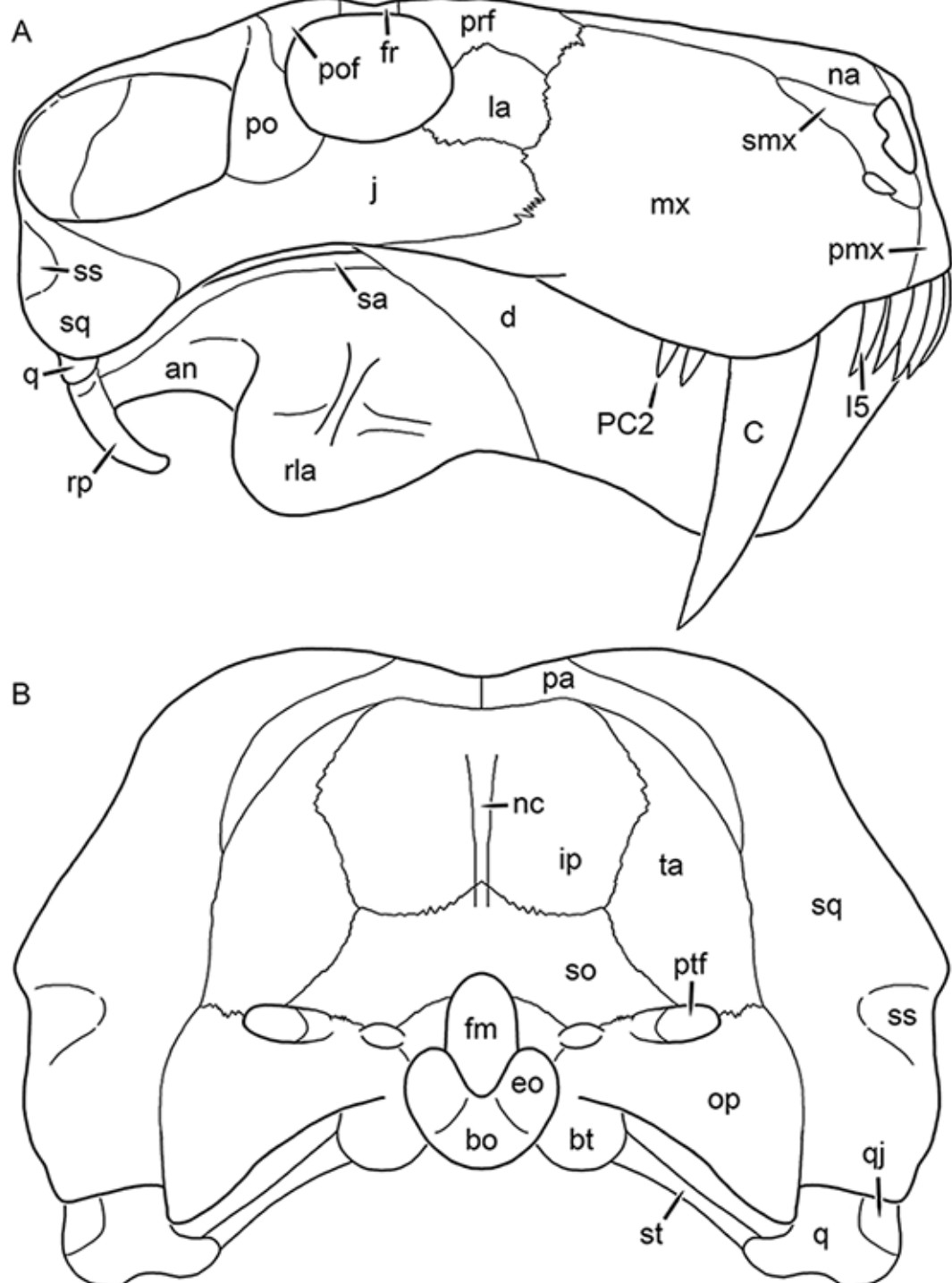

**Figure 55 Reconstruction of the skull of *Smilesaurus ferox* *Broom, 1948* in (A) lateral and (B) occipital views.** Reconstructions based primarily on BP/1/2465, BP/1/4409, and RC 62. Abbreviations: an, angular; bo, basioccipital; bt, basal tuber; C, upper canine; d, dentary; eo, exoccipital; fm, foramen magnum; fr, frontal; I, upper incisor; ip, interparietal; j, jugal; la, lacrimal; mx, maxilla; na, nasal; nc, nuchal crest; op, opisthotic; pa, parietal; PC, upper postcanine; pmx, premaxilla; po, postorbital; pof, postfrontal; prf, prefrontal; ptf, post-temporal fenestra; q, quadrate; qj, quadratojugal; rla, reflected lamina of angular; rp, retroarticular process; sa, surangular; smx, septomaxilla; so, supraoccipital; sq, squamosal; ss, squamosal sulcus; st, stapes; ta, tabular.

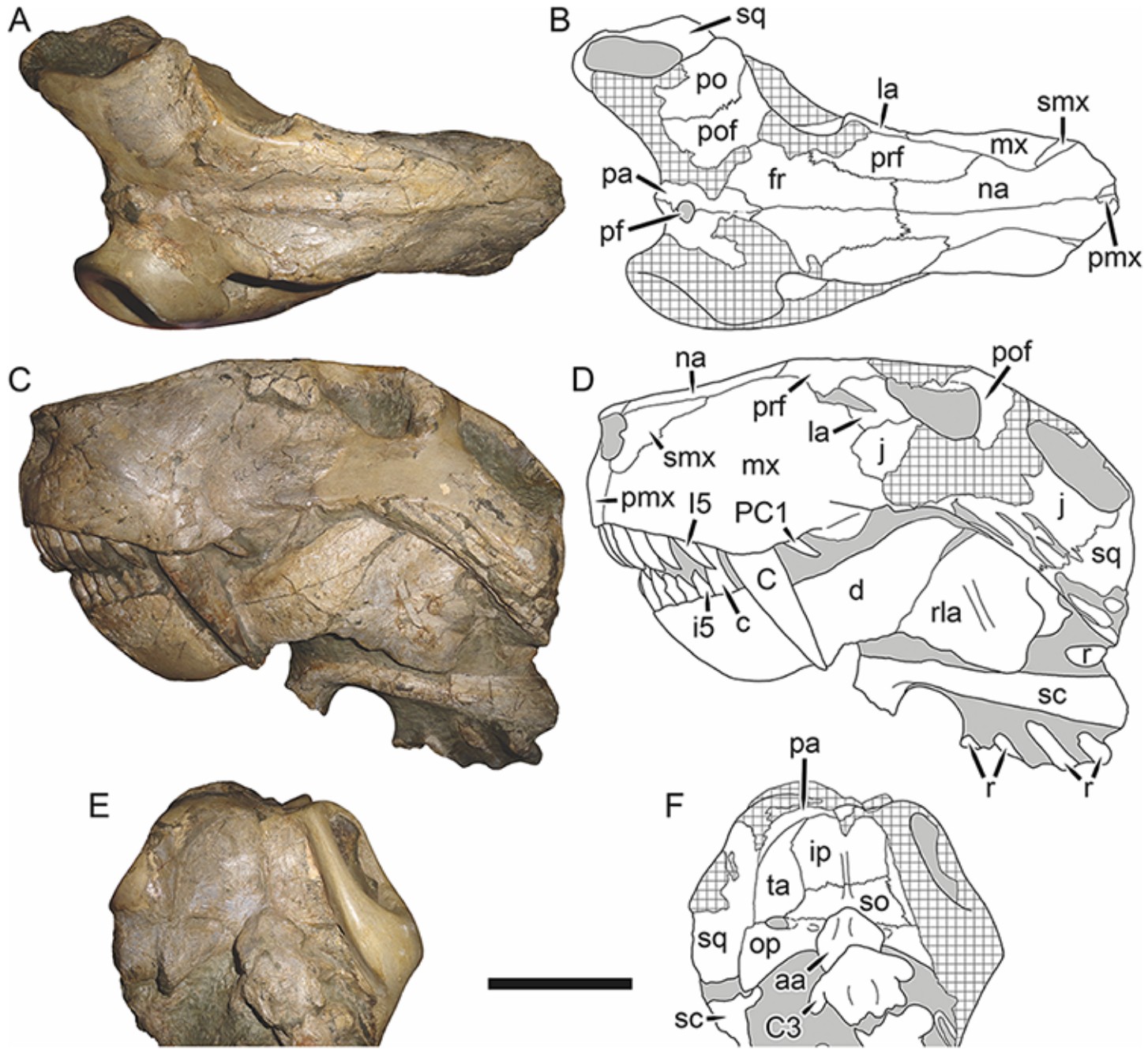

**Figure 56 Holotype (RC 62) of *Smilesaurus ferox Broom, 1948* in (A) dorsal, (C) left lateral, and (E) occipital view (with (B) (D) and (F) interpretive drawings).** Abbreviations: aa, atlas-axis complex; c, lower canine; C, upper canine; C3, third cervical vertebra; d, dentary; fr, frontal; i, lower incisor; I, upper incisor; ip, interparietal; j, jugal; la, lacrimal; mx, maxilla; na, nasal; op, opisthotic; pa, parietal; PC, upper postcanine; pf, pineal foramen; pmx, premaxilla; po, postorbital; pof, postfrontal; prf, prefrontal; r, rib; rla, reflected lamina of angular; sc, scapula; smx, septomaxilla; so, supraoccipital; sq, squamosal; ta, tabular. Gray indicates matrix, hatching indicates plaster. Scale bar equals 10 cm.

*Comments*: *Broom (1948)* described three similar species of large, 'sabre-toothed' gorgonopsian (*Smilesaurus ferox*, *S. maccabei*, and *Pardocephalus wallacei*), which *Sigogneau (1970)* considered conspecific. She tentatively referred this species to the genus *Arctops*, as ?*Arctops ferox*. *Gebauer (2007)* maintained the validity of this species, but

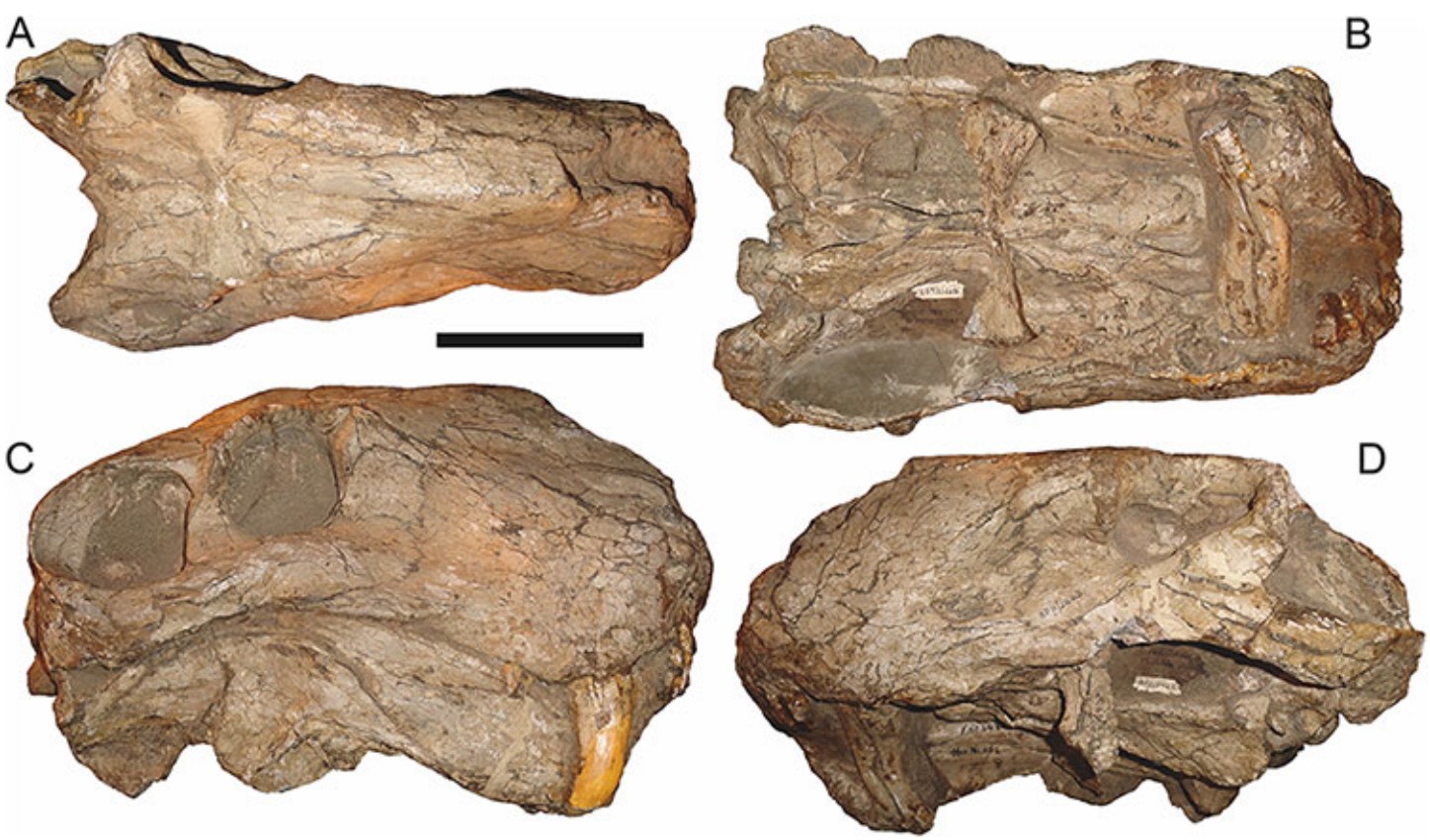

**Figure 57 Referred specimen (BP/1/2465) of *Smilesaurus ferox Broom, 1948* in (A) dorsal, (B) ventral, (C) right lateral, and (D) left lateral view.** Scale bar equals 10 cm.

transferred it to *Aelurognathus*, based on its large size, broad intertemporal region, absence of teeth on the transverse process of the pterygoid, short, broad palatine, large ectopterygoid, and elongate basal tubera. *Norton (2012)* went further, considering *A. ferox* to be a junior synonym of *Aelurognathus tigriceps*.

*Gebauer's (2007)* characters linking this taxon with *Aelurognathus* are broadly distributed throughout Rubidgeinae. *Smilesaurus ferox* is clearly not referable to *Aelurognathus*: it differs from that taxon in the narrow postorbital bar, extremely short parasphenoid rostrum, longer, lower snout, shorter dentary, significant frontal contribution to the orbital rim, reduced (usually two, sometimes three) upper postcanine tooth count, larger canine, and smaller temporal fenestra.

The parasphenoid morphology of *S. ferox* (extremely short, without blade-like rostrum; see Fig. 57B) is herein considered a rubidgeine synapomorphy, but given the numerous differences between this species and other rubidgeines, the genus *Smilesaurus* is resurrected for this species. *Smilesaurus ferox* is similar to *Rubidgea* in the reduced postcanine count (all specimens of *Smilesaurus* have two to three postcanines), but strikingly different in the total lack of cranial pachyostosis and very narrow postorbital bar, as in non-rubidgeine gorgonopsians. Uniquely among gorgonopsians, the pineal boss in *Smilesaurus* is situated in a distinct diamond-shaped depression (Figs. 57A, 58A and

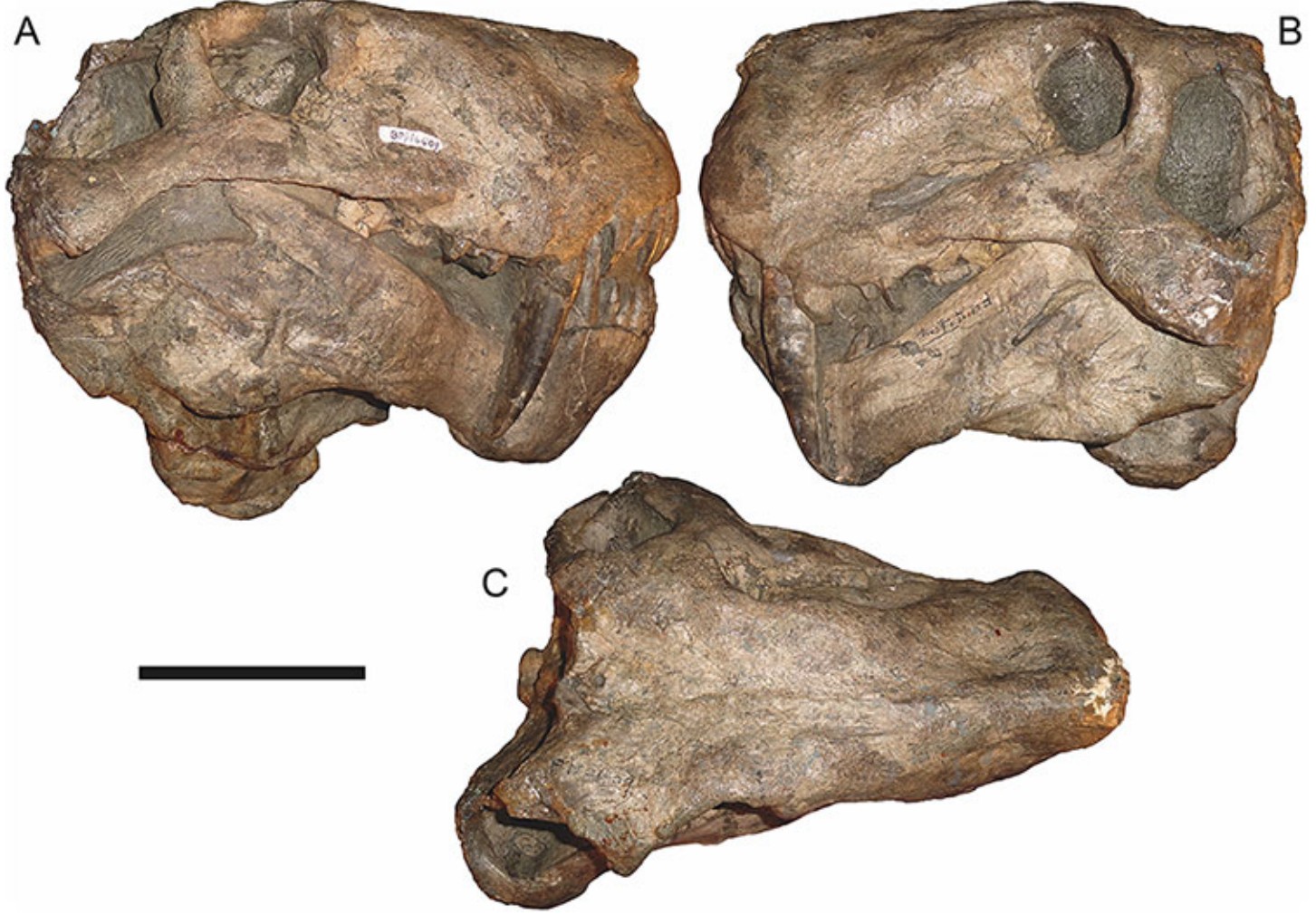

**Figure 58 Referred specimen (BP/1/4409) of *Smilesaurus ferox Broom, 1948* in (A) right lateral, (B) left lateral, and (C) dorsal view.** Scale bar equals 10 cm.

60A), with prominent furrows converging backwards around it that separate the pineal boss from the occiput.

The holotype of *Smilesaurus ferox* (RC 62) is a large (~31 cm basal length) mostly-complete but strongly distorted skull and lower jaws, with seven rib fragments and a scapula pressed up against the left side of the skull (Fig. 56). A single left and two right postcanines are present. The reflected lamina is remarkably deep in this specimen (9.3 cm high at left, 8.8 cm high at right, but damaged at edge). The second species of *Smilesaurus*, *S. maccabei*, is represented by a larger specimen (Fig. 60) that does not differ substantively from the type of *S. ferox*. The holotype (RC 81) has two postcanines on each side of the skull. The holotype of *Pardocephalus wallacei* (RC 82) is an incomplete skull (29.5 cm basal length) missing much of the dorsal surface, but with a well-preserved left mandibular ramus and intact (but poorly-prepared) palate (Fig. 61). Only a single postcanine is visible on each side, and this count can be taken with some confidence given Broom's typically zealous overpreparation of the alveolar margin (the maxilla is ground

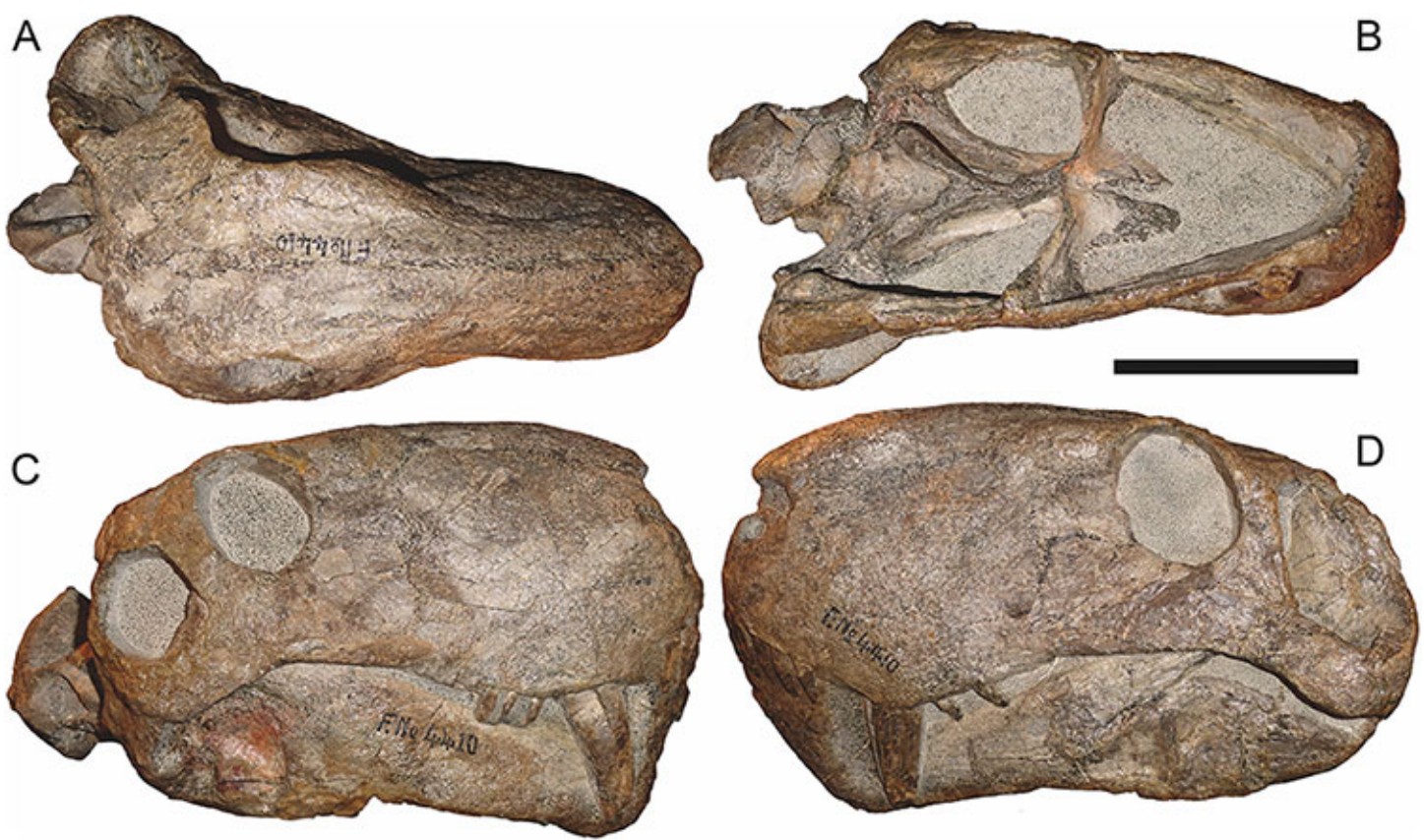

**Figure 59 Referred specimen (BP/1/4410) of *Smilesaurus ferox Broom, 1948* in (A) dorsal, (B) ventral, (C) right lateral, and (D) left lateral view.** Scale bar equals 10 cm.

down around the right tooth, and no other roots are visible). The zygomatic process of RC 82 is exceedingly deep suborbitally (5.3 cm) given the overall size of the skull, which is the reason the plaster-restored orbit on this specimen is so small. Although the orbit is unusually small in *Smilesaurus* in general, the reconstruction is exaggerated in RC 82.

BP/1/4410 is the smallest identifiable skull of *Smilesaurus* (23.0 cm dorsal skull length). This specimen (Fig. 59) shows similar proportions to presumed adults (e.g., BP/1/2465, Fig. 57; BP/1/4409, Fig. 58) and has three upper postcanines (PC1–3 on the right side, PC1 and 3 on the left). Intriguingly, this specimen shows a distinct preparietal bone, which is narrow and lenticular in shape and separated from the pineal foramen by a short (0.8 cm) mid-parietal suture. Whether this indicates loss of the preparietal with ontogeny in *Smilesaurus* or merely individual variation is unknown, although its absence in all larger specimens suggests the former. The position and shape of the preparietal is extremely similar to the condition in *Arctops*, and as *Sigogneau (1970)* noted, there are multiple similarities between these taxa. Besides preparietal morphology, the intertemporal width is very similar between them, and *Arctops* specimens also have three upper postcanines. The type species of *Arctops*, *A. willistoni*, is represented by a very poor back of a skull (NHMUK R4099) in need of further preparation. However, several characters of *Arctops willistoni* suggest that it is not conspecific with *Smilesaurus ferox*.

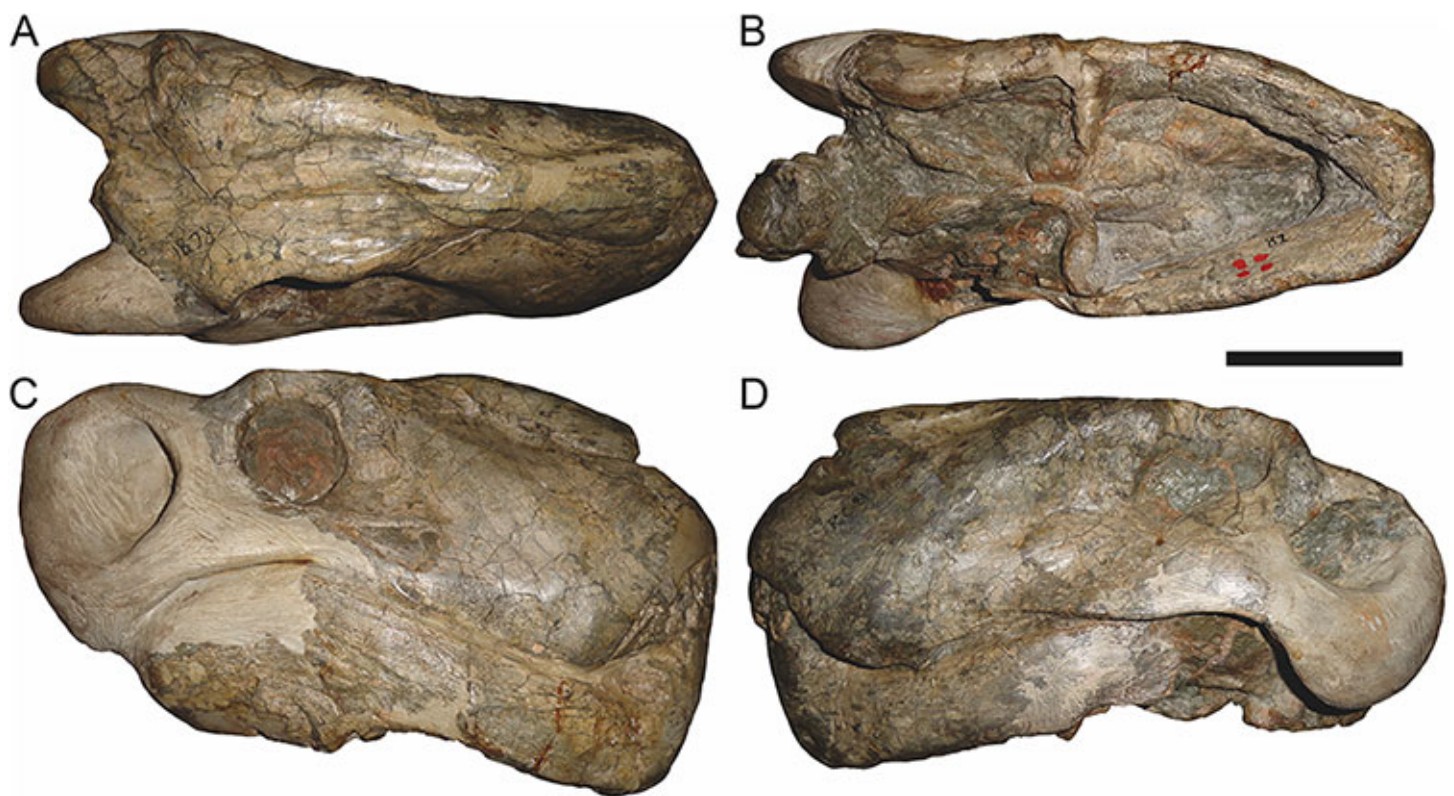

**Figure 60 Referred specimen (RC 81) of _Smilesaurus ferox Broom, 1948_ in (A) dorsal, (B) ventral, (C) right lateral, and (D) left lateral view.** Holotype of _Smilesaurus maccabei Broom, 1948_. Scale bar equals 10 cm.

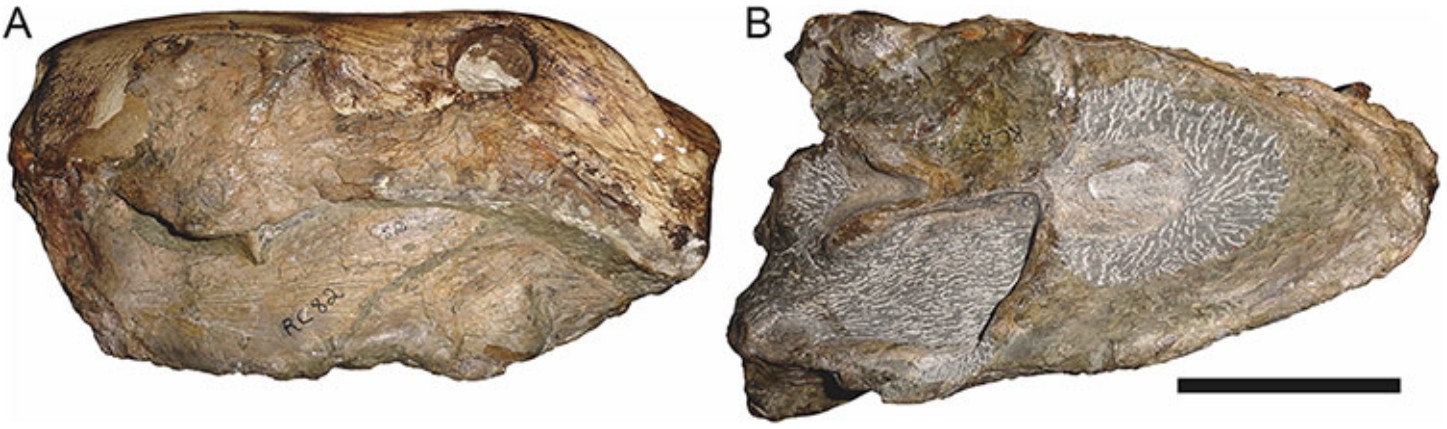

**Figure 61 Referred specimen (RC 82) of _Smilesaurus ferox Broom, 1948_ in (A) left lateral and (B) ventral view.** Holotype of _Pardocephalus wallacei Broom, 1948_. Scale bar equals 10 cm.

The temporal fenestra of _A. willistoni_ is significantly longer, proportionally, than in _S. ferox_. This cannot be explained by size differences between specimens: the holotype of _Arctops watsoni_ (BP/1/698, which is identical in preserved skull morphology to that of _A. willistoni_ and should, as _Sigogneau-Russell (1989)_ suggested, be considered synonymous with it) is of comparable size (23.5 cm dorsal skull length) to BP/1/4410, but in BP/1/698

the distance from the base of the postorbital bar to the rear edge of the fenestra is 7.8 cm, whereas in BP/1/4410 it is 4.0 cm. Nor can it be attributed to taphonomic distortion, as both skulls have undergone relatively little crushing. NHMUK R4099 is only slightly larger than BP/1/698, and shows the same temporal proportions, whereas large *Smilesaurus* skulls have proportionally small temporal fenestrae. Furthermore, BP/1/698 exhibits a long parasphenoid rostrum with relatively short basal tubera (rostrum length 6.3 cm, left basal tuber length 3.3 cm, versus 3.9 cm parasphenoid and 3.7 left tuber lengths in BP/1/4410). Given these differences, it is most parsimonious to consider *Smilesaurus* and *Arctops* distinct taxa. *Arctops* may be related to rubidgeines among gorgonopsians, but this will have to be tested in the context of a complete redescription of *A. willistoni*.

### *Sycosaurus* Haughton, 1924

*Tetraodon* Broili & Schröder, 1936:325 (*non* Linnaeus, 1758)
*Tetraodontonius* Kuhn, 1961:79
*Cephalicustriodus* Parrington, 1974:51

*Type species*: *Sycosaurus laticeps* Haughton, 1924.
   *Included species*: *Sycosaurus nowaki* (Broili & Schröder, 1936) **comb. nov.**
   *Diagnosis*: A large gorgonopsian (up to ~38 cm basal skull length) distinguished from all other rubidgeines by the following autapomorphies: elongate interchoanal body of vomer with relatively gradual transverse expansion anteriorly (reversal to pre-rubidgeine condition), relatively small canines, and narrow zygomatic arch that is tightly constricted beneath the postorbital bar. Also diagnosed by the unique combination of a relatively long snout, numerous, needle-like teeth in a single row on the palatine boss, and pterygoid palatal boss reduced to a thin, edentulous ridge (shared with *Clelandina*, *Dinogorgon*, *Leontosaurus*, and *Rubidgea*). The vomerine morphology of this genus is also unusual in having greater separation than usual between the three vomerine ridges, with the main eminence of the central ridge further forward than in, e.g., *Ruhuhucerberus*.

   *Comments*: *Sycosaurus* was one of the first rubidgeine genera to be named, and has a confused history. It has served as a wastebasket for numerous disparate rubidgeine specimens over the years, including specimens currently referred to *Aelurognathus*, *Leontosaurus*, and *Ruhuhucerberus*. Sigogneau (1970) included three species in *Sycosaurus*: *S. laticeps* (the type), ?*S. kingoriensis*, and *S. vanderhorsti*. Gebauer (2007) recognized four species of *Sycosaurus*: *S. laticeps* (including *Leontosaurus vanderhorsti* as a junior synonym), *S. kingoriensis* (including *Aelurognathus haughtoni* as a junior synonym), *S. terror*, and ?*S. intactus*. Additionally, she considered the holotypes of *Leontocephalus cadlei* and *Broomisaurus rubidgei* to represent specifically indeterminate specimens of *Sycosaurus*.

   My examination of this material indicates that two species of *Sycosaurus* can be recognized: *S. laticeps* in South Africa and Zambia and *S. nowaki* comb. nov. in Tanzania. Members of the genus *Sycosaurus* can be distinguished from similar taxa such as *Aelurognathus* and *Leontosaurus* by the extreme constriction of the zygomatic arch below the postorbital bar and the relatively elongate expanded interchoanal body of the vomer.

This diagnosis excludes *S. vanderhorsti* (=*Leontosaurus vanderhorsti*), *S. haughtoni* and *S. terror* (=*Ruhuhucerberus haughtoni*), *Leontocephalus cadlei* (=*Aelurognathus tigriceps*), and *Broomisaurus rubidgei*. RC 19, the holotype of *Broomisaurus rubidgei* Broom, 1940a, is a problematic specimen that has been extensively reconstructed with plaster. RC 19 represents a short-snouted gorgonopsian with a very large preparietal and anteroposteriorly narrow postorbital bar. These features indicate that it is not referable to *Sycosaurus* and is likely not even a rubidgeine. This specimen requires further study, but will not be considered further in this contribution.

### *Sycosaurus laticeps* Haughton, 1924
### (Reconstruction Figs. 62–63, Specimen Figs. 64–67)

*Holotype*: SAM-PK-4022, a somewhat distorted, complete skull (Fig. 64) from Zuurplaas, Graaff-Reinet, South Africa.

*Referred specimens*: BP/1/1565 (Fig. 65; a dorsoventrally crushed skull and partial lower jaws from Ringsfontein, Murraysburg, South Africa); BP/1/3465 (Fig. 66; a complete skull from Drysdall & Kitching's (1963) Locality 5, Upper Luangwa Valley, Zambia); GPIT/RE/7134 (Fig. 67; an isolated snout from Zuurplaas, Graaff-Reinet, South Africa); NMQR 3535 (a dorsoventrally crushed skull from Vaalkop (Grampian Hills), Free State, South Africa).

*Diagnosis*: A species of *Sycosaurus* that can be distinguished from *S. nowaki* by the presence of 4–5 postcanines, a thick, robust postorbital bar throughout its length, more robust zygomatic portion of squamosal, dorsal margin of snout with downward slope, and transversely narrower palatines behind the canine.

*Comments*: The holotype of *S. laticeps* (SAM-PK-4022) is a small skull (23.8 cm basal skull length), collected by the Rev. J. H. Whaits in 1917. This specimen lacks pachyostosis, although it already shows the anteroposteriorly wide postorbital bar and large, deflected temporal arch typical of rubidgeines. The preparietal is absent, contrary to previous reconstructions (e.g., Sigogneau, 1970)—there is a clear suture at the anterior edge of the pineal foramen that continues forward and indicates the interparietal suture (Fig. 64). The palate is poorly preserved, and the palatine bosses are worn off. From what is left of the palatal bosses of the pterygoid it is clear that they were narrow and elongate. A single tooth root is present on the left palatal boss of the pterygoid, but the transverse processes of the pterygoid are edentulous. The frontal is excluded from the orbital margin by a prefrontal-postfrontal contact slightly anterior to the orbital midpoint. The prefrontal extends outwards, producing a slight brow ridge. The snout is proportionally long (12.4 cm) and narrow. The canine roots in the holotype are remarkably narrow and blade-like, although the shearing of the skull may have distorted them.

The small size of SAM-PK-4022 suggests that it is a juvenile specimen, which complicates comparison with other rubidgeines (and indeed, likely accounts for much of the confusion as to which taxa are referable to *Sycosaurus*). The unique combination of characters exhibited by this skull is present in a handful of other specimens, however. Huene (1938) referred an isolated snout (GPIT/RE/7134; Fig. 67) to *Sycosaurus laticeps*,

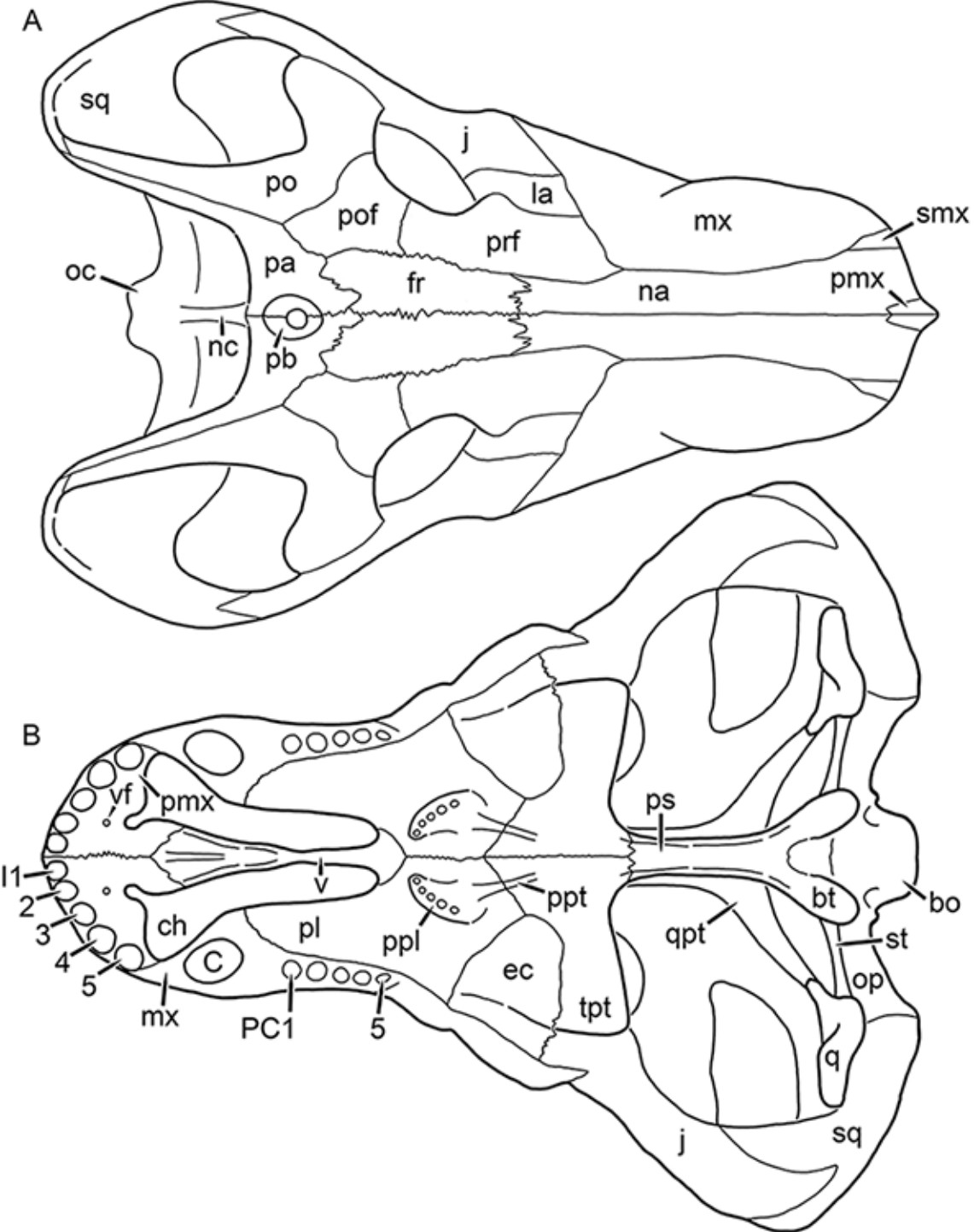

**Figure 62 Reconstruction of the skull of *Sycosaurus laticeps Haughton, 1924* in (A) dorsal and (B) ventral views.** Reconstructions based primarily on BP/1/1565 and SAM-PK-4022. Abbreviations: bo, basioccipital; bt, basal tuber; C, upper canine; ch, choana; ec, ectopterygoid; fr, frontal; I, upper incisor; j, jugal; mx, maxilla; na, nasal; nc, nuchal crest; oc, occipital condyle; op, opisthotic; pa, parietal; pb, pineal boss; PC, upper postcanine; pl, palatine; pmx, premaxilla; po, postorbital; pof, postfrontal; ppl, palatal boss of palatine; ppt, palatal boss of pterygoid; prf, prefrontal; ps, parasphenoid; q, quadrate; qpt, quadrate ramus of pterygoid; smx, septomaxilla; sq, squamosal; st, stapes; tpt, transverse process of pterygoid; v, vomer; vf, ventral premaxillary foramen.

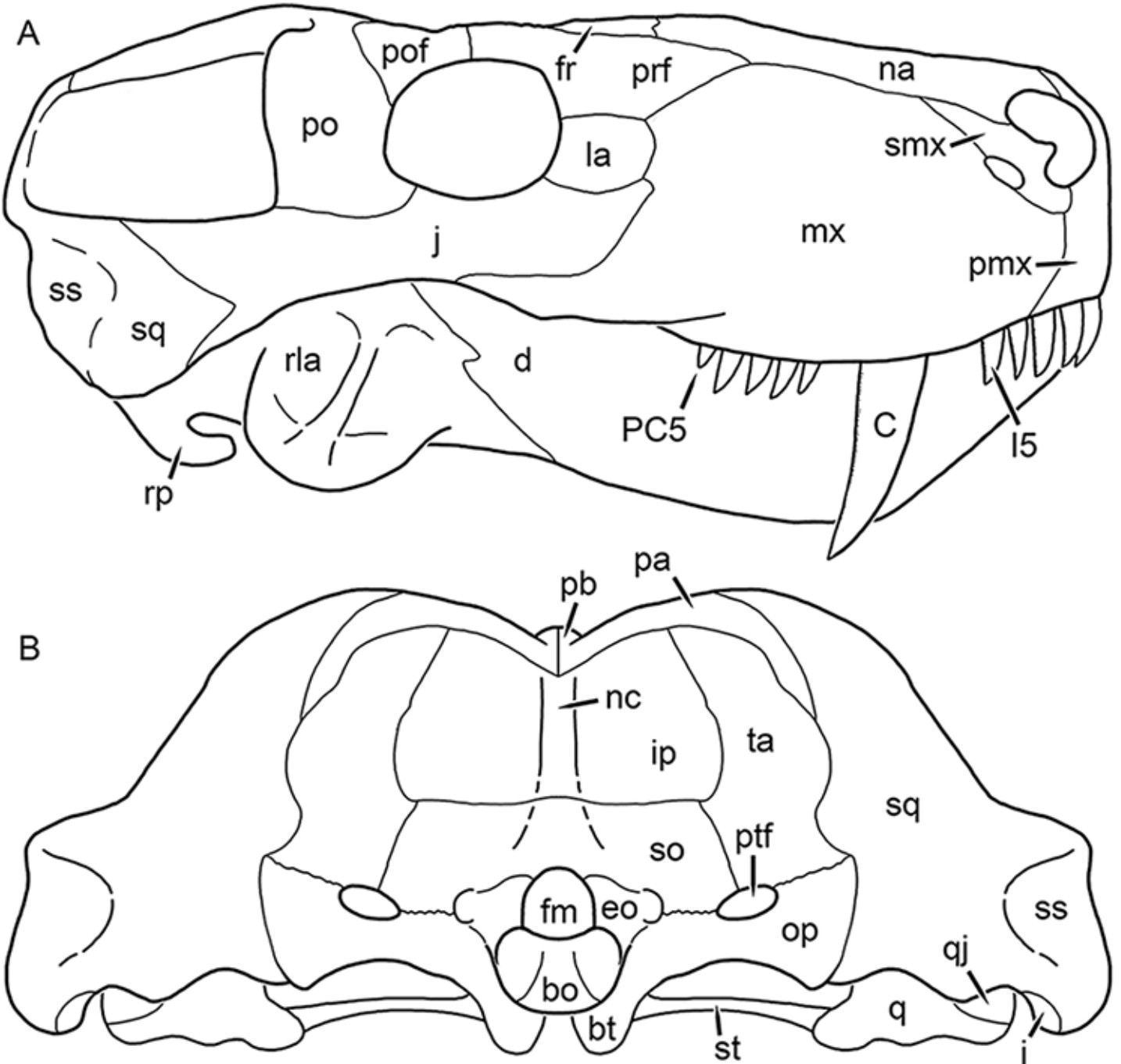

**Figure 63 Reconstruction of the skull of *Sycosaurus laticeps Haughton, 1924* in (A) lateral and (B) occipital views.** Reconstructions based primarily on BP/1/1565, BP/1/3465, and SAM-PK-4022. Abbreviations: bo, basioccipital; bt, basal tuber; C, upper canine; d, dentary; eo, exoccipital; fm, foramen magnum; fr, frontal; I, upper incisor; ip, interparietal; j, jugal; la, lacrimal; mx, maxilla; na, nasal; nc, nuchal crest; op, opisthotic; pa, parietal; pb, pineal boss; PC, upper postcanine; pmx, premaxilla; po, postorbital; pof, postfrontal; prf, prefrontal; q, quadrate; qj, quadratojugal; rla, reflected lamina of angular; rp, retroarticular process; smx, septomaxilla; so, supraoccipital; sq, squamosal; ss, squamosal sulcus; st, stapes; ta, tabular.

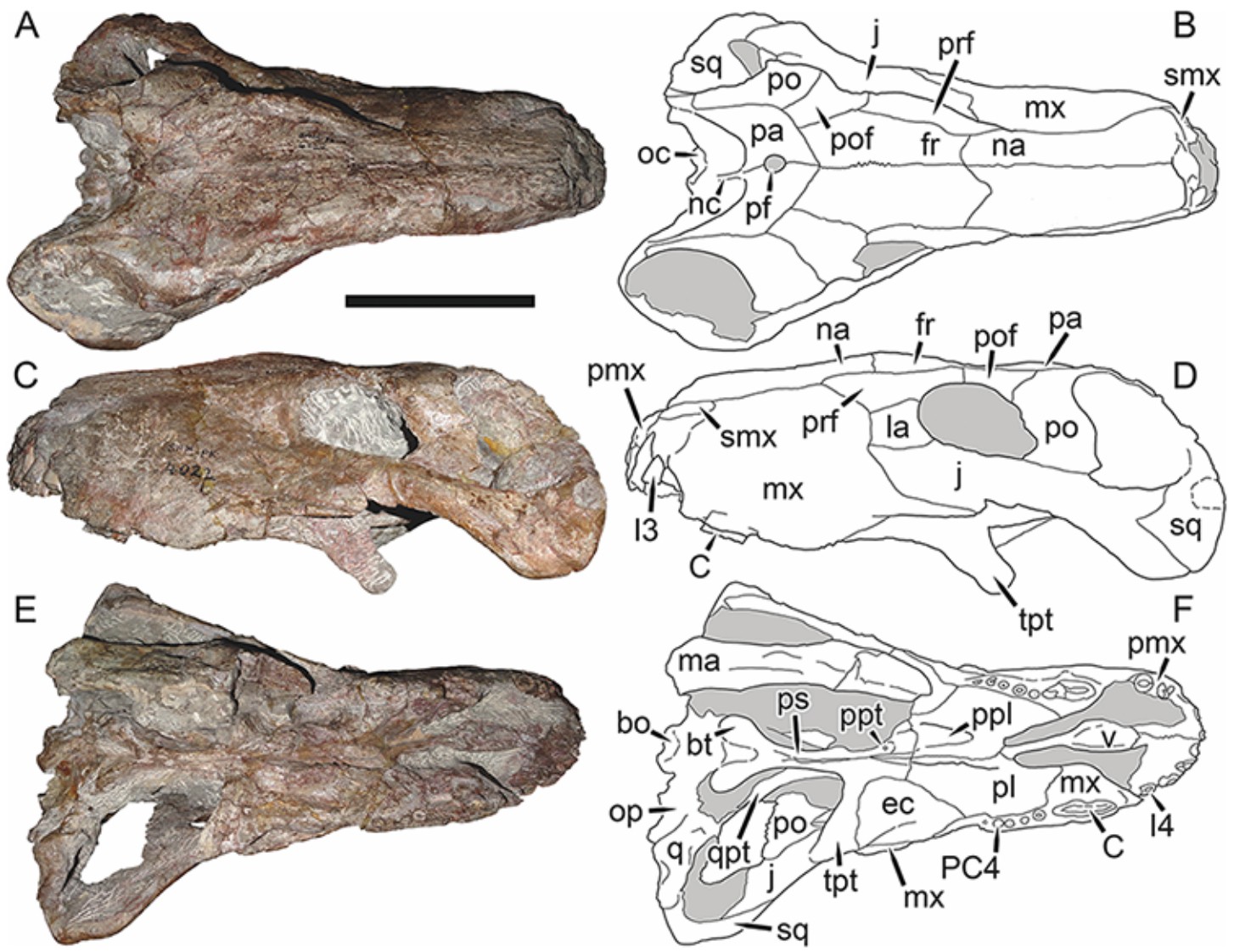

**Figure 64** Holotype (SAM-PK-4022) of *Sycosaurus laticeps Haughton, 1924* in (A) dorsal, (C) left lateral, and (E) ventral view (with (B) (D) and (F) interpretive drawings). Abbreviations: bo, basioccipital; bt, basal tuber; C, upper canine; ec, ectopterygoid; fr, frontal; I, upper incisor; j, jugal; la, lacrimal; ma, mandible; mx, maxilla; na, nasal; nc, nuchal crest; oc, occipital condyle; op, opisthotic; pa, parietal; PC, upper postcanine; pf, pineal foramen; pmx, premaxilla; po, postorbital; pof, postfrontal; ppl, palatal boss of palatine; ppt, palatal boss of pterygoid; prf, prefrontal; ps, parasphenoid; q, quadrate; qpt, quadrate ramus of pterygoid; smx, septomaxilla; sq, squamosal; tpt, transverse process of pterygoid; v, vomer. Gray indicates matrix. Scale bar equals 10 cm.

and this referral is upheld here, based on the presence of four postcanines, a relatively small canine, an elongate expanded interchoanal body of the vomer, and narrow palatines compared to Tanzanian *Sycosaurus* material. More informative is a complete skull from the Luangwa Valley of Zambia (BP/1/3465; Fig. 66), which exhibits all of the features listed above as well as the same morphology of the postorbital bar and zygomatic arch as SAM-PK-4022. Importantly, the larger size of this specimen (38.6 cm basal skull length) indicates that the lack of pachyostosed bosses in the holotype is not attributable to juvenile status, but is characteristic of the species in general.

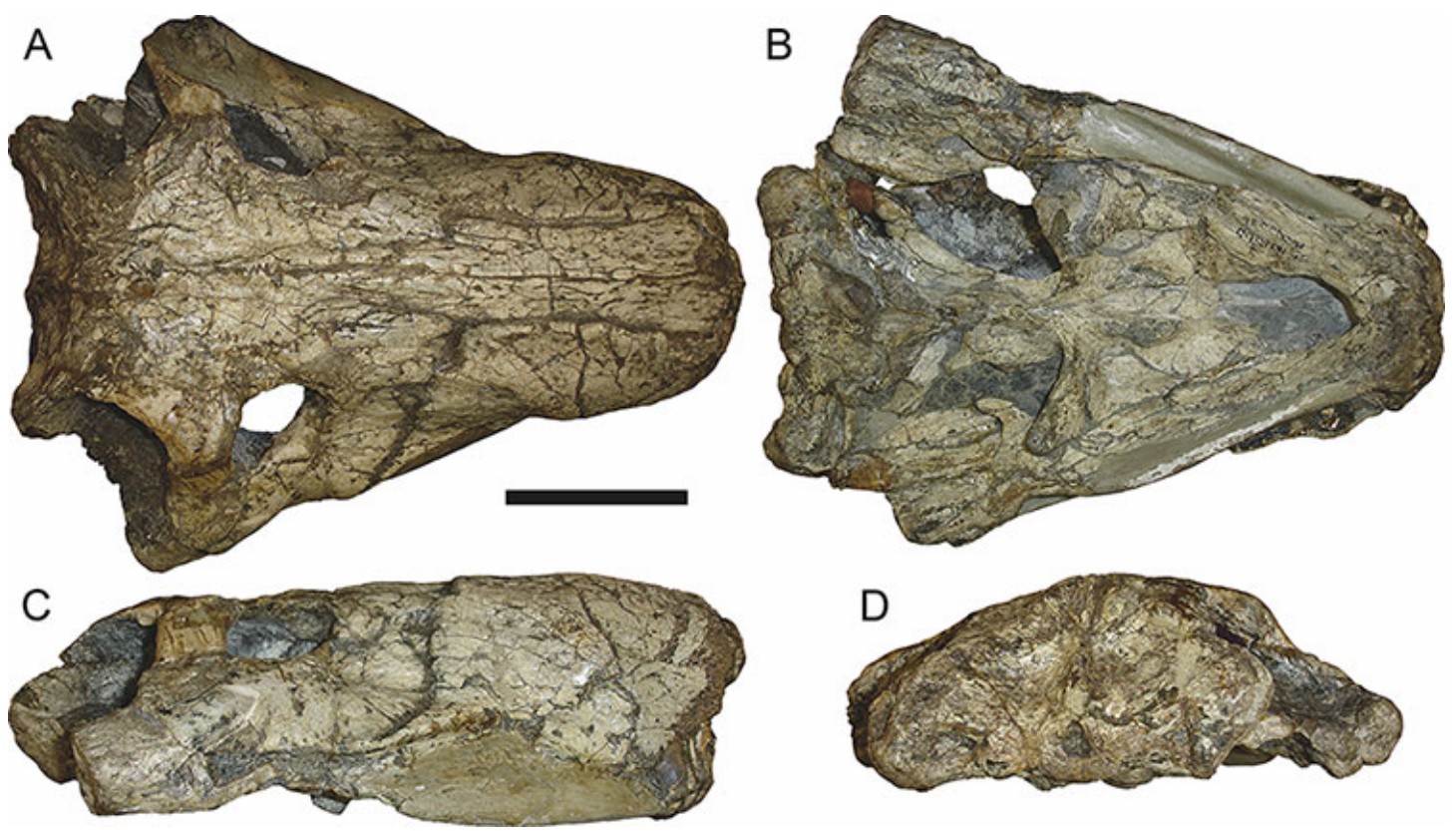

**Figure 65 Referred specimen (BP/1/1565) of *Sycosaurus laticeps Haughton, 1924* in (A) dorsal, (B) ventral, (C) right lateral, and (D) occipital view.** Scale bar equals 10 cm.

### *Sycosaurus nowaki* (*Broili & Schröder, 1936*) comb. nov. (Reconstruction Figs. 68–69, Specimen Figs. 4, 70–73)

*Tetraodon nowaki Broili & Schröder, 1936*:326

*Lycaenops kingoriensis Huene, 1950*:87

*Tetraodontonius nowaki Kuhn, 1961*:79

*Sycosaurus? kingoriensis Sigogneau, 1970*:262

*Leontocephalus intactus Kemp, 1969*:11

*Cephalicustriodus kingoriensis Parrington, 1974*:51 (*partim*)

*Sycosaurus? intactus Gebauer, 2007*:205

*Holotype*: BSPG 1936 III 1, a snout tip (Fig. 70) from Kingori Mountain, Ruhuhu Basin, Tanzania.

    *Referred specimens*: UMZC T877 (Figs. 4 and 71B; a fragmentary, acid-prepared skull and mandible from Stockley's Site B19, between Matamondo and Linyana, Ruhuhu Basin, Tanzania); UMZC T878 (Fig. 72; a complete skull, partial lower jaw, scapulocoracoid, ilium, and other postcranial elements from Stockley's Site B4, Katumbi Viwili, Ruhuhu Basin, Tanzania; holotype of *Leontocephalus intactus*); UMZC T889 (Fig. 71A; a skull in

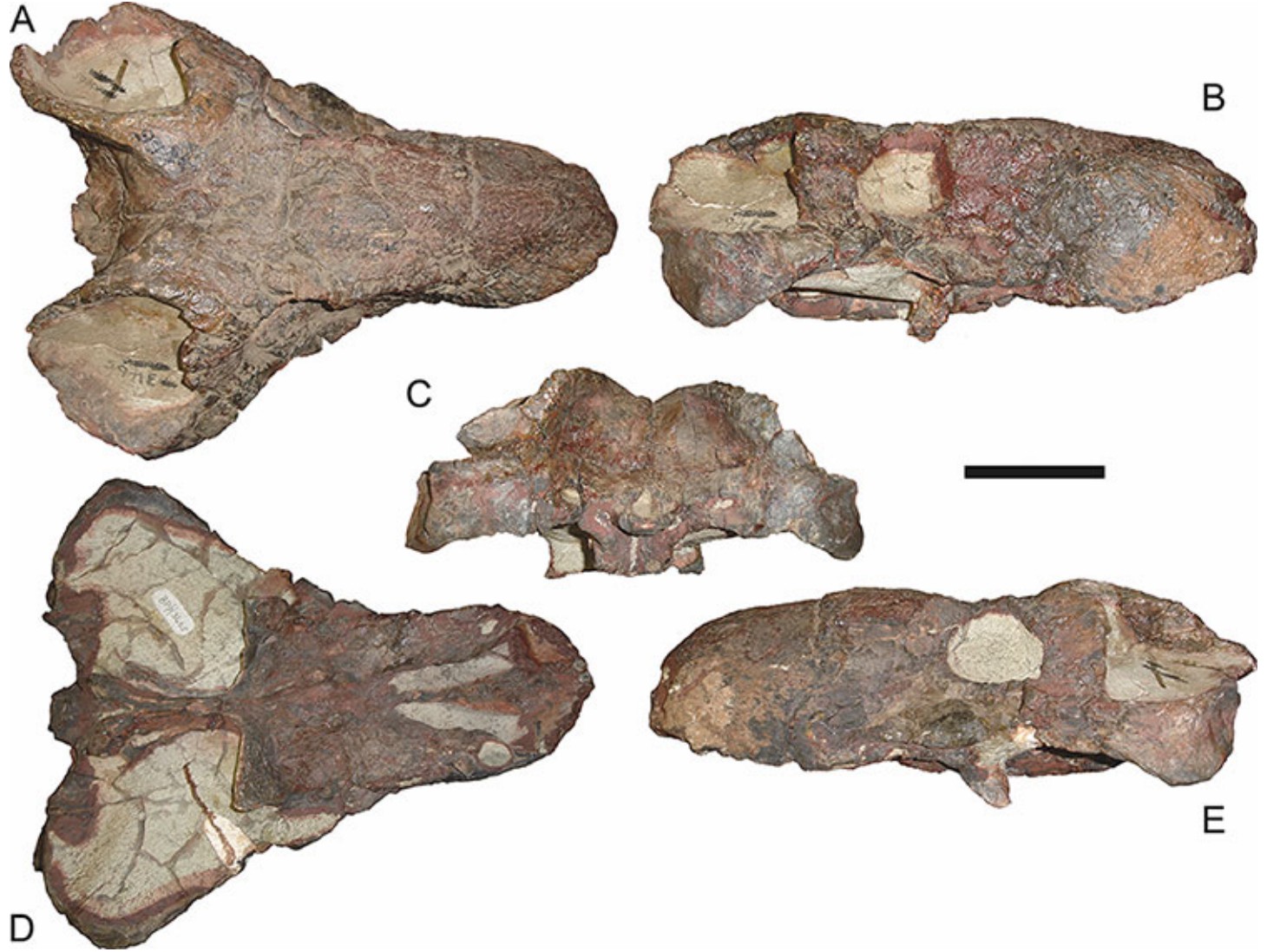

**Figure 66 Referred specimen (BP/1/3465) of *Sycosaurus laticeps Haughton, 1924* in (A) dorsal, (B) right lateral, (C) occipital, (D) ventral, and (E) left lateral view.** Scale bar equals 10 cm.

two pieces, right side badly damaged, from Stockley's Site B16, Matamondo, Ruhuhu Basin, Tanzania); GPIT/RE/7116 (Fig. 73; specimen K47 of *Huene (1950)*, a damaged but largely complete skull from Kingori, Ruhuhu Basin, Tanzania; holotype of *Lycaenops kingoriensis*).

*Diagnosis*: A species of *Sycosaurus* that can be distinguished from *S. laticeps* by the presence of 6–7 postcanines, a shallow postorbital bar with very discrete flange, weaker zygomatic portion of the squamosal, straight dorsal margin of the snout, and transversely broader palatines behind the canines.

*Comments*: *Tetraodon nowaki* was described by *Broili & Schröder (1936)* on the basis of a large, isolated snout (BSPG 1936 III 1; Fig. 70) from Kingori Mountain in the Ruhuhu Basin of Tanzania. The preserved portion of the snout is 20.5 cm long and 10.2 cm wide at

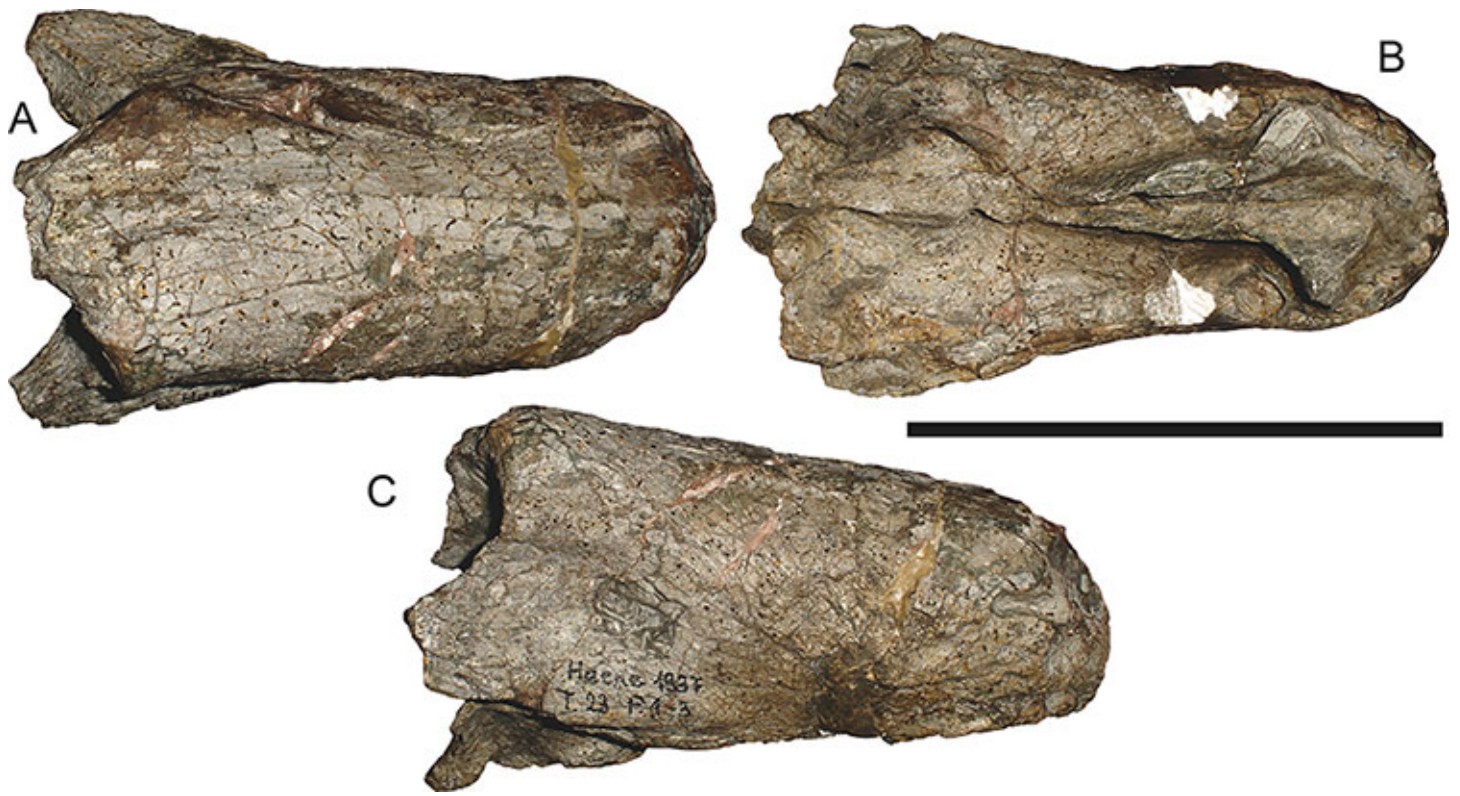

**Figure 67** **Referred specimen (GPIT/RE/7134) of** *Sycosaurus laticeps* *Haughton, 1924* **in (A) dorsal, (B) ventral, and (C) right lateral view.** Scale bar equals 10 cm.

the level of the canines (assuming similar proportions as the *Sycosaurus laticeps* specimen BP/1/3465, the complete skull would be ~33 cm in basal length.) *Broili & Schröder (1936)* diagnosed *T. nowaki* primarily on the basis of its four incisors (as opposed to the usual gorgonopsian five). As the generic name *Tetraodon* was preoccupied by a pufferfish (*Tetraodon Linnaeus, 1758*), *Kuhn (1961)* renamed it *Tetraodontonius*. Following *Broili & Schröder's (1936)* initial description, *Tetraodontonius nowaki* has been largely ignored. *Sigogneau (1970)* regarded this specimen as a *nomen dubium* referable only to Theriodontia *incertae sedis*, but later (*Sigogneau-Russell, 1989*) recognized it as a gorgonopsian, possibly close to *Arctops*. This taxon was not even mentioned by *Gebauer (2007)*, despite her focus on the alpha taxonomy of Tanzanian gorgonopsians.

BSPG 1936 III 1 is highly incomplete and the palatal surface shows significant amounts of wear. In particular, the palatines show extensive surface cracks formed as the snout was weathering out of the rock (Fig. 70A). Luckily, the vomerine surface is very well preserved and well prepared. As in specimens of *S. laticeps* (BP/1/3465, GPIT/RE/7134), the vomer begins its anterior expansion prior to the level of the canine, so the expanded interchoanal body is proportionally much longer than in *Aelurognathus* or *Ruhuhucerberus*. The lateral ridges are tallest posteriorly, right at the point where the vomer begins to expand, and decrease in height anteriorly. The median ridge occurs slightly anterior to the point of expansion, but is relatively low throughout, never as tall as the lateral ridges. Furthermore

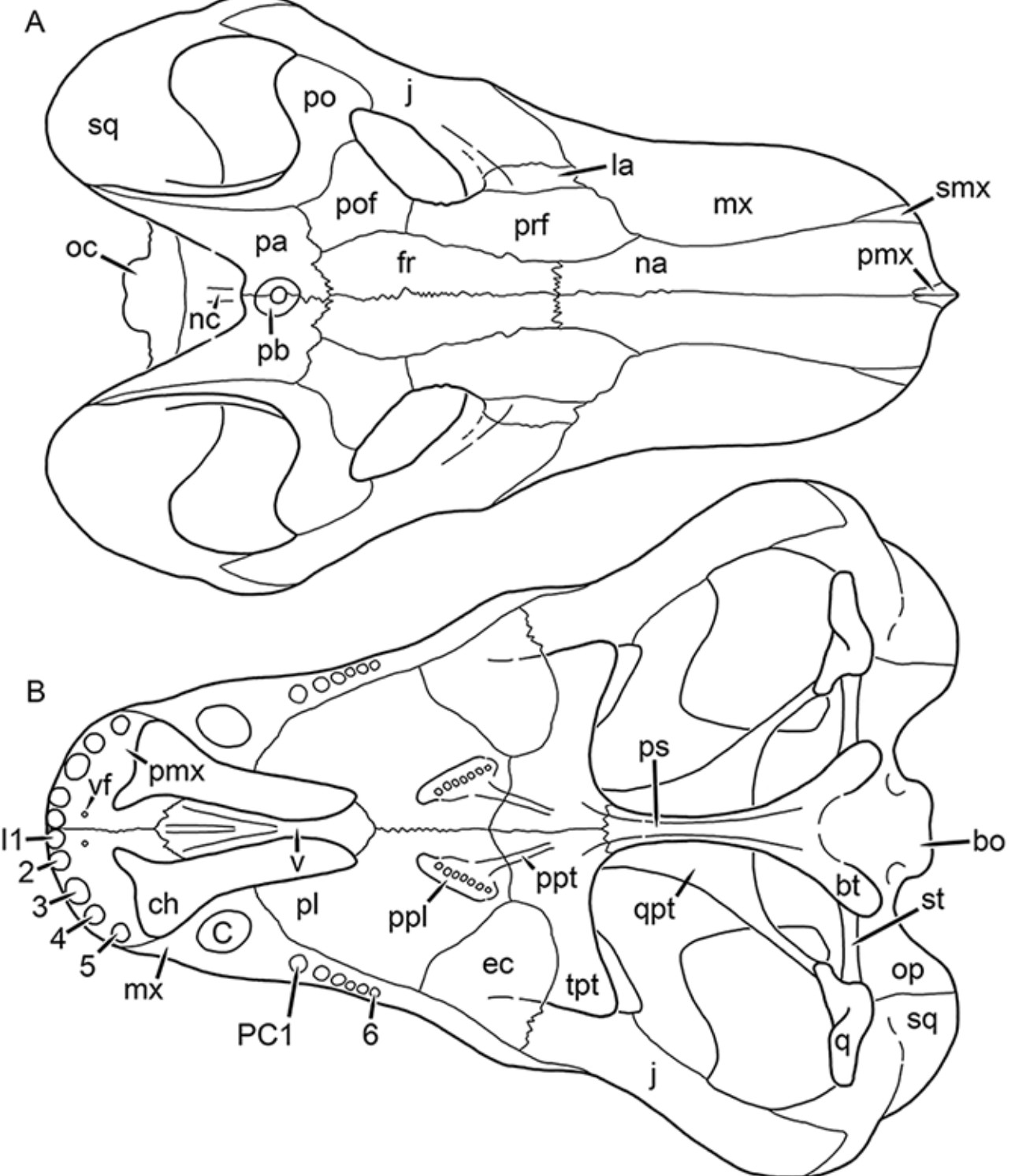

**Figure 68 Reconstruction of the skull of *Sycosaurus nowaki* (*Broili & Schröder, 1936*) in (A) dorsal and (B) ventral views.** Reconstructions based primarily on BSPG 1936 III 1 and UMZC T878. Abbreviations: bo, basioccipital; bt, basal tuber; C, upper canine; ch, choana; ec, ectopterygoid; fr, frontal; I, upper incisor; j, jugal; la, lacrimal; mx, maxilla; na, nasal; nc, nuchal crest; oc, occipital condyle; op, opisthotic; pa, parietal; pb, pineal boss; PC, upper postcanine; pl, palatine; pmx, premaxilla; po, postorbital; pof, postfrontal; ppl, palatal boss of palatine; ppt, palatal boss of pterygoid; prf, prefrontal; ps, parasphenoid; q, quadrate; qpt, quadrate ramus of pterygoid; smx, septomaxilla; sq, squamosal; st, stapes; tpt, transverse process of pterygoid; v, vomer; vf, ventral premaxillary foramen.

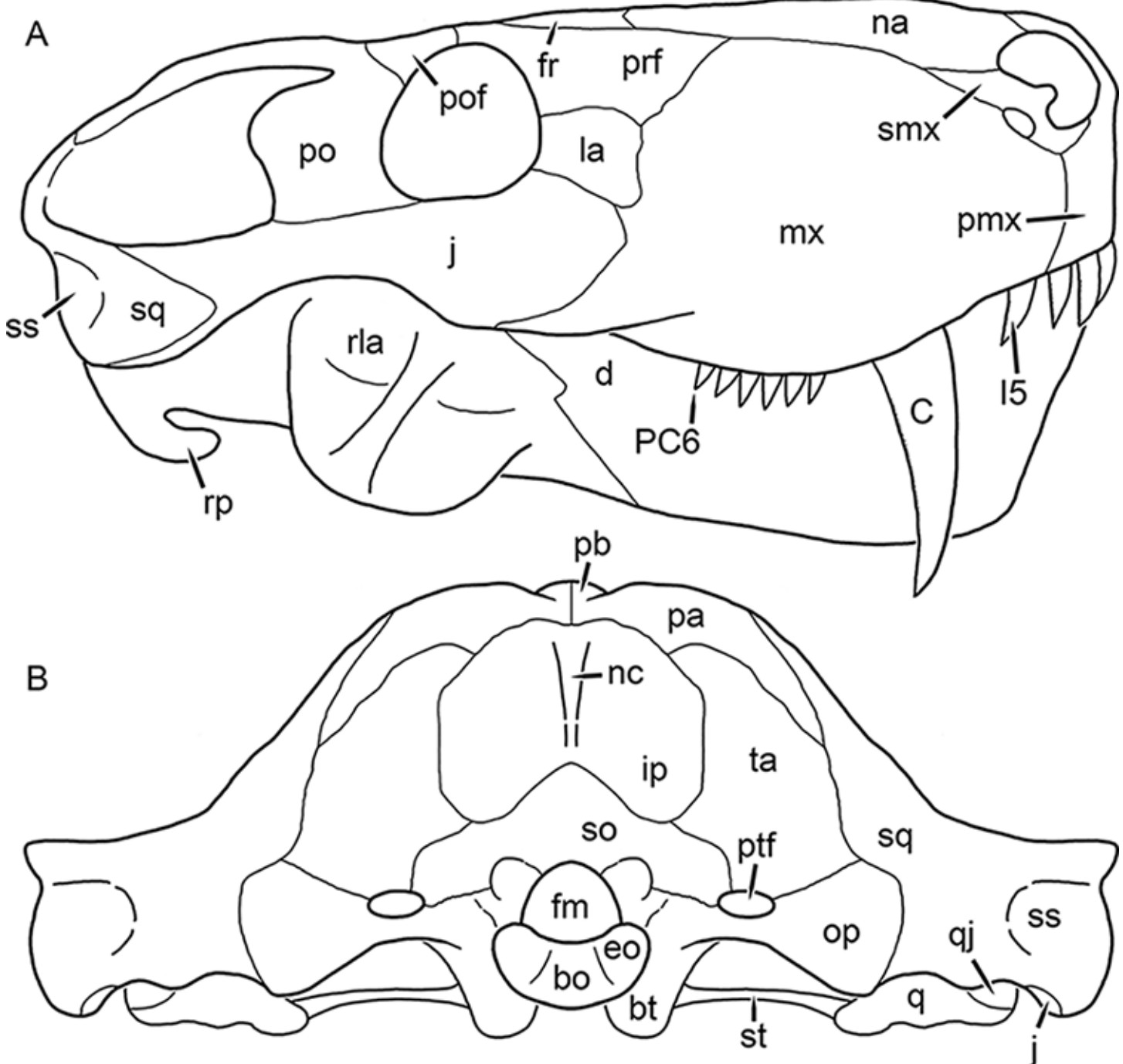

**Figure 69 Reconstruction of the skull of *Sycosaurus nowaki* (*Broili & Schröder, 1936*) in (A) lateral and (B) occipital views.** Reconstructions based primarily on UMZC T878 and UMZC T889. Abbreviations: bo, basioccipital; bt, basal tuber; C, upper canine; d, dentary; ec, ectopterygoid; eo, exoccipital; fm, foramen magnum; fr, frontal; I, upper incisor; ip, interparietal; j, jugal; la, lacrimal; mx, maxilla; na, nasal; nc, nuchal crest; op, opisthotic; pa, parietal; pb, pineal boss; PC, upper postcanine; pmx, premaxilla; po, postorbital; pof, postfrontal; prf, prefrontal; ptf, post-temporal fenestra; q, quadrate; qj, quadratojugal; rla, reflected lamina of angular; rp, retroarticular process; smx, septomaxilla; so, supraoccipital; sq, squamosal; ss, squamosal sulcus; ta, tabular.

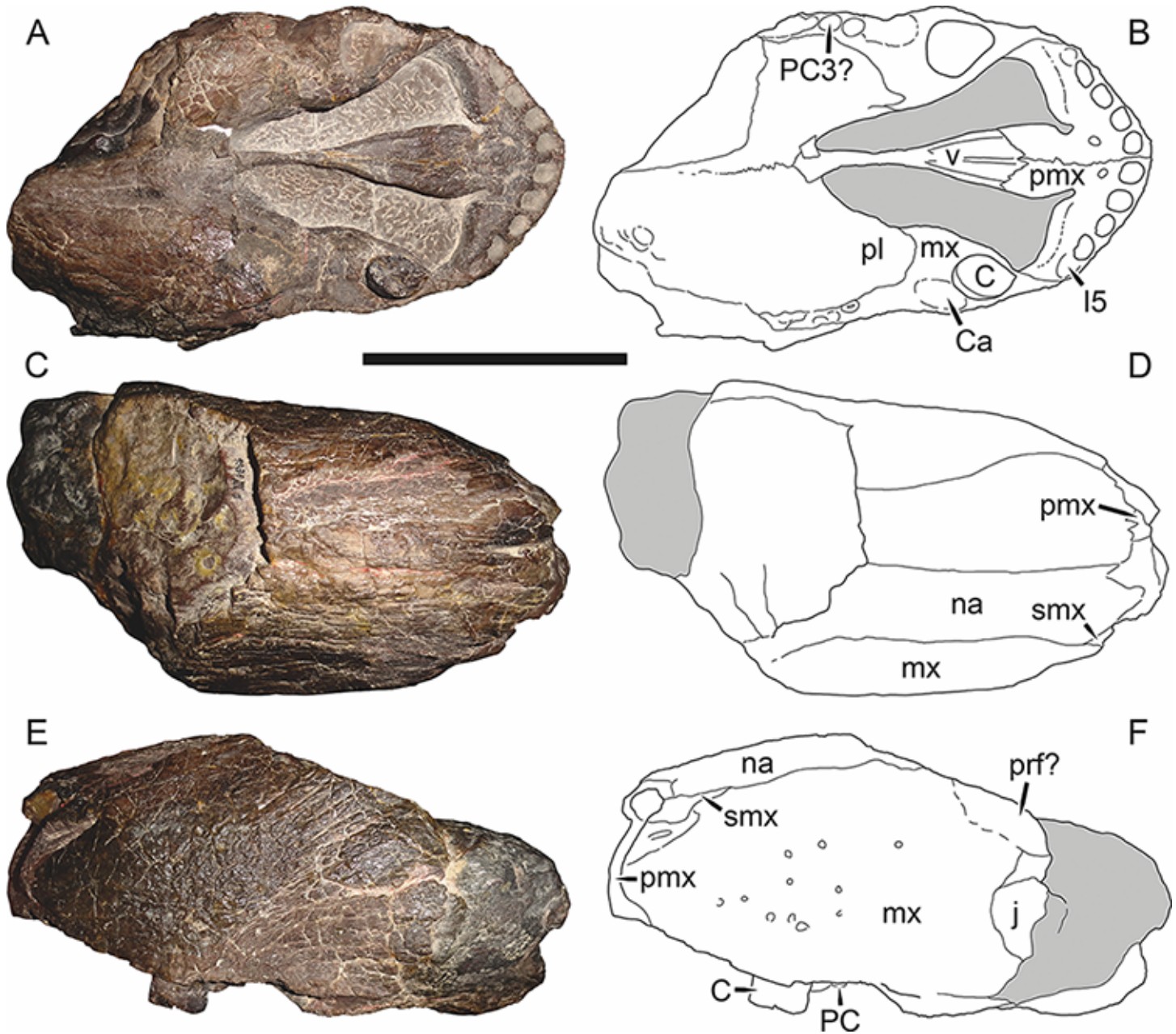

**Figure 70 Holotype (BSPG 1936 III 1) of *Sycosaurus nowaki* (*Broili & Schröder, 1936*) in (A) ventral, (C) dorsal, and (E) left lateral view (with (B) (D) and (F) interpretive drawings).** Abbreviations: C, upper canine; Ca, canine alveolus; I, upper incisor; j, jugal; mx, maxilla; na, nasal; PC, upper postcanine; pmx, premaxilla; prf, prefrontal; smx, septomaxilla; v, vomer. Gray indicates matrix. Scale bar equals 10 cm.

it decreases in height anteriorly near the border with the premaxilla, unlike in *Ruhuhucerberus* in which it is tall and well-developed right at the vomerine-premaxillary suture. As in most gorgonopsians, the vomerine processes of the premaxilla extend posterolaterally and have a distinct invagination on their anterolateral edge, where they contact the premaxillary palatal plate.

*Tetraodon* was named for its supposed possession of only four incisors, an unusual condition in gorgonopsians (although not unique: *Inostrancevia* legitimately has only four

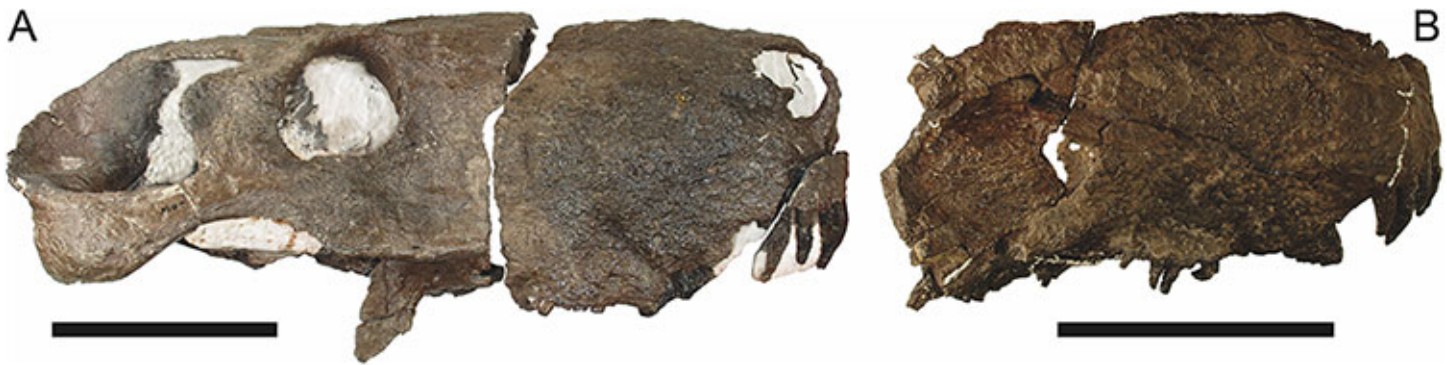

**Figure 71 Referred specimens of *Sycosaurus nowaki* (*Broili & Schröder, 1936*).** (A) UMZC T889 in left lateral view (flipped for comparative purposes). (B) snout of UMZC T877 in right lateral view. Scale bars equal 10 cm.

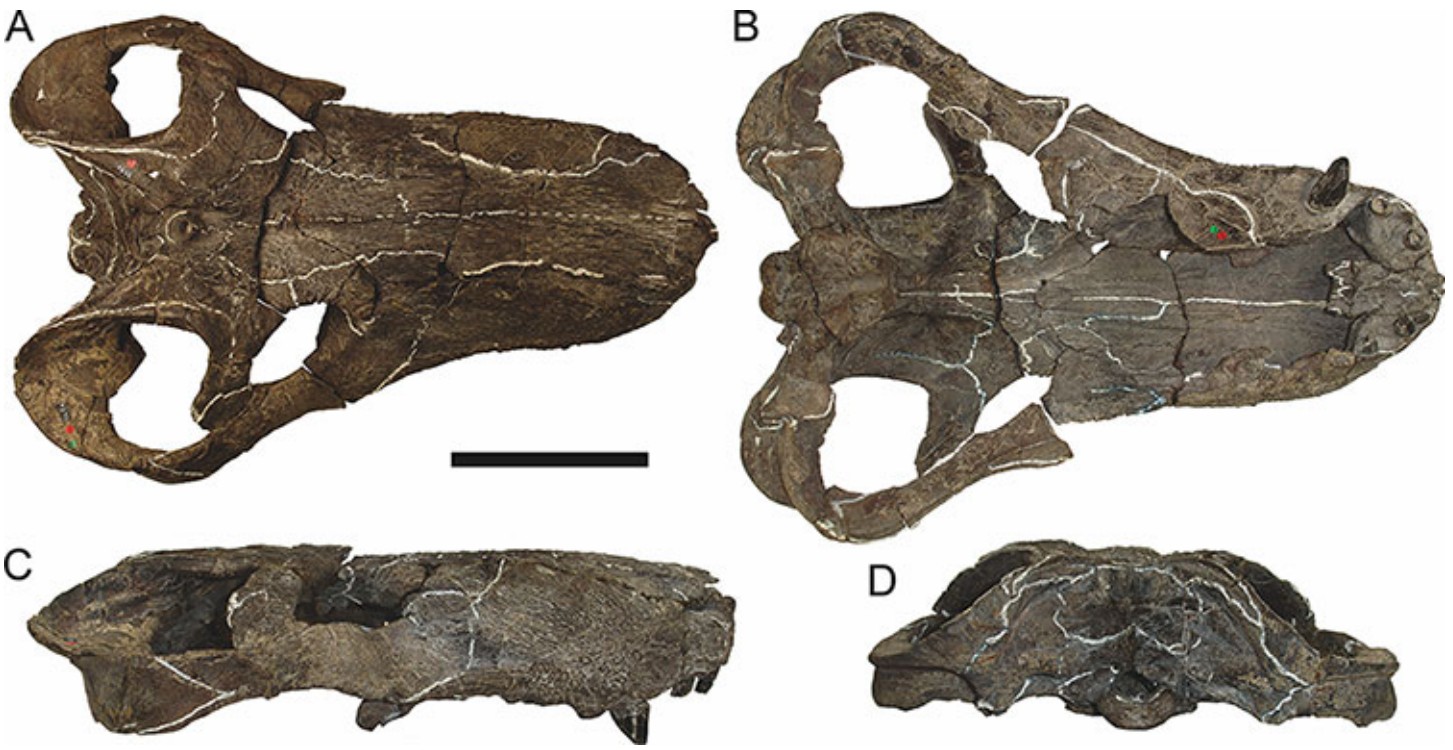

**Figure 72 Referred specimen (UMZC T878) of *Sycosaurus nowaki* (*Broili & Schröder, 1936*) in (A) dorsal, (B) ventral, (C) right lateral, and (D) occipital views. Scale bar equals 10 cm.** Holotype of *Leontocephalus intactus Kemp, 1969*. Scale bar equals 10 cm.

incisors, and several genera were erroneously thought to have only four by previous authors (see discussion in *Kammerer (2014)*). The left premaxilla has been ground down posterior to the I4 alveolus, and no I5 root is preserved. However, while the incisor alveoli in this specimen have been carefully prepared, the alveolar margin of the premaxilla is clearly damaged, likely due to weathering. Although no I5 root is present, all of the incisors have fallen out in this specimen, with only alveoli remaining, and there is clearly both space for a smaller fifth incisor and a weak depression suggestive of an fifth alveolus.

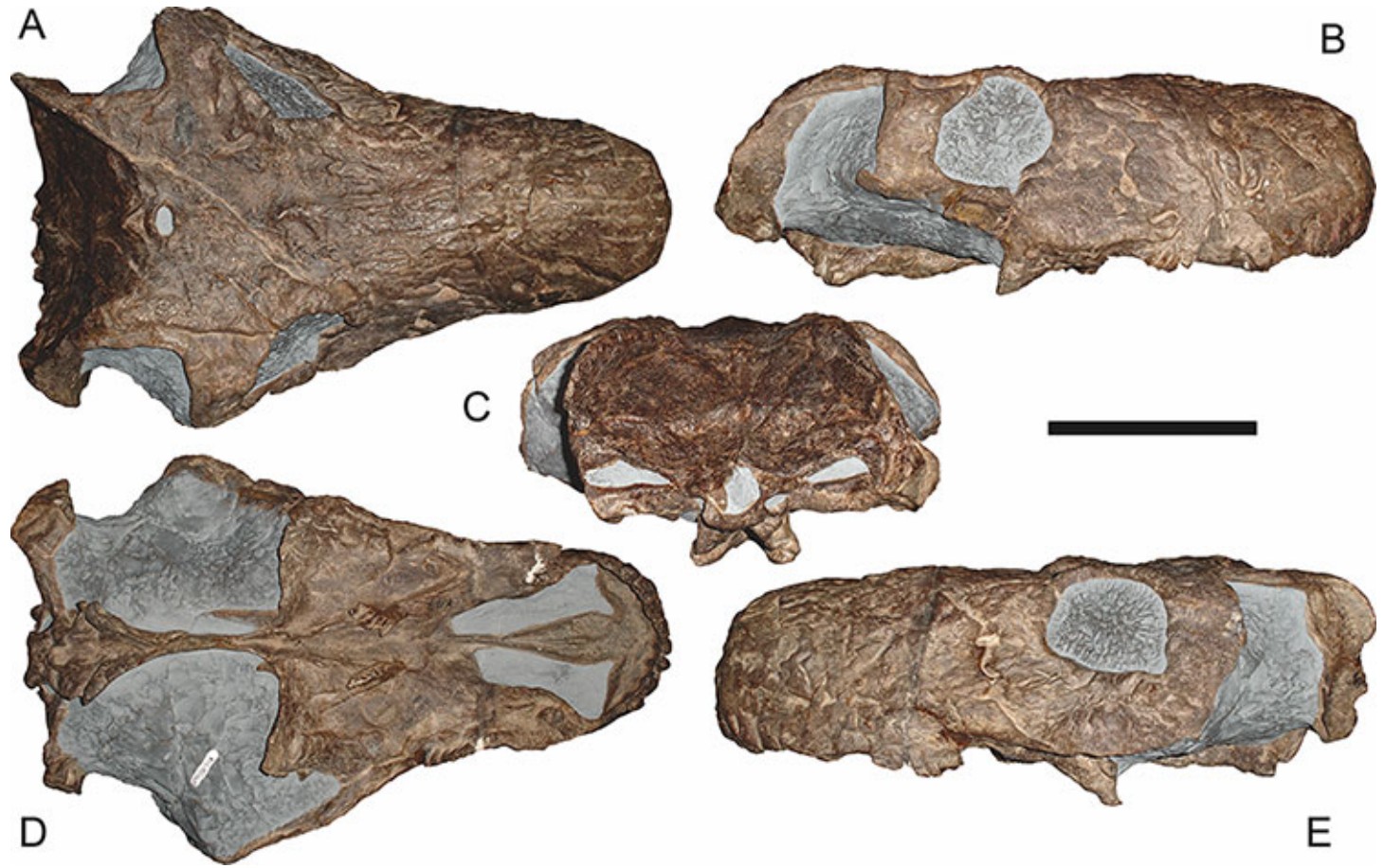

**Figure 73 Referred specimen (GPIT/RE/7116) of *Sycosaurus nowaki* (*Broili & Schröder, 1936*) in (A) dorsal, (B) right lateral, (C) occipital, (D) ventral, and (E) left lateral views.** Holotype of *Lycaenops kingoriensis Huene, 1950*. Scale bar equals 10 cm.

As the I5 is usually the most ventrally-situated of the incisors, it would suffer more than others from erosion of the premaxillary margin. This likely scenario was previously recognized by *Sigogneau-Russell (1989*:fig. 275), who restored this specimen with five incisors.

Four postcanine roots are present in the right maxilla of BSPG 1936 III 1, and three in the left (Fig. 70A). The third postcanine root in the right maxilla is smaller than the second and located posterolaterally to it, so it may represent a replacement PC2 rather than a distinct tooth position. However, on both sides the maxilla is essentially missing behind the posteriormost postcanine, so it is possible that additional teeth were present in life, and a postcanine count of three should not be taken as definitive for BSPG 1936 III 1. Because of its incompleteness, this specimen cannot be distinguished from *S. laticeps* on the basis of postcanine number, but it does exhibit proportionally broader palatines than all known specimens of that species (including the likely adult BP/1/3465), allowing confident distinction of *Sycosaurus nowaki* from *S. laticeps*.

*Sigogneau (1970)*, *Sigogneau-Russell (1989)* and *Gebauer (2007)* all recognized specific distinction for a Tanzanian species of *Sycosaurus*, but considered the valid name for this

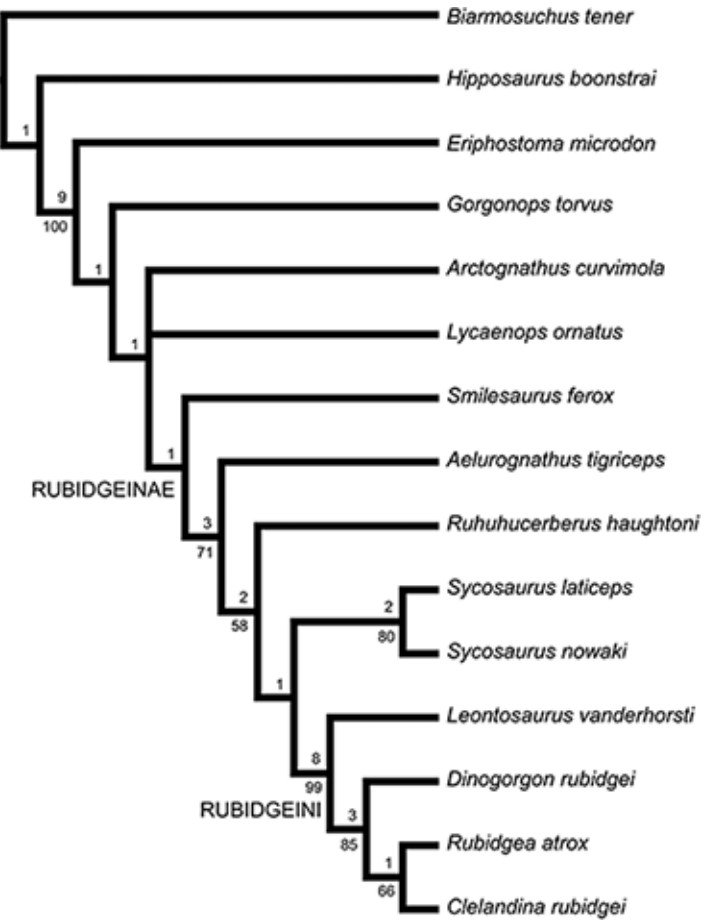

**Figure 74 Strict consensus cladogram of rubidgeine interrelationships.** Numbers under nodes represent symmetric resampling support, numbers above nodes represent Bremer support.

taxon to be *S. kingoriensis*. GPIT/RE/7116, the holotype of *Lycaenops kingoriensis Huene, 1950*, is a weathered but nearly complete skull (Fig. 73). This specimen is extremely similar to BSPG 1936 III 1 where the two overlap: in particular, the vomerine morphology of these specimens is identical. These specimens also share unusually small canines for their size (a general feature of *Sycosaurus* shared with *S. laticeps*). Although small canine size in BSPG 1936 III 1 could be explained by a relatively deep cross-section through the developing canine root, this is definitely not the case in GPIT/RE/7116, in which the alveolar margin of the maxilla is relatively well preserved. GPIT/RE/7116 has a remarkably high upper postcanine count for a rubidgeine. This specimen has at least six and probably seven postcanines, as noted by *Sigogneau (1970)*. *Huene (1950)* and *Gebauer (2007)* mistakenly stated that this specimen has only five upper postcanines, but although five closely-packed postcanine crowns are present in the left maxilla, anterior to these there is a short broken region preceded by at least one definite postcanine root and what may be an additional root anterior to it. This is higher than in any specimens of *S. laticeps* (in which the maximum number is five), lending further support for specific distinction of the Tanzanian species. Finally, the palatal dentition is exceptionally well preserved in

GPIT/RE/7116 (Fig. 73D), and demonstrates that the palatal teeth in this taxon were posteromedially-angled and needle-like.

The majority of Tanzanian gorgonopsian material described by *Kemp (1969)* can also be referred to *Sycosaurus nowaki*. *Leontocephalus intactus Kemp, 1969* was recognized as valid by *Sigogneau-Russell (1989)* and tentatively retained in *Leontocephalus*, but was referred to *Sycosaurus* by *Gebauer (2007)*. The holotype of *L. intactus* (UMZC T878; Fig. 72) is dorsoventrally crushed, but generally accords with identification as *Sycosaurus* based on its relatively narrow zygomatic arch. UMZC T878 has fewer upper postcanines than GPIT/RE/7116 (only five tooth positions are visible in the right maxilla), but this region is damaged in this specimen, and given how closely-packed these teeth are in GPIT/RE/7116 it is possible that more were actually present. *Kemp (1969)* described a second specimen (UMZC T877; Figs. 4 and 71B) as *Arctognathus* sp. based on its high upper postcanine count (6) and robust snout. As noted above, however, this tooth count is also present in GPIT/RE/7116, and UMZC T877 also exhibits the distinctive vomerine morphology of *Sycosaurus*. An additional acid-prepared Tanzanian specimen (UMZC T889; Fig. 71A) can also be referred to *S. nowaki*. Although the palate is not exposed on this specimen, it has an extremely narrow zygomatic arch beneath the postorbital and seven close-packed postcanines.

## PHYLOGENETIC ANALYSIS

In my previous revisionary work on Gorgonopsia (*Kammerer, 2014*; *Kammerer, 2015*; *Kammerer et al., 2015*), I have stated that it would be premature to attempt phylogenetic analysis of this group without a stable alpha taxonomic framework. Now, however, a sufficient number of gorgonopsian species (11, including all valid rubidgeines, *Arctognathus curvimola*, and *Eriphostoma microdon*) have been redescribed to at least begin studying their interrelationships. The only previous phylogenetic analysis of the group was performed as part of *Gebauer's (2007)* PhD thesis, which included 15 gorgonopsian genera and 43 characters. However, I consider this analysis unsatisfactory, as her characters are extremely prone to ontogenetic variation (e.g., *Gebauer, 2007:*230, character 1: "average skull length in adult less than 150 mm (0) up to 300 mm (1) larger than 300 mm (2)"). As an example of potential problems caused by this character set, her analysis recovered the genus *Aloposaurus* (a wastebasket consisting of juvenile and subadult specimens of various gorgonopsian taxa; C Kammerer, personal observations) as the most basal gorgonopsian, with the larger (but stratigraphically earlier) *Eoarctops* (=*Eriphostoma*) being more deeply nested in the tree (despite the extremely primitive palatal morphology of this taxon relative to other gorgonopsians, as discussed by *Kammerer (2014)*).

The phylogenetic analysis presented here is novel, although several characters that are synapomorphies for Gorgonopsia as a whole were previously listed by *Hopson & Barghusen (1986)*. Fifteen species were included, including all rubidgeines herein recognized as valid (*Aelurognathus tigriceps*, *Clelandina rubidgei*, *Dinogorgon rubidgei*, *Leontosaurus vanderhorsti*, *Rubidgea atrox*, *Ruhuhucerberus haughtoni*, *Smilesaurus ferox*, *Sycosaurus laticeps*, and *Sycosaurus nowaki*). The intent of this analysis was to focus on rubidgeine interrelationships, but four non-rubidgeine gorgonopsians were also included:

*Arctognathus curvimola* and *Eriphostoma microdon* (both of which have recently been redescribed: *Kammerer, 2015*; *Kammerer et al., 2015*), *Gorgonops torvus* (the archetypal gorgonopsian), and *Lycaenops ornatus* (the most thoroughly-known gorgonopsian taxon, because of *Colbert's (1948)* monographic treatment). Two biarmosuchians were also included: *Biarmosuchus tener* (used as outgroup) and *Hipposaurus boonstrai* (a taxon historically considered to be an early gorgonopsian). Forty-seven characters were included, all of which are discrete-state, and five of which are ordered (see Appendix for details). Analyses were run in TNT v.1.1 (*Goloboff, Farris & Nixon, 2008*) using the New Technology search parameters. Support metrics were based on symmetric resampling (using 10,000 replicates) and the script bremer.run (based on saved trees from the New Technology search suboptimal by 20 steps) in TNT.

Two most parsimonious trees of length 76 were recovered (CI=0.711, RI=0.852), differing only in the positions of *Lycaenops* and *Arctognathus* (Fig. 74). Rubidgeinae is recovered as monophyletic, although it should be noted that the limited taxon sampling outside of this group biases the results towards this conclusion. Gorgonopsia and a clade consisting of *Clelandina*, *Dinogorgon*, *Leontosaurus*, and *Rubidgea* are extremely robustly supported (with symmetric resampling and Bremer supports of 100/9 and 99/8, respectively). Relatively robust support was also recovered for the genus *Sycosaurus* (80/2), the clade containing *Clelandina*, *Dinogorgon*, and *Rubidgea* (85/3), and the clade containing all rubidgeines other than *Smilesaurus* (71/3).

## DISCUSSION

### Rubidgeine higher-level systematics

A variety of taxonomic schemes for gorgonopsian higher-level taxa have been presented. At opposite extremes are *Sigogneau-Russell (1989)*, who placed all gorgonopsians in a single family (the second 'gorgonopsian' family she recognized was the monotypic Watongiidae, now known to represent a varanopid pelycosaur (*Reisz & Laurin, 2004*)), and *Watson & Romer (1956)*, who split gorgonopsians among twenty families (although three of these, Burnetiidae, Hipposauridae, and Phthinosuchidae, are no longer considered gorgonopsian). At present I follow *Sigogneau-Russell's (1989)* convention and treat Rubidgeinae as a subfamily: although such distinctions are arbitrary, Gorgonopsia as a whole has such a high degree of cranial homomorphism that it is comparable in disparity to the recognized families within Therocephalia or Cynodontia. Rubidgeinae is the only subfamily of Gorgonopidae recognized here, although this is likely to change with additional research.

The composition of Rubidgeinae has also varied over time: *Broom's (1938)* initial conception of the group (originally at family rank) included only *Rubidgea* itself, whereas *Watson & Romer (1956)* also included *Dinogorgon*, *Prorubidgea* (=*Aelurognathus*), and *Tigrisaurus* (=*Clelandina*) and *Sigogneau (1970)* added *Broomicephalus* (=*Rubidgea*) and *Sycosaurus*. *Gebauer (2007)* noted a high degree of similarity between the genera *Aelurognathus* and *Lycaenops*, but considered the former a rubidgeine and not the latter. I follow this break, and tentatively consider Rubidgeinae to include those gorgonopsians

more closely related to *Rubidgea* than *Lycaenops*. I refrain from formally defining this (or any other) gorgonopsian clade, however, pending a comprehensive analysis of the group following the revision of the remaining genera.

*Smilesaurus* is the most atypical rubidgeine, and it remains questionable whether this genus is actually referable to the group. Although it is recovered as a rubidgeine in the analysis (Fig. 74), this position is weakly-supported, and it lacks several important characters present in all other rubidgeines (e.g., exclusion of frontal from the orbital margin, anteroposterior expansion of the postorbital bar, cranial rugosity or pachyostosis). It is possible that the rubidgeine characters of *Smilesaurus* are convergences correlated with large size, and that this genus represents an independent evolution of 'rubidgeine' morphology from an *Arctops*-like ancestor, as suggested by *Sigogneau-Russell (1989)*. This possibility will need to be tested more thoroughly in future iterations of the analysis.

The genera *Clelandina*, *Dinogorgon*, *Leontosaurus*, and *Rubidgea* have previously been considered synonymous to varying degrees, and given that they form an extremely well-supported clade here, it might be wondered whether they should be recognized as species in a single genus (for which the name *Dinogorgon* would have priority). Given the historically continuous usage of the names *Rubidgea* and *Clelandina*, however, I prefer to leave these species in monotypic genera. This also avoids the problem of secondary homonymy caused by inclusion of *D. rubidgei* and *C. rubidgei* in a single genus. Furthermore, although these genera are currently monotypic, it possible that future research will reveal species-level differences between the South African and Tanzanian specimens of *Dinogorgon* and *Rubidgea*, as has been demonstrated here for *Sycosaurus* (and is not uncommon among therapsids: see, e.g., the dicynodonts *Dicynodon* and *Dicynodontoides* (*Angielczyk et al., 2009*; *Kammerer, Angielczyk & Fröbisch, 2011*)). For *Rubidgea* in particular, the east African material is very incomplete, and additional specimens are needed to solidify its conspecificity with *R. atrox*. Should future work support specific distinction for the east African forms, the names *Dinogorgon quinquemolaris* and *Rubidgea maxima* are available for these taxa. Instead of synonymizing these four genera, recognize them in a new tribe, Rubidgeini *Broom, 1938*. This tribe is diagnosed by the combination of a snout that narrows in width immediately posterior to the canine, well-developed maxillary emargination, greatly expanded postorbital bar, massive, strongly deflected subtemporal bar with a lateral ridge and ventral boss, restriction of the squamosal sulcus to the occiput, and restriction of the posterior parietal process to the skull roof (convergent in *Ruhuhucerberus*).

## Ecology of rubidgeines

The large size, serrated teeth, and massive canines of rubidgeines indicates that they were adapted for macropredation, and their cranial morphology is broadly similar to a number of other 'reptilian' top predators. Of particular note is the robust skull roof of most rubidgeines, ranging from rugose dorsal margins of the orbit and temporal fenestra to massive supraorbital bosses in rubidgeins. Comparable supraorbital bosses or horns are widely distributed in 'reptilian' macropredators, e.g., anteosaurian dinocephalians,

'rauisuchians', thalattosuchian crocodiles, mosasaurs, and large theropod dinosaurs, and have been interpreted as a protective 'sink' for stresses inflicted during prey capture (*Young et al., 2010*; *Kammerer, 2011*). The presence of these supraorbital bosses is part of a suite of features (also including ziphodonty) that appear convergently in various large predatory taxa that focus on use of the skull, instead of limbs, in prey capture. *Rubidgea atrox* in particular is extremely convergent on *Anteosaurus magnificus*, right down to the presence of lateral mandibular bosses of unknown function (albeit on the dentary in *Rubidgea* and angular in *Anteosaurus*). Cranial stress and strain sinks in gorgonopsians are also indicated by consistent groups of strongly interdigitated sutures, notably the mid-frontal (which forms a robust interorbital ridge in very large taxa), mid-palatine, and supraoccipital-tabular.

The presence of seven, seemingly coeval large gorgonopsian taxa in the same region is remarkable, and raises the question of how niche partitioning would have occurred between them. The loss of postcanine dentition in some taxa (i.e., *Clelandina* and *Rubidgea*) but not others (e.g., *Dinogorgon*) indicates that these animals were handling prey in different ways, although the functional underpinnings of this difference remain obscure. The complete replacement of the postcanine tooth row with a bony ridge in *Clelandina* is especially intriguing, as it also occurs in the therocephalian *Theriognathus*. This ridge has a smooth bone surface, suggesting that it was not covered in a keratinous sheath in life (as in dicynodonts) to form a blade. Presumably these taxa relied entirely on the incisors and canines to kill and, despite their relatively large size, may have specialized on smaller prey that could be swallowed whole. *Clelandina* is also unusual among gorgonopsians in its remarkably small sclerotic ring relative to orbit size (Fig. 21), indicating that this taxon was highly photopic (*Angielczyk & Schmitz, 2014*). Unfortunately sclerotic rings are not known in other rubidgeines, but it is possible that their niche partitioning was partially related to different cycles of daily activity, with *Clelandina* being a strictly diurnal form.

*Smilesaurus* differs from the remaining rubidgeines in completely lacking any dorsal thickening of the skull, suggesting that it was handling prey in a very different way. This taxon is exceptional in having the proportionally longest canines of any gorgonopsian, which fit into a deep indentation on the lateral surface of the mandible. Although various early therapsids (gorgonopsians especially) are often described as 'sabre-toothed', comparing them to later mammalian predators like *Smilodon* and *Thylacosmilus*, this descriptor refers only to the presence of enlarged canines. Looking at overall cranial morphology in large gorgonopsians, head-focused prey capture in these taxa was probably generally more similar to that of 'reptiles' like crocodiles and dinosaurs (as discussed above) than sabre-toothed mammals. *Smilesaurus*, however, appears to be an exception to this: the possession of exceedingly long, blade-like canines, reduced postcanines, and seemingly little cranial support for dealing with biting down on bones suggests that it was using the canine primarily for slashing large prey (e.g., coeval dicynodonts such as *Rhachiocephalus*). As such, *Smilesaurus* (and the comparably weak-skulled but large canine-bearing Russian taxon *Inostrancevia*) are likely to represent the only 'true' sabre-toothed predators (in the machairodont sense) among early therapsids.

## Biostratigraphy and biogeography of rubidgeines

The earliest known rubidgeines are found in the *Tropidostoma* AZ. Rubidgeines are very rare components of the *Tropidostoma* AZ fauna, with only five specimens currently known (three specimens of *Aelurognathus tigriceps*: SAM-PK-2342, SAM-PK-2672, and SAM-PK-10071 from Dunedin, Beaufort West; and two specimens of *Smilesaurus ferox*: CGS RS 176 from De Hoop, Beaufort West and CGS S 231 from Reiersvlei, Fraserburg). All other rubidgeine fossils (including the majority of known *Aelurognathus* and *Smilesaurus* specimens) occur in the *Cistecephalus* and *Daptocephalus* AZs (or their east African equivalents). Because the majority of rubidgeine specimens were collected without precise stratigraphic data (usually only a farm name), their exact positions within these zones are uncertain. New field work in the *Daptocephalus* AZ is helping to resolve the distribution of gorgonopsians in the latest Permian (*Botha-Brink, Huttenlocker & Modesto, 2014*; *Viglietti et al., 2016*), but biostratigraphy within the *Cistecephalus* AZ is currently poorly resolved, and requires directed, fine-scale stratigraphic analysis.

Rubidgeines are currently known only from Africa, but are found in most of the major basins, with definite records in the Karoo Basin of South Africa, Luangwa Valley of Zambia, Ruhuhu Basin of Tanzania, and Chiweta Beds of Malawi. In general, the gorgonopsian record outside of Africa is poor; for example, no specimens of this group have been collected in the Permian deposits of Brazil or China. Russia has yielded a number of gorgonopsian fossils, however, and *Ivakhnenko (2003)* described one taxon, *Leogorgon klimovensis*, that he referred to Rubidgeinae. The holotype of *L. klimovensis* (PIN 4549/13) is a badly-worn braincase fragment from the Klimovo locality (Late Permian, Vyatkian Horizon). *Ivakhnenko (2003)* considered this specimen referable to Rubidgeinae based on the short, tall paroccipital process, which he considered comparable only to *Dinogorgon* among gorgonopsians. However, several non-rubidgeine gorgonopsians have paroccipital processes similar in height to *Dinogorgon*, such as *Arctognathus* (*Kammerer, 2015*). Furthermore, it is far from certain that PIN 4549/13 even represents a gorgonopsian. *Ivakhnenko (2003)* listed no gorgonopsian synapomorphies to support his identification, and the tall opisthotic of PIN 4549/13 (with a very elevated position for the post-temporal fenestra) is extremely similar to the typical condition in dicynodonts. Further study of this specimen is necessary to determine its relationships, but at present there is zero evidence supporting a rubidgeine referral for PIN 4549/13.

*Ivakhnenko (2003)* also referred an isolated, serrated incisor (PIN 4549/14) to *Leogorgon klimovensis*. This tooth can confidently be identified as gorgonopsian based on its large size, longitudinal striations, and well-developed mesiodistal serrations. However, it is indistinguishable from the incisors of the common Russian taxon *Inostrancevia*. *Ivakhnenko (2003)* noted the general similarity between this specimen and the teeth of *Inostrancevia*, but argued that PIN 4549/14 could be distinguished by a more weakly-faceted crown and better-developed mesial cutting edge. These characters vary with preservation and tooth development in known specimens of *Inostrancevia*, however (C Kammerer, personal observations), and do not permit confident differentiation.

PIN 4549/14 should be considered Gorgonopsia indet., and does not provide evidence of rubidgeines in Russia.

## CONCLUSIONS

Comprehensive taxonomic revision of rubidgeine gorgonopsians has greatly reduced the number of valid taxa from 36 nominal species to the nine recognized here. Rubidgeinae remains a remarkably speciose clade, however, when compared with other groups of non-mammalian therapsid macropredators. Among eutherocephalians and cynodonts, only *Theriognathus*, *Moschorhinus*, *Cynognathus*, and some chiniquodontids would have occupied similar niches. Anteosaurid dinocephalians exhibit similar species richness at the global scale, but lack comparable within-basin diversity (with only two temporally-disjunct species known from African deposits). Middle Permian scylacosaurid therocephalians attained large size and are nominally diverse, but still require taxonomic revision. The presence, in rubidgeines, of so many closely-related large predators in a single basin is unusual for the Permian, and hopefully this result will spur further study of the ecology and stratigraphic distribution of these animals.

I hope that this contribution will provide a useful framework for future study of the Gorgonopsia and encourage renewed systematic attention on this group. Even within rubidgeines, several taxa deserve additional taxonomic scrutiny. As noted in the species accounts, the validity of *Leontosaurus* vis-à-vis *Sycosaurus* and *Rubidgea* is tentative and requires further research. The conspecificity of all the material referred to *Aelurognathus tigriceps* here also deserves a critical eye, particularly as concerns the synonymy of *Prorubidgea maccabei*. My taxonomic conclusions were those best supported by the available data, but many rubidgeine specimens require additional preparation (or CT scanning), and this is absolutely necessary for basic identification in some cases (e.g., the holotypes of *Aelurognathus nyasaensis* and *Gorgonorhinus luckhoffi*). Finally, I would note that the east African rubidgeine fauna remains poorly known compared to that of the main Karoo Basin. Additional collecting in the east African basins is needed, particularly to gauge local variation in broadly-distributed genera such as *Rubidgea*.

## ACKNOWLEDGEMENTS

I thank the reviewers, Robert Reisz and Bruce Rubidge, for their helpful input in improving this manuscript. My sincere thanks to all the collections staff and curators who facilitated my examination of gorgonopsian specimens: Anziske Kayster (B), Bernhard Zipfel and Michael Raath (BP), Ellen de Kock, Linda Karny, and Johann Neveling (CGS), Philipe Havlik and Michael Maisch (GPIT), Paul Barrett and Sandra Chapman (NHMUK), Jennifer Botha-Brink and Elize Butler (NMQR), Andrey Kurkin and the late Mikhail Ivakhnenko (PIN), Sheena Kaal and Roger Smith (SAM), Stephany Potze and Heidi Fourie (TM), Pat Holroyd (UCMP), and Jennifer Clack and Ray Symonds (UMZC). Particular thanks are due to Robert and Marion Rubidge for their kind hospitality at Wellwood (RC). Despite the large reduction in species number that I have undertaken in this paper, it must be noted that a full third of valid rubidgeine species still bear the name of this family that has contributed so much to Karoo paleontology.

# APPENDIX

## Character list

1. **Ascending process of premaxilla:** (0) terminates posterior to the canine; (1) terminates anterior to the canine. A lengthy ascending (also known as the dorsal or internasal) process of the premaxilla is ancestral for therapsids (*Hopson & Barghusen, 1986*). In gorgonopsians, however, this character reverts to the pre-therapsid condition: in all known gorgonopsians the premaxilla has a relatively short ascending process.

2. **Posterior margin of palatal premaxillary surface:** (0) gently rounded; (1) invaginated. In most rubidgeines, there is a deep invagination in the posterior border of the premaxilla between its main body and vomerine process. This invagination is not solely a side-effect of anterior expansion of the vomer (although it is exaggerated by it), as it is also present in some taxa which lack such expansion (e.g., *Arctognathus* (see *Kammerer, 2015*)).

3. **Vomerine process of premaxilla:** (0) medially short, almost totally obscured by vomer in ventral view at midline; (1) long at midline, comprising a substantial portion of the expanded interchoanal body in ventral view. In biarmosuchians, a vomerine process of the premaxilla is present (sheathing the vomer laterally) but is not well-exposed ventrally at the midline: the vomer in these taxa nearly abuts the main body of the premaxilla. In gorgonopsians, by contrast, the vomerine process of the premaxilla is a major contributor to the expanded interchoanal body (i.e., the transversely broad median structure made up of the anterior part of the vomer and the vomerine process of the premaxilla), often making up as much of the midline length of this structure as the vomer. In gorgonopsians where the vomerine process of the premaxilla makes up significantly less of the interchoanal body length than the vomer, it is only because the vomer is expanded for more of its length (as in *Arctognathus* or *Sycosaurus*)—the premaxilla is still longer in these taxa than in biarmosuchians.

4. **Median vomerine ridge:** (0) absent; (1) present.

5. **Lateral vomerine ridges:** (0) absent; (1) present. The vomer in gorgonopsians is always unpaired, with a characteristic 'triple ridge' morphology. A pair of lateral ridges originate at the point where the interchoanal body of the vomer starts to expand; these ridges extend anteriorly, often continuing onto the vomerine process of the premaxilla. The median ridge typically originates at a similar position, although it is located more anteriorly in some taxa (e.g., *Sycosaurus*). Although these three ridges are present in all gorgonopsians in which the vomer is known, this morphology is here split into two characters because of the variable condition in the biarmosuchian outgroups. No vomerine ridges of any kind appear to be present in *Biarmosuchus*, but *Hipposaurus* exhibits a tall, distinct median vomerine ridge (most clearly visible in SAM-PK-K252). Most other biarmosuchians exhibit lateral vomerine ridges but not a median one (e.g., *Bullacephalus jacksoni* (BP/1/5387), *Herpetoskylax hopsoni* (CGP/1/67), *Lemurosaurus pricei* (NMQR 1702), *Lobalopex mordax* (CGP/1/61), *Lophorhinus willodenensis* (SAM-PK-K6655), *Lycaenodon longiceps* (NHMUK R5700), or

*Paraburnetia sneeubergensis* (SAM-PK-K10037)). Given the poor preservation of the palate in known specimens of *Hipposaurus*, it is possible that this genus had the full gorgonopsian complement of three ridges. It is worth noting, however, that the lateral ridges in biarmosuchians are very thin, laminar structures, more like ventral deflection of the vomerine edge than a separate ridge (similar to the 'scroll-like' vomerine morphology of anteosaurs (*Kammerer, 2011*)). By contrast, in gorgonopsians these ridges are relatively robust, discrete structures.

6. **Expanded interchoanal body shape:** (0) elongate and relatively narrow throughout its length, making up nearly all of interchoanal portion of vomer; (1) lobate, with anterior and posterior expansions; (2) elongate but tapering, with broad anterior terminus; (3) bulbous, with broad anterior expansion. Vomerine morphology is remarkably diverse within gorgonopsians, a fact that has traditionally been obscured by poor preparation. The ancestral condition, present in biarmosuchians, is to have nearly the entire interchoanal portion of the vomer be expanded, as opposed to the typical condition in gorgonopsians where the interchoanal vomer is made up of an extremely narrow, rod-like posterior portion and a transversely expanded anterior portion. In biarmosuchians this structure may become slightly wider anteriorly, but is generally quite narrow throughout (in comparison to the condition in rubidgeines). The biarmosuchian vomerine morphology seems to be retained in *Eriphostoma* (based on CT-scans of AMNH FARB 5524 (*Kammerer, 2014*)) and is also present (probably homoplastically) in *Arctognathus* (*Kammerer, 2015*). *Gorgonops* has an unusual vomerine morphology (described here as 'lobate') in which the vomer has double expansions: a posterior one, then a constriction, then an anterior one that becomes confluent with the vomerine process of the premaxilla. The posterior vomerine expansion in *Gorgonops* is also associated with exaggerated development of the lateral vomerine ridges. This character state represents an autapomorphy of *Gorgonops* in the current analysis, but it is also present in other gorgonopsians, such as the Russian taxon *Sauroctonus progressus* (*Tatarinov, 1974*). In *Sycosaurus* and to a lesser extent *Smilesaurus* the interchoanal portion of the vomer is divided into a rod-like posterior and expanded anterior region (unlike biarmosuchians), but expansion begins in a relatively posterior position (compared to other rubidgeines) and proceeds gradually. By contrast, in the remaining rubidgeine taxa the expansion of the vomer is very abrupt and occurs relatively far forward, resulting in a typically 'bulbous' anterior interchoanal body.

7. **Vomerine-pterygoid contact:** (0) present; (1) absent. The presence of a midline palatine suture excluding the vomer from contacting the pterygoids has long been recognized as a key autapomorphy of Gorgonopsia (*Hopson & Barghusen, 1986*; *Sigogneau-Russell, 1989*).

8. **Dentition on palatine boss:** (0) extensive; (1) elongate single row; (2) a few teeth in a restricted position. ORDERED. The primitive condition for synapsids is to have extensive palatal dentition (*Reisz, 1986*). Biarmosuchians typically retain broad patches of teeth across the palatine and pterygoid bosses: in *Biarmosuchus*, each palatine and pterygoid boss bears over fifty densely packed teeth. The palatal dentition is reduced to

varying degrees in all of the 'advanced' groups of therapsids: in gorgonopsians, the palatine dentition usually takes the form of an elongate single row of teeth. In the earliest gorgonopsians (represented by *Eriphostoma* and *Gorgonops* in the current analysis), this row extends along both the lateral and medial sides of the palatine boss, whereas in later gorgonopsians it extends only along the lateral side. In a few rubidgeines (*Clelandina*, *Rubidgea*, and *Smilesaurus*) the palatine dentition is extremely reduced, with only one to three teeth in a small patch at the anterior edge of the boss. This character is treated as ordered, as state 2 just represents an increase in tooth reduction from the previous states.

9. **Palatine boss shape:** (0) delta-shaped; (1) reniform. The palatine bosses of biarmosuchians are large, triangular or delta-shaped structures with extensive dentition. In the early gorgonopsians *Eriphostoma* and *Gorgonops*, the palatine dentition is reduced relative to that of biarmosuchians, but the overall shape of the boss is retained. In later gorgonopsians, the boss is reduced in size and typically is 'bean'-shaped or reniform.

10. **Pterygoid palatal boss:** (0) discrete structure distinct from palatine boss; (1) thin ridge extending posteriorly from palatine boss. Ancestrally in therapsids, the palatine boss and pterygoid palatal boss are dentigerous structures of nearly equivalent size, separated by a weak trough. In gorgonopsians, the palatal boss of the pterygoid is always smaller than the palatine boss, but for the most part is still a discrete, dentigerous structure. In the rubidgeines *Clelandina*, *Dinogorgon*, *Leontosaurus*, *Rubidgea*, *Ruhuhucerberus*, and *Sycosaurus*, however, the palatal pterygoid boss is reduced to a thin ridge extending from the back of the palatine boss towards the midpoint between the transverse processes.

11. **Dentition on palatal boss of pterygoid:** (0) extensive; (1) reduced; (2) absent. ORDERED. As noted above, biarmosuchians have large palatal bosses of the pterygoids with extensive dentition. As for the palatine boss, the amount of pterygoid dentition is reduced in gorgonopsians, and typically constitutes a small, circular patch of three to six teeth. In most rubidgeines the pterygoid dentition is reduced even further, but this character state does not entirely overlap with Character 10. In most gorgonopsians where the palatal boss of the pterygoid is reduced to a thin ridge, it is also edentulous, but *Ruhuhucerberus* exhibits a thin, ridge-like palatal pterygoid boss that clearly still bears teeth. This character is ordered, following the same logic as Character 8.

12. **Dentition on transverse process of pterygoid:** (0) present; (1) absent. Teeth on the transverse process of the pterygoid are absent in many adult gorgonopsians (although they may be present in juveniles). In the current analysis, they are coded as present only in *Eriphostoma*, *Gorgonops*, and the biarmosuchian outgroups.

13. **Parasphenoid morphology:** (0) broad parasphenoid with edges separated by a narrow median groove; (1) parasphenoid rostrum forming tall, narrow blade. A blade-like parasphenoid rostrum is characteristic of gorgonopsians. In rubidgeines, however, there is a reversal to the pre-gorgonopsian morphology, with loss of the 'blade' and presence of an elongate groove between the edges.

14. **Parasphenoid length:** (0) short (<20% basal skull length); (1) long (>20% basal skull length). The parasphenoid and basisphenoid are fused in gorgonopsians to form a parabasisphenoid. For the purposes of this character, parasphenoid length is taken as the length between the parasphenoid-pterygoid suture and the anterior edges of the basal tubera (which are formed by the basisphenoid anteriorly and basioccipital posteriorly). In 'pelycosaurs' and biarmosuchians, the braincase is relatively short, with the distance between the transverse processes of the pterygoids and the basal tubera being less than 20% of the basal skull length. In gorgonopsians, the parasphenoid rostrum is typically elongate, making up more than 20% of the basal skull length. Reversal to a short parasphenoid is observed in some rubidgeines, however, including *Clelandina*, *Dinogorgon*, *Leontosaurus*, *Rubidgea*, and *Sycosaurus*. Shortening of the parasphenoid in gorgonopsians is correlated with increasing length of the basal tubera (so that is not included as a distinct character). An exceptionally short parasphenoid and elongate basal tuber is present in *Smilesaurus*.

15. **Transverse lamina of septomaxilla:** (0) absent; (1) present, bifurcating naris into dorsal and ventral compartments visible in lateral view. A foramen between the septomaxilla and maxilla within the naris (usually visible in anterior view) is broadly present in early therapsids, but the main body of the septomaxilla remains tightly appressed to the maxilla overall. In gorgonopsians, the septomaxilla produces a transverse lamina set well above the dorsal edge of the maxilla, which divides the external naris into dorsal and ventral portions in lateral view.

16. **Long axis of facial process of septomaxilla:** (0) sloping posterodorsally; (1) subhorizontal. In biarmosuchians and most gorgonopsians, the facial process of the septomaxilla (which extends between the nasal and maxilla) is angled upwards, tapering posterodorsally. In *Clelandina*, *Dinogorgon*, and *Rubidgea*, however, this process extends more posteriorly, with limited dorsal angulation.

17. **Snout width:** (0) equal or greater posterior to canine as across canine; (1) narrower posterior to canine than across canine. Taxa with large canines (as in many early therapsids, and gorgonopsians in particular) necessarily have swollen maxillae to house the roots of these enlarged teeth, typically resulting in a broad snout relative to similar taxa in which the canines are small or absent. In most of these taxa, however, the transverse length between the canines is nevertheless not the broadest point in the snout: the snout either continues to expand posteriorly or remains equally broad. The rubidgeines *Clelandina*, *Dinogorgon*, *Leontosaurus*, and *Rubidgea* are exceptions, however: in these taxa the snout is distinctly constricted posterior to the canines, before expanding again anterolateral to the orbits.

18. **Maxillary emargination:** (0) absent; (1) present. In nearly all gorgonopsians, the postcanine-bearing region of the maxilla is somewhat inset from the zygomatic arch, but this separation is typically minor. In some taxa, however, there is a distinct maxillary emargination in which the lateral surface of the maxilla is deeply concave and situated strongly medial to the zygoma. This is developed to the most extreme

degree in the taxa which have lost most or all of their postcanines (*Clelandina* and *Rubidgea*) but is also present in *Dinogorgon* and *Leontosaurus* (indicating that it is not solely a correlate of tooth loss). Outside of rubidgeines, a well-developed maxillary emargination is also present in *Eriphostoma* (*Kammerer et al., 2015*).

19. **Lacrimal foramen:** (0) confined to orbit; (1) also exits on lateral surface of lacrimal. In biarmosuchians and most gorgonopsians a single lacrimal foramen occurs on the posterior surface of the lacrimal, within the orbital rim. In *Clelandina*, *Dinogorgon*, and *Rubidgea*, however, there is a large, second lacrimal foramen exiting onto the facial surface of the lacrimal.

20. **Frontal contribution to orbital margin:** (0) present; (1) absent. In therapsids, the ancestral condition is to have the orbit bordered dorsally by the prefrontal, frontal, and postfrontal. In rubidgeines (with the exception of *Smilesaurus*), the prefrontal and postfrontal are expanded and contact each other, excluding the frontals from the orbital margin (although it can still make up part of the dorsomedial orbital wall). Prefrontal-postfrontal contact is weakly developed (and may be intraspecifically variable) in *Aelurognathus* and *Ruhuhucerberus*, but is extremely well developed in the remaining rubidgeines, with the frontals broadly separated from the orbits dorsally.

21. **Interorbital ridge:** (0) absent or weakly-developed; (1) well-developed. A median interorbital ridge is present in many early therapsids (see, e.g., *Kammerer, 2011*), but the condition in gorgonopsians is a special case. In nearly all gorgonopsians there is a span near the mid-length of the frontals with an increased degree of interdigitation of the mid-frontal suture. In some taxa this is expanded into a low boss. In the rubidgeines *Clelandina*, *Dinogorgon*, *Rubidgea*, and *Ruhuhucerberus* this span forms a well-developed, elongate ridge extending near the anterior margin of the orbits.

22. **Cranial pachyostosis:** (0) absent; (1) restricted to thickened edges of the orbital and temporal margins; (2) extensive, lateral margins of prefrontal, postfrontal, and postorbital form swollen, rugose bosses. Pachyostosis of the skull occurs independently in several therapsid clades, with particularly extreme cases in burnetiamorphs (*Rubidge & Sidor, 2002*) and dinocephalians (*Boonstra, 1969*). Among gorgonopsians, cranial pachyostosis is only present in rubidgeines. Pachyostosis is relatively weakly developed in *Aelurognathus*, *Ruhuhucerberus*, and *Sycosaurus*, taking the form of a thickened dorsal rim of the orbit and anterior edge of the temporal fenestra. The condition in *Leontosaurus* is similar, although in that taxon a thicker boss is present at the posterodorsal corner of the orbit. In *Clelandina*, *Dinogorgon*, and *Rubidgea*, however, pachyostosis is taken to an extreme degree comparable to that of anteosaurine dinocephalians (*Kammerer, 2011*): large supraorbital and postorbital bosses are present and the bone surface is highly rugose.

23. **Preparietal:** (0) absent; (1) present. A small median preparietal bone, situated anterior to the pineal boss, is present in anomodonts, gorgonopsians, and biarmosuchians (absent in *Biarmosuchus* itself, but present in *Hipposaurus*) among

therapsids. Loss of the preparietal has occurred in all of those groups; in gorgonopsians the preparietal is absent in *Arctognathus* (*Kammerer, 2015*) and rubidgeines. A preparietal is absent in some specimens of *Aelurognathus*, *Ruhuhucerberus*, and *Smilesaurus*, but clearly present in other (probably subadult) specimens, and may fuse late in ontogeny in those taxa. All known specimens of *Clelandina*, *Dinogorgon*, *Leontosaurus*, *Rubidgea*, and *Sycosaurus* (including likely juveniles for all of these genera) lack a preparietal.

24. **Postorbital bar:** (0) unexpanded; (1) expanded (>10% of basal skull length); (2) greatly expanded (>20% of basal skull length). ORDERED. Anteroposterior expansion of the postorbital bar relative to other gorgonopsians is present in all rubidgeines but *Smilesaurus*. In most of these taxa, the postorbital bar remains transversely narrow, but is broadened laterally such that it exceeds 10% of the basal skull length. In *Clelandina*, *Dinogorgon*, *Leontosaurus*, and *Rubidgea*, the postorbital bar is extremely expanded (greater than 20% of skull length) laterally and also thickened transversely, often augmented by the presence of postorbital rugosities or bosses.

25. **Facial portion of jugal:** (0) confluent with suborbital zygomatic portion; (1) depressed relative to zygomatic portion. The facial portion of the jugal often has a concave surface in gorgonopsians, but usually this depression smoothly attenuates onto the zygomatic arch. In a few taxa (*Clelandina*, *Dinogorgon*, *Rubidgea*, and *Ruhuhucerberus*), however, there is a sharp break in slope between the zygoma and a deeply depressed facial portion of the jugal.

26. **Jugal height below postorbital bar:** (0) tall (>10% of basal skull length); (1) short (5–7% of basal skull length). The jugal contribution to the zygomatic arch is slightly constricted beneath the postorbital bar in most gorgonopsians, but this is taken to an extreme in the two species of *Sycosaurus*. In these taxa, there is a deep ventral concavity in the zygoma.

27. **Subtemporal bar angle:** (0) straight; (1) deflected (>20° from long axis of skull); (2) strongly deflected (>45° from long axis of skull). ORDERED. Ventral deflection of the subtemporal bar is present in all rubidgeines and the non-rubidgeine gorgonopsian *Lycaenops*, but the angle of deflection is usually weak (20–30°). In *Clelandina*, *Dinogorgon*, *Leontosaurus*, and *Rubidgea*, however the bar is deflected to an extreme degree, exceeding 45° relative to the long axis of the skull.

28. **Subtemporal bar width:** (0) narrow; (1) transversely expanded. In addition to being dorsoventrally expanded in the majority of rubidgeines, the subtemporal bar is massive, with significant transverse expansion, in *Clelandina*, *Dinogorgon*, *Leontosaurus*, and *Rubidgea*.

29. **Zygomatic process of squamosal:** (0) terminates under temporal fenestra; (1) terminates under postorbital bar. The zygomatic ramus of the squamosal usually terminates near the midpoint of the temporal fenestra in gorgonopsians, even in taxa in which the postorbital bar is anteroposteriorly expanded (e.g., *Aelurognathus*, *Sycosaurus*). In *Arctognathus* and *Ruhuhucerberus*, this ramus is unusually long, reaching

the level of the postorbital bar. In *Clelandina*, *Dinogorgon*, *Leontosaurus*, and *Rubidgea*, the squamosal curves anterodorsally, such that it also reaches the postorbital bar.

30. **Zygomatic ridge:** (0) absent; (1) present. In *Clelandina*, *Dinogorgon*, *Leontosaurus*, and *Rubidgea*, a thickened ridge runs from the tip of the squamosal up the jugal, ending beneath the posteroventral edge of the orbit.

31. **Zygomatic boss:** (0) absent; (1) present. A massive, rounded boss at the ventral edge of the subtemporal bar is present in *Clelandina*, *Dinogorgon*, *Leontosaurus*, and *Rubidgea* among gorgonopsians.

32. **Squamosal sulcus:** (0) extends laterally onto subtemporal bar; (1) restricted to occiput. The squamosal sulcus is homologous with the external auditory meatus of mammals. In most gorgonopsians it extends laterally, extending onto the zygomatic arch. The degree of lateral coverage is variable, but is almost always visible at least in part in lateral view. The exceptions to this are *Clelandina*, *Dinogorgon*, *Leontosaurus*, and *Rubidgea*, in which the sulcus is restricted to the occiput.

33. **Subtemporal bar lateral margin:** (0) smoothly rounded; (1) with concavity between squamosal and jugal. A distinct concavity between the squamosal and jugal is visible in dorsal view in *Clelandina*, *Dinogorgon*, *Leontosaurus*, and *Rubidgea*. This concavity is not simply a result of the zygomatic ridge, as it is present even in subadult specimens where this ridge is not yet developed (e.g., RC 101).

34. **Temporal fenestra:** (0) taller than wide; (1) wider than tall. A narrow temporal fenestra is ancestral for therapsids and is present in all biarmosuchians. In most gorgonopsians, the temporal fenestra is anteroposteriorly expanded by comparison. In the rubidgeines *Clelandina*, *Dinogorgon*, *Leontosaurus*, and *Rubidgea*, this fenestra reverts to being relatively short. Although accentuated by cranial pachyostosis and expansion of the postorbital bar, once again subadult or juvenile specimens illustrate that even before extreme expansion of the postorbital bar occurs, the fenestra is relatively short (e.g., RC 101).

35. **Squamosal contribution to occiput:** (0) narrower than tabular; (1) broader than tabular. The squamosal is a dorsoventrally elongate but transversely narrow contributor to the occiput in biarmosuchians, such that it is thinner in posterior view than the tabular. In gorgonopsians, the squamosal portion of the occiput is greatly expanded, and is always transversely broader than the tabular.

36. **Parietal midline:** (0) consists entirely of pineal boss; (1) extends beyond pineal boss. The parietal is extremely anteroposteriorly short in 'pelycosaurs' (*Reisz, 1986*), and similar proportions are retained in biarmosuchians. Indeed, in biarmosuchians the midline of the parietal only encompasses the pineal boss. Gorgonopsians have relatively short parietals compared to other 'advanced' therapsid clades, but they are still significantly longer than in biarmosuchians, extending beyond the pineal boss in all known taxa.

37. **Posterior process of parietal:** (0) elongate, extending between tabular and squamosal; (1) confined to skull roof. An elongate, tapering posterior process of the parietal

makes up part of the occiput in biarmosuchians and most gorgonopsians, extending between the tabular and squamosal. In *Clelandina*, *Dinogorgon*, *Leontosaurus*, *Rubidgea*, and *Ruhuhucerberus*, the parietal terminates dorsal to the interparietal, being effectively restricted to the skull roof.

38. **Occipital height:** (0) tall (greater or equal to 50% of maximum occipital width, measured from lateral edge of squamosal); (2) short and broad (less than 40% of maximum occipital width). A relatively low, broad occiput, with prominent transverse expansion of the squamosals, is present in the majority of rubidgeines, with only *Smilesaurus* and *Aelurognathus* exhibiting a tall, 'boxy' occiput. Among non-rubidgeine gorgonopsians, a low, broad occiput is also present in *Gorgonops*.

39. **Dorsal edge of tabular:** (0) narrow, tapering tip; (1) broad. The tabular remains fairly broad throughout its height in biarmosuchians, whereas in gorgonopsians the dorsal portion of the tabular is elongate, strongly tapering upwards. A reversal to the primitive condition occurs in various rubidgeines, however, including *Clelandina*, *Dinogorgon*, *Leontosaurus*, *Rubidgea*, and *Ruhuhucerberus*.

40. **Supraoccipital shape:** (0) height roughly half of width; (1) very low (less than third of width) and ribbon-like. The supraoccipital is generally roughly rectangular in biarmosuchians and gorgonopsians, but in most rubidgeines is extraordinarily low and broad, such that it forms a ribbon-like structure in posterior view.

41. **Mandibular ramus morphology:** (0) straight; (1) sinusoidal. The mandibular ramus in biarmosuchians is essentially straight, with only minor curvature between the symphysis and point of articulation. Gorgonopsians, by contrast, have a distinctly sinusoidal angulation to the jaw in ventral view, curving around the canine and the transition between the dentary and postdentary bones.

42. **Splenial process:** (0) absent; (1) present. The mandibular symphysis of gorgonopsians is very robust compared with that of biarmosuchians and therocephalians. In most gorgonopsians, the posteroventral tip of the symphysis (i.e., the midline suture of the splenials) forms a thickened, posteriorly-directed process. This process is particularly well-developed in *Clelandina* and *Rubidgea*.

43. **Dentary postcanines:** (0) present; (1) absent. Lower postcanine dentition is completely absent in *Clelandina*, *Leontosaurus*, and *Rubidgea*.

44. **Coronoid process of dentary:** (0) tightly appressed to dorsal margin of mandible; (1) free-standing process. Separation of the dentary coronoid process from the main body of the mandible is a synapomorphy of Theriodontia: a free-standing process is present in gorgonopsians and therocephalians, but not biarmosuchians.

45. **Lateral surface of reflected lamina:** (0) lobate sculpturing; (1) well-developed dorsoventrally-oriented bar, with weakly-developed crossbar. The major therapsid clades each display a unique morphology of the reflected lamina. In gorgonopsians this takes the form of a vaguely cruciate pattern, with a well-developed, dorsoventrally-oriented bar above a weaker anteroposteriorly-oriented one.

46. **Dorsal edge of reflected lamina:** (0) free; (1) attached to angular body. The reflected lamina is typically attached to the main body of the angular at its anterior edge in therapsids. In gorgonopsians, it is more extensively attached, such that both the anterior and dorsal margins are attached to the rest of the mandible.

47. **Reflected lamina position:** (0) immediately adjacent to the point of jaw articulation; (1) broadly separated from point of jaw articulation by the body of the angular. In biarmosuchians, the reflected lamina is so close to the articular that it almost overlaps the jaw articulation laterally. Gorgonopsians differ from this in having a broad lateral exposure of the main body of the angular, separating the reflected lamina from the articular.

## Character matrix

| | |
|---|---|
| *Biarmosuchus tener* | 0000000000000000000001000100000000000100000000 |
| *Hipposaurus boonstrai* | 0001000000000000000000001000000000100100000000 |
| *Eriphostoma microdon* | 10111011001011001000000000000?0111?0001001111 |
| *Gorgonops torvus* | 111111110010111000000000000000000011101011101111 |
| *Arctognathus curvimola* | 1111101111211110000000100000100001110000011?1111 |
| *Lycaenops ornatus* | 1?111?11101111100000000000010000001111000011011111 |
| *Smilesaurus ferox* | 11111212101100100000000[01]000100000011100001101111 |
| *Aelurognathus tigriceps* | 11111311101101100001[01]1[01]1001000000011100001101111 |
| *Ruhuhucerberus haughtoni* | 11111311111110110000111[01]11010100001111110?????? |
| *Sycosaurus laticeps* | 1111121111210010000101110111000001110101111???? |
| *Sycosaurus nowaki* | 1111121111210010000101110111[01]10000011101011101111 |
| *Leontosaurus vanderhorsti* | 11111311112100101101011200211111101111111111111 |
| *Dinogorgon rubidgei* | 1???1?11112100111111121210211111101111111101111 |
| *Rubidgea atrox* | 1?????1211210011111121210211111101111111111111 |
| *Clelandina rubidgei* | 11111312112100111111121210211111101111111111111 |

## Funding

Support for my research was provided by the Deutsche Forschungsgemeinschaft (KA 4133/1-1), the Museum für Naturkunde (Berlin), a Gerstner-Kalbfleisch fellowship at the American Museum of Natural History and Richard Gilder Graduate School, and a Sofja Kovalevskaja award from the Alexander von Humboldt Foundation (donated by the German Federal Ministry for Education and Research) to Jörg Fröbisch. The funders had no role in study design, data collection and analysis, decision to publish, or preparation of the manuscript.

## Grant Disclosures

The following grant information was disclosed by the authors:
Deutsche Forschungsgemeinschaft: KA 4133/1-1.

## Competing Interests

The authors declare that they have no competing interests.

## Author Contributions

- Christian F. Kammerer analyzed the data, wrote the paper, prepared figures and/or tables.

## Data Deposition

The research in this article did not generate any raw data beyond the phylogenetic character matrix, which is included in the current contribution both within the text and as a supplemental file.

This publication was registered for an LSID: urn:lsid:zoobank.org:pub:B3A78FFF-D388-413C-B9E9-1AFD7BEC195E.

## Supplemental Information

Supplemental information for this article can be found online at http://dx.doi.org/10.7717/peerj.1608#supplemental-information.

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
