# Peer review of "Systematics of the Rubidgeinae (Therapsida: Gorgonopsia)"

_PeerJ, doi:10.7717/peerj.1608_

## Round 0.1 · original submission · Minor Revisions

This is generally an excellent paper that requires only modifications in some of the illustrations. As suggested by one reviewer, cranial openings, the outline of the cranium, and certain projecting features should be shown in heavier lines on the outline reconstructions. The author might also consider some shading to show the cranial openings. One minor detail: Citations to Huene's papers in the text should read "Huene" rather than "Von Huene" (the references in the bibliography are correctly cited).

·

Basic reporting

This is a monumental piece of work, well worth publishing. Overall, the manuscript is well written, well thought out, the phylogenetic portion is fine, and the overall interpretations and conclusions are fine.
I have a problem with differentiating "large gorgonopsians" from "gigantic gorgonopsians" in the diagnosis. I would avoid such differentiations as unnecessary, labeling 35cm skulled taxa "large", but 40 cm skulled taxa "gigantic". Numerical values are also generally to be avoided in a diagnosis, since a smaller or larger member of each taxon may appear in the future.
However, the most serious problems reside in the reconstructions of the various taxa. 1) The author should orient the dorsal and ventral views in the direction for direct comparisons. In fact, ideally, all four views should be in one figure, all at the same scale, so that the can provide a holistic view of the animal. 2) No scale is provided for the reconstructions. 3) Occcipital views are not at the same scale as the other views. 3) Line weight hierarchy is not maintained in the figures. Thus, the thickness of edges of bones should be different from internal features like bosses and ridges. Openings, like orbits, internal nares, external nares, temporal fenestrae should have the thickest lines. 4) THE DIFFERENT VIEW OF THE SKULL RECONSTRUCTIONS DO NOT MATCH, in fact they are very inconsistent not only terms of the particular bones, but also skull outlines, snout-occipital condyle lengths, etc. This is probably the most serious issue with the paper, and does need extensive reworking, since there are reconstructions for most of the taxa described in this monograph. 5) I may not be very familiar with the details of the rubidgeine skull, but it is very unlikely that the quadrate and quadratojugal would not be visible in any reconstructed view. Instead, a bland ventral shelf of the squamosal is presented in these reconstructions in ventral view, and in occipital views. VERY UNLIKELY. If the author does not have the information, then this aspect of the morphology should be dotted in.

Experimental design

NA

Validity of the findings

The results of this research are sound and well thought out.

Additional comments

Same as the Basic Reporting. I know that the issue that I am bringing up is difficult to resolve, but the paper, as is, suffers significantly in the reconstructed representations of the taxa that are being reconsidered.

·

Basic reporting

The paper undertakes a thorough taxonomic revision of Rubidgeia gorgonopsians. The author has personally examined most material in different museum collections around the world. This is a huge undertaking and this author is to be commended on his very diligent and thorough paper. The paper also provides the first cladistic phylogenetic analysis of Rubidgeia gorgonopsians.
The paper is well constructed and written in an accessible style, and the text is richly supported by useful photographs and interpretive diagrams.

Experimental design

The author has made a concerted effort to study as much cranial material as possible of Rubidgeia gorgonopsians in museum collections around the world. Good and diligent observations on the morphology of the specimens have been well recorded and the attributes of the different specimens have been compared with each other.

Validity of the findings

The taxonomy of gorgonopsid therapsids has been a problem in therapsid palaeobiology as the morphology of the different specimens is remarkably similar. The paper, which results from the ambition to personally study almost all the Rubidgeia gorgonopsids in museum collections around the world, is a huge contribution to resolve and understand the – up till now – relatively poorly understood gorgonopsian taxonomy, palaeobiology and phylogenetic relationships. This mammoth undertaking is an important contribution to therapsid palaeontology and should be published. For the size of the project there are relatively few typographical and editorial modifications and these are indicated in the attached manuscript.

Additional comments

This paper has been well executed and should be published subject to minor corrections which are indicated in the attached pd of the text.
Figures are useful and relevant. Please label characters on line drawings such as premax foramen and interpterygoidal vacuity which are referred to in the text.

---

## Round 0.2 · Minor Revisions

The author's responses to the reviewers' comments are thoughtful and satisfactory. As Editor I would just request that the author include a brief paragraph with his entire appropriate comments about the reconstructions and the decisions regarding figure layout. With that small addition, the manuscript will be recommended for acceptance.

---

## Round 0.3 · accepted · Accept

The revised version of the manuscript is acceptable for publication.